# Unified Analyses for Hierarchical Federated Learning: Topology Selection under Data Heterogeneity

**Ziyi Zhou**[1], **Yipeng Li**[2] **& Xinchen Lyu**[1*]
[1]National Engineering Research Center for Mobile Network Technologies,
Beijing University of Posts and Telecommunications
[2]Tsinghua Shenzhen International Graduate School, Tsinghua University
`{zhouzy0821, lvxinchen}@bupt.edu.cn, liyp25@mails.tsinghua.edu.cn`

## Abstract

Hierarchical Federated Learning (HFL) addresses critical scalability limitations in conventional federated learning by incorporating intermediate aggregation layers, yet optimal topology selection across varying data heterogeneity conditions and network conditions remains an open challenge. This paper establishes the first unified convergence framework for all four HFL topologies (Star-Star, Star-Ring, Ring-Star, and Ring-Ring) with full/partial client participation under non-convex objectives and different intra/inter-group data heterogeneity. Our theoretical analysis reveals three fundamental principles for topology selection: (1) The top-tier aggregation topology exerts greater influence on convergence than the intra-group topology, with ring-based top-tier configurations generally outperforming star-based alternatives; (2) Optimal topology strongly depends on client grouping characteristics, where Ring-Star excels with numerous small groups while Star-Ring is superior for large, client-dense clusters; and (3) Inter-group heterogeneity dominates convergence dynamics across all topologies, necessitating clustering strategies that minimize inter-group divergence. Extensive experiments on CIFAR-10/CINIC-10/Fashion-MNIST/SST-2 with ResNet-18/VGG-9/ResNet-10/MLP validate these insights, and provide practitioners with theoretically grounded guidance for HFL system design in real-world deployments.

## 1 Introduction

Federated Learning (FL) (McMahan et al., 2017) has revolutionized collaborative machine learning by enabling distributed model training across decentralized devices while preserving data privacy. However, conventional single-tier FL faces critical scalability challenges in large-scale deployments, including communication bottlenecks, synchronization latency, and vulnerability to single-point failures. Hierarchical Federated Learning (HFL) (Liu et al., 2020) has emerged as a promising paradigm, introducing intermediate aggregation layers (such as edge servers or cluster heads) to form a two/multi-tier architecture that distributes the coordination burden for massive deployment. Despite its promise, the theoretical understanding of HFL remains nascent, particularly under realistic conditions of data heterogeneity and diverse hierarchical topologies.

In two-tier HFL frameworks, each level of aggregation can be performed via *parallel updates (star topology) or sequential updates (ring topology)*, yielding four distinct configurations: Star-Star, Star-Ring, Ring-Star, and Ring-Ring (see Figure 1). These topological choices fundamentally influence the convergence dynamics, robustness to data heterogeneity, and communication efficiency. For instance, star aggregation enables parallel client updates but may suffer from abrupt synchronization of divergent models, while ring updates propagate sequentially, potentially mitigating client drift in non-IID settings (Li & Lyu, 2023; 2025).

**Literature Review.** Existing theoretical analyses of HFL have largely focused on the Star-Star topology on non-convex functions (Zhou & Cong, 2019; Castiglia et al., 2021; Wang et al., 2022),

---

*Xinchen Lyu is the corresponding author.

data heterogeneity (Wang et al., 2022), partial client participation (Jiang & Zhu, 2024), and other variants (Liu et al., 2022; Yang et al., 2023). Some recent works explore Star-Ring topology (Chen et al., 2020; Lee et al., 2020; Fang et al., 2022; Ding et al., 2024) and the Ring-Star topology (Chaoyang et al., 2020; Huang et al., 2024; Yan et al., 2025). However, a unified convergence analysis that compares all four topologies is still lacking. This theoretical gap impedes informed topology selection in practical deployments, where system performance is highly sensitive to data distribution and network conditions.

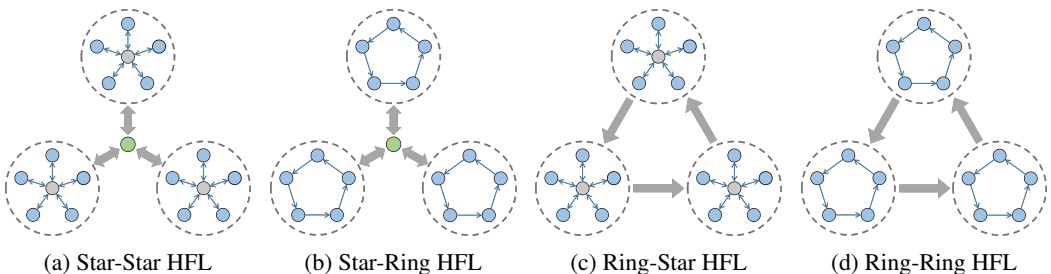

|  (a) Star-Star HFL | (b) Star-Ring HFL | (c) Ring-Star HFL | (d) Ring-Ring HFL |

Figure 1: Different topology configurations of HFL

**Research Question.** The central research problem is:

*How should practitioners select the optimal HFL topology configuration when facing varying degrees of intra-group and inter-group data heterogeneity, diverse client grouping characteristics, and constrained network conditions?*

This research problem is of practical importance for system convergence. For example, in systems with high inter-group heterogeneity (such as clients clustered by geographic region with distinct data distributions), selecting star topology at the top tier may amplify inter-group divergence through abrupt parallel synchronization. Conversely, ring aggregation at the top tier enables gradual, sequential alignment that may better accommodate distributional differences. This topology selection problem is further complicated when considering diverse client grouping characteristics in HFL.

**Analytical Challenges.** Establishing a comprehensive convergence framework encompassing all four HFL topologies presents key challenges:

- HFL exhibits a cascading heterogeneity where inter- and intra-group data distributions interact non-trivially. Since local model divergence directly exacerbates global inconsistency, convergence bounds must explicitly account for the interaction between intra- and inter-group divergence.

- The two-tier aggregation architecture creates a cross-tier dependency where updates at one tier directly influence error propagation at the other, making the effective learning rate topology-dependent.

- Star and ring topologies introduce fundamentally different statistical properties (unbiased high-variance vs. biased low-variance sequential updates) that compound across layers. These topology-specific error propagation patterns require sophisticated analysis to track how topology choices modulate the cross-tier bias-variance tradeoff.

**Contributions.** This paper establishes a unified theoretical framework for analyzing and comparing all four HFL topology configurations (Star-Star, Star-Ring, Ring-Star, and Ring-Ring) under non-convex optimization and varying data heterogeneity conditions (see Theorem 1 and Table 1). Our convergence bounds explicitly quantify the effects of key system parameters, including number of groups ($G$), clients per group ($M$), local steps ($K$), and group rounds ($P$), and reveal how topology choices interact with intra/inter-group heterogeneity to shape convergence behavior. Furthermore, we provide a proof extending Theorem 1 to settings with a random-shuffle execution order. The framework also extends to partial client participation settings with corresponding convergence guarantees (see Theorem 2). Our analysis yields three insights for HFL system design:

*(1) HFL prioritizes scalability over convergence acceleration.* Counterintuitively, HFL is primarily valuable for enabling large-scale deployments where single-tier FL becomes impractical, rather than inherently accelerating convergence. Crucially, HFL with ring aggregation at the top tier (Ring-Star,

Table 1: Comparison of convergence rates under full participation. We omit constants and some terms. Notation: $R$ (global rounds); $G$ (groups); $P$ (group rounds); $M$ (clients/group); $K$ (local steps); $\sigma$ (variance); $\delta, \zeta$ (inter/intra-heterogeneity); $L$ (smoothness).

| Topology | | Convergence Rate[1] |
|---|---|---|
| **PFL** | (Koloskova et al., 2020)[2] | $\frac{L^{1/2}\sigma}{\sqrt{MKR}} + \frac{L^{2/3}\sigma^{2/3}}{(KR^2)^{\frac{1}{3}}} + \frac{L^{2/3}\delta^{2/3}}{(R^2)^{\frac{1}{3}}}$ |
| **SFL** | (Li & Lyu, 2023) | $\frac{L^{1/2}\sigma}{\sqrt{MKR}} + \frac{L^{2/3}\sigma^{2/3}}{(MKR^2)^{\frac{1}{3}}} + \frac{L^{2/3}\delta^{2/3}}{(R^2)^{\frac{1}{3}}}$ |
| **Star-Star** | Theorem 1 | $\frac{L^{1/2}\sigma}{\sqrt{GPMKR}} + \frac{L^{2/3}\sigma^{2/3}}{(P^2KR^2)^{\frac{1}{3}}} + \frac{L^{2/3}\sigma^{2/3}}{(PMKR^2)^{\frac{1}{3}}} + \frac{L^{2/3}\zeta^{2/3}}{(P^2R^2)^{\frac{1}{3}}} + \frac{L^{2/3}\delta^{2/3}}{(R^2)^{\frac{1}{3}}}$ |
| | (Wang et al., 2022) | $\frac{L^{1/2}\sigma}{\sqrt{GPMKR}} + \frac{L^{2/3}\sigma^{2/3}}{(P^2KR^2)^{\frac{1}{3}}} + \frac{L^{2/3}\sigma^{2/3}}{(PMKR^2)^{\frac{1}{3}}} + \frac{L^{2/3}\zeta^{2/3}}{(P^2R^2)^{\frac{1}{3}}} + \frac{L^{2/3}\delta^{2/3}}{(R^2)^{\frac{1}{3}}}$ |
| **Star-Ring**[3] | Theorem 1 | $\frac{L^{1/2}\sigma}{\sqrt{GPMKR}} + \frac{L^{2/3}\sigma^{2/3}}{(PMKR^2)^{\frac{1}{3}}} + \frac{L^{2/3}\zeta^{2/3}}{(P^2R^2)^{\frac{1}{3}}} + \frac{L^{2/3}\delta^{2/3}}{(R^2)^{\frac{1}{3}}}$ |
| **Ring-Star**[3] | Theorem 1 | $\frac{L^{1/2}\sigma}{\sqrt{GPMKR}} + \frac{L^{2/3}\sigma^{2/3}}{(G^2P^2KR^2)^{\frac{1}{3}}} + \frac{L^{2/3}\sigma^{2/3}}{(GPMKR^2)^{\frac{1}{3}}} + \frac{L^{2/3}\zeta^{2/3}}{(G^2P^2R^2)^{\frac{1}{3}}} + \frac{L^{2/3}\delta^{2/3}}{(R^2)^{\frac{1}{3}}}$ |
| **Ring-Ring** | Theorem 1 | $\frac{L^{1/2}\sigma}{\sqrt{GPMKR}} + \frac{L^{2/3}\sigma^{2/3}}{(GPMKR^2)^{\frac{1}{3}}} + \frac{L^{2/3}\zeta^{2/3}}{(G^2P^2R^2)^{\frac{1}{3}}} + \frac{L^{2/3}\delta^{2/3}}{(R^2)^{\frac{1}{3}}}$ |

[1] Terms highlighted in color, e.g., $G$ and $G^2$, denote the additional denominator terms introduced by the top-level Ring topology (i.e., Ring-Star vs. Star-Star, Ring-Ring vs. Star-Ring).
[2] FedAvg, non-IID, non-convex. If there is only one client in each group (i.e., $P = 1$, $M = 1$ and $\zeta = 0$), Star-Star reduces to FedAvg. In this case, our rate becomes $\mathcal{O}\left(\frac{L^{1/2}\sigma}{\sqrt{MKR}} + \frac{L^{2/3}\sigma^{2/3}}{(KR^2)^{1/3}} + \frac{L^{2/3}\delta^{2/3}}{(R^2)^{1/3}}\right)$ which matches that of FedAvg.
[3] Even though similar algorithms and some preliminary convergence analyses have been studied, the advanced convergence analyses are still missing for Star-Ring and Ring-Star. For example, Chen et al. (2020) and Yan et al. (2025) used the bounded gradient assumption, meaning the stochastic gradient is uniformly bounded by a constant.

Ring-Ring) consistently outperforms star-based counterparts under data heterogeneity. This reveals that carefully selected single-tier FL configurations may actually converge faster than two-tier HFL, positioning HFL as a solution for scalability constraints rather than a convergence accelerator.

*(2) Inter-group heterogeneity dominates convergence dynamics.* We establish that inter-group data divergence ($\delta$) exerts a more significant impact on convergence than intra-group heterogeneity ($\zeta$) across all four topologies. This finding fundamentally reshapes client clustering strategies, indicating that minimizing inter-group distributional differences should take precedence over optimizing intra-group homogeneity. Effective grouping, such as forming clusters with approximately IID inter-group distributions, would converge faster than fine-tuning intra-group training dynamics.

*(3) Optimal topology selection depends critically on group structure.* Our analysis reveals optimal topology selection depends critically on group structure. Ring-Star excels when numerous small groups exist, as sequential inter-group updates benefit from increased parallelism at the lower tier and fine-grained global alignment. Star-Ring is preferable for few large, client-dense clusters, where intra-group ring aggregation enables deep local refinement before global synchronization.

We validate these theoretical insights through extensive experiments using ResNet-18/VGG-9 on CIFAR-10, ResNet-18 on CINIC-10, ResNet-10 on Fashion-MNIST, and MLP on SST-2 under four distinct heterogeneity scenarios. The results consistently demonstrate accuracy gains from informed topology selection, with ring-based top-tier configurations showing particular advantages in heterogeneous environments. Beyond establishing the first convergence analysis for the previously unstudied Ring-Ring topology, our framework overcomes critical limitations of prior work by: (i) adopting consistent, general non-convex assumptions across all topologies, whereas previous studies relied on heterogeneous assumptions that prevented direct comparison; and (ii) deriving tighter convergence bounds of $\mathcal{O}(1/\sqrt{GPMKR})$ for Star-Ring and Ring-Star topologies, versus $\mathcal{O}(1/\sqrt{R})$ in prior

literature (Lee et al., 2020; Yan et al., 2025). These contributions provide system designers with theoretically grounded principles for topology selection based on specific deployment constraints.

## 2 CONVERGENCE THEORY

This section presents a unified convergence analysis of HFL under non-convex optimization objectives. In the following, we formalize the setup of HFL with four different topologies, introduce general assumptions, derive the convergence bounds, and extract actionable insights for topology selection in practical deployments.

### 2.1 SETUP

We begin by formalizing the HFL framework and the update mechanisms for each topology configuration. In two-tier HFL, the global objective is to minimize:

$$\min_{\mathbf{x} \in \mathbb{R}^d} \left\{ F(\mathbf{x}) = \frac{1}{G} \sum_{g=1}^{G} F_g(\mathbf{x}) = \frac{1}{G} \sum_{g=1}^{G} \frac{1}{M} \sum_{m=1}^{M} F_{g,m}(\mathbf{x}) \right\} \tag{1}$$

where $F_g$ denotes the average local objective function over all clients in group $g$ ($g \in [G]$), and $F_{g,m}$ denotes the local objective function of client $m$ ($m \in [M]$) in group $g$. Specifically, it is defined as $F_{g,m}(x) = \mathbb{E}_{\xi \sim \mathcal{D}_m}[f_m(x; \xi)]$, where $\mathcal{D}_m$ is the local dataset of client $m$.

In the full participation setting, the HFL process with four topology configurations operates according to distinct update rules:

*(1) Star-Star.* Each group $g$ initializes its model as $\mathbf{x}_{g,0}^{(r)} = \mathbf{x}^{(r)}$. Within each group, clients initialize their models as $\mathbf{x}_{g,p,m,0}^{(r)} = \mathbf{x}_{g,p}^{(r)}$, perform $K$ parallel local updates, and send updates to the group server for aggregation. After $P$ group updates, the global server aggregates group parameters to generate the next global parameters $\mathbf{x}^{(r+1)}$.

*(2) Star-Ring.* Each group $g$ initializes its model as $\mathbf{x}_{g,0}^{(r)} = \mathbf{x}^{(r)}$. Within each group, clients initialize their models from the previous client in sequence and perform $K$ local updates. The group server receives the latest parameters from the last client. After $P$ group updates, group servers send their updated parameters to the global server for aggregation.

*(3) Ring-Star.* Each group $g$ initializes its model with the latest parameters from the previous group. Within each group, clients initialize their models as $\mathbf{x}_{g,p,m,0}^{(r)} = \mathbf{x}_{g,p}^{(r)}$, perform $K$ parallel local updates, and send updates to the group server for aggregation. After $P$ group updates, group servers send their updated parameters to the next group in sequence.

*(4) Ring-Ring.* Each group $g$ initializes its model with the latest parameters from the previous group. Within each group, clients initialize their models from the previous client and perform $K$ local updates. The group server aggregates the latest parameters from the last client. After $P$ group updates, group servers send their updated parameters to the next group in sequence.

Crucially, for ring-based topologies, we distinguish two cases: (i) under *full participation*, the execution sequence is *fixed* (see Appendix A.1); and (ii) under *partial participation* (where $S_1$ groups and $S_2$ clients are active), the execution sequence is *random*.

### 2.2 ASSUMPTIONS

**Assumption 1** (*L*-Smoothness)**.** *Each local objective function $F_{g,m}$ is L-smooth, i.e., there exists one constant $L$ such that $\|\nabla F_{g,m}(\mathbf{x}) - \nabla F_{g,m}(\mathbf{y})\| \leq L\|\mathbf{x} - \mathbf{y}\|$, for all $g \in \{1, \ldots, G\}$, $m \in \{1, \ldots, M\}$, and $\mathbf{x}, \mathbf{y} \in \mathbb{R}^d$.*

**Assumption 2** (Bounded variance)**.** *For the local objective function $F_{g,m}$, the stochastic gradient $\nabla F_{g,m}(\mathbf{x}, \xi)$ computed using a mini-batch $\xi$, sampled from local dataset $\mathcal{D}_{g,m}$, is an unbiased estimate, i.e., $\mathbb{E}_{\xi \sim \mathcal{D}_{g,m}}[\nabla F_{g,m}(\mathbf{x}, \xi)] = \nabla F_{g,m}(\mathbf{x})$. The variance of the stochastic gradient at each client is bounded: $\mathbb{E}_{\xi \sim \mathcal{D}_{g,m}}[\|\nabla F_{g,m}(\mathbf{x}, \xi) - \nabla F_{g,m}(\mathbf{x})\|^2] \leq \sigma^2$.*

**Assumption 3** (Bounded Inter-Group Heterogeneity). *There exists one constant $\delta^2$ such that*

$$\frac{1}{G} \sum_{g=1}^{G} \|\nabla F(\mathbf{x}) - \nabla F_g(\mathbf{x})\|^2 \leq \delta^2. \tag{2}$$

**Assumption 4** (Bounded Intra-Group Heterogeneity). *For any $g \in \{1, 2, \ldots, G\}$, there exist constants $\zeta_g^2$ such that*

$$\frac{1}{M} \sum_{m=1}^{M} \|\nabla F_{g,m}(\mathbf{x}) - \nabla F_g(\mathbf{x})\|^2 \leq \zeta_g^2. \tag{3}$$

*Furthermore, we define the average intra-group heterogeneity as $\zeta^2 := \frac{1}{G} \sum_{g=1}^{G} \zeta_g^2$.*

The first two assumptions are standard in non-convex optimization (Ghadimi & Lan, 2013; Bottou et al., 2018). Assumptions 3 and 4 extend standard FL analysis to the hierarchical setting, explicitly bounding inter- and intra-group data heterogeneity (Wang & Ji, 2022). Notably, the global heterogeneity can be naturally decomposed into the inter-group and intra-group terms defined above. Specifically, we have: $\frac{1}{G} \sum_{g=1}^{G} \frac{1}{M} \sum_{m=1}^{M} \|\nabla F_{g,m}(\mathbf{x}) - \nabla F(\mathbf{x})\|^2 = \frac{1}{G} \sum_{g=1}^{G} \|\nabla F_g(\mathbf{x}) - \nabla F(\mathbf{x})\|^2 + \frac{1}{G} \sum_{g=1}^{G} \frac{1}{M} \sum_{m=1}^{M} \|\nabla F_{g,m}(\mathbf{x}) - \nabla F_g(\mathbf{x})\|^2$. This equality is an implementation of the law of total variance. This shows that inter- and intra-group heterogeneity are, in fact, a partition of the global heterogeneity.

## 2.3 CONVERGENCE ANALYSIS

**Theorem 1.** *(Convergence with Full Participation). Let $A = F(x^{(0)}) - F^*$ denote the initial optimality gap. Here $\bar{x}^R$ is defined as a model uniformly sampled from the $x^{(0)}, \ldots, x^{(R-1)}$ of previous iterations, and $\lesssim$ hides universal constants. Under Assumptions 2–4, the following convergence bounds hold for each HFL topology.*

**Star-Star**: *There exist $\tilde{\eta} = PK\eta$, and $\tilde{\eta} \leq \frac{1}{\sqrt{30}L}$, such that*

$$\mathbb{E}[\|\nabla F(\bar{x}^{(R)})\|^2] \lesssim \frac{A}{\tilde{\eta}R} + \frac{L\tilde{\eta}\sigma^2}{GPMK} + \frac{L^2\tilde{\eta}^2\sigma^2}{P^2K} + \frac{L^2\tilde{\eta}^2\sigma^2}{PMK} + \frac{L^2\tilde{\eta}^2\zeta^2}{P^2} + L^2\tilde{\eta}^2\delta^2. \tag{4}$$

**Star-Ring**: *There exist $\tilde{\eta} = PMK\eta$, and $\tilde{\eta} \leq \frac{1}{\sqrt{60}L}$, such that*

$$\mathbb{E}[\|\nabla F(\bar{x}^{(R)})\|^2] \lesssim \frac{A}{\tilde{\eta}R} + \frac{L\tilde{\eta}\sigma^2}{GPMK} + \frac{L^2\tilde{\eta}^2\sigma^2}{PMK} + \frac{L^2\tilde{\eta}^2\zeta^2}{P^2} + L^2\tilde{\eta}^2\delta^2. \tag{5}$$

**Ring-Star**: *There exist $\tilde{\eta} = GPK\eta$, and $\tilde{\eta} \leq \frac{1}{\sqrt{60}L}$, such that*

$$\mathbb{E}[\|\nabla F(\bar{x}^{(R)})\|^2] \lesssim \frac{A}{\tilde{\eta}R} + \frac{L\tilde{\eta}\sigma^2}{GPMK} + \frac{L^2\tilde{\eta}^2\sigma^2}{G^2P^2K} + \frac{L^2\tilde{\eta}^2\sigma^2}{GPMK} + \frac{L^2\tilde{\eta}^2\zeta^2}{G^2P^2} + L^2\tilde{\eta}^2\delta^2. \tag{6}$$

**Ring-Ring**: *There exist $\tilde{\eta} = GPMK\eta$, and $\tilde{\eta} \leq \frac{1}{10L}$, such that*

$$\mathbb{E}[\|\nabla F(\bar{x}^{(R)})\|^2] \lesssim \frac{A}{\tilde{\eta}R} + \frac{L\tilde{\eta}\sigma^2}{GPMK} + \frac{L^2\tilde{\eta}^2\sigma^2}{GPMK} + \frac{L^2\tilde{\eta}^2\zeta^2}{G^2P^2} + L^2\tilde{\eta}^2\delta^2. \tag{7}$$

**Theorem 2.** *(Convergence with Partial Participation). Under Assumptions 2–4, where in each global round, a subset of $S_1$ groups are sampled uniformly at random from the $G$ total groups, and within each selected group, a subset of $S_2$ clients are sampled uniformly at random from the $M$ total clients, the following convergence bounds hold for each HFL topology.*

**Star-Star**: *There exist $\tilde{\eta} = PK\eta$, and $\tilde{\eta} \leq \frac{1}{10L}$, such that*

$$\mathbb{E}[\|\nabla F(\bar{x}^{(R)})\|^2] \lesssim \frac{A}{\tilde{\eta}R} + \frac{L\tilde{\eta}\sigma^2}{S_1PS_2K} + \frac{L\tilde{\eta}\zeta^2}{S_2} + \frac{L\tilde{\eta}\delta^2}{S_1} \tag{8}$$

$$+ \frac{L^2\tilde{\eta}^2\sigma^2}{P^2K} + \frac{L^2\tilde{\eta}^2\sigma^2}{PS_2K} + \frac{L^2\tilde{\eta}^2\zeta^2}{P^2} + \frac{L^2\tilde{\eta}^2\zeta^2}{S_2} + L^2\tilde{\eta}^2\delta^2. \tag{9}$$

**Star-Ring**: *There exist $\tilde{\eta} = PMK\eta$, and $\tilde{\eta} \leq \frac{1}{15L}$, such that*

$$\mathbb{E}[\|\nabla F(\bar{x}^{(R)})\|^2] \lesssim \frac{A}{\tilde{\eta}R} + \frac{L\tilde{\eta}\sigma^2}{S_1PS_2K} + \frac{L\tilde{\eta}\zeta^2}{S_2} + \frac{L\tilde{\eta}\delta^2}{S_1} + \frac{L^2\tilde{\eta}^2\sigma^2}{PS_2K} + \frac{L^2\tilde{\eta}^2\zeta^2}{S_2} + L^2\tilde{\eta}^2\delta^2. \tag{10}$$

**Ring-Star**: *There exist $\tilde{\eta} = GPK\eta$, and $\tilde{\eta} \leq \frac{1}{15L}$, such that*

$$\mathbb{E}[\|\nabla F(\bar{x}^{(R)})\|^2] \lesssim \frac{A}{\tilde{\eta}R} + \frac{L\tilde{\eta}\sigma^2}{S_1 P S_2 K} + \frac{L\tilde{\eta}\zeta^2}{S_2} + \frac{L\tilde{\eta}\delta^2}{S_1} \tag{11}$$

$$+ \frac{L^2\tilde{\eta}^2\sigma^2}{S_1^2 P^2 K} + \frac{L^2\tilde{\eta}^2\sigma^2}{S_1 P S_2 K} + \frac{L^2\tilde{\eta}^2\zeta^2}{S_1^2 P^2} + \frac{L^2\tilde{\eta}^2\zeta^2}{S_2} + \frac{L^2\tilde{\eta}^2\delta^2}{S_1}. \tag{12}$$

**Ring-Ring**: *There exist $\tilde{\eta} = GPMK\eta$, and $\tilde{\eta} \leq \frac{1}{20L}$, such that*

$$\mathbb{E}[\|\nabla F(\bar{x}^{(R)})\|^2] \lesssim \frac{A}{\tilde{\eta}R} + \frac{L\tilde{\eta}\sigma^2}{S_1 P S_2 K} + \frac{L\tilde{\eta}\zeta^2}{S_2} + \frac{L\tilde{\eta}\delta^2}{S_1} + \frac{L^2\tilde{\eta}^2\sigma^2}{S_1 P S_2 K} + \frac{L^2\tilde{\eta}^2\zeta^2}{S_2} + \frac{L^2\tilde{\eta}^2\delta^2}{S_1}. \tag{13}$$

**Effective Learning Rate.** Theorems 1 and 2 introduce a topology-dependent effective learning rate, denoted by $\tilde{\eta}$, which incorporates key architectural parameters: the number of groups $G$, group rounds $P$, clients per group $M$, local update steps $K$, and global rounds $R$. This effective learning rate captures how hierarchical updates affect convergence. The derived bounds comprise an *optimization term* that decreases with $R$, and *error terms* from stochastic noise and data heterogeneity. A larger $\tilde{\eta}$ accelerates optimization but amplifies errors. Corollary 1 and Corollary 2 specify an optimal $\tilde{\eta}$ that minimizes the overall bound.

**Corollary 1.** *(Convergence with full participation under effective learning rate). With the learning rate $\tilde{\eta}$, the convergence rate satisfies the following, where $\mathcal{O}(\cdot)$ hides absolute constants:*

$$\mathbb{E}[\|\nabla F(\bar{x}^{(R)})\|^2] = \mathcal{O}\left(\frac{LA}{R} + \frac{(L\sigma^2 A)^{1/2}}{\sqrt{GPMKR}} + \mathcal{T}\right), \tag{14}$$

*and $\mathcal{T}$ denotes the topology-dependent terms defined as follows:*

**Star-Star**:

$$\mathcal{T} = \frac{(L^2 A^2 \sigma^2)^{1/3}}{(P^2 K R^2)^{1/3}} + \frac{(L^2 A^2 \sigma^2)^{1/3}}{(PMKR^2)^{1/3}} + \frac{(L^2 A^2 \zeta^2)^{1/3}}{(P^2 R^2)^{1/3}} + \frac{(L^2 A^2 \delta^2)^{1/3}}{(R^2)^{1/3}}. \tag{15}$$

**Star-Ring**:

$$\mathcal{T} = \frac{(L^2 A^2 \sigma^2)^{1/3}}{(PMKR^2)^{1/3}} + \frac{(L^2 A^2 \zeta^2)^{1/3}}{(P^2 R^2)^{1/3}} + \frac{(L^2 A^2 \delta^2)^{1/3}}{(R^2)^{1/3}}. \tag{16}$$

**Ring-Star**:

$$\mathcal{T} = \frac{(L^2 A^2 \sigma^2)^{1/3}}{(G^2 P^2 K R^2)^{1/3}} + \frac{(L^2 A^2 \sigma^2)^{1/3}}{(GPMKR^2)^{1/3}} + \frac{(L^2 A^2 \zeta^2)^{1/3}}{(G^2 P^2 R^2)^{1/3}} + \frac{(L^2 A^2 \delta^2)^{1/3}}{(R^2)^{1/3}}. \tag{17}$$

**Ring-Ring**:

$$\mathcal{T} = \frac{(L^2 A^2 \sigma^2)^{1/3}}{(GPMKR^2)^{1/3}} + \frac{(L^2 A^2 \zeta^2)^{1/3}}{(G^2 P^2 R^2)^{1/3}} + \frac{(L^2 A^2 \delta^2)^{1/3}}{(R^2)^{1/3}}. \tag{18}$$

**Corollary 2.** *(Convergence with partial participation under effective learning rate). With the learning rate $\tilde{\eta}$, the convergence rate satisfies the following, where $\mathcal{O}(\cdot)$ hides absolute constants:*

$$\mathbb{E}[\|\nabla F(\bar{x}^{(R)})\|^2] = \mathcal{O}\left(\frac{LA}{R} + \frac{(LA\sigma^2)^{1/2}}{\sqrt{S_1 P S_2 K R}} + \frac{(LA\zeta^2)^{1/2}}{\sqrt{S_2 R}} + \frac{(LA\delta^2)^{1/2}}{\sqrt{S_1 R}} + \mathcal{T}\right), \tag{19}$$

*and $\mathcal{T}$ denotes the topology-dependent terms defined as follows:*

**Star-Star**:

$$\mathcal{T} = \frac{(L^2 A^2 \sigma^2)^{1/3}}{(P^2 K R^2)^{1/3}} + \frac{(L^2 A^2 \sigma^2)^{1/3}}{(PS_2 K R^2)^{1/3}} + \frac{(L^2 A^2 \zeta^2)^{1/3}}{(P^2 R^2)^{1/3}} + \frac{(L^2 A^2 \zeta^2)^{1/3}}{(S_2 R^2)^{1/3}} + \frac{(L^2 A^2 \delta^2)^{1/3}}{(R^2)^{1/3}}. \tag{20}$$

*Star-Ring*:

$$\mathcal{T} = \frac{(L^2 A^2 \sigma^2)^{1/3}}{(PS_2 KR^2)^{1/3}} + \frac{(L^2 A^2 \zeta^2)^{1/3}}{(S_2 R^2)^{1/3}} + \frac{(L^2 A^2 \delta^2)^{1/3}}{(R^2)^{1/3}}. \tag{21}$$

*Ring-Star*:

$$\mathcal{T} = \frac{(L^2 A^2 \sigma^2)^{1/3}}{(S_1^2 P^2 KR^2)^{1/3}} + \frac{(L^2 A^2 \sigma^2)^{1/3}}{(S_1 PS_2 KR^2)^{1/3}} + \frac{(L^2 A^2 \zeta^2)^{1/3}}{(S_1^2 P^2 R^2)^{1/3}} + \frac{(L^2 A^2 \zeta^2)^{1/3}}{(S_2 R^2)^{1/3}} + \frac{(L^2 A^2 \delta^2)^{1/3}}{(S_1 R^2)^{1/3}}. \tag{22}$$

*Ring-Ring*:

$$\mathcal{T} = \frac{(L^2 A^2 \sigma^2)^{1/3}}{(S_1 PS_2 KR^2)^{1/3}} + \frac{(L^2 A^2 \zeta^2)^{1/3}}{(S_2 R^2)^{1/3}} + \frac{(L^2 A^2 \delta^2)^{1/3}}{(S_1 R^2)^{1/3}}. \tag{23}$$

## 2.4 KEY IMPLICATIONS

**The Top-Tier Dominance Principle.** Contrary to intuition, the synchronization mechanism at the global tier influences convergence significantly more than intra-group topology. This principle is quantitatively demonstrated in Corollaries 1 and 2. Specifically, for ring-based top-tier topologies, the error terms include an additional scaling factor of $G$ or $S_1$ in the denominators of the SGD variance along with the intra-group heterogeneity terms, under full or partial participation respectively. This also implies that under partial participation, increasing the number of participating groups or clients accelerates convergence. Furthermore, random shuffling introduces additional factors $S_1$ and $S_2$ to the heterogeneity terms (Li & Lyu, 2023; 2025). Consequently, ring-based updates are inherently more robust to both stochastic noise and data heterogeneity. Such robustness manifests in two crucial ways: (i) Ring-based top-tier configurations (Ring-Star, Ring-Ring) consistently outperform star-based alternatives under data heterogeneity, with the gap widening as inter-group divergence increases. (ii) The performance difference between top-tier topologies exceeds that between lower-tier configurations. For example, Ring-Star typically outperforms Star-Star by a larger margin than Star-Ring outperforms Star-Star, despite both differing only in the lower tier.

**Inter-Group Heterogeneity as the Fundamental Bottleneck.** Our analysis quantitatively establishes that inter-group heterogeneity ($\delta$) is the primary convergence bottleneck across all topologies. For instance, under full participation, while all topologies share the same asymptotic convergence rate of $\mathcal{O}(1/\sqrt{GPMKR})$, the practical convergence speed is dominated by inter-group divergence, which decays slowly at $\mathcal{O}(\frac{(L^2 A^2 \delta^2)^{1/3}}{R^{2/3}})$ regardless of topology choice. Intra-group heterogeneity ($\zeta$) decays significantly faster—particularly in ring-based top-tier configurations $\mathcal{O}(\frac{(L^2 A^2 \zeta^2)^{1/3}}{R^{2/3} G^{2/3} P^{2/3}})$—making it a secondary concern compared to inter-group divergence. This insight provides a principled foundation for system design: to accelerate convergence in heterogeneous environments, minimizing inter-group divergence should be prioritized. Practical strategies such as intelligent client clustering, e.g., grouping clients with complementary data distributions to approximate the global distribution (Zeng et al., 2022), are more impactful than optimizing local training dynamics within groups.

**Topology-Structure Compatibility Principle.** The optimal topology selection depends critically on the underlying client grouping structure, creating a fundamental design trade-off. Ring-Star excels with numerous small groups. When clients naturally form many small clusters (e.g., IoT devices, retail outlets), Ring-Star leverages parallelism at the lower tier while benefiting from the smoothing effect of sequential global updates. Its convergence rate improves dramatically with increasing $G$, making it ideal for deployments with abundant but sparse client clusters. Star-Ring dominates with few large clusters. In settings with limited but data-rich clusters, Star-Ring's intra-group ring aggregation enables deeper local refinement before global synchronization, producing higher-quality group models. This topology shows diminishing returns as $G$ increases beyond a certain point. Star-Star consistently underperforms. Despite its conceptual simplicity, the double averaging in Star-Star significantly dampens the effective learning rate, making it the least efficient configuration across all heterogeneity scenarios.

**Wall-Clock Time Considerations in Practical Deployments.** While convergence properties guide theoretical topology selection, real-world deployments must account for wall-clock execution time

under bandwidth constraints. Our analysis in Appendix A.2 reveals that star topologies don't always achieve ideal $\mathcal{O}(1)$ parallelism in practice due to straggler effects and diluted bandwidth resources. The total wall-clock time follows Eq. (24), where communication overheads scale with group and client counts. In bandwidth-constrained edge environments, ring-based topologies often demonstrate competitive performance despite sequential updates, as they avoid synchronization bottlenecks and uplink transmission constraints. This insight suggests that in networks with heterogeneous bandwidth capabilities or significant straggler effects, ring-based top-tier configurations (Ring-Star, Ring-Ring) may offer better practical performance than theoretically optimal but communication-intensive star-based alternatives, especially when inter-group heterogeneity is high.

## 3 EXPERIMENTS

### 3.1 EXPERIMENTAL SETTINGS

We employ ResNet-18 (Lin et al., 2020) and VGG-9 (Acar et al., 2021) for CIFAR-10 (Krizhevsky et al., 2009), ResNet-18 for CINIC-10 (Darlow et al., 2018), ResNet-10 for Fashion-MNIST (Xiao et al., 2017), and MLP for SST-2 (Socher et al., 2013). We simulate a hierarchical setup with $N = 100$ clients distributed across $G = 10$ groups under four data partitioning schemes: (1) IID-IID, (2) Non-IID-IID, (3) IID-Non-IID, and (4) Non-IID-Non-IID. To ensure fair comparison, the learning rate ($\eta$) was tuned individually for each topology. We consistently observed that to maintain comparable convergence, the required raw learning rates follow the pattern: $\eta_{\text{Star-Star}} > \eta_{\text{Star-Ring}} \approx \eta_{\text{Ring-Star}} > \eta_{\text{Ring-Ring}}$. This empirical observation aligns perfectly with our theoretical findings. Detailed descriptions of the hyperparameters (e.g., batch size, momentum), specific definitions of the data partitioning schemes, and the complete search space for learning rates are provided in Appendix B.1.

### 3.2 EFFECT OF TOPOLOGY

Figure 2 presents the test accuracy curves for the four HFL topologies under the four heterogeneity settings. The results consistently show that topologies with a ring-based top-tier aggregation (i.e., Ring-Star and Ring-Ring) achieve superior convergence speed and higher final accuracy compared to their star-based counterparts. Notably, the classical Star-Star configuration (equivalent to standard HFedAvg) performs the worst across all settings. This is attributed to its conservative update mechanism, i.e., the double averaging at both group and global levels dampens the effective learning rate, slowing convergence. In contrast, ring-based top-tier updates propagate changes sequentially, enabling more aggressive and continuous model refinement. This allows the global model to traverse the loss landscape more rapidly, especially in heterogeneous environments. However, this performance advantage necessitates careful operational considerations. Empirically, ring topologies may exhibit greater sensitivity to data distribution skewness, manifesting as slightly higher volatility in training curves compared to other topologies. This suggests that a client with highly skewed data could influence the update chain, implying that hyperparameter tuning (e.g., learning rate calibration) becomes more critical to maintain stability.

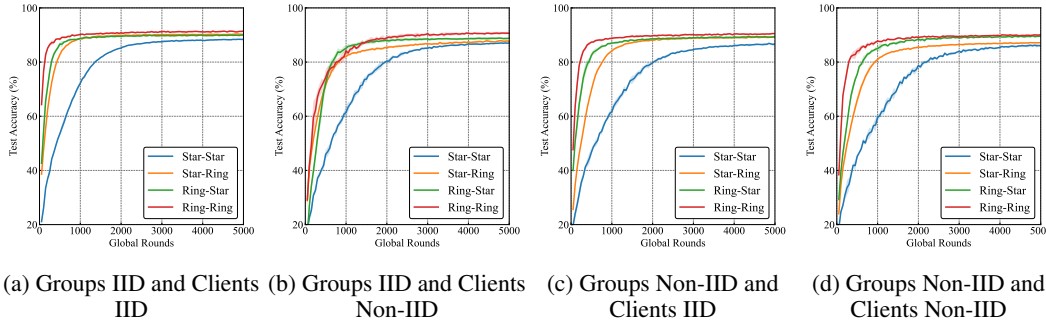

(a) Groups IID and Clients IID    (b) Groups IID and Clients Non-IID    (c) Groups Non-IID and Clients IID    (d) Groups Non-IID and Clients Non-IID

Figure 2: Comparison of the four HFL topologies on CIFAR-10 Dataset

Table 2: Test accuracy (%) on CIFAR-10, CINIC-10, Fashion-MNIST, and SST-2 under various HFL topologies and data partitioning approaches. The non-IID partitions are generated using a Dirichlet distribution (Yurochkin et al., 2019; Hsu et al., 2019).

| Dataset | Model | Heterogeneity | | Topology | | | |
|---|---|---|---|---|---|---|---|
| | | Inter | Intra | Star-Star | Star-Ring | Ring-Star | Ring-Ring |
| CIFAR-10 | ResNet-18 | IID | IID | 88.48 | 90.30 | 90.40 | **91.53** |
| | | | Non-IID | 87.01 | 89.55 | 89.75 | **91.10** |
| | | Non-IID | IID | 87.03 | 88.22 | 89.15 | **90.94** |
| | | | Non-IID | 86.78 | 87.40 | 90.01 | **90.33** |
| | VGG-9 | IID | IID | 84.83 | 87.30 | 87.77 | **89.10** |
| | | | Non-IID | 84.21 | 85.33 | 87.81 | **88.17** |
| | | Non-IID | IID | 85.00 | 85.42 | 86.16 | **88.12** |
| | | | Non-IID | 83.80 | 85.04 | 87.05 | **87.63** |
| CINIC-10 | ResNet-18 | IID | IID | 76.88 | 78.59 | 78.70 | **79.56** |
| | | | Non-IID | 74.25 | 76.09 | 78.23 | **78.35** |
| | | Non-IID | IID | 74.20 | 72.53 | 75.83 | **76.23** |
| | | | Non-IID | 73.63 | 74.21 | **77.11** | 76.78 |
| Fashion-MNIST | ResNet-10 | IID | IID | 89.70 | 92.59 | 92.67 | **93.01** |
| | | | Non-IID | 87.45 | 92.33 | 92.76 | **93.07** |
| | | Non-IID | IID | 88.21 | 89.41 | 91.40 | **91.33** |
| | | | Non-IID | 88.04 | 92.18 | 92.27 | **93.33** |
| SST-2 | MLP | IID | IID | 69.61 | 72.36 | 80.50 | **81.42** |
| | | | Non-IID | 69.95 | 73.17 | 79.59 | **80.50** |
| | | Non-IID | IID | 68.12 | 73.85 | 79.13 | **81.08** |
| | | | Non-IID | 68.12 | 73.97 | 78.44 | **81.65** |

## 3.3 EFFECT OF DATA HETEROGENEITY

Table 2 reports the final test accuracy for all topology and data partition combinations. A key observation is that inter-group heterogeneity has a more detrimental effect on model performance than intra-group heterogeneity. For instance, on CIFAR-10 with ResNet-18 under the Star-Ring topology, shifting to Non-IID group distributions causes a 2.08% accuracy drop (from 90.30% to 88.22%), whereas Non-IID client distributions lead to a smaller drop of only 0.75% (to 89.55%). The effect is even more pronounced on CINIC-10, where in the same setup, inter-group heterogeneity results in a substantial 6.06% performance degradation (from 78.59% to 72.53%), compared to a 2.50% drop for intra-group heterogeneity. This trend also holds for Fashion-MNIST with ResNet-10, where inter-group heterogeneity causes a significant 3.18% accuracy drop (from 92.59% to 89.41%), while the impact of intra-group heterogeneity is a negligible 0.26% decrease.

This empirical finding strongly supports our theoretical conclusion that inter-group divergence$(\delta)$ is the dominant bottleneck in HFL convergence. It suggests that system designers should prioritize clustering strategies that minimize distributional differences between groups even at the expense of increased intra-group heterogeneity. For example, grouping clients by semantic similarity of data (e.g., geographic region, user demographics) rather than arbitrary network proximity can significantly improve convergence.

## 3.4 EFFECT OF GROUPS

We further investigate how the number of groups $G$ influences performance, focusing on the hybrid topologies (i.e., Star-Ring and Ring-Star) as they offer a practical balance between convergence efficiency and stability. With the total number of clients $N = 100$, we vary the number of groups $G \in \{1, 5, 10, 20, 100\}$ and tune the learning rate for each configuration to ensure optimal perfor-

mance. Figure 3 illustrates the convergence of Star-Ring and Ring-Star under both IID and Non-IID settings. We can find two distinct patterns:

(1) Star-Ring performs best with fewer, larger groups, i.e., small values of $G$. This is because intra-group ring aggregation benefits from longer update chains: sequential updates allow for deeper local refinement before global synchronization, producing higher-quality group models.

(2) Ring-Star, in contrast, excels with more, smaller groups, i.e., large values of $G$. Here, parallel intra-group aggregation (star) is less effective in large groups due to the averaging of divergent local updates, which can dilute valuable gradients. Smaller groups reduce this averaging effect, and the sequential inter-group updates in Ring-Star enable fine-grained global alignment.

It is worth noting that our conclusion continues to hold even in the extreme cases—Ring-Star with $G = N$ and Star-Ring with $G = 1$, both degenerating to a pure ring topology. This does not conflict with the notion of "catastrophic forgetting" in sequential federated learning, because the step size is not fixed. We scale it with the number of groups to keep the effective learning rate constant. Under the same effective learning rate, selecting an appropriate $G$ therefore yields optimal performance. These results highlight a critical design principle: *optimal topology selection depends on the underlying group structure*. In applications with a few large, data-rich clusters (e.g., hospital networks), Star-Ring is preferable. In contrast, systems with many small or independent units (e.g., IoT devices, retail outlets) benefit more from the Ring-Star topology.

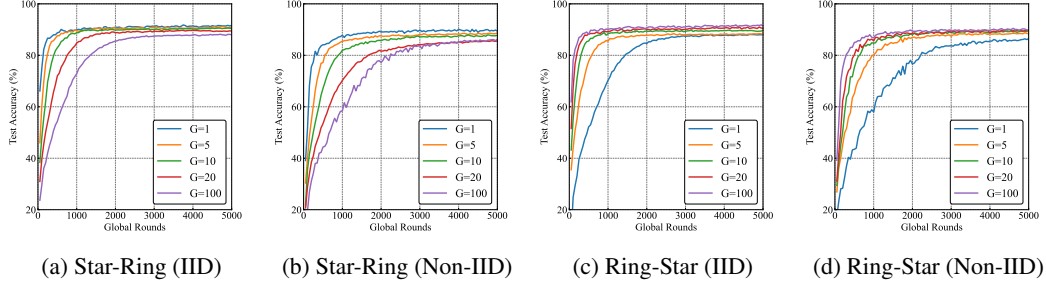

(a) Star-Ring (IID)  (b) Star-Ring (Non-IID)  (c) Ring-Star (IID)  (d) Ring-Star (Non-IID)

Figure 3: Comparison of Star-Ring and Ring-Star topologies with different numbers of groups on CIFAR10 Dataset

# 4    CONCLUSION

This paper presents the first unified convergence analysis for all four HFL topologies under non-convex objectives and intra/inter-group data heterogeneity. Our results reveal that: (1) top-tier topology dictates convergence behavior, and ring-based top-tier aggregation generally converges faster than star-based methods; (2) inter-group heterogeneity is the dominant bottleneck, outweighing intra-group effects; and (3) optimal topology depends on group structure, where Ring-Star suits many small groups, while Star-Ring excels with few large clusters. These findings enable system designers to move beyond heuristic topology choices and instead make informed, theoretically grounded decisions based on deployment-specific constraints such as network scale, client distribution, and data heterogeneity profiles.

## ACKNOWLEDGMENTS

This work is supported in part by National Science and Technology Major Project of China under Grant 2025ZD1302200, and in part by National Natural Science Foundation of China under Grant 62371059.

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

# Appendix

## A    DISCUSSION

### A.1    ALGORITHM DETAILS

For clarity and completeness, this appendix provides the detailed pseudocode for the four HFL topologies mentioned in the main body of our paper. Each algorithm outlines a different communication pattern for both inter-group and intra-group model aggregation.

The Star-Star topology (Algorithm 1) represents a fully parallel framework. Both the groups at the server level and the clients within each group perform their training and updates in parallel, synchronizing with their respective servers before aggregation.

---

**Algorithm 1** Star-Star Hierarchical FL

1: **for** global rounds $r = 0, 1, \ldots, R - 1$ **do**
2:     **for** groups $g = 1, 2, \ldots, G$ **in parallel do**
3:         Initialize group model: $\mathbf{x}_{g,0}^{(r)} = \mathbf{x}^{(r)}$
4:         **for** group rounds $p = 0, 1, \ldots, P - 1$ **do**
5:             **for** clients $m = 1, 2, \ldots, M$ **in parallel do**
6:                 Initialize local model: $\mathbf{x}_{g,p,m,0}^{(r)} = \mathbf{x}_{g,p}^{(r)}$
7:                 **for** local steps $k = 0, 1, \ldots, K - 1$ **do**
8:                     $\mathbf{x}_{g,p,m,k+1}^{(r)} = \mathbf{x}_{g,p,m,k}^{(r)} - \eta \mathbf{g}_{g,p,m,k}^{(r)}$
9:                 **end for**
10:             **end for**
11:             Group aggregation: $\mathbf{x}_{g,p+1}^{(r)} = \frac{1}{M} \sum_{m=1}^{M} \mathbf{x}_{g,p,m,K}^{(r)}$
12:         **end for**
13:     **end for**
14:     Global aggregation: $\mathbf{x}^{(r+1)} = \frac{1}{G} \sum_{g=1}^{G} \mathbf{x}_{g,P}^{(r)}$
15: **end for**

---

The Star-Ring topology (Algorithm 2) combines parallel inter-group communication with sequential intra-group updates. While groups update in parallel with the global server, clients within each group form a ring, passing the model sequentially from one client to the next.

---

**Algorithm 2** Star-Ring Hierarchical FL

1: **for** global rounds $r = 0, 1, \ldots, R - 1$ **do**
2:     **for** groups $g = 1, 2, \ldots, G$ **in parallel do**
3:         Initialize group model: $\mathbf{x}_{g,0}^{(r)} = \mathbf{x}^{(r)}$
4:         **for** group rounds $p = 0, 1, \ldots, P - 1$ **do**
5:             **for** clients $m = 1, 2, \ldots, M$ **in sequence do**
6:                 Initialize local model: $\mathbf{x}_{g,p,m,0}^{(r)} = \begin{cases} \mathbf{x}_{g,p}^{(r)} & \text{if } m = 1 \\ \mathbf{x}_{g,p,m-1,K}^{(r)} & \text{if } m > 1 \end{cases}$
7:                 **for** local steps $k = 0, 1, \ldots, K - 1$ **do**
8:                     $\mathbf{x}_{g,p,m,k+1}^{(r)} = \mathbf{x}_{g,p,m,k}^{(r)} - \eta \mathbf{g}_{g,p,m,k}^{(r)}$
9:                 **end for**
10:             **end for**
11:             Group model: $\mathbf{x}_{g,p+1}^{(r)} = \mathbf{x}_{g,p,M,K}^{(r)}$
12:         **end for**
13:     **end for**
14:     Global aggregation: $\mathbf{x}^{(r+1)} = \frac{1}{G} \sum_{g=1}^{G} \mathbf{x}_{g,P}^{(r)}$
15: **end for**

---

Conversely, the Ring-Star topology (Algorithm 3) employs sequential communication among groups and parallel updates within them. The groups form a ring at the global level, while clients inside each group operate in a standard star configuration.

---

**Algorithm 3** Ring-Star Hierarchical FL

---

1: **for** global rounds $r = 0, 1, \ldots, R-1$ **do**
2:     **for** groups $g = 1, 2, \ldots, G$ **in sequence do**
3:         Initialize group model: $\mathbf{x}_{g,0}^{(r)} = \begin{cases} \mathbf{x}^{(r)} & \text{if } g = 1 \\ \mathbf{x}_{g-1,P}^{(r)} & \text{if } g > 1 \end{cases}$
4:         **for** group rounds $p = 0, 1, \ldots, P-1$ **do**
5:             **for** clients $m = 1, 2, \ldots, M$ **in parallel do**
6:                 Initialize local model: $\mathbf{x}_{g,p,m,0}^{(r)} = \mathbf{x}_{g,p}^{(r)}$
7:                 **for** local steps $k = 0, 1, \ldots, K-1$ **do**
8:                     $\mathbf{x}_{g,p,m,k+1}^{(r)} = \mathbf{x}_{g,p,m,k}^{(r)} - \eta \mathbf{g}_{g,p,m,k}^{(r)}$
9:                 **end for**
10:             **end for**
11:             Group aggregation: $\mathbf{x}_{g,p+1}^{(r)} = \frac{1}{M} \sum_{m=1}^{M} \mathbf{x}_{g,p,m,K}^{(r)}$
12:         **end for**
13:     **end for**
14:     Global model: $\mathbf{x}^{(r+1)} = \mathbf{x}_{G,P}^{(r)}$
15: **end for**

---

The Ring-Ring topology (Algorithm 4) implements a fully sequential communication protocol. Both the groups at the global level and the clients within each group update their models in a sequential, ring-based manner.

---

**Algorithm 4** Ring-Ring Hierarchical FL

---

1: **for** global rounds $r = 0, 1, \ldots, R-1$ **do**
2:     **for** groups $g = 1, 2, \ldots, G$ **in sequence do**
3:         Initialize group model: $\mathbf{x}_{g,0}^{(r)} = \begin{cases} \mathbf{x}^{(r)} & \text{if } g = 1 \\ \mathbf{x}_{g-1,P}^{(r)} & \text{if } g > 1 \end{cases}$
4:         **for** group rounds $p = 0, 1, \ldots, P-1$ **do**
5:             **for** clients $m = 1, 2, \ldots, M$ **in sequence do**
6:                 Initialize local model: $\mathbf{x}_{g,p,m,0}^{(r)} = \begin{cases} \mathbf{x}_{g,p}^{(r)} & \text{if } m = 1 \\ \mathbf{x}_{g,p,m-1,K}^{(r)} & \text{if } m > 1 \end{cases}$
7:                 **for** local steps $k = 0, 1, \ldots, K-1$ **do**
8:                     $\mathbf{x}_{g,p,m,k+1}^{(r)} = \mathbf{x}_{g,p,m,k}^{(r)} - \eta \mathbf{g}_{g,p,m,k}^{(r)}$
9:                 **end for**
10:             **end for**
11:             Group model: $\mathbf{x}_{g,p+1}^{(r)} = \mathbf{x}_{g,p,M,K}^{(r)}$
12:         **end for**
13:     **end for**
14:     Global model: $\mathbf{x}^{(r+1)} = \mathbf{x}_{G,P}^{(r)}$
15: **end for**

---

## A.2   WALL-CLOCK TIME ANALYSIS

To bridge the gap between convergence rounds and real-world training latency, we analyze the total wall-clock time $T$. This is defined as the sum of computation and communication overheads: $T = T_{\text{Comp}} + T_{\text{Comm}}$. Let $S$ denote the model size, $\tau_x$ the computation time for a node $x$, and $r_{A \to B}$ the transmission rate from node $A$ to $B$.

**Single-Layer Analysis.** In a single-layer setting with $M$ clients, the Star topology is limited by the straggler, i.e., $T^{\text{Star}} = \max_m \{ \frac{S}{r_{0 \to m}} + \tau_m + \frac{S}{r_{m \to 0}} \}$, where the index 0 denotes the aggregation server. Conversely, the Ring topology operates sequentially, accumulating latencies: $T^{\text{Ring}} = \frac{S}{r_{0 \to 1}} +$

$\sum_m \tau_m + \sum_{m=2}^{M} \frac{S}{r_{m-1 \to m}} + \frac{S}{r_{M \to 0}}$. While $T^{\text{Star}}$ theoretically offers $O(1)$ communication scaling versus $T^{\text{Ring}}$'s $O(M)$, this assumes unlimited bandwidth, which rarely holds in edge scenarios.

**HFL Topology Analysis.** The total wall-clock time for HFL over $R$ global rounds, with $P$ group rounds, can be unified as:

$$T_{\text{total}} = R \cdot (T_{\text{inter}} + P \cdot T_{\text{intra}}), \tag{24}$$

where $T_{\text{inter}}$ represents the wall-clock time for inter-group updates, and $T_{\text{intra}}$ represents the wall-clock time for intra-group updates. Table 3 details these components for four topologies.

Table 3: Wall-clock Time Components for HFL Topologies. Here $\mathcal{G}$ denotes the Intra-group Aggregation Server (Group Server).

| Topology | Inter-Group Time ($T_{\text{inter}}$) | Intra-Group Time ($T_{\text{intra}}$) |
|---|---|---|
| Star-Star | $\max_g \left\{ \frac{S}{r_{0 \to g}} + \tau_g + \frac{S}{r_{g \to 0}} \right\}$ | $\max_{g,m} \left\{ \frac{S}{r_{\mathcal{G} \to m,g}} + \tau_{g,m} + \frac{S}{r_{m \to \mathcal{G},g}} \right\}$ |
| Star-Ring | $\max_g \left\{ \frac{S}{r_{0 \to g}} + \tau_g + \frac{S}{r_{g \to 0}} \right\}$ | $\max_g \left\{ \frac{S}{r_{\mathcal{G} \to 1,g}} + \sum_{m=1}^{M} \tau_{g,m} + \sum_{m=2}^{M} \frac{S}{r_{m-1 \to m,g}} + \frac{S}{r_{M \to \mathcal{G},g}} \right\}$ |
| Ring-Star | $\frac{S}{r_{0 \to 1}} + \sum_{g=1}^{G} \tau_g + \sum_{g=2}^{G} \frac{S}{r_{g-1 \to g}} + \frac{S}{r_{G \to 0}}$ | $\sum_{g=1}^{G} \max_m \left\{ \frac{S}{r_{\mathcal{G} \to m,g}} + \tau_{g,m} + \frac{S}{r_{m \to \mathcal{G},g}} \right\}$ |
| Ring-Ring | $\frac{S}{r_{0 \to 1}} + \sum_{g=1}^{G} \tau_g + \sum_{g=2}^{G} \frac{S}{r_{g-1 \to g}} + \frac{S}{r_{G \to 0}}$ | $\sum_{g=1}^{G} \left\{ \frac{S}{r_{\mathcal{G} \to 1,g}} + \sum_{m=1}^{M} \tau_{g,m} + \sum_{m=2}^{M} \frac{S}{r_{m-1 \to m,g}} + \frac{S}{r_{M \to \mathcal{G},g}} \right\}$ |

**Bandwidth Constraints and Practical Latency.** Theoretically, the star topology offers advantages in parallel computation; however, in real-world scenarios, it often fails to achieve the idea $O(1)$ level of parallelism. In HFL, the total wall-clock time faces challenges similar to those in single-layer FL. Many studies (Lim et al., 2021; Liu et al., 2025) on HFL communication optimization highlight that bandwidth resources are diluted as the number of users increases, e.g., $r_{m-1 \to m,g} \sim \frac{R_g}{M}$, where $R_g$ represents the total bandwidth resources of the group server and $M$ denotes the total number of clients connected to that server. Substituting this relationship into the total wall-clock time for the Star-Star topology yields $T_{total}^{Star-Star} = R \left( \max_{g=1}^{G} \left\{ \frac{2SG}{R} + \tau_g \right\} + P \cdot \max_{g=1}^{G} \left\{ \max_{m=1}^{M} \left\{ \frac{2SM}{R_g} + \tau_{g,m} \right\} \right\} \right)$, where the communication latency in the upper layer approaches $O(G)$, while the communication overhead in the lower layer approaches $O(M)$. In particular, the uplink transmission rate of workers is a major bottleneck in the training process that can lead to the straggler's effect (Lim et al., 2020). Therefore, in bandwidth-constrained real-world scenarios, the ring topology remains a competitive option.

# B EXPERIMENTAL DETAILS AND ADDITIONAL RESULTS

## B.1 DETAILED EXPERIMENTAL SETTINGS

All batch normalization layers in the models (ResNet-18, VGG-9, ResNet-10) are removed to ensure cleaner validation of our convergence bounds. We use SGD with a constant learning rate, zero momentum, mini-batch size of 20, and gradient clipping. The global model is updated over $R = 5000$ rounds ($R = 1000$ for SST-2), with each group performing $P = 1$ group-level updates and each client conducting $K = 2$ local steps.

**Data Partitioning Schemes.** We simulate a hierarchical setup with $N = 100$ clients evenly distributed across $G = 10$ groups, unless otherwise specified, and examine four data partitioning schemes (Fang et al., 2024): (1) IID within groups & IID between groups, where data is uniformly and randomly partitioned at both group and client levels; (2) Non-IID within groups & IID between groups, where groups receive statistically similar data distributions, but clients within each group are assigned non-IID partitions via a Dirichlet distribution with parameter $\alpha = 0.1$ ($\alpha = 0.3$ for SST-2); (3) IID within groups & Non-IID between groups, where clients within a group share IID data, but group-level distributions differ significantly, again using a Dirichlet split across groups;

and (4) Non-IID within groups & Non-IID between groups, where the entire dataset is partitioned using a Dirichlet distribution, resulting in heterogeneous data at both intra- and inter-group levels.

**Learning Rate Tuning.** The learning rate ($\eta$) was tuned individually for each of the four topologies to ensure they all operate near their optimal convergence speed. Taking CIFAR-10 as an example, we explored a search space of $\{2, 1, 0.5, 0.2, 0.05, 0.01\}$ to identify the optimal $\eta$. We found the optimal learning rates to be $1.0$ for Star-Star, $0.2$ for both Star-Ring and Ring-Star, and $0.05$ for Ring-Ring

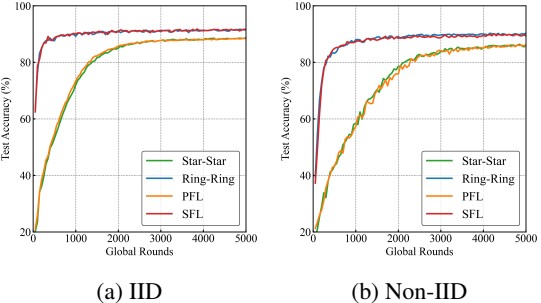

Figure 4: Test accuracies on CIFAR-10 after training for 1000 training rounds with different learning rates.

## B.2 COMPARISON WITH SINGLE-TIER FL

To evaluate the performance of HFL against traditional single-tier architectures, we conducted comparative experiments using the CIFAR-10 dataset with a ResNet-18 model. The total number of clients was set to $N = 100$.
We compared our Star-Star and Ring-Ring HFL ($G$=10, $M$=10) against standard single-tier FedAvg (denoted as Parallel FL, PFL) and single-tier Sequential FL (Li & Lyu, 2023) (denoted as SFL).

The models were trained for 5,000 global rounds. The experimental results reveal distinct convergence correlations: (i) PFL demonstrates a convergence speed and final accuracy highly similar to that of the Star-Star HFL topology across both partitions; (ii) SFL exhibits performance characteristics comparable to that of the Ring-Ring HFL topology.

The convergence curves for the IID and Non-IID settings are visualized in Figure 5. These comparisons empirically validate that the hierarchical structure itself does not inherently accelerate convergence speed. Instead, the convergence characteristics of HFL topolo-

(a) IID      (b) Non-IID

Figure 5: Comparison with Single-Tier FL on CIFAR-10 Dataset

gies strongly correlate with the synchronization mechanism (parallel vs. sequential) of their single-tier analogues. Consequently, the primary advantage of HFL lies in mitigating communication bottlenecks at the central server rather than improving convergence rates.

## B.3 EFFECT OF GROUP ROUNDS $P$ AND LOCAL STEPS $K$

To investigate the impact of computation and communication frequencies on convergence dynamics, we evaluated all four HFL topologies on the CIFAR-10 dataset (Non-IID) under varying numbers of local steps ($K \in \{2, 5, 10\}$) and group rounds ($P \in \{1, 2, 5\}$). The convergence curves over the first 3,000 global rounds are presented in Figure 6 below.

**Effect of Local Steps.** Comparing the curves with fixed $P = 1$ (Green: $K = 2$, Blue: $K = 5$, Orange: $K = 10$), we observe that increasing $K$ consistently accelerates the initial convergence speed across all topologies. Increasing $K$ increases the computational density per communication round, allowing the model to traverse the loss landscape further before aggregation. This confirms that greater local computation can effectively trade off communication rounds for faster convergence, particularly in the initial phase.

**Effect of Group Rounds.** Comparing the curves with fixed $K = 2$ (Green: $P = 1$, Red: $P = 2$, Purple: $P = 5$), increasing the number of group-level updates yields a substantial improvement in convergence efficiency. Increasing $P$ allows for more thorough intra-group refinement before global synchronization. For Ring-Star, a larger $P$ means the model circulates among groups more times (or effectively allows more groups to participate sequentially if viewed as virtual steps), drastically reducing inter-group heterogeneity impact per global round.

While both increasing $P$ and $K$ improve convergence speed, increasing group rounds ($P$) generally provides a stronger acceleration effect than increasing local steps ($K$) in our experiments. This suggests that for HFL, promoting consensus at the group level (via larger $P$) is a highly effective strategy for combating heterogeneity, especially for architectures with a Ring-based top tier.

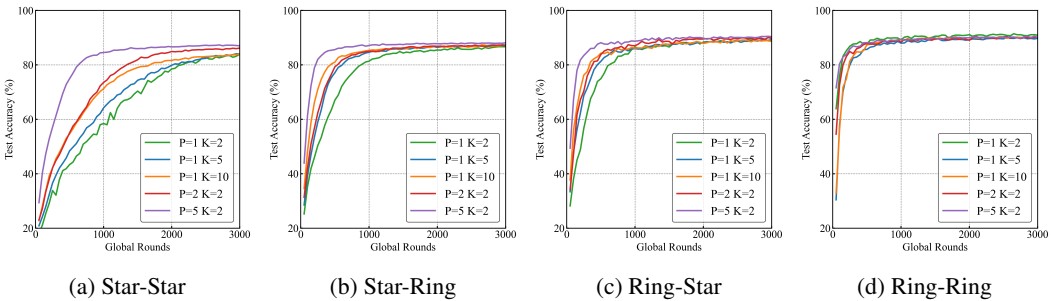

(a) Star-Star  (b) Star-Ring  (c) Ring-Star  (d) Ring-Ring

Figure 6: Impact of P and K on Convergence of Four HFL Topologies

## C  NOTATIONS

Table 4 summarizes the notations appearing in this paper.

Table 4: Key notations for HFL algorithm.

| Symbol | Description |
|---|---|
| $R, r$ | number, index of training rounds |
| $G, g$ | number, index of groups |
| $M, m$ | number, index of clients in each group |
| $S_1, s_1$ | number, index of selected groups under partial participation |
| $S_2, s_2$ | number, index of selected clients in each group under partial participation |
| $P, p$ | number, index of group steps |
| $K, k$ | number, index of local update steps |
| $\eta$ | learning rate (or stepsize) |
| $\tilde{\eta}$ | effective learning rate |
| $L$ | $L$-smoothness constant (Assumption 1) |
| $\sigma$ | upper bound on variance of stochastic gradients at each client (Assumption 2) |
| $\delta$ | constant in Assumption 3 to bound inter-group heterogeneity |
| $\zeta_g$ | constants in Assumption 4 to bound intra-group heterogeneity |
| $\zeta$ | average intra-group heterogeneity constant, defined as $\zeta^2 = \frac{1}{G} \sum_{g=1}^{G} \zeta_g^2$ |
| $F/F_g/F_{g,m}$ | global objective/group $p$ objective/local objective of client $m$ in group $p$ |
| $\mathbf{x}^{(r)}$ | global model parameters in the $r$-th round |
| $\mathbf{x}_{g,p,m,k}^{(r)}$ | local model parameters of the $m$-th client after $k$ local steps in the $g$-th group after $p$ group steps in the $r$-th round |
| $\mathbf{g}_{g,p,m,k}^{(r)}$ | $\mathbf{g}_{g,p,m,k}^{(r)} := \nabla f_{g,m}(\mathbf{x}_{g,p,m,k}^{(r)}; \xi)$ denotes the stochastic gradients of $F_{g,m}$ regarding $\mathbf{x}_{g,p,m,k}^{(r)}$ |

## D  PROOF OF THEOREM 1

In the following proof, we consider the full participation setting with a fixed execution order. Specifically, all $G$ groups and all $M$ clients participate in every training round. We assume the execution follows a deterministic sequence, corresponding to the natural indices $\{1, 2, \ldots, G\}$ for groups and $\{1, 2, \ldots, M\}$ for clients, without random permutation.

### D.1  FIND THE PER-ROUND RECURSION

**Lemma 1.** *Let Assumptions 1, 2 hold. If the learning rate satisfies $\eta \leq \frac{1}{2Lc_0GPMK}$, then*

$$
\mathbb{E}\left[F(\mathbf{x}^{(r+1)}) - F(\mathbf{x}^{(r)})\right] \leq -\frac{1}{2}\eta c_0 GMPK\mathbb{E}[\|\nabla F(\mathbf{x}^{(r)})\|^2] + \eta^2 L c_0{}^2 GMPK\sigma^2
$$
$$
+ \frac{1}{2}\eta L^2 c_0 \sum_{g=1}^{G}\sum_{p=0}^{P-1}\sum_{m=1}^{M}\sum_{k=0}^{K-1}\mathbb{E}\left[\left\|\mathbf{x}_{g,p,m,k}^{(r)} - \mathbf{x}^{(r)}\right\|^2\right],
$$

*where $c_0$ is a topology-dependent coefficient that takes values of $1/(GM)$, $1/G$, $1/M$, and $1$ for the Star-Star, Star-Ring, Ring-Star, and Ring-Ring topologies, respectively.*

*Proof.* In the following, we focus on a single training round, and hence we drop the superscripts $r$ for a while, e.g., writing $\mathbf{x}_{g,p,m,k}$ to replace $\mathbf{x}_{g,p,m,k}^{(r)}$. Specifically, we would like to use $\mathbf{x}$ to replace $\mathbf{x}_{1,0,1,0}^{(r)}$. Unless otherwise stated, the expectation is conditioned on $\mathbf{x}^{(r)}$.

Starting from the smoothness of $F$ (applying Assumption 1, $\|\nabla F(\mathbf{x}) - \nabla F(\mathbf{y})\| \leq L\|\mathbf{x} - \mathbf{y}\|$), we have

$$
\mathbb{E}\left[F\left(\mathbf{x} + \Delta\mathbf{x}\right) - F\left(\mathbf{x}\right)\right] \leq \underbrace{\mathbb{E}\left\langle \nabla F\left(\mathbf{x}\right), \Delta\mathbf{x}\right\rangle}_{A_1} + \underbrace{\frac{L}{2}\mathbb{E}\|\Delta\mathbf{x}\|^2}_{A_2}
$$

The model updates within a single global round for the four topologies can be represented by a unified format,

$$
\Delta\mathbf{x} = \mathbf{x}^{(R+1)} - \mathbf{x}^{(R)} = -c_0\eta \sum_{g,p,m,k} \mathbf{g}_{g,p,m,k},
$$

where $c_0$ denotes the topology-dependent coefficient as defined in Lemma 1.

After substituting the overall updates $\Delta\mathbf{x}$, we can get $A_1$

$$
\mathbb{E}\left[\langle \nabla F(\mathbf{x}^{(r)}), \Delta\mathbf{x}^{(r)}\rangle\right]
$$
$$
= \mathbb{E}\left[\left\langle \nabla F(\mathbf{x}^{(r)}), -\eta c_0 \sum_{g=1}^{G}\sum_{p=0}^{P-1}\sum_{m=1}^{M}\sum_{k=0}^{K-1} \mathbf{g}_{g,p,m,k}^{(r)}\right\rangle\right]
$$
$$
= \mathbb{E}\left[\left\langle \nabla F(\mathbf{x}^{(r)}), -\eta c_0 \sum_{g=1}^{G}\sum_{p=0}^{P-1}\sum_{m=1}^{M}\sum_{k=0}^{K-1} \nabla F_{g,m}(\mathbf{x}_{g,p,m,k}^{(r)})\right\rangle\right]
$$
$$
= \mathbb{E}\left[\left\langle \nabla F(\mathbf{x}^{(r)}), -\eta c_0 \sum_{g=1}^{G}\sum_{p=0}^{P-1}\sum_{m=1}^{M}\sum_{k=0}^{K-1} \left(\nabla F_{g,m}(\mathbf{x}_{g,p,m,k}^{(r)}) - \nabla F(\mathbf{x}^{(r)})\right)\right\rangle\right]
$$
$$
+ \mathbb{E}\left[\left\langle \nabla F(\mathbf{x}^{(r)}), -\eta c_0 \sum_{g=1}^{G}\sum_{p=0}^{P-1}\sum_{m=1}^{M}\sum_{k=0}^{K-1} \nabla F(\mathbf{x}^{(r)})\right\rangle\right]
$$
$$
= -\eta c_0 GMPK\mathbb{E}[\|\nabla F(\mathbf{x}^{(r)})\|^2] + \eta c_0 GMPK\mathbb{E}\left[\left\langle \nabla F(\mathbf{x}^{(r)}),\right.\right.
$$
$$
\left.\left. c_0 \sum_{g=1}^{G}\sum_{p=0}^{P-1}\sum_{m=1}^{M}\sum_{k=0}^{K-1} \left(\nabla F_{g,m}(\mathbf{x}_{g,p,m,k}^{(r)}) - \nabla F_{g,m}(\mathbf{x}^{(r)})\right)\right\rangle\right]
$$

$$\leq -\eta c_0 GMPK\mathbb{E}[\|\nabla F(\mathbf{x}^{(r)})\|^2] + \frac{1}{2}\eta c_0 GMPK\mathbb{E}[\|\nabla F(\mathbf{x}^{(r)})\|^2]$$

$$- \frac{1}{2}\eta c_0 \frac{1}{GPMK}\mathbb{E}\left\|\sum_{g=1}^{G}\sum_{p=0}^{P-1}\sum_{m=1}^{M}\sum_{k=0}^{K-1}\left(\nabla F_{g,m}(\mathbf{x}_{g,p,m,k}^{(r)})\right)\right\|^2$$

$$+ \frac{1}{2}\eta c_0 GMPK\mathbb{E}\left[\left\|c_0\sum_{g=1}^{G}\sum_{p=0}^{P-1}\sum_{m=1}^{M}\sum_{k=0}^{K-1}\left(\nabla F_{g,m}(\mathbf{x}_{g,p,m,k}^{(r)}) - \nabla F_{g,m}(\mathbf{x}^{(r)})\right)\right\|^2\right]$$

$$= -\frac{1}{2}\eta c_0 GMPK\mathbb{E}[\|\nabla F(\mathbf{x}^{(r)})\|^2]$$

$$+ \frac{1}{2}\eta c_0 GMPK\mathbb{E}\left\|c_0\sum_{g=1}^{G}\sum_{p=0}^{P-1}\sum_{m=1}^{M}\sum_{k=0}^{K-1}\left(\nabla F_{g,m}(\mathbf{x}_{g,p,m,k}^{(r)}) - \nabla F_{g,m}(\mathbf{x}^{(r)})\right)\right\|^2$$

$$- \frac{1}{2}\eta c_0 \frac{1}{GPMK}\mathbb{E}\left\|\sum_{g=1}^{G}\sum_{p=0}^{P-1}\sum_{m=1}^{M}\sum_{k=0}^{K-1}\left(\nabla F_{g,m}(\mathbf{x}_{g,p,m,k}^{(r)})\right)\right\|^2$$

$$\leq -\frac{1}{2}\eta c_0 GMPK\mathbb{E}[\|\nabla F(\mathbf{x}^{(r)})\|^2] + \frac{1}{2}\eta L^2 c_0 \sum_{g=1}^{G}\sum_{p=0}^{P-1}\sum_{m=1}^{M}\sum_{k=0}^{K-1}\mathbb{E}\left[\left\|\mathbf{x}_{g,p,m,k}^{(r)} - \mathbf{x}^{(r)}\right\|^2\right]$$

$$- \frac{1}{2}\eta c_0 \frac{1}{GPMK}\mathbb{E}\left\|\sum_{g=1}^{G}\sum_{p=0}^{P-1}\sum_{m=1}^{M}\sum_{k=0}^{K-1}\left(\nabla F_{g,m}(\mathbf{x}_{g,p,m,k}^{(r)})\right)\right\|^2$$

where we use $\langle a, b \rangle = \frac{1}{2}\left(\|a\|^2 + \|b\|^2 - \|a - b\|^2\right)$ for the first inequality and use Jensen's inequality, $\|\sum_{i=1}^{n}\mathbf{a}_i\|^2 \leq n\sum_{i=1}^{n}\|\mathbf{a}_i\|^2$, for the last inequality.

Next, we bound the term $A_2$,

$$\frac{1}{2}L\mathbb{E}[\|\Delta\mathbf{x}\|^2] = \frac{1}{2}L\mathbb{E}\left[\left\|\eta c_0\sum_{g=1}^{G}\sum_{p=0}^{P-1}\sum_{m=1}^{M}\sum_{k=0}^{K-1}\mathbf{g}_{g,p,m,k}^{(r)}\right\|^2\right]$$

$$\leq \eta^2 L\mathbb{E}\left[\left\|c_0\sum_{g=1}^{G}\sum_{p=0}^{P-1}\sum_{m=1}^{M}\sum_{k=0}^{K-1}\left(\mathbf{g}_{g,p,m,k}^{(r)} - \nabla F_{g,m}(\mathbf{x}_{g,p,m,k}^{(r)})\right)\right\|^2\right]$$

$$+ \eta^2 L\mathbb{E}\left[\left\|c_0\sum_{g=1}^{G}\sum_{p=0}^{P-1}\sum_{m=1}^{M}\sum_{k=0}^{K-1}\nabla F_{g,m}(\mathbf{x}_{g,p,m,k}^{(r)})\right\|^2\right].$$

Let $\eta \leq \frac{1}{2Lc_0 GMPK}$, we have

$$\mathbb{E}\left[F(\mathbf{x}^{(r+1)}) - F(\mathbf{x}^{(r)})\right]$$

$$\leq \mathbb{E}\left[\langle\nabla F(\mathbf{x}^{(r)}), \Delta\mathbf{x}^{(r)}\rangle\right] + \frac{1}{2}L\mathbb{E}[\|\Delta\mathbf{x}^{(r)}\|^2]$$

$$= -\frac{1}{2}\eta c_0 GMPK\mathbb{E}[\|\nabla F(\mathbf{x}^{(r)})\|^2] + \eta^2 Lc_0^2 GMPK\sigma^2$$

$$+ \frac{1}{2}\eta L^2 c_0 \sum_{g=1}^{G}\sum_{p=0}^{P-1}\sum_{m=1}^{M}\sum_{k=0}^{K-1}\mathbb{E}\left[\left\|\mathbf{x}_{g,p,m,k}^{(r)} - \mathbf{x}^{(r)}\right\|^2\right]$$

$$- \frac{\left(\frac{1}{2} - Lc_0 GMPK\eta\right)c_0\eta}{GPMK}\mathbb{E}\left\|\sum_{g=1}^{G}\sum_{p=0}^{P-1}\sum_{m=1}^{M}\sum_{k=0}^{K-1}\nabla F_{g,m}(\mathbf{x}_{g,p,m,k}^{(r)})\right\|^2$$

$$\leq -\frac{1}{2}\eta c_0 GMPK\mathbb{E}[\|\nabla F(\mathbf{x}^{(r)})\|^2] + \eta^2 Lc_0^2 GMPK\sigma^2$$

$$+ \frac{1}{2}\eta L^2 c_0 \sum_{g=1}^{G} \sum_{p=0}^{P-1} \sum_{m=1}^{M} \sum_{k=0}^{K-1} \mathbb{E}\left[\left\|\mathbf{x}_{g,p,m,k}^{(r)} - \mathbf{x}^{(r)}\right\|^2\right].$$

We define the client drift in HFL:

$$E_r := \sum_{g=1}^{G} \sum_{p=0}^{P-1} \sum_{m=1}^{M} \sum_{k=0}^{K-1} \mathbb{E}\|\mathbf{x}_{g,p,m,k}^{(r)} - \mathbf{x}^{(r)}\|^2.$$

### D.2 PROOF OF THEOREM 1 FOR THE STAR-STAR CASE

For Star-Star,

$$\mathbf{x}_{g,p,m,k} - \mathbf{x} = \sum_{k'=0}^{k-1} \mathbf{g}_{g,p,m,k'} + \sum_{p'=0}^{p-1} \frac{1}{M} \sum_{m'=1}^{M} \sum_{k'=0}^{K-1} \mathbf{g}_{g,p',m',k'}.$$

To bound $E_r$, we first bound $\mathbb{E}\|\mathbf{x}_{g,p,m,k} - \mathbf{x}\|^2$.

$\mathbb{E}\|\mathbf{x}_{g,p,m,k} - \mathbf{x}\|^2$

$$= \eta^2 \mathbb{E}\| \sum_{k'=0}^{k-1} \mathbf{g}_{g,p,m,k'} + \sum_{p'=0}^{p-1} \frac{1}{M} \sum_{m'=1}^{M} \sum_{k'=0}^{K-1} \mathbf{g}_{g,p',m',k'} \|^2$$

$$\leq 5\eta^2 \mathbb{E}\| \sum_{k'=0}^{k-1} \mathbf{g}_{g,p,m,k'} + \sum_{p'=0}^{p-1} \frac{1}{M} \sum_{m'=1}^{M} \sum_{k'=0}^{K-1} \mathbf{g}_{g,p',m',k'} - \sum_{k'=0}^{k-1} \nabla F_{g,m}\left(\mathbf{x}_{g,p,m,k'}\right)$$

$$- \sum_{p'=0}^{p-1} \frac{1}{M} \sum_{m'=1}^{M} \sum_{k'=0}^{K-1} \nabla F_{g,m'}\left(\mathbf{x}_{g,p',m',k'}\right) \|^2$$

$$+ 5\eta^2 \mathbb{E}\| \sum_{k'=0}^{k-1} \nabla F_{g,m}\left(\mathbf{x}_{g,p,m,k'}\right) + \sum_{p'=0}^{p-1} \frac{1}{M} \sum_{m'=1}^{M} \sum_{k'=0}^{K-1} \nabla F_{g,m'}\left(\mathbf{x}_{g,p',m',k'}\right) - \sum_{k'=0}^{k-1} \nabla F_{g,m}\left(\mathbf{x}\right)$$

$$- \sum_{p'=0}^{p-1} \frac{1}{M} \sum_{m'=1}^{M} \sum_{k'=0}^{K-1} \nabla F_{g,m'}\left(\mathbf{x}\right) \|^2$$

$$+ 5\eta^2 \mathbb{E}\| \sum_{k'=0}^{k-1} \nabla F_{g,m}\left(\mathbf{x}\right) + \sum_{p'=0}^{p-1} \frac{1}{M} \sum_{m'=1}^{M} \sum_{k'=0}^{K-1} \nabla F_{g,m'}\left(\mathbf{x}\right) - \sum_{k'=0}^{k-1} \nabla F_{g}\left(\mathbf{x}\right)$$

$$- \sum_{p'=0}^{p-1} \frac{1}{M} \sum_{m'=1}^{M} \sum_{k'=0}^{K-1} \nabla F_{g}\left(\mathbf{x}\right) \|^2$$

$$+ 5\eta^2 \mathbb{E}\| \sum_{k'=0}^{k-1} \nabla F_{g}\left(\mathbf{x}\right) + \sum_{p'=0}^{p-1} \frac{1}{M} \sum_{m'=1}^{M} \sum_{k'=0}^{K-1} \nabla F_{g}\left(\mathbf{x}\right) - \sum_{k'=0}^{k-1} \nabla F\left(\mathbf{x}\right) - \sum_{p'=0}^{p-1} \frac{1}{M} \sum_{m'=1}^{M} \sum_{k'=0}^{K-1} \nabla F\left(\mathbf{x}\right) \|^2$$

$$+ 5\eta^2 \mathbb{E}\| \sum_{k'=0}^{k-1} \nabla F\left(\mathbf{x}\right) + \sum_{p'=0}^{p-1} \frac{1}{M} \sum_{m'=1}^{M} \sum_{k'=0}^{K-1} \nabla F\left(\mathbf{x}\right) \|^2,$$

where we apply the Jensen's Inequality for the first inequality. The term $\mathbb{E}\|\mathbf{x}_{g,p,m,k} - \mathbf{x}\|^2$ is decomposed into 5 components (e.g., local SGD noise, intra-group drift, inter-group drift, and global model discrepancy).

Bounding the first term in the left-hand inequality,

$$5\eta^2 \mathbb{E}\| \sum_{k'=0}^{k-1} \mathbf{g}_{g,p,m,k'} + \sum_{p'=0}^{p-1} \frac{1}{M} \sum_{m'=1}^{M} \sum_{k'=0}^{K-1} \mathbf{g}_{g,p',m',k'} - \sum_{k'=0}^{k-1} \nabla F_{g,m}\left(\mathbf{x}_{g,p,m,k'}\right)$$

$$- \sum_{p'=0}^{p-1} \frac{1}{M} \sum_{m'=1}^{M} \sum_{k'=0}^{K-1} \nabla F_{g,m'}\left(\mathbf{x}_{g,p',m',k'}\right) \|^2$$

$$\leq 10\eta^2 \sum_{k'=0}^{k-1} \mathbb{E}\|\mathbf{g}_{g,p,m,k'} - \nabla F_{g,m}\left(\mathbf{x}_{g,p,m,k'}\right)\|^2$$

$$+ 10\eta^2 \sum_{p'=0}^{p-1} \frac{1}{M^2} \sum_{m'=1}^{M} \sum_{k'=0}^{K-1} \mathbb{E}\|\mathbf{g}_{g,p',m',k'} - \nabla F_{g,m'}\left(\mathbf{x}_{g,p',m',k'}\right)\|^2$$

$$\leq 10\eta^2 k\sigma^2 + 10\eta^2 \frac{pK}{M}\sigma^2$$

Bounding the second term in the left-hand inequality,

$$5\eta^2 \mathbb{E}\| \sum_{k'=0}^{k-1} \nabla F_{g,m}\left(\mathbf{x}_{g,p,m,k'}\right) + \sum_{p'=0}^{p-1} \frac{1}{M} \sum_{m'=1}^{M} \sum_{k'=0}^{K-1} \nabla F_{g,m'}\left(\mathbf{x}_{g,p',m',k'}\right)$$

$$- \sum_{k'=0}^{k-1} \nabla F_{g,m}\left(\mathbf{x}\right) - \sum_{p'=0}^{p-1} \frac{1}{M} \sum_{m'=1}^{M} \sum_{k'=0}^{K-1} \nabla F_{g,m'}\left(\mathbf{x}\right)\|^2$$

$$\leq 10\eta^2 k \sum_{k'=0}^{k-1} \mathbb{E}\|\nabla F_{g,m}\left(\mathbf{x}_{g,p,m,k'}\right) - \nabla F_{g,m}\left(\mathbf{x}\right)\|^2$$

$$+ 10\eta^2 pMK \sum_{p'=0}^{p-1} \frac{1}{M^2} \sum_{m'=1}^{M} \sum_{k'=0}^{K-1} \mathbb{E}\|\nabla F_{g,m'}\left(\mathbf{x}_{g,p',m',k'}\right) - \nabla F_{g,m'}\left(\mathbf{x}\right)\|^2$$

$$\leq 10L^2\eta^2 k \sum_{k'=0}^{k-1} \mathbb{E}\|\mathbf{x}_{g,p,m,k'} - \mathbf{x}\|^2 + 10L^2\eta^2 \frac{pK}{M} \sum_{p'=0}^{p-1} \sum_{m'=1}^{M} \sum_{k'=0}^{K-1} \mathbb{E}\|\mathbf{x}_{g,p',m',k'} - \mathbf{x}\|^2$$

Bounding the third term in the left-hand inequality,

$$5\eta^2 \mathbb{E}\| \sum_{p'=0}^{p-1} \frac{1}{M} \sum_{m'=1}^{M} \sum_{k'=0}^{K-1} \nabla F_{g,m'}\left(\mathbf{x}\right) - \sum_{p'=0}^{p-1} \frac{1}{M} \sum_{m'=1}^{M} \sum_{k'=0}^{K-1} \nabla F_{g}\left(\mathbf{x}\right)$$

$$+ \sum_{k'=0}^{k-1} \nabla F_{g,m}\left(\mathbf{x}\right) - \sum_{k'=0}^{k-1} \nabla F_{g}\left(\mathbf{x}\right)\|^2 \leq 10\eta^2 k \sum_{k'=0}^{k-1} \mathbb{E}\|\nabla F_{g,m}\left(\mathbf{x}\right) - \nabla F_{g}\left(\mathbf{x}\right)\|^2$$

The group objective function is defined as the aggregation of local functions, i.e., $F_g = \frac{1}{M} \sum_{m'=1}^{M} F_{g,m'}$. Consequently, the summation of the deviations of local gradients from the group gradient vanishes.

Bounding the fourth term in the left-hand inequality,

$$5\eta^2 \mathbb{E}\| \sum_{k'=0}^{k-1} \nabla F_{g}\left(\mathbf{x}\right) + \sum_{p'=0}^{p-1} \frac{1}{M} \sum_{m'=1}^{M} \sum_{k'=0}^{K-1} \nabla F_{g}\left(\mathbf{x}\right) - \sum_{k'=0}^{k-1} \nabla F\left(\mathbf{x}\right) - \sum_{p'=0}^{p-1} \frac{1}{M} \sum_{m'=1}^{M} \sum_{k'=0}^{K-1} \nabla F\left(\mathbf{x}\right)\|^2$$

$$\leq 10\eta^2 k \sum_{k'=0}^{k-1} \mathbb{E}\|\nabla F_{g}\left(\mathbf{x}\right) - \nabla F\left(\mathbf{x}\right)\|^2 + 10\eta^2 \frac{pK}{M} \sum_{p'=0}^{p-1} \sum_{m'=1}^{M} \sum_{k'=0}^{K-1} \mathbb{E}\|\nabla F_{g}\left(\mathbf{x}\right) - \nabla F\left(\mathbf{x}\right)\|^2$$

Bounding the fifth term in the left-hand inequality,

$$5\eta^2 \mathbb{E}\| \sum_{k'=0}^{k-1} \nabla F\left(\mathbf{x}\right) + \sum_{p'=0}^{p-1} \frac{1}{M} \sum_{m'=1}^{M} \sum_{k'=0}^{K-1} \nabla F\left(\mathbf{x}\right)\|^2$$

$$\leq 10\eta^2 k \sum_{k'=0}^{k-1} \mathbb{E}\|\nabla F\left(\mathbf{x}\right)\|^2 + 10\eta^2 pMK \sum_{p'=0}^{p-1} \frac{1}{M^2} \sum_{m'=1}^{M} \sum_{k'=0}^{K-1} \mathbb{E}\|\nabla F\left(\mathbf{x}\right)\|^2$$

Substitute these terms into $E_r$,

$$E_r \leq 10\eta^2 \sum_{g,p,m,k} \left( k\sigma^2 + \frac{pK}{M}\sigma^2 + L^2 k \sum_{k'=0}^{k-1} \mathbb{E}\|\mathbf{x}_{g,p,m,k'} - \mathbf{x}\|^2 \right.$$

$$+ L^2 \frac{pK}{M} \sum_{p'=0}^{p-1} \sum_{m'=1}^{M} \sum_{k'=0}^{K-1} \mathbb{E}\|\mathbf{x}_{g,p',m',k'} - \mathbf{x}\|^2 + k \sum_{k'=0}^{k-1} \mathbb{E}\|\nabla F_{g,m}(\mathbf{x}) - \nabla F_g(\mathbf{x})\|^2$$

$$k \sum_{k'=0}^{k-1} \mathbb{E}\|\nabla F_g(\mathbf{x}) - \nabla F(\mathbf{x})\|^2 + \frac{pK}{M} \sum_{p'=0}^{p-1} \sum_{m'=1}^{M} \sum_{k'=0}^{K-1} \mathbb{E}\|\nabla F_g(\mathbf{x}) - \nabla F(\mathbf{x})\|^2$$

$$\left. k \sum_{k'=0}^{k-1} \mathbb{E}\|\nabla F(\mathbf{x})\|^2 + pMK \sum_{p'=0}^{p-1} \frac{1}{M^2} \sum_{m'=1}^{M} \sum_{k'=0}^{K-1} \mathbb{E}\|\nabla F(\mathbf{x})\|^2 \right)$$

Let $c_1 = \frac{1}{1 - 10L^2 P^2 K^2 \eta^2}$, we have

$$E_r \leq c_1 10\eta^2 GPMK^2\sigma^2 + c_1 10\eta^2 GP^2 K^2 \sigma^2 + c_1 10\eta^2 GPMK^3\zeta^2$$
$$+ c_1 10\eta^2 GPMK^3\delta^2 + c_1 10\eta^2 GP^3 MK^3\delta^2$$
$$+ c_1 10\eta^2 GPMK^3\|\nabla F(\mathbf{x})\|^2 + c_1 10\eta^2 GP^3 MK^3\|\nabla F(\mathbf{x})\|^2$$
$$\leq 10c_1\eta^2 GPMK^2\sigma^2 + 10c_1\eta^2 GP^2 K^2\sigma^2 + 10c_1\eta^2 GPMK^3\zeta^2 + 20c_1\eta^2 GP^3 MK^3\delta^2$$
$$+ 20c_1\eta^2 GP^3 MK^3\|\nabla F(\mathbf{x})\|^2$$

After substituting $E_r$ into $\mathbb{E}[F(\mathbf{x}^{(r+1)}) - F(\mathbf{x}^{(r)})]$, we can obtain

$$\mathbb{E}[F(\mathbf{x}^{(r+1)}) - F(\mathbf{x}^{(r)})]$$
$$\leq -\eta PK\left(\frac{1}{2} - 10c_1\eta^2 L^2 P^2 K^2\right)\mathbb{E}[\|\nabla F(\mathbf{x}^{(r)})\|^2] + \eta^2 L \frac{PK}{GM}\sigma^2$$
$$+ 5c_1\eta^3 L^2 PK^2\sigma^2 + 5c_1\eta^3 L^2 \frac{P^2 K^2}{M}\sigma^2 + 5c_1\eta^3 L^2 PK^3\zeta^2 + 10c_1\eta^3 L^2 P^3 K^3\delta^2.$$

Let $c_2 = \frac{1}{\frac{1}{2} - 10c_1\eta^2 L^2 P^2 K^2}$, then

$$\frac{1}{R}\sum_{r=0}^{R-1} \mathbb{E}[\|\nabla F(\mathbf{x}^{(r)})\|^2] \leq \frac{c_2\left(F(\mathbf{x}^{(0)}) - F(\mathbf{x}^{(R)})\right)}{\eta RPK} + c_2\eta L \frac{1}{GM}\sigma^2 + 5c_1 c_2\eta^2 L^2 K\sigma^2$$

$$+ 5c_1 c_2\eta^2 L^2 \frac{PK}{M}\sigma^2 + 5c_1 c_2\eta^2 L^2 K^2\zeta^2 + 10c_1 c_2\eta^2 L^2 P^2 K^2\delta^2.$$

Let $\tilde{\eta} = \eta PK$, and $\tilde{\eta} \leq \frac{1}{\sqrt{30}L}$, we have

$$\mathbb{E}[\|\nabla F(\bar{x}^{(R)})\|^2] \lesssim \frac{A}{\tilde{\eta}R} + \frac{L\tilde{\eta}\sigma^2}{GPMK} + \frac{L^2\tilde{\eta}^2\sigma^2}{P^2 K} + \frac{L^2\tilde{\eta}^2\sigma^2}{PMK} + \frac{L^2\tilde{\eta}^2\zeta^2}{P^2} + L^2\tilde{\eta}^2\delta^2$$

$$\mathbb{E}[\|\nabla F(\bar{x}^{(R)})\|^2] = \mathcal{O}\left( \frac{LA}{R} + \frac{(L\sigma^2 A)^{1/2}}{\sqrt{GPMKR}} + \frac{(L^2 A^2\sigma^2)^{1/3}}{(P^2 KR^2)^{1/3}} + \frac{(L^2 A^2\sigma^2)^{1/3}}{(PMKR^2)^{1/3}} \right.$$
$$\left. + \frac{(L^2 A^2\zeta^2)^{1/3}}{(P^2 R^2)^{1/3}} + \frac{(L^2 A^2\delta^2)^{1/3}}{(R^2)^{1/3}} \right)$$

where $\bar{x}^R$ is defined as a model uniformly sampled from the $x^{(0)}, \dots, x^{(R-1)}$ of previous iterations, and $\lesssim$ hides universal constants.

### D.3 PROOF OF THEOREM 1 FOR THE STAR-RING CASE

For Star-Ring,

$$\mathbf{x}_{g,p,m,k} - \mathbf{x} = \sum_{k'=0}^{k-1} \mathbf{g}_{g,p,m,k'} + \sum_{m'=1}^{m-1}\sum_{k'=0}^{K-1} \mathbf{g}_{g,p,m',k'} + \sum_{p'=0}^{p-1}\sum_{m'=1}^{M}\sum_{k=0}^{K-1} \mathbf{g}_{g,p',m',k'}.$$

To bound $E_r$, we first bound $\mathbb{E}\|\mathbf{x}_{g,p,m,k} - \mathbf{x}\|^2$.

$$\mathbb{E}\|\mathbf{x}_{g,p,m,k} - \mathbf{x}\|^2$$

$$= \eta^2 \mathbb{E}\| \sum_{k'=0}^{k-1} \mathbf{g}_{g,p,m,k'} + \sum_{m'=1}^{m-1}\sum_{k'=0}^{K-1} \mathbf{g}_{g,p,m',k'} + \sum_{p'=0}^{p-1}\sum_{m'=1}^{M}\sum_{k=0}^{K-1} \mathbf{g}_{g,p',m',k'}\|^2$$

$$\leq 5\eta^2 \mathbb{E}\| \sum_{k'=0}^{k-1} \mathbf{g}_{g,p,m,k'} + \sum_{m'=1}^{m-1}\sum_{k'=0}^{K-1} \mathbf{g}_{g,p,m',k'} + \sum_{p'=0}^{p-1}\sum_{m'=1}^{M}\sum_{k=0}^{K-1} \mathbf{g}_{g,p',m',k'} - \sum_{k'=0}^{k-1} \nabla F_{g,m}\left(\mathbf{x}_{g,p,m,k'}\right)$$

$$- \sum_{m'=1}^{m-1}\sum_{k'=0}^{K-1} \nabla F_{g,m'}\left(\mathbf{x}_{g,p,m',k'}\right) - \sum_{p'=0}^{p-1}\sum_{m'=1}^{M}\sum_{k'=0}^{K-1} \nabla F_{g,m'}\left(\mathbf{x}_{g,p',m',k'}\right)\|^2$$

$$+ 5\eta^2 \mathbb{E}\| \sum_{k'=0}^{k-1} \nabla F_{g,m}\left(\mathbf{x}_{g,p,m,k'}\right) - \sum_{k'=0}^{k-1} \nabla F_{g,m}\left(\mathbf{x}\right) + \sum_{m'=1}^{m-1}\sum_{k'=0}^{K-1} \nabla F_{g,m'}\left(\mathbf{x}_{g,p,m',k'}\right)$$

$$- \sum_{m'=1}^{m-1}\sum_{k'=0}^{K-1} \nabla F_{g,m'}\left(\mathbf{x}\right) + \sum_{p'=0}^{p-1}\sum_{m'=1}^{M}\sum_{k'=0}^{K-1} \nabla F_{g,m'}\left(\mathbf{x}_{g,p',m',k'}\right) - \sum_{p'=0}^{p-1}\sum_{m'=1}^{M}\sum_{k'=0}^{K-1} \nabla F_{g,m'}\left(\mathbf{x}\right)\|^2$$

$$+ 5\eta^2 \mathbb{E}\| \sum_{k'=0}^{k-1} \nabla F_{g,m}\left(\mathbf{x}\right) + \sum_{m'=1}^{m-1}\sum_{k'=0}^{K-1} \nabla F_{g,m'}\left(\mathbf{x}\right) + \sum_{p'=0}^{p-1}\sum_{m'=1}^{M}\sum_{k'=0}^{K-1} \nabla F_{g,m'}\left(\mathbf{x}\right) - \sum_{k'=0}^{k-1} \nabla F_g\left(\mathbf{x}\right)$$

$$- \sum_{m'=1}^{m-1}\sum_{k'=0}^{K-1} \nabla F_g\left(\mathbf{x}\right) - \sum_{p'=0}^{p-1}\sum_{m'=1}^{M}\sum_{k'=0}^{K-1} \nabla F_g\left(\mathbf{x}\right)\|^2$$

$$+ 5\eta^2 \mathbb{E}\| \sum_{k'=0}^{k-1} \nabla F_g\left(\mathbf{x}\right) + \sum_{m'=1}^{m-1}\sum_{k'=0}^{K-1} \nabla F_g\left(\mathbf{x}\right) + \sum_{p'=0}^{p-1}\sum_{m'=1}^{M}\sum_{k'=0}^{K-1} \nabla F_g\left(\mathbf{x}\right) - \sum_{k'=0}^{k-1} \nabla F\left(\mathbf{x}\right)$$

$$- \sum_{m'=1}^{m-1}\sum_{k'=0}^{K-1} \nabla F\left(\mathbf{x}\right) - \sum_{p'=0}^{p-1}\sum_{m'=1}^{M}\sum_{k'=0}^{K-1} \nabla F\left(\mathbf{x}\right)\|^2$$

$$+ 5\eta^2 \mathbb{E}\| \sum_{k'=0}^{k-1} \nabla F\left(\mathbf{x}\right) + \sum_{m'=1}^{m-1}\sum_{k'=0}^{K-1} \nabla F\left(\mathbf{x}\right) + \sum_{p'=0}^{p-1}\sum_{m'=1}^{M}\sum_{k'=0}^{K-1} \nabla F\left(\mathbf{x}\right)\|^2$$

Bounding the first term in the left-hand inequality,

$$5\eta^2 \mathbb{E}\| \sum_{k'=0}^{k-1} \mathbf{g}_{g,p,m,k'} + \sum_{m'=1}^{m-1}\sum_{k'=0}^{K-1} \mathbf{g}_{g,p,m',k'} + \sum_{p'=0}^{p-1}\sum_{m'=1}^{M}\sum_{k=0}^{K-1} \mathbf{g}_{g,p',m',k'} - \sum_{k'=0}^{k-1} \nabla F_{g,m}\left(\mathbf{x}_{g,p,m,k'}\right)$$

$$- \sum_{m'=1}^{m-1}\sum_{k'=0}^{K-1} \nabla F_{g,m'}\left(\mathbf{x}_{g,p,m',k'}\right) - \sum_{p'=0}^{p-1}\sum_{m'=1}^{M}\sum_{k'=0}^{K-1} \nabla F_{g,m'}\left(\mathbf{x}_{g,p',m',k'}\right)\|^2$$

$$\leq 15\eta^2 \sum_{k'=0}^{k-1} \mathbb{E}\|\mathbf{g}_{g,p,m,k'} - \nabla F_{g,m}\left(\mathbf{x}_{g,p,m,k'}\right)\|^2$$

$$+ 15\eta^2 \sum_{m'=1}^{m-1}\sum_{k'=0}^{K-1} \mathbb{E}\|\mathbf{g}_{g,p,m',k'} - \nabla F_{g,m'}\left(\mathbf{x}_{g,p,m',k'}\right)\|^2$$

$$+ 15\eta^2 \sum_{p'=0}^{p-1} \sum_{m'=1}^{M} \sum_{k'=0}^{K-1} \mathbb{E}\|\mathbf{g}_{g,p',m',k'} - \nabla F_{g,m'}\left(\mathbf{x}_{g,p',m',k'}\right)\|^2$$

$$\leq 15\eta^2 k\sigma^2 + 15\eta^2 mK\sigma^2 + 15\eta^2 pMK\sigma^2$$

Bounding the second term in the left-hand inequality,

$$5\eta^2 \mathbb{E}\| \sum_{k'=0}^{k-1} \nabla F_{g,m}\left(\mathbf{x}_{g,p,m,k'}\right) - \sum_{k'=0}^{k-1} \nabla F_{g,m}\left(\mathbf{x}\right) + \sum_{m'=1}^{m-1}\sum_{k'=0}^{K-1} \nabla F_{g,m'}\left(\mathbf{x}_{g,p,m',k'}\right)$$

$$- \sum_{m'=1}^{m-1}\sum_{k'=0}^{K-1} \nabla F_{g,m'}\left(\mathbf{x}\right) + \sum_{p'=0}^{p-1}\sum_{m'=1}^{M}\sum_{k'=0}^{K-1} \nabla F_{g,m'}\left(\mathbf{x}_{g,p',m',k'}\right) - \sum_{p'=0}^{p-1}\sum_{m'=1}^{M}\sum_{k'=0}^{K-1} \nabla F_{g,m'}\left(\mathbf{x}\right) \|^2$$

$$\leq 15\eta^2 k \sum_{k'=0}^{k-1} \mathbb{E}\|\nabla F_{g,m}\left(\mathbf{x}_{g,p,m,k'}\right) - \nabla F_{g,m}\left(\mathbf{x}\right)\|^2$$

$$+ 15\eta^2 mK \sum_{m'=1}^{m-1}\sum_{k'=0}^{K-1} \mathbb{E}\|\nabla F_{g,m'}\left(\mathbf{x}_{g,p,m',k'}\right) - \nabla F_{g,m'}\left(\mathbf{x}\right)\|^2$$

$$+ 15\eta^2 pMK \sum_{p'=0}^{p-1}\sum_{m'=1}^{M}\sum_{k'=0}^{K-1} \mathbb{E}\|\nabla F_{g,m'}\left(\mathbf{x}_{g,p',m',k'}\right) - \nabla F_{g,m'}\left(\mathbf{x}\right)\|^2$$

$$\leq 15L^2\eta^2 k \sum_{k'=0}^{k-1} \mathbb{E}\|\mathbf{x}_{g,p,m,k'} - \mathbf{x}\|^2 + 15L^2\eta^2 mK \sum_{m'=1}^{m-1}\sum_{k'=0}^{K-1} \mathbb{E}\|\mathbf{x}_{g,p,m',k'} - \mathbf{x}\|^2$$

$$+ 15L^2\eta^2 pMK \sum_{p'=0}^{p-1}\sum_{m'=1}^{M}\sum_{k'=0}^{K-1} \mathbb{E}\|\mathbf{x}_{g,p',m',k'} - \mathbf{x}\|^2$$

Bounding the third term in the left-hand inequality,

$$5\eta^2 \mathbb{E}\| \sum_{k'=0}^{k-1} \nabla F_{g,m}\left(\mathbf{x}\right) + \sum_{m'=1}^{m-1}\sum_{k'=0}^{K-1} \nabla F_{g,m'}\left(\mathbf{x}\right) + \sum_{p'=0}^{p-1}\sum_{m'=1}^{M}\sum_{k'=0}^{K-1} \nabla F_{g,m'}\left(\mathbf{x}\right) - \sum_{k'=0}^{k-1} \nabla F_{g}\left(\mathbf{x}\right)$$

$$- \sum_{m'=1}^{m-1}\sum_{k'=0}^{K-1} \nabla F_{g}\left(\mathbf{x}\right) - \sum_{p'=0}^{p-1}\sum_{m'=1}^{M}\sum_{k'=0}^{K-1} \nabla F_{g}\left(\mathbf{x}\right) \|^2$$

$$\leq 15\eta^2 k \sum_{k'=0}^{k-1} \mathbb{E}\|\nabla F_{g,m}\left(\mathbf{x}\right) - \nabla F_{g}\left(\mathbf{x}\right)\|^2 + 15\eta^2 mK \sum_{m'=1}^{m-1}\sum_{k'=0}^{K-1} \mathbb{E}\|\nabla F_{g,m'}\left(\mathbf{x}\right) - \nabla F_{g}\left(\mathbf{x}\right)\|^2$$

$$+ 15\eta^2 pMK \sum_{p'=0}^{p-1}\sum_{m'=1}^{M}\sum_{k'=0}^{K-1} \mathbb{E}\|\nabla F_{g,m'}\left(\mathbf{x}\right) - \nabla F_{g}\left(\mathbf{x}\right)\|^2$$

Bounding the fourth term in the left-hand inequality,

$$5\eta^2 \mathbb{E}\| \sum_{k'=0}^{k-1} \nabla F_{g}\left(\mathbf{x}\right) + \sum_{m'=1}^{m-1}\sum_{k'=0}^{K-1} \nabla F_{g}\left(\mathbf{x}\right) + \sum_{p'=0}^{p-1}\sum_{m'=1}^{M}\sum_{k'=0}^{K-1} \nabla F_{g}\left(\mathbf{x}\right)$$

$$- \sum_{k'=0}^{k-1} \nabla F\left(\mathbf{x}\right) - \sum_{m'=1}^{m-1}\sum_{k'=0}^{K-1} \nabla F\left(\mathbf{x}\right) - \sum_{p'=0}^{p-1}\sum_{m'=1}^{M}\sum_{k'=0}^{K-1} \nabla F\left(\mathbf{x}\right) \|^2$$

$$\leq 15\eta^2 k \sum_{k'=0}^{k-1} \mathbb{E}\|\nabla F_{g}\left(\mathbf{x}\right) - \nabla F\left(\mathbf{x}\right)\|^2 + 15\eta^2 mK \sum_{m'=1}^{m-1}\sum_{k'=0}^{K-1} \mathbb{E}\|\nabla F_{g}\left(\mathbf{x}\right) - \nabla F\left(\mathbf{x}\right)\|^2$$

$$+ 15\eta^2 pMK \sum_{p'=0}^{p-1} \sum_{m'=1}^{M} \sum_{k'=0}^{K-1} \mathbb{E}\|\nabla F_g(\mathbf{x}) - \nabla F(\mathbf{x})\|^2$$

Bounding the fifth term in the left-hand inequality,

$$5\eta^2 \mathbb{E}\| \sum_{k'=0}^{k-1} \nabla F(\mathbf{x}) + \sum_{m'=1}^{m-1} \sum_{k'=0}^{K-1} \nabla F(\mathbf{x}) + \sum_{p'=0}^{p-1} \sum_{m'=1}^{M} \sum_{k'=0}^{K-1} \nabla F(\mathbf{x})\|^2$$

$$\leq 15\eta^2 k \sum_{k'=0}^{k-1} \mathbb{E}\|\nabla F(\mathbf{x})\|^2 + 15\eta^2 mK \sum_{p'=0}^{p-1} \sum_{m'=1}^{m-1} \sum_{k'=0}^{K-1} \mathbb{E}\|\nabla F(\mathbf{x})\|^2$$

$$+ 15\eta^2 pMK \sum_{p'=0}^{p-1} \sum_{m'=1}^{M} \sum_{k'=0}^{K-1} \mathbb{E}\|\nabla F(\mathbf{x})\|^2$$

Substitute these terms into $E_r$,

$$E_r \leq 15\eta^2 \sum_{g,p,m,k} \Bigg( k\sigma^2 + mK\sigma^2 + pMK\sigma^2 + L^2 k \sum_{k'=0}^{k-1} \mathbb{E}\|\mathbf{x}_{g,p,m,k'} - \mathbf{x}\|^2$$

$$+ L^2 mK \sum_{m'=1}^{m-1} \sum_{k'=0}^{K-1} \mathbb{E}\|\mathbf{x}_{g,p,m',k'} - \mathbf{x}\|^2 + L^2 pMK \sum_{p'=0}^{p-1} \sum_{m'=1}^{M} \sum_{k'=0}^{K-1} \mathbb{E}\|\mathbf{x}_{g,p',m',k'} - \mathbf{x}\|^2$$

$$+ k \sum_{k'=0}^{k-1} \mathbb{E}\|\nabla F_{g,m}(\mathbf{x}) - \nabla F_g(\mathbf{x})\|^2 + mK \sum_{m'=1}^{m-1} \sum_{k'=0}^{K-1} \mathbb{E}\|\nabla F_{g,m'}(\mathbf{x}) - \nabla F_g(\mathbf{x})\|^2$$

$$+ k \sum_{k'=0}^{k-1} \mathbb{E}\|\nabla F_g(\mathbf{x}) - \nabla F(\mathbf{x})\|^2 + mK \sum_{m'=1}^{m-1} \sum_{k'=0}^{K-1} \mathbb{E}\|\nabla F_g(\mathbf{x}) - \nabla F(\mathbf{x})\|^2$$

$$+ pMK \sum_{p'=0}^{p-1} \sum_{m'=1}^{M} \sum_{k'=0}^{K-1} \mathbb{E}\|\nabla F_g(\mathbf{x}) - \nabla F(\mathbf{x})\|^2$$

$$+ k \sum_{k'=0}^{k-1} \mathbb{E}\|\nabla F(\mathbf{x})\|^2 + mK \sum_{m'=1}^{m-1} \sum_{k'=0}^{K-1} \mathbb{E}\|\nabla F(\mathbf{x})\|^2 + pMK \sum_{p'=0}^{p-1} \sum_{m'=1}^{M} \sum_{k'=0}^{K-1} \mathbb{E}\|\nabla F(\mathbf{x})\|^2 \Bigg)$$

$$\leq 15\eta^2 GPMK^2\sigma^2 + 15\eta^2 GPM^2K^2\sigma^2 + 15\eta^2 GP^2M^2K^2\sigma^2$$

$$+ 15\eta^2 GPMK^3\zeta^2 + 15\eta^2 GPM^3K^3\zeta^2 + 15\eta^2 GPMK^3\delta^2$$

$$+ 15\eta^2 GPM^3K^3\delta^2 + 15\eta^2 GP^3M^3K^3\delta^2 + 15\eta^2 GPMK^3\|\nabla F(\mathbf{x})\|^2$$

$$+ 15\eta^2 GPM^3K^3\|\nabla F(\mathbf{x})\|^2 + 15\eta^2 P^2M^2K^2\|\nabla F(\mathbf{x})\|^2 + 15L^2P^2M^2K^2\eta^2 E_r$$

Let $c_1 = \frac{1}{1-15L^2P^2M^2K^2\eta^2}$, we have

$$E_r \leq 45c_1\eta^2 GP^2M^2K^2\sigma^2 + 30c_1\eta^2 GPM^3K^3\zeta^2 + 45c_1\eta^2 GP^3M^3K^3\delta^2$$

$$+ 45c_1\eta^2 GP^3M^3K^3\|\nabla F(\mathbf{x})\|^2$$

After substituting $E_r$ into $\mathbb{E}[F(\mathbf{x}^{(r+1)}) - F(\mathbf{x}^{(r)})]$, we can obtain

$$\mathbb{E}[F(\mathbf{x}^{(r+1)}) - F(\mathbf{x}^{(r)})]$$

$$\leq -\eta MPK(\frac{1}{2} - \frac{45}{2}c_1\eta^2 L^2 P^2 M^2 K^2)\mathbb{E}[\|\nabla F(\mathbf{x}^{(r)})\|^2] + \eta^2 L\frac{PMK}{G}\sigma^2$$

$$+ \frac{45}{2}c_1\eta^3 L^2 P^2 M^2 K^2\sigma^2 + 15c_1\eta^3 L^2 PM^3K^3\zeta^2 + \frac{45}{2}c_1\eta^3 L^2 P^3 M^3 K^3\delta^2.$$

Let $c_2 = \frac{1}{\frac{1}{2} - \frac{45}{2} c_1 \eta^2 L^2 P^2 M^2 K^2}$, then

$$\frac{1}{R} \sum_{r=0}^{R-1} \mathbb{E}[\|\nabla F(\mathbf{x}^{(r)})\|^2] \leq \frac{c_2 \left( F(\mathbf{x}^{(0)}) - F(\mathbf{x}^{(R)}) \right)}{\eta RPMK} + c_2 \eta L \frac{1}{G} \sigma^2 + \frac{45}{2} c_1 c_2 \eta^2 L^2 PMK\sigma^2$$

$$+ 15 c_1 c_2 \eta^2 L^2 M^2 K^2 \zeta^2 + \frac{45}{2} c_1 c_2 \eta^2 L^2 P^2 M^2 K^2 \delta^2.$$

Let $\tilde{\eta} = \eta PMK$, and $\tilde{\eta} \leq \frac{1}{\sqrt{60}L}$, we have

$$\mathbb{E}[\|\nabla F(\bar{x}^{(R)})\|^2] \lesssim \frac{A}{\tilde{\eta}R} + \frac{L\tilde{\eta}\sigma^2}{GPMK} + \frac{L^2 \tilde{\eta}^2 \sigma^2}{PMK} + \frac{L^2 \tilde{\eta}^2 \zeta^2}{P^2} + L^2 \tilde{\eta}^2 \delta^2$$

$$\mathbb{E}[\|\nabla F(\bar{x}^{(R)})\|^2] = \mathcal{O}\left( \frac{LA}{R} + \frac{(L\sigma^2 A)^{1/2}}{\sqrt{GPMKR}} + \frac{(L^2 A^2 \sigma^2)^{1/3}}{(PMKR^2)^{1/3}} + \frac{(L^2 A^2 \zeta^2)^{1/3}}{(P^2 R^2)^{1/3}} \right.$$

$$\left. + \frac{(L^2 A^2 \delta^2)^{1/3}}{(R^2)^{1/3}} \right)$$

where $\bar{x}^R$ is defined as a model uniformly sampled from the $x^{(0)}, \ldots, x^{(R-1)}$ of previous iterations, and $\lesssim$ hides universal constants.

### D.4 PROOF OF THEOREM 1 FOR THE RING-STAR CASE

For Ring-Star,

$$\mathbf{x}_{g,p,m,k} - \mathbf{x} = \sum_{k'=0}^{k-1} \mathbf{g}_{g,p,m,k'} + \sum_{p'=0}^{p-1} \frac{1}{M} \sum_{m=1}^{M} \sum_{k=0}^{K-1} \mathbf{g}_{g,p',m',k'} + \sum_{g'=1}^{g-1} \sum_{p'=0}^{P-1} \frac{1}{M} \sum_{m=1}^{M} \sum_{k=0}^{K-1} \mathbf{g}_{g',p',m',k'}.$$

To bound $E_r$, we first bound $\mathbb{E}\|\mathbf{x}_{g,p,m,k} - \mathbf{x}\|^2$.

$$\mathbb{E}\|\mathbf{x}_{g,p,m,k} - \mathbf{x}\|^2$$

$$= \eta^2 \mathbb{E}\| \sum_{k'=0}^{k-1} \mathbf{g}_{g,p,m,k'} + \sum_{p'=0}^{p-1} \frac{1}{M} \sum_{m=1}^{M} \sum_{k=0}^{K-1} \mathbf{g}_{g,p',m',k'} + \sum_{g'=1}^{g-1} \sum_{p'=0}^{P-1} \frac{1}{M} \sum_{m=1}^{M} \sum_{k=0}^{K-1} \mathbf{g}_{g',p',m',k'} \|^2$$

$$\leq 5\eta^2 \mathbb{E}\| \sum_{k'=0}^{k-1} \mathbf{g}_{g,p,m,k'} + \sum_{p'=0}^{p-1} \frac{1}{M} \sum_{m=1}^{M} \sum_{k=0}^{K-1} \mathbf{g}_{g,p',m',k'} + \sum_{g'=1}^{g-1} \sum_{p'=0}^{P-1} \frac{1}{M} \sum_{m=1}^{M} \sum_{k=0}^{K-1} \mathbf{g}_{g',p',m',k'}$$

$$- \sum_{k'=0}^{k-1} \nabla F_{g,m}(\mathbf{x}_{g,p,m,k'}) - \sum_{p'=0}^{p-1} \frac{1}{M} \sum_{m=1}^{M} \sum_{k=0}^{K-1} \nabla F_{g,m'}(\mathbf{x}_{g,p',m',k'})$$

$$- \sum_{g'=1}^{g-1} \sum_{p'=0}^{P-1} \frac{1}{M} \sum_{m=1}^{M} \sum_{k=0}^{K-1} \nabla F_{g',m'}(\mathbf{x}_{g',p',m',k'}) \|^2$$

$$+ 5\eta^2 \mathbb{E}\| \sum_{k'=0}^{k-1} \nabla F_{g,m}(\mathbf{x}_{g,p,m,k'}) + \sum_{p'=0}^{p-1} \frac{1}{M} \sum_{m=1}^{M} \sum_{k=0}^{K-1} \nabla F_{g,m'}(\mathbf{x}_{g,p',m',k'})$$

$$+ \sum_{g'=1}^{g-1} \sum_{p'=0}^{P-1} \frac{1}{M} \sum_{m=1}^{M} \sum_{k=0}^{K-1} \nabla F_{g',m'}(\mathbf{x}_{g',p',m',k'}) - \sum_{k'=0}^{k-1} \nabla F_{g,m}(\mathbf{x})$$

$$- \sum_{p'=0}^{p-1} \frac{1}{M} \sum_{m=1}^{M} \sum_{k=0}^{K-1} \nabla F_{g,m'}(\mathbf{x}) - \sum_{g'=1}^{g-1} \sum_{p'=0}^{P-1} \frac{1}{M} \sum_{m=1}^{M} \sum_{k=0}^{K-1} \nabla F_{g',m'}(\mathbf{x}) \|^2$$

$$+ 5\eta^2 \mathbb{E}\| \sum_{k'=0}^{k-1} \nabla F_{g,m}(\mathbf{x}) + \sum_{p'=0}^{p-1} \frac{1}{M} \sum_{m=1}^{M} \sum_{k=0}^{K-1} \nabla F_{g,m'}(\mathbf{x}) + \sum_{g'=1}^{g-1} \sum_{p'=0}^{P-1} \frac{1}{M} \sum_{m=1}^{M} \sum_{k=0}^{K-1} \nabla F_{g',m'}(\mathbf{x})$$

$$-\sum_{k'=0}^{k-1}\nabla F_g\left(\mathbf{x}\right)-\sum_{p'=0}^{p-1}\frac{1}{M}\sum_{m=1}^{M}\sum_{k=0}^{K-1}\nabla F_g\left(\mathbf{x}\right)-\sum_{g'=1}^{g-1}\sum_{p'=0}^{P-1}\frac{1}{M}\sum_{m=1}^{M}\sum_{k=0}^{K-1}\nabla F_{g'}\left(\mathbf{x}\right)\|^2$$

$$+5\eta^2\mathbb{E}\|\sum_{k'=0}^{k-1}\nabla F_g\left(\mathbf{x}\right)+\sum_{p'=0}^{p-1}\frac{1}{M}\sum_{m=1}^{M}\sum_{k=0}^{K-1}\nabla F_g\left(\mathbf{x}\right)+\sum_{g'=1}^{g-1}\sum_{p'=0}^{P-1}\frac{1}{M}\sum_{m=1}^{M}\sum_{k=0}^{K-1}\nabla F_{g'}\left(\mathbf{x}\right)$$

$$-\sum_{k'=0}^{k-1}\nabla F\left(\mathbf{x}\right)-\sum_{p'=0}^{p-1}\frac{1}{M}\sum_{m=1}^{M}\sum_{k=0}^{K-1}\nabla F\left(\mathbf{x}\right)-\sum_{g'=1}^{g-1}\sum_{p'=0}^{P-1}\frac{1}{M}\sum_{m=1}^{M}\sum_{k=0}^{K-1}\nabla F\left(\mathbf{x}\right)\|^2$$

$$+5\eta^2\mathbb{E}\|\sum_{k'=0}^{k-1}\nabla F\left(\mathbf{x}\right)+\sum_{p'=0}^{p-1}\frac{1}{M}\sum_{m=1}^{M}\sum_{k=0}^{K-1}\nabla F\left(\mathbf{x}\right)+\sum_{g'=1}^{g-1}\sum_{p'=0}^{P-1}\frac{1}{M}\sum_{m=1}^{M}\sum_{k=0}^{K-1}\nabla F\left(\mathbf{x}\right)\|^2$$

Bounding the first term in the left-hand inequality,

$$5\eta^2\mathbb{E}\|\sum_{k'=0}^{k-1}\mathbf{g}_{g,p,m,k'}+\sum_{p'=0}^{p-1}\frac{1}{M}\sum_{m=1}^{M}\sum_{k=0}^{K-1}\mathbf{g}_{g,p',m',k'}+\sum_{g'=1}^{g-1}\sum_{p'=0}^{P-1}\frac{1}{M}\sum_{m=1}^{M}\sum_{k=0}^{K-1}\mathbf{g}_{g',p',m',k'}$$

$$-\sum_{k'=0}^{k-1}\nabla F_{g,m}\left(\mathbf{x}_{g,p,m,k'}\right)-\sum_{p'=0}^{p-1}\frac{1}{M}\sum_{m=1}^{M}\sum_{k=0}^{K-1}\nabla F_{g,m'}\left(\mathbf{x}_{g,p',m',k'}\right)$$

$$-\sum_{g'=1}^{g-1}\sum_{p'=0}^{P-1}\frac{1}{M}\sum_{m=1}^{M}\sum_{k=0}^{K-1}\nabla F_{g',m'}\left(\mathbf{x}_{g',p',m',k'}\right)\|^2$$

$$\leq 15\eta^2\sum_{k'=0}^{k-1}\mathbb{E}\|\mathbf{g}_{g,p,m,k'}-\nabla F_{g,m}\left(\mathbf{x}_{g,p,m,k'}\right)\|^2$$

$$+15\eta^2\sum_{p'=0}^{p-1}\frac{1}{M^2}\sum_{m'=1}^{M}\sum_{k'=0}^{K-1}\mathbb{E}\|\mathbf{g}_{g,p',m',k'}-\nabla F_{g,m'}\left(\mathbf{x}_{g,p',m',k'}\right)\|^2$$

$$+15\eta^2\sum_{g'=1}^{g-1}\sum_{p'=0}^{P-1}\frac{1}{M^2}\sum_{m=1}^{M}\sum_{k=0}^{K-1}\mathbb{E}\|\mathbf{g}_{g',p',m',k'}-\nabla F_{g',m'}\left(\mathbf{x}_{g',p',m',k'}\right)\|^2$$

$$\leq 15\eta^2 k\sigma^2+15\eta^2\frac{pK}{M}\sigma^2+15\eta^2\frac{gPK}{M}\sigma^2$$

Bounding the second term in the left-hand inequality,

$$5\eta^2\mathbb{E}\|\sum_{k'=0}^{k-1}\nabla F_{g,m}\left(\mathbf{x}_{g,p,m,k'}\right)+\sum_{p'=0}^{p-1}\frac{1}{M}\sum_{m=1}^{M}\sum_{k=0}^{K-1}\nabla F_{g,m'}\left(\mathbf{x}_{g,p',m',k'}\right)$$

$$+\sum_{g'=1}^{g-1}\sum_{p'=0}^{P-1}\frac{1}{M}\sum_{m=1}^{M}\sum_{k=0}^{K-1}\nabla F_{g',m'}\left(\mathbf{x}_{g',p',m',k'}\right)-\sum_{k'=0}^{k-1}\nabla F_{g,m}\left(\mathbf{x}\right)$$

$$-\sum_{p'=0}^{p-1}\frac{1}{M}\sum_{m=1}^{M}\sum_{k=0}^{K-1}\nabla F_{g,m'}\left(\mathbf{x}\right)-\sum_{g'=1}^{g-1}\sum_{p'=0}^{P-1}\frac{1}{M}\sum_{m=1}^{M}\sum_{k=0}^{K-1}\nabla F_{g',m'}\left(\mathbf{x}\right)\|^2$$

$$\leq 15\eta^2 k\sum_{k'=0}^{k-1}\mathbb{E}\|\nabla F_{g,m}\left(\mathbf{x}_{g,p,m,k'}\right)-\nabla F_{g,m}\left(\mathbf{x}\right)\|^2$$

$$+15\eta^2 pMK\sum_{p'=0}^{p-1}\frac{1}{M^2}\sum_{m'=1}^{M}\sum_{k'=0}^{K-1}\mathbb{E}\|\nabla F_{g,m'}\left(\mathbf{x}_{g,p',m',k'}\right)-\nabla F_{g,m'}\left(\mathbf{x}\right)\|^2$$

$$+ 15\eta^2 g P M K \sum_{g'=1}^{g-1} \sum_{p'=0}^{P-1} \frac{1}{M^2} \sum_{m'=1}^{M} \sum_{k'=0}^{K-1} \mathbb{E}\|\nabla F_{g,m'}\left(\mathbf{x}_{g,p',m',k'}\right) - \nabla F_{g,m'}\left(\mathbf{x}\right)\|^2$$

$$\leq 15L^2\eta^2 k \sum_{k'=0}^{k-1} \mathbb{E}\|\mathbf{x}_{g,p,m,k'} - \mathbf{x}\|^2 + 15L^2\eta^2 \frac{pK}{M} \sum_{p'=0}^{p-1} \sum_{m'=1}^{M} \sum_{k'=0}^{K-1} \mathbb{E}\|\mathbf{x}_{g,p',m',k'} - \mathbf{x}\|^2$$

$$+ 15L^2\eta^2 \frac{gPK}{M} \sum_{g'=1}^{g-1} \sum_{p'=0}^{P-1} \sum_{m'=1}^{M} \sum_{k'=0}^{K-1} \mathbb{E}\|\mathbf{x}_{g,p',m',k'} - \mathbf{x}\|^2$$

Bounding the third term in the left-hand inequality,

$$5\eta^2 \mathbb{E}\| \sum_{k'=0}^{k-1} \nabla F_{g,m}\left(\mathbf{x}\right) + \sum_{p'=0}^{p-1} \frac{1}{M} \sum_{m=1}^{M} \sum_{k=0}^{K-1} \nabla F_{g,m'}\left(\mathbf{x}\right) + \sum_{g'=1}^{g-1} \sum_{p'=0}^{P-1} \frac{1}{M} \sum_{m=1}^{M} \sum_{k=0}^{K-1} \nabla F_{g',m'}\left(\mathbf{x}\right)$$

$$- \sum_{k'=0}^{k-1} \nabla F_g\left(\mathbf{x}\right) - \sum_{p'=0}^{p-1} \frac{1}{M} \sum_{m=1}^{M} \sum_{k=0}^{K-1} \nabla F_g\left(\mathbf{x}\right) - \sum_{g'=1}^{g-1} \sum_{p'=0}^{P-1} \frac{1}{M} \sum_{m=1}^{M} \sum_{k=0}^{K-1} \nabla F_{g'}\left(\mathbf{x}\right)\|^2$$

$$\leq 15\eta^2 k \sum_{k'=0}^{k-1} \mathbb{E}\|\nabla F_{g,m}\left(\mathbf{x}\right) - \nabla F_g\left(\mathbf{x}\right)\|^2$$

$$+ 15\eta^2 \frac{pK}{M} \sum_{p'=0}^{p-1} \sum_{m'=1}^{M} \sum_{k'=0}^{K-1} \mathbb{E}\|\nabla F_{g,m'}\left(\mathbf{x}\right) - \nabla F_g\left(\mathbf{x}\right)\|^2$$

$$+ 15\eta^2 \frac{gPK}{M} \sum_{g'=1}^{g-1} \sum_{p'=0}^{P-1} \sum_{m'=1}^{M} \sum_{k'=0}^{K-1} \mathbb{E}\|\nabla F_{g',m'}\left(\mathbf{x}\right) - \nabla F_{g'}\left(\mathbf{x}\right)\|^2$$

Bounding the fourth term in the left-hand inequality,

$$5\eta^2 \mathbb{E}\| \sum_{k'=0}^{k-1} \nabla F_g\left(\mathbf{x}\right) + \sum_{p'=0}^{p-1} \frac{1}{M} \sum_{m=1}^{M} \sum_{k=0}^{K-1} \nabla F_g\left(\mathbf{x}\right) + \sum_{g'=1}^{g-1} \sum_{p'=0}^{P-1} \frac{1}{M} \sum_{m=1}^{M} \sum_{k=0}^{K-1} \nabla F_{g'}\left(\mathbf{x}\right)$$

$$- \sum_{k'=0}^{k-1} \nabla F\left(\mathbf{x}\right) - \sum_{p'=0}^{p-1} \frac{1}{M} \sum_{m=1}^{M} \sum_{k=0}^{K-1} \nabla F\left(\mathbf{x}\right) - \sum_{g'=1}^{g-1} \sum_{p'=0}^{P-1} \frac{1}{M} \sum_{m=1}^{M} \sum_{k=0}^{K-1} \nabla F\left(\mathbf{x}\right)\|^2$$

$$\leq 15\eta^2 k \sum_{k'=0}^{k-1} \mathbb{E}\|\nabla F_g\left(\mathbf{x}\right) - \nabla F\left(\mathbf{x}\right)\|^2 + 15\eta^2 \frac{pK}{M} \sum_{p'=0}^{p-1} \sum_{m'=1}^{M} \sum_{k'=0}^{K-1} \mathbb{E}\|\nabla F_g\left(\mathbf{x}\right) - \nabla F\left(\mathbf{x}\right)\|^2$$

$$+ 15\eta^2 \frac{gPK}{M} \sum_{g'=1}^{g-1} \sum_{p'=0}^{P-1} \sum_{m'=1}^{M} \sum_{k'=0}^{K-1} \mathbb{E}\|\nabla F_{g'}\left(\mathbf{x}\right) - \nabla F\left(\mathbf{x}\right)\|^2$$

Bounding the fifth term in the left-hand inequality,

$$5\eta^2 \mathbb{E}\| \sum_{k'=0}^{k-1} \nabla F\left(\mathbf{x}\right) + \sum_{p'=0}^{p-1} \frac{1}{M} \sum_{m=1}^{M} \sum_{k=0}^{K-1} \nabla F\left(\mathbf{x}\right) + \sum_{g'=1}^{g-1} \sum_{p'=0}^{P-1} \frac{1}{M} \sum_{m=1}^{M} \sum_{k=0}^{K-1} \nabla F\left(\mathbf{x}\right)\|^2$$

$$\leq 15\eta^2 k \sum_{k'=0}^{k-1} \mathbb{E}\|\nabla F\left(\mathbf{x}\right)\|^2 + 15\eta^2 pMK \sum_{p'=0}^{p-1} \frac{1}{M^2} \sum_{m'=1}^{M} \sum_{k'=0}^{K-1} \mathbb{E}\|\nabla F\left(\mathbf{x}\right)\|^2$$

$$+ 15\eta^2 g P M K \sum_{g'=1}^{g-1} \sum_{p'=0}^{P-1} \frac{1}{M^2} \sum_{m'=1}^{M} \sum_{k'=0}^{K-1} \mathbb{E}\|\nabla F\left(\mathbf{x}\right)\|^2$$

Substitute these terms into $E_r$,

$$
\begin{aligned}
E_r \leq 15\eta^2 \sum_{g,p,m,k} &\left( k\sigma^2 + \frac{pK}{M}\sigma^2 + \frac{gPK}{M}\sigma^2 \right.\\
&+ L^2 k \sum_{k'=0}^{k-1} \mathbb{E}\|\mathbf{x}_{g,p,m,k'} - \mathbf{x}\|^2 + L^2 \frac{pK}{M} \sum_{p'=0}^{p-1} \sum_{m'=1}^{M} \sum_{k'=0}^{K-1} \mathbb{E}\|\mathbf{x}_{g,p',m',k'} - \mathbf{x}\|^2 \\
&+ L^2 \frac{gPK}{M} \sum_{g'=1}^{g-1} \sum_{p'=0}^{P-1} \sum_{m'=1}^{M} \sum_{k'=0}^{K-1} \mathbb{E}\|\mathbf{x}_{g,p',m',k'} - \mathbf{x}\|^2 + k \sum_{k'=0}^{k-1} \mathbb{E}\|\nabla F_{g,m}(\mathbf{x}) - \nabla F_g(\mathbf{x})\|^2 \\
&+ k \sum_{k'=0}^{k-1} \mathbb{E}\|\nabla F_g(\mathbf{x}) - \nabla F(\mathbf{x})\|^2 + \frac{pK}{M} \sum_{p'=0}^{p-1} \sum_{m'=1}^{M} \sum_{k'=0}^{K-1} \mathbb{E}\|\nabla F_g(\mathbf{x}) - \nabla F(\mathbf{x})\|^2 \\
&+ \frac{gPK}{M} \sum_{g'=1}^{g-1} \sum_{p'=0}^{P-1} \sum_{m'=1}^{M} \sum_{k'=0}^{K-1} \mathbb{E}\|\nabla F_g(\mathbf{x}) - \nabla F(\mathbf{x})\|^2 + k \sum_{k'=0}^{k-1} \mathbb{E}\|\nabla F(\mathbf{x})\|^2 \\
&\left. + pMK \sum_{p'=0}^{p-1} \frac{1}{M^2} \sum_{m'=1}^{M} \sum_{k'=0}^{K-1} \mathbb{E}\|\nabla F(\mathbf{x})\|^2 + gPMK \sum_{g'=1}^{g-1} \sum_{p'=0}^{P-1} \frac{1}{M^2} \sum_{m'=1}^{M} \sum_{k'=0}^{K-1} \mathbb{E}\|\nabla F(\mathbf{x})\|^2 \right) \\
\leq\ & 15\eta^2 GPMK^2\sigma^2 + 15\eta^2 GP^2K^2\sigma^2 + 15\eta^2 G^2P^2K^2\sigma^2 \\
&+ 15\eta^2 GPMK^3\zeta^2 + 15\eta^2 GPMK^3\delta^2 + 15\eta^2 GP^3MK^3\delta^2 + 15\eta^2 G^3P^3MK^3\delta^2 \\
&+ 15\eta^2 GPMK^3\|\nabla F(\mathbf{x})\|^2 + 15\eta^2 GP^3MK^3\|\nabla F(\mathbf{x})\|^2 + 15\eta^2 G^3P^3MK^3\|\nabla F(\mathbf{x})\|^2 \\
&+ 15L^2 G^2P^2K^2\eta^2 E_r.
\end{aligned}
$$

Let $c_1 = \frac{1}{1-15L^2G^2P^2K^2\eta^2}$, we have

$$
\begin{aligned}
E_r \leq\ & 15c_1\eta^2 GPMK^2\sigma^2 + 30c_1\eta^2 G^2P^2K^2\sigma^2 + 15c_1\eta^2 GPMK^3\zeta^2 + 45c_1\eta^2 G^3P^3MK^3\delta^2 \\
&+ 45c_1\eta^2 G^3P^3MK^3\|\nabla F(\mathbf{x})\|^2
\end{aligned}
$$

After substituting $E_r$ into $\mathbb{E}[F(\mathbf{x}^{(r+1)}) - F(\mathbf{x}^{(r)})]$, we can obtain

$$
\begin{aligned}
&\mathbb{E}[F(\mathbf{x}^{(r+1)}) - F(\mathbf{x}^{(r)})] \\
&\leq -\eta GPK\left(\frac{1}{2} - \frac{45}{2}c_1\eta^2 L^2G^2P^2K^2\right)\mathbb{E}[\|\nabla F(\mathbf{x}^{(r)})\|^2] + \eta^2 L\frac{GPK}{M}\sigma^2 \\
&\quad + \frac{15}{2}c_1\eta^3 L^2 GPK^2\sigma^2 + 15c_1\eta^3 L^2\frac{G^2P^2K^2}{M}\sigma^2 + \frac{15}{2}c_1\eta^3 L^2 GPK^3\zeta^2 + \frac{45}{2}c_1\eta^3 L^2 G^3P^3K^3\delta^2.
\end{aligned}
$$

Let $c_2 = \frac{1}{\frac{1}{2} - \frac{45}{2}c_1\eta^2 L^2G^2P^2K^2}$, then

$$
\begin{aligned}
\frac{1}{R}\sum_{r=0}^{R-1}\mathbb{E}[\|\nabla F(\mathbf{x}^{(r)})\|^2] \leq\ & \frac{c_2\left(F(\mathbf{x}^{(0)}) - F(\mathbf{x}^{(R)})\right)}{\eta RGPK} + c_2\eta L\frac{1}{M}\sigma^2 + \frac{15}{2}c_1c_2\eta^2 L^2 K\sigma^2 \\
&+ 15c_1c_2\eta^2 L^2\frac{GPK}{M}\sigma^2 + \frac{15}{2}c_1c_2\eta^2 L^2 K^2\zeta^2 + \frac{45}{2}c_1c_2\eta^2 L^2 G^2P^2K^2\delta^2.
\end{aligned}
$$

Let $\tilde{\eta} = \eta GPK$, and $\tilde{\eta} \leq \frac{1}{\sqrt{60}L}$, we have

$$
\mathbb{E}[\|\nabla F(\bar{x}^{(R)})\|^2] \lesssim \frac{A}{\tilde{\eta}R} + \frac{L\tilde{\eta}\sigma^2}{GPMK} + \frac{L^2\tilde{\eta}^2\sigma^2}{G^2P^2K} + \frac{L^2\tilde{\eta}^2\sigma^2}{GPMK} + \frac{L^2\tilde{\eta}^2\zeta^2}{G^2P^2} + L^2\tilde{\eta}^2\delta^2.
$$

$$
\mathbb{E}[\|\nabla F(\bar{x}^{(R)})\|^2] = \mathcal{O}\left( \frac{LA}{R} + \frac{(L\sigma^2 A)^{1/2}}{\sqrt{GPMKR}} + \frac{(L^2 A^2\sigma^2)^{1/3}}{(G^2P^2KR^2)^{1/3}} + \frac{(L^2 A^2\sigma^2)^{1/3}}{(GPMKR^2)^{1/3}} \right)
$$

$$+ \frac{(L^2 A^2 \zeta^2)^{1/3}}{(G^2 P^2 R^2)^{1/3}} + \frac{(L^2 A^2 \delta^2)^{1/3}}{(R^2)^{1/3}} \Bigg)$$

where $\bar{x}^R$ is defined as a model uniformly sampled from the $x^{(0)}, \ldots, x^{(R-1)}$ of previous iterations, and $\lesssim$ hides universal constants.

## D.5 PROOF OF THEOREM 1 FOR THE RING-RING CASE

For Ring-Ring,

$$\mathbf{x}_{g,p,m,k} - \mathbf{x} = \sum_{k'=0}^{k-1} \mathbf{g}_{g,p,m,k'} + \sum_{m'=1}^{m-1}\sum_{k'=0}^{K-1} \mathbf{g}_{g,p,m',k'} + \sum_{p'=0}^{p-1}\sum_{m'=1}^{M}\sum_{k=0}^{K-1} \mathbf{g}_{g,p',m',k'}$$
$$+ \sum_{g'=1}^{g-1}\sum_{p'=0}^{P-1}\sum_{m'=1}^{M}\sum_{k=0}^{K-1} \mathbf{g}_{g',p',m',k'}.$$

To bound $E_r$, we first bound $\mathbb{E}\|\mathbf{x}_{g,p,m,k} - \mathbf{x}\|^2$.

$\mathbb{E}\|\mathbf{x}_{g,p,m,k} - \mathbf{x}\|^2$

$$= \eta^2 \mathbb{E}\Big\| \sum_{k'=0}^{k-1} \mathbf{g}_{g,p,m,k'} + \sum_{m'=1}^{m-1}\sum_{k'=0}^{K-1} \mathbf{g}_{g,p,m',k'} + \sum_{p'=0}^{p-1}\sum_{m'=1}^{M}\sum_{k=0}^{K-1} \mathbf{g}_{g,p',m',k'}$$
$$+ \sum_{g'=1}^{g-1}\sum_{p'=0}^{P-1}\sum_{m'=1}^{M}\sum_{k=0}^{K-1} \mathbf{g}_{g',p',m',k'} \Big\|^2$$

$$\leq 5\eta^2 \mathbb{E}\Big\| \sum_{k'=0}^{k-1} \mathbf{g}_{g,p,m,k'} + \sum_{m'=1}^{m-1}\sum_{k'=0}^{K-1} \mathbf{g}_{g,p,m',k'} + \sum_{p'=0}^{p-1}\sum_{m'=1}^{M}\sum_{k=0}^{K-1} \mathbf{g}_{g,p',m',k'}$$
$$+ \sum_{g'=1}^{g-1}\sum_{p'=0}^{P-1}\sum_{m'=1}^{M}\sum_{k=0}^{K-1} \mathbf{g}_{g',p',m',k'} - \sum_{k'=0}^{k-1} \nabla F_{g,m}\left(\mathbf{x}_{g,p,m,k'}\right) - \sum_{m'=1}^{m-1}\sum_{k'=0}^{K-1} \nabla F_{g,m'}\left(\mathbf{x}_{g,p,m',k'}\right)$$
$$- \sum_{p'=0}^{p-1}\sum_{m'=1}^{M}\sum_{k'=0}^{K-1} \nabla F_{g,m'}\left(\mathbf{x}_{g,p',m',k'}\right) - \sum_{g'=1}^{g-1}\sum_{p'=0}^{P-1}\sum_{m'=1}^{M}\sum_{k'=0}^{K-1} \nabla F_{g,m'}\left(\mathbf{x}_{g',p',m',k'}\right) \Big\|^2$$

$$+ 5\eta^2 \mathbb{E}\Big\| \sum_{k'=0}^{k-1} \nabla F_{g,m}\left(\mathbf{x}_{g,p,m,k'}\right) + \sum_{m'=1}^{m-1}\sum_{k'=0}^{K-1} \nabla F_{g,m'}\left(\mathbf{x}_{g,p,m',k'}\right)$$
$$+ \sum_{p'=0}^{p-1}\sum_{m'=1}^{M}\sum_{k'=0}^{K-1} \nabla F_{g,m'}\left(\mathbf{x}_{g,p',m',k'}\right) + \sum_{g'=1}^{g-1}\sum_{p'=0}^{P-1}\sum_{m'=1}^{M}\sum_{k'=0}^{K-1} \nabla F_{g,m'}\left(\mathbf{x}_{g',p',m',k'}\right)$$
$$- \sum_{k'=0}^{k-1} \nabla F_{g,m}\left(\mathbf{x}\right) - \sum_{m'=1}^{m-1}\sum_{k'=0}^{K-1} \nabla F_{g,m'}\left(\mathbf{x}\right) - \sum_{p'=0}^{p-1}\sum_{m'=1}^{M}\sum_{k'=0}^{K-1} \nabla F_{g,m'}\left(\mathbf{x}\right)$$
$$- \sum_{g'=1}^{g-1}\sum_{p'=0}^{P-1}\sum_{m'=1}^{M}\sum_{k'=0}^{K-1} \nabla F_{g',m'}\left(\mathbf{x}\right) \Big\|^2$$

$$+ 5\eta^2 \mathbb{E}\Big\| \sum_{k'=0}^{k-1} \nabla F_{g,m}\left(\mathbf{x}\right) + \sum_{m'=1}^{m-1}\sum_{k'=0}^{K-1} \nabla F_{g,m'}\left(\mathbf{x}\right) + \sum_{p'=0}^{p-1}\sum_{m'=1}^{M}\sum_{k'=0}^{K-1} \nabla F_{g,m'}\left(\mathbf{x}\right)$$
$$+ \sum_{g'=1}^{g-1}\sum_{p'=0}^{P-1}\sum_{m'=1}^{M}\sum_{k'=0}^{K-1} \nabla F_{g',m'}\left(\mathbf{x}\right) - \sum_{k'=0}^{k-1} \nabla F_g\left(\mathbf{x}\right) - \sum_{m'=1}^{m-1}\sum_{k'=0}^{K-1} \nabla F_g\left(\mathbf{x}\right)$$
$$- \sum_{p'=0}^{p-1}\sum_{m'=1}^{M}\sum_{k'=0}^{K-1} \nabla F_g\left(\mathbf{x}\right) - \sum_{g'=1}^{g-1}\sum_{p'=0}^{P-1}\sum_{m'=1}^{M}\sum_{k'=0}^{K-1} \nabla F_{g'}\left(\mathbf{x}\right) \Big\|^2$$

$$+ 5\eta^2 \mathbb{E}\| \sum_{k'=0}^{k-1} \nabla F_g(\mathbf{x}) + \sum_{m'=1}^{m-1}\sum_{k'=0}^{K-1} \nabla F_g(\mathbf{x}) + \sum_{p'=0}^{p-1}\sum_{m'=1}^{M}\sum_{k'=0}^{K-1} \nabla F_g(\mathbf{x})$$

$$+ \sum_{g'=1}^{g-1}\sum_{p'=0}^{P-1}\sum_{m'=1}^{M}\sum_{k'=0}^{K-1} \nabla F_{g'}(\mathbf{x}) - \sum_{k'=0}^{k-1} \nabla F(\mathbf{x}) - \sum_{m'=1}^{m-1}\sum_{k'=0}^{K-1} \nabla F(\mathbf{x})$$

$$- \sum_{p'=0}^{p-1}\sum_{m'=1}^{M}\sum_{k'=0}^{K-1} \nabla F(\mathbf{x}) - \sum_{g'=1}^{g-1}\sum_{p'=0}^{P-1}\sum_{m'=1}^{M}\sum_{k'=0}^{K-1} \nabla F(\mathbf{x}) \|^2$$

$$+ 5\eta^2 \mathbb{E}\| \sum_{k'=0}^{k-1} \nabla F(\mathbf{x}) + \sum_{m'=1}^{m-1}\sum_{k'=0}^{K-1} \nabla F(\mathbf{x}) + \sum_{p'=0}^{p-1}\sum_{m'=1}^{M}\sum_{k'=0}^{K-1} \nabla F(\mathbf{x}) + \sum_{g'=1}^{g-1}\sum_{p'=0}^{P-1}\sum_{m'=1}^{M}\sum_{k'=0}^{K-1} \nabla F(\mathbf{x})\|^2$$

Bounding the first term in the left-hand inequality,

$$5\eta^2 \mathbb{E}\| \sum_{k'=0}^{k-1} \mathbf{g}_{g,p,m,k'} + \sum_{m'=1}^{m-1}\sum_{k'=0}^{K-1} \mathbf{g}_{g,p,m',k'} + \sum_{p'=0}^{p-1}\sum_{m'=1}^{M}\sum_{k=0}^{K-1} \mathbf{g}_{g,p',m',k'}$$

$$+ \sum_{g'=1}^{g-1}\sum_{p'=0}^{P-1}\sum_{m'=1}^{M}\sum_{k=0}^{K-1} \mathbf{g}_{g',p',m',k'} - \sum_{k'=0}^{k-1} \nabla F_{g,m}(\mathbf{x}_{g,p,m,k'}) - \sum_{m'=1}^{m-1}\sum_{k'=0}^{K-1} \nabla F_{g,m'}(\mathbf{x}_{g,p,m',k'})$$

$$- \sum_{p'=0}^{p-1}\sum_{m'=1}^{M}\sum_{k'=0}^{K-1} \nabla F_{g,m'}(\mathbf{x}_{g,p',m',k'}) - \sum_{g'=1}^{g-1}\sum_{p'=0}^{P-1}\sum_{m'=1}^{M}\sum_{k'=0}^{K-1} \nabla F_{g,m'}(\mathbf{x}_{g',p',m',k'}) \|^2$$

$$\leq 20\eta^2 \sum_{k'=0}^{k-1} \mathbb{E}\|\mathbf{g}_{g,p,m,k'} - \nabla F_{g,m}(\mathbf{x}_{g,p,m,k'})\|^2$$

$$+ 20\eta^2 \sum_{m'=1}^{m-1}\sum_{k'=0}^{K-1} \mathbb{E}\|\mathbf{g}_{g,p,m',k'} - \nabla F_{g,m'}(\mathbf{x}_{g,p,m',k'})\|^2$$

$$+ 20\eta^2 \sum_{p'=0}^{p-1}\sum_{m'=1}^{M}\sum_{k'=0}^{K-1} \mathbb{E}\|\mathbf{g}_{g,p',m',k'} - \nabla F_{g,m'}(\mathbf{x}_{g,p',m',k'})\|^2$$

$$+ 20\eta^2 \sum_{g'=1}^{g-1}\sum_{p'=0}^{P-1}\sum_{m'=1}^{M}\sum_{k'=0}^{K-1} \mathbb{E}\|\mathbf{g}_{g',p',m',k'} - \nabla F_{g,m'}(\mathbf{x}_{g',p',m',k'})\|^2$$

$$\leq 20\eta^2 k\sigma^2 + 20\eta^2 mK\sigma^2 + 20\eta^2 pMK\sigma^2 + 20\eta^2 gPMK\sigma^2$$

Bounding the second term in the left-hand inequality,

$$5\eta^2 \mathbb{E}\| \sum_{k'=0}^{k-1} \nabla F_{g,m}(\mathbf{x}_{g,p,m,k'}) + \sum_{m'=1}^{m-1}\sum_{k'=0}^{K-1} \nabla F_{g,m'}(\mathbf{x}_{g,p,m',k'})$$

$$+ \sum_{p'=0}^{p-1}\sum_{m'=1}^{M}\sum_{k'=0}^{K-1} \nabla F_{g,m'}(\mathbf{x}_{g,p',m',k'}) + \sum_{g'=1}^{g-1}\sum_{p'=0}^{P-1}\sum_{m'=1}^{M}\sum_{k'=0}^{K-1} \nabla F_{g,m'}(\mathbf{x}_{g',p',m',k'})$$

$$- \sum_{k'=0}^{k-1} \nabla F_{g,m}(\mathbf{x}) - \sum_{m'=1}^{m-1}\sum_{k'=0}^{K-1} \nabla F_{g,m'}(\mathbf{x}) - \sum_{p'=0}^{p-1}\sum_{m'=1}^{M}\sum_{k'=0}^{K-1} \nabla F_{g,m'}(\mathbf{x})$$

$$- \sum_{g'=1}^{g-1}\sum_{p'=0}^{P-1}\sum_{m'=1}^{M}\sum_{k'=0}^{K-1} \nabla F_{g',m'}(\mathbf{x}) \|^2$$

$$\leq 20\eta^2 k \sum_{k'=0}^{k-1} \mathbb{E}\|\nabla F_{g,m}(\mathbf{x}_{g,p,m,k'}) - \nabla F_{g,m}(\mathbf{x})\|^2$$

$$+ 20\eta^2 mK \sum_{m'=1}^{m-1} \sum_{k'=0}^{K-1} \mathbb{E}\|\nabla F_{g,m'}(\mathbf{x}_{g,p,m',k'}) - \nabla F_{g,m'}(\mathbf{x})\|^2$$

$$+ 20\eta^2 pMK \sum_{p'=0}^{p-1} \sum_{m'=1}^{M} \sum_{k'=0}^{K-1} \mathbb{E}\|\nabla F_{g,m'}(\mathbf{x}_{g,p',m',k'}) - \nabla F_{g,m'}(\mathbf{x})\|^2$$

$$+ 20\eta^2 gPMK \sum_{g'=1}^{g-1} \sum_{p'=0}^{P-1} \sum_{m'=1}^{M} \sum_{k'=0}^{K-1} \mathbb{E}\|\nabla F_{g',m'}(\mathbf{x}_{g',p',m',k'}) - \nabla F_{g',m'}(\mathbf{x})\|^2$$

$$\leq 20L^2\eta^2 k \sum_{k'=0}^{k-1} \mathbb{E}\|\mathbf{x}_{g,p,m,k'} - \mathbf{x}\|^2 + 20L^2\eta^2 mK \sum_{m'=1}^{m-1} \sum_{k'=0}^{K-1} \mathbb{E}\|\mathbf{x}_{g,p',m',k'} - \mathbf{x}\|^2$$

$$+ 20L^2\eta^2 pMK \sum_{p'=0}^{p-1} \sum_{m'=1}^{M} \sum_{k'=0}^{K-1} \mathbb{E}\|\mathbf{x}_{g,p',m',k'} - \mathbf{x}\|^2$$

$$+ 20L^2\eta^2 gPMK \sum_{g'=1}^{g-1} \sum_{p'=0}^{P-1} \sum_{m'=1}^{M} \sum_{k'=0}^{K-1} \mathbb{E}\|\mathbf{x}_{g',p',m',k'} - \mathbf{x}\|^2$$

Bounding the third term in the left-hand inequality,

$$5\eta^2 \mathbb{E}\| \sum_{k'=0}^{k-1} \nabla F_{g,m}(\mathbf{x}) + \sum_{m'=1}^{m-1} \sum_{k'=0}^{K-1} \nabla F_{g,m'}(\mathbf{x}) + \sum_{p'=0}^{p-1} \sum_{m'=1}^{M} \sum_{k'=0}^{K-1} \nabla F_{g,m'}(\mathbf{x})$$

$$+ \sum_{g'=1}^{g-1} \sum_{p'=0}^{P-1} \sum_{m'=1}^{M} \sum_{k'=0}^{K-1} \nabla F_{g',m'}(\mathbf{x}) - \sum_{k'=0}^{k-1} \nabla F_g(\mathbf{x}) - \sum_{m'=1}^{m-1} \sum_{k'=0}^{K-1} \nabla F_g(\mathbf{x})$$

$$- \sum_{p'=0}^{p-1} \sum_{m'=1}^{M} \sum_{k'=0}^{K-1} \nabla F_g(\mathbf{x}) - \sum_{g'=1}^{g-1} \sum_{p'=0}^{P-1} \sum_{m'=1}^{M} \sum_{k'=0}^{K-1} \nabla F_{g'}(\mathbf{x}) \|^2$$

$$\leq 20\eta^2 k \sum_{k'=0}^{k-1} \mathbb{E}\|\nabla F_{g,m}(\mathbf{x}) - \nabla F_g(\mathbf{x})\|^2 + 20\eta^2 mK \sum_{m'=1}^{m-1} \sum_{k'=0}^{K-1} \mathbb{E}\|\nabla F_{g,m'}(\mathbf{x}) - \nabla F_g(\mathbf{x})\|^2$$

$$+ 20\eta^2 pMK \sum_{p'=0}^{p-1} \sum_{m'=1}^{M} \sum_{k'=0}^{K-1} \mathbb{E}\|\nabla F_{g,m'}(\mathbf{x}) - \nabla F_g(\mathbf{x})\|^2$$

$$+ 20\eta^2 gPMK \sum_{g'=1}^{g-1} \sum_{p'=0}^{P-1} \sum_{m'=1}^{M} \sum_{k'=0}^{K-1} \mathbb{E}\|\nabla F_{g',m'}(\mathbf{x}) - \nabla F_{g'}(\mathbf{x})\|^2$$

Bounding the fourth term in the left-hand inequality,

$$5\eta^2 \mathbb{E}\| \sum_{k'=0}^{k-1} \nabla F_g(\mathbf{x}) + \sum_{m'=1}^{m-1} \sum_{k'=0}^{K-1} \nabla F_g(\mathbf{x}) + \sum_{p'=0}^{p-1} \sum_{m'=1}^{M} \sum_{k'=0}^{K-1} \nabla F_g(\mathbf{x})$$

$$+ \sum_{g'=1}^{g-1} \sum_{p'=0}^{P-1} \sum_{m'=1}^{M} \sum_{k'=0}^{K-1} \nabla F_{g'}(\mathbf{x}) - \sum_{k'=0}^{k-1} \nabla F(\mathbf{x}) - \sum_{m'=1}^{m-1} \sum_{k'=0}^{K-1} \nabla F(\mathbf{x})$$

$$- \sum_{p'=0}^{p-1} \sum_{m'=1}^{M} \sum_{k'=0}^{K-1} \nabla F(\mathbf{x}) - \sum_{g'=1}^{g-1} \sum_{p'=0}^{P-1} \sum_{m'=1}^{M} \sum_{k'=0}^{K-1} \nabla F(\mathbf{x}) \|^2$$

$$\leq 20\eta^2 k \sum_{k'=0}^{k-1} \mathbb{E}\|\nabla F_g(\mathbf{x}) - \nabla F(\mathbf{x})\|^2 + 20\eta^2 mK \sum_{m'=1}^{m-1} \sum_{k'=0}^{K-1} \mathbb{E}\|\nabla F_g(\mathbf{x}) - \nabla F(\mathbf{x})\|^2$$

$$+ 20\eta^2 pMK \sum_{p'=0}^{p-1} \sum_{m'=1}^{M} \sum_{k'=0}^{K-1} \mathbb{E}\|\nabla F_g\left(\mathbf{x}\right) - \nabla F\left(\mathbf{x}\right)\|^2$$

$$+ 20\eta^2 gPMK \sum_{g'=1}^{g-1} \sum_{p'=0}^{P-1} \sum_{m'=1}^{M} \sum_{k'=0}^{K-1} \mathbb{E}\|\nabla F_{g'}\left(\mathbf{x}\right) - \nabla F\left(\mathbf{x}\right)\|^2$$

Bounding the fifth term in the left-hand inequality,

$$5\eta^2 \mathbb{E}\| \sum_{k'=0}^{k-1} \nabla F\left(\mathbf{x}\right) + \sum_{m'=1}^{m-1} \sum_{k'=0}^{K-1} \nabla F\left(\mathbf{x}\right) + \sum_{p'=0}^{p-1} \sum_{m'=1}^{M} \sum_{k'=0}^{K-1} \nabla F\left(\mathbf{x}\right) + \sum_{g'=1}^{g-1} \sum_{p'=0}^{P-1} \sum_{m'=1}^{M} \sum_{k'=0}^{K-1} \nabla F\left(\mathbf{x}\right)\|^2$$

$$\leq 20\eta^2 k \sum_{k'=0}^{k-1} \mathbb{E}\|\nabla F\left(\mathbf{x}\right)\|^2 + 20\eta^2 mK \sum_{p'=0}^{p-1} \sum_{m'=1}^{m-1} \sum_{k'=0}^{K-1} \mathbb{E}\|\nabla F\left(\mathbf{x}\right)\|^2$$

$$+ 20\eta^2 pMK \sum_{p'=0}^{p-1} \sum_{m'=1}^{M} \sum_{k'=0}^{K-1} \mathbb{E}\|\nabla F\left(\mathbf{x}\right)\|^2 + 20\eta^2 gPMK \sum_{g'=1}^{g-1} \sum_{p'=0}^{P-1} \sum_{m'=1}^{M} \sum_{k'=0}^{K-1} \mathbb{E}\|\nabla F\left(\mathbf{x}\right)\|^2$$

Substitute these terms into $E_r$,

$$E_r \leq 20\eta^2 \sum_{g,p,m,k} \left( k\sigma^2 + mK\sigma^2 + pMK\sigma^2 + gPMK\sigma^2 + L^2 k \sum_{k'=0}^{k-1} \mathbb{E}\|\mathbf{x}_{g,p,m,k'} - \mathbf{x}\|^2 \right.$$

$$+ L^2 mK \sum_{m'=1}^{m-1} \sum_{k'=0}^{K-1} \mathbb{E}\|\mathbf{x}_{g,p,m',k'} - \mathbf{x}\|^2 + L^2 pMK \sum_{p'=0}^{p-1} \sum_{m'=1}^{M} \sum_{k'=0}^{K-1} \mathbb{E}\|\mathbf{x}_{g,p',m',k'} - \mathbf{x}\|^2$$

$$+ L^2 gPMK \sum_{g'=1}^{g-1} \sum_{p'=0}^{P-1} \sum_{m'=1}^{M} \sum_{k'=0}^{K-1} \mathbb{E}\|\mathbf{x}_{g',p',m',k'} - \mathbf{x}\|^2 + k \sum_{k'=0}^{k-1} \mathbb{E}\|\nabla F_{g,m}(\mathbf{x}) - \nabla F_g(\mathbf{x})\|^2$$

$$+ mK \sum_{m'=1}^{m-1} \sum_{k'=0}^{K-1} \mathbb{E}\|\nabla F_{g,m'}(\mathbf{x}) - \nabla F_g(\mathbf{x})\|^2 + k \sum_{k'=0}^{k-1} \mathbb{E}\|\nabla F_g(\mathbf{x}) - \nabla F(\mathbf{x})\|^2$$

$$+ mK \sum_{m'=1}^{m-1} \sum_{k'=0}^{K-1} \mathbb{E}\|\nabla F_g(\mathbf{x}) - \nabla F(\mathbf{x})\|^2 + pMK \sum_{p'=0}^{p-1} \sum_{m'=1}^{M} \sum_{k'=0}^{K-1} \mathbb{E}\|\nabla F_g(\mathbf{x}) - \nabla F(\mathbf{x})\|^2$$

$$+ gPMK \sum_{g'=1}^{g-1} \sum_{p'=0}^{P-1} \sum_{m'=1}^{M} \sum_{k'=0}^{K-1} \mathbb{E}\|\nabla F_{g'}(\mathbf{x}) - \nabla F(\mathbf{x})\|^2 + k \sum_{k'=0}^{k-1} \mathbb{E}\|\nabla F(\mathbf{x})\|^2$$

$$+ mK \sum_{m'=1}^{m-1} \sum_{k'=0}^{K-1} \mathbb{E}\|\nabla F(\mathbf{x})\|^2 + pMK \sum_{p'=0}^{p-1} \sum_{m'=1}^{M} \sum_{k'=0}^{K-1} \mathbb{E}\|\nabla F(\mathbf{x})\|^2$$

$$+ gpMK \sum_{g'=1}^{g-1} \sum_{p'=0}^{P-1} \sum_{m'=1}^{M} \sum_{k'=0}^{K-1} \mathbb{E}\|\nabla F(\mathbf{x})\|^2 \Bigg)$$

$$\leq 20\eta^2 GPMK^2\sigma^2 + 20\eta^2 GPM^2K^2\sigma^2 + 20\eta^2 GP^2M^2K^2\sigma^2 + 20\eta^2 G^2P^2M^2K^2\sigma^2$$

$$+ 20\eta^2 GPMK^3\zeta^2 + 20\eta^2 GPM^3K^3\zeta^2 + 20\eta^2 GPMK^3\delta^2$$

$$+ 20\eta^2 GPM^3K^3\delta^2 + 20\eta^2 GP^3M^3K^3\delta^2 + 20\eta^2 G^3P^3M^3K^3\delta^2$$

$$+ 20\eta^2 GPMK^3\|\nabla F\left(\mathbf{x}\right)\|^2 + 20\eta^2 GPM^3K^3\|\nabla F\left(\mathbf{x}\right)\|^2 + 20\eta^2 GP^3M^3K^3\|\nabla F\left(\mathbf{x}\right)\|^2$$

$$+ 20\eta^2 G^3P^3M^3K^3\|\nabla F\left(\mathbf{x}\right)\|^2 + 20L^2 G^2P^2M^2K^2\eta^2 E_r$$

Let $c_1 = \frac{1}{1-20L^2G^2P^2M^2K^2\eta^2}$, we have

$$E_r \leq 80c_1\eta^2 G^2P^2M^2K^2\sigma^2 + 40c_1\eta^2 GPM^3K^3\zeta^2 + 80c_1\eta^2 G^3P^3M^3K^3\delta^2$$

$$+ 80c_1\eta^2 G^3 P^3 M^3 K^3 \|\nabla F(\mathbf{x})\|^2$$

After substituting $E_r$ into $\mathbb{E}[F(\mathbf{x}^{(r+1)}) - F(\mathbf{x}^{(r)})]$, we can obtain

$$\mathbb{E}[F(\mathbf{x}^{(r+1)}) - F(\mathbf{x}^{(r)})]$$

$$\leq -\eta GPMK(\frac{1}{2} - 40c_1\eta^2 L^2 G^2 P^2 M^2 K^2)\mathbb{E}[\|\nabla F(\mathbf{x}^{(r)})\|^2] + \eta^2 LGPMK\sigma^2$$

$$+ 40c_1\eta^3 L^2 GP^2 M^2 K^2\sigma^2 + 20c_1\eta^3 L^2 GPM^3 K^3\zeta^2 + 40c_1\eta^3 L^2 G^3 P^3 M^3 K^3\delta^2.$$

Let $c_2 = \frac{1}{\frac{1}{2} - 40c_1\eta^2 L^2 G^2 P^2 M^2 K^2}$, then

$$\frac{1}{R}\sum_{r=0}^{R-1}\mathbb{E}[\|\nabla F(\mathbf{x}^{(r)})\|^2] \leq \frac{c_2\left(F(\mathbf{x}^{(0)}) - F(\mathbf{x}^{(R)})\right)}{\eta RGPMK} + c_2\eta L\sigma^2 + 40c_1 c_2\eta^2 L^2 GPMK\sigma^2$$

$$+ 20c_1 c_2\eta^2 L^2 M^2 K^2\zeta^2 + 40c_1 c_2\eta^2 L^2 G^2 P^2 M^2 K^2\delta^2.$$

Let $\tilde{\eta} = \eta GPMK$, and $\tilde{\eta} \leq \frac{1}{10L}$, we have

$$\mathbb{E}[\|\nabla F(\bar{x}^{(R)})\|^2] \lesssim \frac{A}{\tilde{\eta}R} + \frac{L\tilde{\eta}\sigma^2}{GPMK} + \frac{L^2\tilde{\eta}^2\sigma^2}{GPMK} + \frac{L^2\tilde{\eta}^2\zeta^2}{G^2 P^2} + L^2\tilde{\eta}^2\delta^2$$

$$\mathbb{E}[\|\nabla F(\bar{x}^{(R)})\|^2] = \mathcal{O}\left(\frac{LA}{R} + \frac{(L\sigma^2 A)^{1/2}}{\sqrt{GPMKR}} + \frac{(L^2 A^2\sigma^2)^{1/3}}{(GPMKR^2)^{1/3}} + \frac{(L^2 A^2\zeta^2)^{1/3}}{(G^2 P^2 R^2)^{1/3}}\right.$$

$$\left.+ \frac{(L^2 A^2\delta^2)^{1/3}}{(R^2)^{1/3}}\right)$$

where $\bar{x}^R$ is defined as a model uniformly sampled from the $x^{(0)}, \ldots, x^{(R-1)}$ of previous iterations, and $\lesssim$ hides universal constants.

## E  RANDOM-ORDER EXTENSION OF THEOREM 1

In the following, we consider the full participation setting under a random execution order. The overall proof strategy parallels that of Theorem 1, and in particular shares the same auxiliary result (Lemma 1).

The main difference between the fixed-order and random-order settings lies in the treatment of the heterogeneity terms $\sum_{m'=1}^{m-1}\mathbb{E}\|\nabla F_{g,m'}(\mathbf{x}) - \nabla F_g(\mathbf{x})\|^2$ and $\sum_{g'=1}^{g-1}\mathbb{E}\|\nabla F_{g,m'}(\mathbf{x}) - \nabla F_g(\mathbf{x})\|^2$. The distinction arises because under a random order, the expectation no longer scales with this prefix length, since the execution position of each client is itself random. The constants differ significantly due to the random-order execution, a random shuffle of the participating clients at the beginning of each round. A comprehensive comparison of the resulting convergence rates under both fixed-order ($F$) and random-shuffle ($S$) settings is summarized in Table 5. As observed in the bounds, the random shuffling mechanism actively mitigates the accumulation of topology-specific bias inherent in sequential updates. Mathematically, this is reflected by the emergence of additional scaling factors, $G$ and $M$, in the denominators of the heterogeneity and variance terms for configurations involving ring topologies. Because these error terms are further suppressed by $G$ and $M$, the random-order execution yields a strictly tighter convergence upper bound compared to the fixed-order setting, demonstrating its superior robustness against data heterogeneity.

Since Lemma 1 remains applicable, the proof proceeds by first bounding the client drift under different topologies, after which we derive the corresponding recursion. Notably, the per-round recursion retains the same structural form as in Theorem 1. For completeness, we provide the full proof below.

### E.1  RANDOM-ORDER EXTENSION OF THEOREM 1 FOR THE STAR-STAR CASE

For Star-Star,

$$\mathbf{x}_{g,p,m,k} - \mathbf{x} = \sum_{k'=0}^{k-1}\mathbf{g}_{g,p,m,k'} + \sum_{p'=0}^{p-1}\frac{1}{M}\sum_{m'=1}^{M}\sum_{k'=0}^{K-1}\mathbf{g}_{g,p',m',k'}.$$

Table 5: Comparison between Fixed-Order and Random-Shuffle convergence bounds.

| Topology | Convergence rate |
|---|---|
| **Star-Star**($F$) | $\frac{LA}{R} + \frac{(L\sigma^2 A)^{\frac{1}{2}}}{\sqrt{GPMKR}} + \frac{(L^2 A^2 \sigma^2)^{\frac{1}{3}}}{(P^2 KR^2)^{\frac{1}{3}}} + \frac{(L^2 A^2 \sigma^2)^{\frac{1}{3}}}{(PMKR^2)^{\frac{1}{3}}} + \frac{(L^2 A^2 \zeta^2)^{\frac{1}{3}}}{(P^2 R^2)^{\frac{1}{3}}} + \frac{(L^2 A^2 \delta^2)^{\frac{1}{3}}}{(R^2)^{\frac{1}{3}}}$ |
| **Star-Star**($S$) | $\frac{LA}{R} + \frac{(L\sigma^2 A)^{\frac{1}{2}}}{\sqrt{GPMKR}} + \frac{(L^2 A^2 \sigma^2)^{\frac{1}{3}}}{(P^2 KR^2)^{\frac{1}{3}}} + \frac{(L^2 A^2 \sigma^2)^{\frac{1}{3}}}{(PMKR^2)^{\frac{1}{3}}} + \frac{(L^2 A^2 \zeta^2)^{\frac{1}{3}}}{(P^2 R^2)^{\frac{1}{3}}} + \frac{(L^2 A^2 \delta^2)^{\frac{1}{3}}}{(R^2)^{\frac{1}{3}}}$ |
| **Star-Ring** ($F$) | $\frac{LA}{R} + \frac{(L\sigma^2 A)^{\frac{1}{2}}}{\sqrt{GPMKR}} + \frac{(L^2 A^2 \sigma^2)^{\frac{1}{3}}}{(PKR^2)^{\frac{1}{3}}} + \frac{(L^2 A^2 \zeta^2)^{\frac{1}{3}}}{(P^2 MR^2)^{\frac{1}{3}}} + \frac{(L^2 A^2 \delta^2)^{\frac{1}{3}}}{(R^2)^{\frac{1}{3}}}$ |
| **Star-Ring**($S$) | $\frac{LA}{R} + \frac{(L\sigma^2 A)^{\frac{1}{2}}}{\sqrt{GPMKR}} + \frac{(L^2 A^2 \sigma^2)^{\frac{1}{3}}}{(P{\color{red}M}KR^2)^{\frac{1}{3}}} + \frac{(L^2 A^2 \zeta^2)^{\frac{1}{3}}}{(P^2 MR^2)^{\frac{1}{3}}} + \frac{(L^2 A^2 \delta^2)^{\frac{1}{3}}}{(R^2)^{\frac{1}{3}}}$ |
| **Ring-Star** ($F$) | $\frac{LA}{R} + \frac{(L\sigma^2 A)^{\frac{1}{2}}}{\sqrt{GPMKR}} + \frac{(L^2 A^2 \sigma^2)^{\frac{1}{3}}}{(G^2 P^2 KR^2)^{\frac{1}{3}}} + \frac{(L^2 A^2 \sigma^2)^{\frac{1}{3}}}{(GPMKR^2)^{\frac{1}{3}}} + \frac{(L^2 A^2 \zeta^2)^{\frac{1}{3}}}{(G^2 P^2 R^2)^{\frac{1}{3}}} + \frac{(L^2 A^2 \delta^2)^{\frac{1}{3}}}{(R^2)^{\frac{1}{3}}}$ |
| **Ring-Star**($S$) | $\frac{LA}{R} + \frac{(L\sigma^2 A)^{\frac{1}{2}}}{\sqrt{GPMKR}} + \frac{(L^2 A^2 \sigma^2)^{\frac{1}{3}}}{(G^2 P^2 KR^2)^{\frac{1}{3}}} + \frac{(L^2 A^2 \sigma^2)^{\frac{1}{3}}}{(GPMKR^2)^{\frac{1}{3}}} + \frac{(L^2 A^2 \zeta^2)^{\frac{1}{3}}}{(G^2 P^2 R^2)^{\frac{1}{3}}} + \frac{(L^2 A^2 \delta^2)^{\frac{1}{3}}}{({\color{red}G}R^2)^{\frac{1}{3}}}$ |
| **Ring-Ring** ($F$) | $\frac{LA}{R} + \frac{(L\sigma^2 A)^{\frac{1}{2}}}{\sqrt{GPMKR}} + \frac{(L^2 A^2 \sigma^2)^{\frac{1}{3}}}{(GPMKR^2)^{\frac{1}{3}}} + \frac{(L^2 A^2 \zeta^2)^{\frac{1}{3}}}{(G^2 P^2 R^2)^{\frac{1}{3}}} + \frac{(L^2 A^2 \delta^2)^{\frac{1}{3}}}{(R^2)^{\frac{1}{3}}}$ |
| **Ring-Ring**($S$) | $\frac{LA}{R} + \frac{(L\sigma^2 A)^{\frac{1}{2}}}{\sqrt{GPMKR}} + \frac{(L^2 A^2 \sigma^2)^{\frac{1}{3}}}{(GPMKR^2)^{\frac{1}{3}}} + \frac{(L^2 A^2 \zeta^2)^{\frac{1}{3}}}{(G^2 P^2 {\color{red}M}R^2)^{\frac{1}{3}}} + \frac{(L^2 A^2 \delta^2)^{\frac{1}{3}}}{({\color{red}G}R^2)^{\frac{1}{3}}}$ |

To bound $E_r$, we first bound $\mathbb{E}\|\mathbf{x}_{g,p,m,k} - \mathbf{x}\|^2$.

$$\mathbb{E}\|\mathbf{x}_{g,p,m,k} - \mathbf{x}\|^2$$

$$= \eta^2 \mathbb{E}\| \sum_{k'=0}^{k-1} \mathbf{g}_{g,p,m,k'} + \sum_{p'=0}^{p-1} \frac{1}{M} \sum_{m'=1}^{M} \sum_{k'=0}^{K-1} \mathbf{g}_{g,p',m',k'} \|^2$$

$$\leq 5\eta^2 \mathbb{E}\| \sum_{k'=0}^{k-1} \mathbf{g}_{g,p,m,k'} + \sum_{p'=0}^{p-1} \frac{1}{M} \sum_{m'=1}^{M} \sum_{k'=0}^{K-1} \mathbf{g}_{g,p',m',k'} - \sum_{k'=0}^{k-1} \nabla F_{g,m}\left(\mathbf{x}_{g,p,m,k'}\right)$$

$$- \sum_{p'=0}^{p-1} \frac{1}{M} \sum_{m'=1}^{M} \sum_{k'=0}^{K-1} \nabla F_{g,m'}\left(\mathbf{x}_{g,p',m',k'}\right) \|^2$$

$$+ 5\eta^2 \mathbb{E}\| \sum_{k'=0}^{k-1} \nabla F_{g,m}\left(\mathbf{x}_{g,p,m,k'}\right) + \sum_{p'=0}^{p-1} \frac{1}{M} \sum_{m'=1}^{M} \sum_{k'=0}^{K-1} \nabla F_{g,m'}\left(\mathbf{x}_{g,p',m',k'}\right) - \sum_{k'=0}^{k-1} \nabla F_{g,m}\left(\mathbf{x}\right)$$

$$- \sum_{p'=0}^{p-1} \frac{1}{M} \sum_{m'=1}^{M} \sum_{k'=0}^{K-1} \nabla F_{g,m'}\left(\mathbf{x}\right) \|^2$$

$$+ 5\eta^2 \mathbb{E}\| \sum_{k'=0}^{k-1} \nabla F_{g,m}\left(\mathbf{x}\right) + \sum_{p'=0}^{p-1} \frac{1}{M} \sum_{m'=1}^{M} \sum_{k'=0}^{K-1} \nabla F_{g,m'}\left(\mathbf{x}\right) - \sum_{k'=0}^{k-1} \nabla F_g\left(\mathbf{x}\right)$$

$$- \sum_{p'=0}^{p-1} \frac{1}{M} \sum_{m'=1}^{M} \sum_{k'=0}^{K-1} \nabla F_g\left(\mathbf{x}\right) \|^2$$

$$+ 5\eta^2 \mathbb{E}\| \sum_{k'=0}^{k-1} \nabla F_g\left(\mathbf{x}\right) + \sum_{p'=0}^{p-1} \frac{1}{M} \sum_{m'=1}^{M} \sum_{k'=0}^{K-1} \nabla F_g\left(\mathbf{x}\right) - \sum_{k'=0}^{k-1} \nabla F\left(\mathbf{x}\right) - \sum_{p'=0}^{p-1} \frac{1}{M} \sum_{m'=1}^{M} \sum_{k'=0}^{K-1} \nabla F\left(\mathbf{x}\right) \|^2$$

$$+ 5\eta^2 \mathbb{E}\| \sum_{k'=0}^{k-1} \nabla F(\mathbf{x}) + \sum_{p'=0}^{p-1} \frac{1}{M} \sum_{m'=1}^{M} \sum_{k'=0}^{K-1} \nabla F(\mathbf{x}) \|^2,$$

where we apply the Jensen's Inequality for the first inequality. The term $\mathbb{E}\|\mathbf{x}_{g,p,m,k} - \mathbf{x}\|^2$ is decomposed into 5 components (e.g., local SGD noise, intra-group drift, inter-group drift, and global model discrepancy).

Bounding the first term in the left-hand inequality,

$$5\eta^2 \mathbb{E}\| \sum_{k'=0}^{k-1} \mathbf{g}_{g,p,m,k'} + \sum_{p'=0}^{p-1} \frac{1}{M} \sum_{m'=1}^{M} \sum_{k'=0}^{K-1} \mathbf{g}_{g,p',m',k'} - \sum_{k'=0}^{k-1} \nabla F_{g,m}(\mathbf{x}_{g,p,m,k'})$$

$$- \sum_{p'=0}^{p-1} \frac{1}{M} \sum_{m'=1}^{M} \sum_{k'=0}^{K-1} \nabla F_{g,m'}(\mathbf{x}_{g,p',m',k'}) \|^2$$

$$\leq 10\eta^2 \sum_{k'=0}^{k-1} \mathbb{E}\|\mathbf{g}_{g,p,m,k'} - \nabla F_{g,m}(\mathbf{x}_{g,p,m,k'})\|^2$$

$$+ 10\eta^2 \sum_{p'=0}^{p-1} \frac{1}{M^2} \sum_{m'=1}^{M} \sum_{k'=0}^{K-1} \mathbb{E}\|\mathbf{g}_{g,p',m',k'} - \nabla F_{g,m'}(\mathbf{x}_{g,p',m',k'}) \|^2$$

$$\leq 10\eta^2 k \sigma^2 + 10\eta^2 \frac{pK}{M} \sigma^2$$

Bounding the second term in the left-hand inequality,

$$5\eta^2 \mathbb{E}\| \sum_{k'=0}^{k-1} \nabla F_{g,m}(\mathbf{x}_{g,p,m,k'}) + \sum_{p'=0}^{p-1} \frac{1}{M} \sum_{m'=1}^{M} \sum_{k'=0}^{K-1} \nabla F_{g,m'}(\mathbf{x}_{g,p',m',k'})$$

$$- \sum_{k'=0}^{k-1} \nabla F_{g,m}(\mathbf{x}) - \sum_{p'=0}^{p-1} \frac{1}{M} \sum_{m'=1}^{M} \sum_{k'=0}^{K-1} \nabla F_{g,m'}(\mathbf{x}) \|^2$$

$$\leq 10\eta^2 k \sum_{k'=0}^{k-1} \mathbb{E}\|\nabla F_{g,m}(\mathbf{x}_{g,p,m,k'}) - \nabla F_{g,m}(\mathbf{x})\|^2$$

$$+ 10\eta^2 pMK \sum_{p'=0}^{p-1} \frac{1}{M^2} \sum_{m'=1}^{M} \sum_{k'=0}^{K-1} \mathbb{E}\|\nabla F_{g,m'}(\mathbf{x}_{g,p',m',k'}) - \nabla F_{g,m'}(\mathbf{x}) \|^2$$

$$\leq 10L^2\eta^2 k \sum_{k'=0}^{k-1} \mathbb{E}\|\mathbf{x}_{g,p,m,k'} - \mathbf{x}\|^2 + 10L^2\eta^2 \frac{pK}{M} \sum_{p'=0}^{p-1} \sum_{m'=1}^{M} \sum_{k'=0}^{K-1} \mathbb{E}\|\mathbf{x}_{g,p',m',k'} - \mathbf{x}\|^2$$

Bounding the third term in the left-hand inequality,

$$5\eta^2 \mathbb{E}\| \sum_{p'=0}^{p-1} \frac{1}{M} \sum_{m'=1}^{M} \sum_{k'=0}^{K-1} \nabla F_{g,m'}(\mathbf{x}) - \sum_{p'=0}^{p-1} \frac{1}{M} \sum_{m'=1}^{M} \sum_{k'=0}^{K-1} \nabla F_g(\mathbf{x})$$

$$+ \sum_{k'=0}^{k-1} \nabla F_{g,m}(\mathbf{x}) - \sum_{k'=0}^{k-1} \nabla F_g(\mathbf{x}) \|^2 \leq 10\eta^2 k \sum_{k'=0}^{k-1} \mathbb{E}\|\nabla F_{g,m}(\mathbf{x}) - \nabla F_g(\mathbf{x})\|^2$$

The group objective function is defined as the aggregation of local functions, i.e., $F_g = \frac{1}{M}\sum_{m'=1}^{M} F_{g,m'}$. Consequently, the summation of the deviations of local gradients from the group gradient vanishes.

Bounding the fourth term in the left-hand inequality,

$$5\eta^2\mathbb{E}\|\sum_{k'=0}^{k-1}\nabla F_g\left(\mathbf{x}\right)+\sum_{p'=0}^{p-1}\frac{1}{M}\sum_{m'=1}^{M}\sum_{k'=0}^{K-1}\nabla F_g\left(\mathbf{x}\right)-\sum_{k'=0}^{k-1}\nabla F\left(\mathbf{x}\right)-\sum_{p'=0}^{p-1}\frac{1}{M}\sum_{m'=1}^{M}\sum_{k'=0}^{K-1}\nabla F\left(\mathbf{x}\right)\|^2$$

$$\leq 10\eta^2 k\sum_{k'=0}^{k-1}\mathbb{E}\|\nabla F_g\left(\mathbf{x}\right)-\nabla F\left(\mathbf{x}\right)\|^2+10\eta^2\frac{pK}{M}\sum_{p'=0}^{p-1}\sum_{m'=1}^{M}\sum_{k'=0}^{K-1}\mathbb{E}\|\nabla F_g\left(\mathbf{x}\right)-\nabla F\left(\mathbf{x}\right)\|^2$$

Bounding the fifth term in the left-hand inequality,

$$5\eta^2\mathbb{E}\|\sum_{k'=0}^{k-1}\nabla F\left(\mathbf{x}\right)+\sum_{p'=0}^{p-1}\frac{1}{M}\sum_{m'=1}^{M}\sum_{k'=0}^{K-1}\nabla F\left(\mathbf{x}\right)\|^2$$

$$\leq 10\eta^2 k\sum_{k'=0}^{k-1}\mathbb{E}\|\nabla F\left(\mathbf{x}\right)\|^2+10\eta^2 pMK\sum_{p'=0}^{p-1}\frac{1}{M^2}\sum_{m'=1}^{M}\sum_{k'=0}^{K-1}\mathbb{E}\|\nabla F\left(\mathbf{x}\right)\|^2$$

Substitute these terms into $E_r$,

$$E_r\leq 10\eta^2\sum_{g,p,m,k}\left(k\sigma^2+\frac{pK}{M}\sigma^2+L^2 k\sum_{k'=0}^{k-1}\mathbb{E}\|\mathbf{x}_{g,p,m,k'}-\mathbf{x}\|^2\right.$$

$$+L^2\frac{pK}{M}\sum_{p'=0}^{p-1}\sum_{m'=1}^{M}\sum_{k'=0}^{K-1}\mathbb{E}\|\mathbf{x}_{g,p',m',k'}-\mathbf{x}\|^2+k\sum_{k'=0}^{k-1}\mathbb{E}\|\nabla F_{g,m}\left(\mathbf{x}\right)-\nabla F_g\left(\mathbf{x}\right)\|^2$$

$$k\sum_{k'=0}^{k-1}\mathbb{E}\|\nabla F_g\left(\mathbf{x}\right)-\nabla F\left(\mathbf{x}\right)\|^2+\frac{pK}{M}\sum_{p'=0}^{p-1}\sum_{m'=1}^{M}\sum_{k'=0}^{K-1}\mathbb{E}\|\nabla F_g\left(\mathbf{x}\right)-\nabla F\left(\mathbf{x}\right)\|^2$$

$$\left.k\sum_{k'=0}^{k-1}\mathbb{E}\|\nabla F\left(\mathbf{x}\right)\|^2+pMK\sum_{p'=0}^{p-1}\frac{1}{M^2}\sum_{m'=1}^{M}\sum_{k'=0}^{K-1}\mathbb{E}\|\nabla F\left(\mathbf{x}\right)\|^2\right)$$

Let $c_1=\frac{1}{1-10L^2P^2K^2\eta^2}$, we have

$$E_r\leq c_1 10\eta^2 GPMK^2\sigma^2+c_1 10\eta^2 GP^2K^2\sigma^2+c_1 10\eta^2 GPMK^3\zeta^2$$
$$+c_1 10\eta^2 GPMK^3\delta^2+c_1 10\eta^2 GP^3MK^3\delta^2$$
$$+c_1 10\eta^2 GPMK^3\|\nabla F\left(\mathbf{x}\right)\|^2+c_1 10\eta^2 GP^3MK^3\|\nabla F\left(\mathbf{x}\right)\|^2$$
$$\leq 10c_1\eta^2 GPMK^2\sigma^2+10c_1\eta^2 GP^2K^2\sigma^2+10c_1\eta^2 GPMK^3\zeta^2+20c_1\eta^2 GP^3MK^3\delta^2$$
$$+20c_1\eta^2 GP^3MK^3\|\nabla F\left(\mathbf{x}\right)\|^2$$

After substituting $E_r$ into $\mathbb{E}[F(\mathbf{x}^{(r+1)})-F(\mathbf{x}^{(r)})]$, we can obtain

$$\mathbb{E}[F(\mathbf{x}^{(r+1)})-F(\mathbf{x}^{(r)})]$$
$$\leq-\eta PK(\frac{1}{2}-10c_1\eta^2 L^2 P^2 K^2)\mathbb{E}[\|\nabla F(\mathbf{x}^{(r)})\|^2]+\eta^2 L\frac{PK}{GM}\sigma^2$$
$$+5c_1\eta^3 L^2 PK^2\sigma^2+5c_1\eta^3 L^2\frac{P^2K^2}{M}\sigma^2+5c_1\eta^3 L^2 PK^3\zeta^2+10c_1\eta^3 L^2 P^3K^3\delta^2.$$

Let $c_2=\frac{1}{\frac{1}{2}-10c_1\eta^2 L^2 P^2 K^2}$, then

$$\frac{1}{R}\sum_{r=0}^{R-1}\mathbb{E}[\|\nabla F(\mathbf{x}^{(r)})\|^2]\leq\frac{c_2\left(F(\mathbf{x}^{(0)})-F(\mathbf{x}^{(R)})\right)}{\eta RPK}+c_2\eta L\frac{1}{GM}\sigma^2+5c_1 c_2\eta^2 L^2 K\sigma^2$$

$$+ 5c_1c_2\eta^2 L^2 \frac{PK}{M}\sigma^2 + 5c_1c_2\eta^2 L^2 K^2\zeta^2 + 10c_1c_2\eta^2 L^2 P^2 K^2\delta^2.$$

Let $\tilde{\eta} = \eta PK$, and $\tilde{\eta} \leq \frac{1}{\sqrt{30L}}$, we have

$$\mathbb{E}[\|\nabla F(\bar{x}^{(R)})\|^2] \lesssim \frac{A}{\tilde{\eta}R} + \frac{L\tilde{\eta}\sigma^2}{GPMK} + \frac{L^2\tilde{\eta}^2\sigma^2}{P^2K} + \frac{L^2\tilde{\eta}^2\sigma^2}{PMK} + \frac{L^2\tilde{\eta}^2\zeta^2}{P^2} + L^2\tilde{\eta}^2\delta^2$$

$$\mathbb{E}[\|\nabla F(\bar{x}^{(R)})\|^2] = \mathcal{O}\left(\frac{LA}{R} + \frac{(L\sigma^2 A)^{1/2}}{\sqrt{GPMKR}} + \frac{(L^2 A^2\sigma^2)^{1/3}}{(P^2KR^2)^{1/3}} + \frac{(L^2 A^2\sigma^2)^{1/3}}{(PMKR^2)^{1/3}}\right.$$
$$\left. + \frac{(L^2 A^2\zeta^2)^{1/3}}{(P^2R^2)^{1/3}} + \frac{(L^2 A^2\delta^2)^{1/3}}{(R^2)^{1/3}}\right)$$

where $\bar{x}^R$ is defined as a model uniformly sampled from the $x^{(0)}, \ldots, x^{(R-1)}$ of previous iterations, and $\lesssim$ hides universal constants.

## E.2 RANDOM-ORDER EXTENSION OF THEOREM 1 FOR THE STAR-RING CASE

For Star-Ring,

$$\mathbf{x}_{g,p,m,k} - \mathbf{x} = \sum_{k'=0}^{k-1} \mathbf{g}_{g,p,m,k'} + \sum_{m'=1}^{m-1}\sum_{k'=0}^{K-1} \mathbf{g}_{g,p,m',k'} + \sum_{p'=0}^{p-1}\sum_{m'=1}^{M}\sum_{k=0}^{K-1} \mathbf{g}_{g,p',m',k'}.$$

To bound $E_r$, we first bound $\mathbb{E}\|\mathbf{x}_{g,p,m,k} - \mathbf{x}\|^2$.

$$\mathbb{E}\|\mathbf{x}_{g,p,m,k} - \mathbf{x}\|^2$$
$$= \eta^2\mathbb{E}\| \sum_{k'=0}^{k-1} \mathbf{g}_{g,p,m,k'} + \sum_{m'=1}^{m-1}\sum_{k'=0}^{K-1} \mathbf{g}_{g,p,m',k'} + \sum_{p'=0}^{p-1}\sum_{m'=1}^{M}\sum_{k=0}^{K-1} \mathbf{g}_{g,p',m',k'}\|^2$$
$$\leq 5\eta^2\mathbb{E}\| \sum_{k'=0}^{k-1} \mathbf{g}_{g,p,m,k'} + \sum_{m'=1}^{m-1}\sum_{k'=0}^{K-1} \mathbf{g}_{g,p,m',k'} + \sum_{p'=0}^{p-1}\sum_{m'=1}^{M}\sum_{k=0}^{K-1} \mathbf{g}_{g,p',m',k'} - \sum_{k'=0}^{k-1} \nabla F_{g,m}\left(\mathbf{x}_{g,p,m,k'}\right)$$
$$- \sum_{m'=1}^{m-1}\sum_{k'=0}^{K-1} \nabla F_{g,m'}\left(\mathbf{x}_{g,p,m',k'}\right) - \sum_{p'=0}^{p-1}\sum_{m'=1}^{M}\sum_{k'=0}^{K-1} \nabla F_{g,m'}\left(\mathbf{x}_{g,p',m',k'}\right)\|^2$$
$$+ 5\eta^2\mathbb{E}\| \sum_{k'=0}^{k-1} \nabla F_{g,m}\left(\mathbf{x}_{g,p,m,k'}\right) - \sum_{k'=0}^{k-1} \nabla F_{g,m}\left(\mathbf{x}\right) + \sum_{m'=1}^{m-1}\sum_{k'=0}^{K-1} \nabla F_{g,m'}\left(\mathbf{x}_{g,p,m',k'}\right)$$
$$- \sum_{m'=1}^{m-1}\sum_{k'=0}^{K-1} \nabla F_{g,m'}\left(\mathbf{x}\right) + \sum_{p'=0}^{p-1}\sum_{m'=1}^{M}\sum_{k'=0}^{K-1} \nabla F_{g,m'}\left(\mathbf{x}_{g,p',m',k'}\right) - \sum_{p'=0}^{p-1}\sum_{m'=1}^{M}\sum_{k'=0}^{K-1} \nabla F_{g,m'}\left(\mathbf{x}\right)\|^2$$
$$+ 5\eta^2\mathbb{E}\| \sum_{k'=0}^{k-1} \nabla F_{g,m}\left(\mathbf{x}\right) + \sum_{m'=1}^{m-1}\sum_{k'=0}^{K-1} \nabla F_{g,m'}\left(\mathbf{x}\right) + \sum_{p'=0}^{p-1}\sum_{m'=1}^{M}\sum_{k'=0}^{K-1} \nabla F_{g,m'}\left(\mathbf{x}\right) - \sum_{k'=0}^{k-1} \nabla F_g\left(\mathbf{x}\right)$$
$$- \sum_{m'=1}^{m-1}\sum_{k'=0}^{K-1} \nabla F_g\left(\mathbf{x}\right) - \sum_{p'=0}^{p-1}\sum_{m'=1}^{M}\sum_{k'=0}^{K-1} \nabla F_g\left(\mathbf{x}\right)\|^2$$
$$+ 5\eta^2\mathbb{E}\| \sum_{k'=0}^{k-1} \nabla F_g\left(\mathbf{x}\right) + \sum_{m'=1}^{m-1}\sum_{k'=0}^{K-1} \nabla F_g\left(\mathbf{x}\right) + \sum_{p'=0}^{p-1}\sum_{m'=1}^{M}\sum_{k'=0}^{K-1} \nabla F_g\left(\mathbf{x}\right) - \sum_{k'=0}^{k-1} \nabla F\left(\mathbf{x}\right)$$
$$- \sum_{m'=1}^{m-1}\sum_{k'=0}^{K-1} \nabla F\left(\mathbf{x}\right) - \sum_{p'=0}^{p-1}\sum_{m'=1}^{M}\sum_{k'=0}^{K-1} \nabla F\left(\mathbf{x}\right)\|^2$$
$$+ 5\eta^2\mathbb{E}\| \sum_{k'=0}^{k-1} \nabla F\left(\mathbf{x}\right) + \sum_{m'=1}^{m-1}\sum_{k'=0}^{K-1} \nabla F\left(\mathbf{x}\right) + \sum_{p'=0}^{p-1}\sum_{m'=1}^{M}\sum_{k'=0}^{K-1} \nabla F\left(\mathbf{x}\right)\|^2$$

Bounding the first term in the left-hand inequality,

$$5\eta^2\mathbb{E}\|\sum_{k'=0}^{k-1}\mathbf{g}_{g,p,m,k'}+\sum_{m'=1}^{m-1}\sum_{k'=0}^{K-1}\mathbf{g}_{g,p,m',k'}+\sum_{p'=0}^{p-1}\sum_{m'=1}^{M}\sum_{k=0}^{K-1}\mathbf{g}_{g,p',m',k'}-\sum_{k'=0}^{k-1}\nabla F_{g,m}\left(\mathbf{x}_{g,p,m,k'}\right)$$

$$-\sum_{m'=1}^{m-1}\sum_{k'=0}^{K-1}\nabla F_{g,m'}\left(\mathbf{x}_{g,p,m',k'}\right)-\sum_{p'=0}^{p-1}\sum_{m'=1}^{M}\sum_{k'=0}^{K-1}\nabla F_{g,m'}\left(\mathbf{x}_{g,p',m',k'}\right)\|^2$$

$$\leq 15\eta^2\sum_{k'=0}^{k-1}\mathbb{E}\|\mathbf{g}_{g,p,m,k'}-\nabla F_{g,m}\left(\mathbf{x}_{g,p,m,k'}\right)\|^2$$

$$+15\eta^2\sum_{m'=1}^{m-1}\sum_{k'=0}^{K-1}\mathbb{E}\|\mathbf{g}_{g,p,m',k'}-\nabla F_{g,m'}\left(\mathbf{x}_{g,p,m',k'}\right)\|^2$$

$$+15\eta^2\sum_{p'=0}^{p-1}\sum_{m'=1}^{M}\sum_{k'=0}^{K-1}\mathbb{E}\|\mathbf{g}_{g,p',m',k'}-\nabla F_{g,m'}\left(\mathbf{x}_{g,p',m',k'}\right)\|^2$$

$$\leq 15\eta^2 k\sigma^2+15\eta^2 mK\sigma^2+15\eta^2 pMK\sigma^2$$

Bounding the second term in the left-hand inequality,

$$5\eta^2\mathbb{E}\|\sum_{k'=0}^{k-1}\nabla F_{g,m}\left(\mathbf{x}_{g,p,m,k'}\right)-\sum_{k'=0}^{k-1}\nabla F_{g,m}\left(\mathbf{x}\right)+\sum_{m'=1}^{m-1}\sum_{k'=0}^{K-1}\nabla F_{g,m'}\left(\mathbf{x}_{g,p,m',k'}\right)$$

$$-\sum_{m'=1}^{m-1}\sum_{k'=0}^{K-1}\nabla F_{g,m'}\left(\mathbf{x}\right)+\sum_{p'=0}^{p-1}\sum_{m'=1}^{M}\sum_{k'=0}^{K-1}\nabla F_{g,m'}\left(\mathbf{x}_{g,p',m',k'}\right)-\sum_{p'=0}^{p-1}\sum_{m'=1}^{M}\sum_{k'=0}^{K-1}\nabla F_{g,m'}\left(\mathbf{x}\right)\|^2$$

$$\leq 15\eta^2 k\sum_{k'=0}^{k-1}\mathbb{E}\|\nabla F_{g,m}\left(\mathbf{x}_{g,p,m,k'}\right)-\nabla F_{g,m}\left(\mathbf{x}\right)\|^2$$

$$+15\eta^2 mK\sum_{m'=1}^{m-1}\sum_{k'=0}^{K-1}\mathbb{E}\|\nabla F_{g,m'}\left(\mathbf{x}_{g,p,m',k'}\right)-\nabla F_{g,m'}\left(\mathbf{x}\right)\|^2$$

$$+15\eta^2 pMK\sum_{p'=0}^{p-1}\sum_{m'=1}^{M}\sum_{k'=0}^{K-1}\mathbb{E}\|\nabla F_{g,m'}\left(\mathbf{x}_{g,p',m',k'}\right)-\nabla F_{g,m'}\left(\mathbf{x}\right)\|^2$$

$$\leq 15L^2\eta^2 k\sum_{k'=0}^{k-1}\mathbb{E}\|\mathbf{x}_{g,p,m,k'}-\mathbf{x}\|^2+15L^2\eta^2 mK\sum_{m'=1}^{m-1}\sum_{k'=0}^{K-1}\mathbb{E}\|\mathbf{x}_{g,p',m',k'}-\mathbf{x}\|^2$$

$$+15L^2\eta^2 pMK\sum_{p'=0}^{p-1}\sum_{m'=1}^{M}\sum_{k'=0}^{K-1}\mathbb{E}\|\mathbf{x}_{g,p',m',k'}-\mathbf{x}\|^2$$

Bounding the third term in the left-hand inequality,

$$5\eta^2\mathbb{E}\|\sum_{k'=0}^{k-1}\nabla F_{g,m}\left(\mathbf{x}\right)+\sum_{m'=1}^{m-1}\sum_{k'=0}^{K-1}\nabla F_{g,m'}\left(\mathbf{x}\right)+\sum_{p'=0}^{p-1}\sum_{m'=1}^{M}\sum_{k'=0}^{K-1}\nabla F_{g,m'}\left(\mathbf{x}\right)-\sum_{k'=0}^{k-1}\nabla F_{g}\left(\mathbf{x}\right)$$

$$-\sum_{m'=1}^{m-1}\sum_{k'=0}^{K-1}\nabla F_{g}\left(\mathbf{x}\right)-\sum_{p'=0}^{p-1}\sum_{m'=1}^{M}\sum_{k'=0}^{K-1}\nabla F_{g}\left(\mathbf{x}\right)\|^2$$

$$\leq 15\eta^2 k\sum_{k'=0}^{k-1}\mathbb{E}\|\nabla F_{g,m}\left(\mathbf{x}\right)-\nabla F_{g}\left(\mathbf{x}\right)\|^2+15\eta^2 K\sum_{m'=1}^{m-1}\sum_{k'=0}^{K-1}\mathbb{E}\|\nabla F_{g,m'}\left(\mathbf{x}\right)-\nabla F_{g}\left(\mathbf{x}\right)\|^2$$

$$+ 15\eta^2 pMK \sum_{p'=0}^{p-1} \sum_{m'=1}^{M} \sum_{k'=0}^{K-1} \mathbb{E}\|\nabla F_{g,m'}(\mathbf{x}) - \nabla F_g(\mathbf{x})\|^2$$

Compared to the fixed-order case, the second term of the above equation $15\eta^2 K \sum_{m'=1}^{m-1} \sum_{k'=0}^{K-1} \mathbb{E}\|\nabla F_{g,m'}(\mathbf{x}) - \nabla F_g(\mathbf{x})\|^2$ is missing a factor of $m$.

Bounding the fourth term in the left-hand inequality,

$$5\eta^2 \mathbb{E}\| \sum_{k'=0}^{k-1} \nabla F_g(\mathbf{x}) + \sum_{m'=1}^{m-1} \sum_{k'=0}^{K-1} \nabla F_g(\mathbf{x}) + \sum_{p'=0}^{p-1} \sum_{m'=1}^{M} \sum_{k'=0}^{K-1} \nabla F_g(\mathbf{x})$$
$$- \sum_{k'=0}^{k-1} \nabla F(\mathbf{x}) - \sum_{m'=1}^{m-1} \sum_{k'=0}^{K-1} \nabla F(\mathbf{x}) - \sum_{p'=0}^{p-1} \sum_{m'=1}^{M} \sum_{k'=0}^{K-1} \nabla F(\mathbf{x}) \|^2$$
$$\leq 15\eta^2 k \sum_{k'=0}^{k-1} \mathbb{E}\|\nabla F_g(\mathbf{x}) - \nabla F(\mathbf{x})\|^2 + 15\eta^2 mK \sum_{m'=1}^{m-1} \sum_{k'=0}^{K-1} \mathbb{E}\|\nabla F_g(\mathbf{x}) - \nabla F(\mathbf{x})\|^2$$
$$+ 15\eta^2 pMK \sum_{p'=0}^{p-1} \sum_{m'=1}^{M} \sum_{k'=0}^{K-1} \mathbb{E}\|\nabla F_g(\mathbf{x}) - \nabla F(\mathbf{x})\|^2$$

Bounding the fifth term in the left-hand inequality,

$$5\eta^2 \mathbb{E}\| \sum_{k'=0}^{k-1} \nabla F(\mathbf{x}) + \sum_{m'=1}^{m-1} \sum_{k'=0}^{K-1} \nabla F(\mathbf{x}) + \sum_{p'=0}^{p-1} \sum_{m'=1}^{M} \sum_{k'=0}^{K-1} \nabla F(\mathbf{x}) \|^2$$
$$\leq 15\eta^2 k \sum_{k'=0}^{k-1} \mathbb{E}\|\nabla F(\mathbf{x})\|^2 + 15\eta^2 mK \sum_{p'=0}^{p-1} \sum_{m'=1}^{m-1} \sum_{k'=0}^{K-1} \mathbb{E}\|\nabla F(\mathbf{x})\|^2$$
$$+ 15\eta^2 pMK \sum_{p'=0}^{p-1} \sum_{m'=1}^{M} \sum_{k'=0}^{K-1} \mathbb{E}\|\nabla F(\mathbf{x})\|^2$$

Substitute these terms into $E_r$,

$$E_r \leq 15\eta^2 \sum_{g,p,m,k} \left( k\sigma^2 + mK\sigma^2 + pMK\sigma^2 + L^2 k \sum_{k'=0}^{k-1} \mathbb{E}\|\mathbf{x}_{g,p,m,k'} - \mathbf{x}\|^2 \right.$$
$$+ L^2 mK \sum_{m'=1}^{m-1} \sum_{k'=0}^{K-1} \mathbb{E}\|\mathbf{x}_{g,p,m',k'} - \mathbf{x}\|^2 + L^2 pMK \sum_{p'=0}^{p-1} \sum_{m'=1}^{M} \sum_{k'=0}^{K-1} \mathbb{E}\|\mathbf{x}_{g,p',m',k'} - \mathbf{x}\|^2$$
$$+ k \sum_{k'=0}^{k-1} \mathbb{E}\|\nabla F_{g,m}(\mathbf{x}) - \nabla F_g(\mathbf{x})\|^2 + K \sum_{m'=1}^{m-1} \sum_{k'=0}^{K-1} \mathbb{E}\|\nabla F_{g,m'}(\mathbf{x}) - \nabla F_g(\mathbf{x})\|^2$$
$$+ k \sum_{k'=0}^{k-1} \mathbb{E}\|\nabla F_g(\mathbf{x}) - \nabla F(\mathbf{x})\|^2 + mK \sum_{m'=1}^{m-1} \sum_{k'=0}^{K-1} \mathbb{E}\|\nabla F_g(\mathbf{x}) - \nabla F(\mathbf{x})\|^2$$
$$+ pMK \sum_{p'=0}^{p-1} \sum_{m'=1}^{M} \sum_{k'=0}^{K-1} \mathbb{E}\|\nabla F_g(\mathbf{x}) - \nabla F(\mathbf{x})\|^2$$
$$+ k \sum_{k'=0}^{k-1} \mathbb{E}\|\nabla F(\mathbf{x})\|^2 + mK \sum_{m'=1}^{m-1} \sum_{k'=0}^{K-1} \mathbb{E}\|\nabla F(\mathbf{x})\|^2 + pMK \sum_{p'=0}^{p-1} \sum_{m'=1}^{M} \sum_{k'=0}^{K-1} \mathbb{E}\|\nabla F(\mathbf{x})\|^2 \right)$$
$$\leq 15\eta^2 GPMK^2\sigma^2 + 15\eta^2 GPM^2K^2\sigma^2 + 15\eta^2 GP^2M^2K^2\sigma^2$$

$$+ 15\eta^2 GPMK^3\zeta^2 + 15\eta^2 GPM^2K^3\zeta^2 + 15\eta^2 GPMK^3\delta^2$$
$$+ 15\eta^2 GPM^3K^3\delta^2 + 15\eta^2 GP^3M^3K^3\delta^2 + 15\eta^2 GPMK^3\|\nabla F(\mathbf{x})\|^2$$
$$+ 15\eta^2 GPM^3K^3\|\nabla F(\mathbf{x})\|^2 + 15\eta^2 P^2M^2K^2\|\nabla F(\mathbf{x})\|^2 + 15L^2P^2M^2K^2\eta^2 E_r$$

Let $c_1 = \frac{1}{1 - 15L^2P^2M^2K^2\eta^2}$, we have

$$E_r \leq 45c_1\eta^2 GP^2M^2K^2\sigma^2 + 30c_1\eta^2 GPM^2K^3\zeta^2 + 45c_1\eta^2 GP^3M^3K^3\delta^2$$
$$+ 45c_1\eta^2 GP^3M^3K^3\|\nabla F(\mathbf{x})\|^2$$

After substituting $E_r$ into $\mathbb{E}[F(\mathbf{x}^{(r+1)}) - F(\mathbf{x}^{(r)})]$, we can obtain

$$\mathbb{E}[F(\mathbf{x}^{(r+1)}) - F(\mathbf{x}^{(r)})]$$
$$\leq -\eta MPK(\frac{1}{2} - \frac{45}{2}c_1\eta^2 L^2P^2M^2K^2)\mathbb{E}[\|\nabla F(\mathbf{x}^{(r)})\|^2] + \eta^2 L\frac{PMK}{G}\sigma^2$$
$$+ \frac{45}{2}c_1\eta^3 L^2P^2M^2K^2\sigma^2 + 15c_1\eta^3 L^2PM^2K^3\zeta^2 + \frac{45}{2}c_1\eta^3 L^2P^3M^3K^3\delta^2.$$

Let $c_2 = \frac{1}{\frac{1}{2} - \frac{45}{2}c_1\eta^2 L^2P^2M^2K^2}$, then

$$\frac{1}{R}\sum_{r=0}^{R-1}\mathbb{E}[\|\nabla F(\mathbf{x}^{(r)})\|^2] \leq \frac{c_2\left(F(\mathbf{x}^{(0)}) - F(\mathbf{x}^{(R)})\right)}{\eta RPMK} + c_2\eta L\frac{1}{G}\sigma^2 + \frac{45}{2}c_1c_2\eta^2 L^2PMK\sigma^2$$
$$+ 15c_1c_2\eta^2 L^2MK^2\zeta^2 + \frac{45}{2}c_1c_2\eta^2 L^2P^2M^2K^2\delta^2.$$

Let $\tilde{\eta} = \eta PMK$, and $\tilde{\eta} \leq \frac{1}{\sqrt{60L}}$, we have

$$\mathbb{E}[\|\nabla F(\bar{x}^{(R)})\|^2] \lesssim \frac{A}{\tilde{\eta}R} + \frac{L\tilde{\eta}\sigma^2}{GPMK} + \frac{L^2\tilde{\eta}^2\sigma^2}{PMK} + \frac{L^2\tilde{\eta}^2\zeta^2}{P^2M} + L^2\tilde{\eta}^2\delta^2$$

$$\mathbb{E}[\|\nabla F(\bar{x}^{(R)})\|^2] = \mathcal{O}\left(\frac{LA}{R} + \frac{(L\sigma^2A)^{1/2}}{\sqrt{GPMKR}} + \frac{(L^2A^2\sigma^2)^{1/3}}{(PMKR^2)^{1/3}} + \frac{(L^2A^2\zeta^2)^{1/3}}{(P^2MR^2)^{1/3}}\right.$$
$$\left. + \frac{(L^2A^2\delta^2)^{1/3}}{(R^2)^{1/3}}\right)$$

where $\bar{x}^R$ is defined as a model uniformly sampled from the $x^{(0)}, \ldots, x^{(R-1)}$ of previous iterations, and $\lesssim$ hides universal constants.

### E.3 RANDOM-ORDER EXTENSION OF THEOREM 1 FOR THE RING-STAR CASE

For Ring-Star,

$$\mathbf{x}_{g,p,m,k} - \mathbf{x} = \sum_{k'=0}^{k-1}\mathbf{g}_{g,p,m,k'} + \sum_{p'=0}^{p-1}\frac{1}{M}\sum_{m=1}^{M}\sum_{k=0}^{K-1}\mathbf{g}_{g,p',m',k'} + \sum_{g'=1}^{g-1}\sum_{p'=0}^{P-1}\frac{1}{M}\sum_{m=1}^{M}\sum_{k=0}^{K-1}\mathbf{g}_{g',p',m',k'}.$$

To bound $E_r$, we first bound $\mathbb{E}\|\mathbf{x}_{g,p,m,k} - \mathbf{x}\|^2$.

$$\mathbb{E}\|\mathbf{x}_{g,p,m,k} - \mathbf{x}\|^2$$
$$= \eta^2\mathbb{E}\|\sum_{k'=0}^{k-1}\mathbf{g}_{g,p,m,k'} + \sum_{p'=0}^{p-1}\frac{1}{M}\sum_{m=1}^{M}\sum_{k=0}^{K-1}\mathbf{g}_{g,p',m',k'} + \sum_{g'=1}^{g-1}\sum_{p'=0}^{P-1}\frac{1}{M}\sum_{m=1}^{M}\sum_{k=0}^{K-1}\mathbf{g}_{g',p',m',k'}\|^2$$
$$\leq 5\eta^2\mathbb{E}\|\sum_{k'=0}^{k-1}\mathbf{g}_{g,p,m,k'} + \sum_{p'=0}^{p-1}\frac{1}{M}\sum_{m=1}^{M}\sum_{k=0}^{K-1}\mathbf{g}_{g,p',m',k'} + \sum_{g'=1}^{g-1}\sum_{p'=0}^{P-1}\frac{1}{M}\sum_{m=1}^{M}\sum_{k=0}^{K-1}\mathbf{g}_{g',p',m',k'}$$

$$-\sum_{k'=0}^{k-1}\nabla F_{g,m}\left(\mathbf{x}_{g,p,m,k'}\right)-\sum_{p'=0}^{p-1}\frac{1}{M}\sum_{m=1}^{M}\sum_{k=0}^{K-1}\nabla F_{g,m'}\left(\mathbf{x}_{g,p',m',k'}\right)$$

$$-\sum_{g'=1}^{g-1}\sum_{p'=0}^{P-1}\frac{1}{M}\sum_{m=1}^{M}\sum_{k=0}^{K-1}\nabla F_{g',m'}\left(\mathbf{x}_{g',p',m',k'}\right)\|^2$$

$$+5\eta^2\mathbb{E}\|\sum_{k'=0}^{k-1}\nabla F_{g,m}\left(\mathbf{x}_{g,p,m,k'}\right)+\sum_{p'=0}^{p-1}\frac{1}{M}\sum_{m=1}^{M}\sum_{k=0}^{K-1}\nabla F_{g,m'}\left(\mathbf{x}_{g,p',m',k'}\right)$$

$$+\sum_{g'=1}^{g-1}\sum_{p'=0}^{P-1}\frac{1}{M}\sum_{m=1}^{M}\sum_{k=0}^{K-1}\nabla F_{g',m'}\left(\mathbf{x}_{g',p',m',k'}\right)-\sum_{k'=0}^{k-1}\nabla F_{g,m}\left(\mathbf{x}\right)$$

$$-\sum_{p'=0}^{p-1}\frac{1}{M}\sum_{m=1}^{M}\sum_{k=0}^{K-1}\nabla F_{g,m'}\left(\mathbf{x}\right)-\sum_{g'=1}^{g-1}\sum_{p'=0}^{P-1}\frac{1}{M}\sum_{m=1}^{M}\sum_{k=0}^{K-1}\nabla F_{g',m'}\left(\mathbf{x}\right)\|^2$$

$$+5\eta^2\mathbb{E}\|\sum_{k'=0}^{k-1}\nabla F_{g,m}\left(\mathbf{x}\right)+\sum_{p'=0}^{p-1}\frac{1}{M}\sum_{m=1}^{M}\sum_{k=0}^{K-1}\nabla F_{g,m'}\left(\mathbf{x}\right)+\sum_{g'=1}^{g-1}\sum_{p'=0}^{P-1}\frac{1}{M}\sum_{m=1}^{M}\sum_{k=0}^{K-1}\nabla F_{g',m'}\left(\mathbf{x}\right)$$

$$-\sum_{k'=0}^{k-1}\nabla F_g\left(\mathbf{x}\right)-\sum_{p'=0}^{p-1}\frac{1}{M}\sum_{m=1}^{M}\sum_{k=0}^{K-1}\nabla F_g\left(\mathbf{x}\right)-\sum_{g'=1}^{g-1}\sum_{p'=0}^{P-1}\frac{1}{M}\sum_{m=1}^{M}\sum_{k=0}^{K-1}\nabla F_{g'}\left(\mathbf{x}\right)\|^2$$

$$+5\eta^2\mathbb{E}\|\sum_{k'=0}^{k-1}\nabla F_g\left(\mathbf{x}\right)+\sum_{p'=0}^{p-1}\frac{1}{M}\sum_{m=1}^{M}\sum_{k=0}^{K-1}\nabla F_g\left(\mathbf{x}\right)+\sum_{g'=1}^{g-1}\sum_{p'=0}^{P-1}\frac{1}{M}\sum_{m=1}^{M}\sum_{k=0}^{K-1}\nabla F_{g'}\left(\mathbf{x}\right)$$

$$-\sum_{k'=0}^{k-1}\nabla F\left(\mathbf{x}\right)-\sum_{p'=0}^{p-1}\frac{1}{M}\sum_{m=1}^{M}\sum_{k=0}^{K-1}\nabla F\left(\mathbf{x}\right)-\sum_{g'=1}^{g-1}\sum_{p'=0}^{P-1}\frac{1}{M}\sum_{m=1}^{M}\sum_{k=0}^{K-1}\nabla F\left(\mathbf{x}\right)\|^2$$

$$+5\eta^2\mathbb{E}\|\sum_{k'=0}^{k-1}\nabla F\left(\mathbf{x}\right)+\sum_{p'=0}^{p-1}\frac{1}{M}\sum_{m=1}^{M}\sum_{k=0}^{K-1}\nabla F\left(\mathbf{x}\right)+\sum_{g'=1}^{g-1}\sum_{p'=0}^{P-1}\frac{1}{M}\sum_{m=1}^{M}\sum_{k=0}^{K-1}\nabla F\left(\mathbf{x}\right)\|^2$$

Bounding the first term in the left-hand inequality,

$$5\eta^2\mathbb{E}\|\sum_{k'=0}^{k-1}\mathbf{g}_{g,p,m,k'}+\sum_{p'=0}^{p-1}\frac{1}{M}\sum_{m=1}^{M}\sum_{k=0}^{K-1}\mathbf{g}_{g,p',m',k'}+\sum_{g'=1}^{g-1}\sum_{p'=0}^{P-1}\frac{1}{M}\sum_{m=1}^{M}\sum_{k=0}^{K-1}\mathbf{g}_{g',p',m',k'}$$

$$-\sum_{k'=0}^{k-1}\nabla F_{g,m}\left(\mathbf{x}_{g,p,m,k'}\right)-\sum_{p'=0}^{p-1}\frac{1}{M}\sum_{m=1}^{M}\sum_{k=0}^{K-1}\nabla F_{g,m'}\left(\mathbf{x}_{g,p',m',k'}\right)$$

$$-\sum_{g'=1}^{g-1}\sum_{p'=0}^{P-1}\frac{1}{M}\sum_{m=1}^{M}\sum_{k=0}^{K-1}\nabla F_{g',m'}\left(\mathbf{x}_{g',p',m',k'}\right)\|^2$$

$$\leq 15\eta^2\sum_{k'=0}^{k-1}\mathbb{E}\|\mathbf{g}_{g,p,m,k'}-\nabla F_{g,m}\left(\mathbf{x}_{g,p,m,k'}\right)\|^2$$

$$+15\eta^2\sum_{p'=0}^{p-1}\frac{1}{M^2}\sum_{m'=1}^{M}\sum_{k'=0}^{K-1}\mathbb{E}\|\mathbf{g}_{g,p',m',k'}-\nabla F_{g,m'}\left(\mathbf{x}_{g,p',m',k'}\right)\|^2$$

$$+15\eta^2\sum_{g'=1}^{g-1}\sum_{p'=0}^{P-1}\frac{1}{M^2}\sum_{m=1}^{M}\sum_{k=0}^{K-1}\mathbb{E}\|\mathbf{g}_{g',p',m',k'}-\nabla F_{g',m'}\left(\mathbf{x}_{g',p',m',k'}\right)\|^2$$

$$\leq 15\eta^2 k\sigma^2+15\eta^2\frac{pK}{M}\sigma^2+15\eta^2\frac{gPK}{M}\sigma^2$$

Bounding the second term in the left-hand inequality,

$$5\eta^2 \mathbb{E}\| \sum_{k'=0}^{k-1} \nabla F_{g,m}\left(\mathbf{x}_{g,p,m,k'}\right) + \sum_{p'=0}^{p-1} \frac{1}{M} \sum_{m=1}^{M} \sum_{k=0}^{K-1} \nabla F_{g,m'}\left(\mathbf{x}_{g,p',m',k'}\right)$$

$$+ \sum_{g'=1}^{g-1} \sum_{p'=0}^{P-1} \frac{1}{M} \sum_{m=1}^{M} \sum_{k=0}^{K-1} \nabla F_{g',m'}\left(\mathbf{x}_{g',p',m',k'}\right) - \sum_{k'=0}^{k-1} \nabla F_{g,m}\left(\mathbf{x}\right)$$

$$- \sum_{p'=0}^{p-1} \frac{1}{M} \sum_{m=1}^{M} \sum_{k=0}^{K-1} \nabla F_{g,m'}\left(\mathbf{x}\right) - \sum_{g'=1}^{g-1} \sum_{p'=0}^{P-1} \frac{1}{M} \sum_{m=1}^{M} \sum_{k=0}^{K-1} \nabla F_{g',m'}\left(\mathbf{x}\right) \|^2$$

$$\leq 15\eta^2 k \sum_{k'=0}^{k-1} \mathbb{E}\|\nabla F_{g,m}\left(\mathbf{x}_{g,p,m,k'}\right) - \nabla F_{g,m}\left(\mathbf{x}\right)\|^2$$

$$+ 15\eta^2 pMK \sum_{p'=0}^{p-1} \frac{1}{M^2} \sum_{m'=1}^{M} \sum_{k'=0}^{K-1} \mathbb{E}\|\nabla F_{g,m'}\left(\mathbf{x}_{g,p',m',k'}\right) - \nabla F_{g,m'}\left(\mathbf{x}\right)\|^2$$

$$+ 15\eta^2 gPMK \sum_{g'=1}^{g-1} \sum_{p'=0}^{P-1} \frac{1}{M^2} \sum_{m'=1}^{M} \sum_{k'=0}^{K-1} \mathbb{E}\|\nabla F_{g,m'}\left(\mathbf{x}_{g,p',m',k'}\right) - \nabla F_{g,m'}\left(\mathbf{x}\right)\|^2$$

$$\leq 15L^2\eta^2 k \sum_{k'=0}^{k-1} \mathbb{E}\|\mathbf{x}_{g,p,m,k'} - \mathbf{x}\|^2 + 15L^2\eta^2 \frac{pK}{M} \sum_{p'=0}^{p-1} \sum_{m'=1}^{M} \sum_{k'=0}^{K-1} \mathbb{E}\|\mathbf{x}_{g,p',m',k'} - \mathbf{x}\|^2$$

$$+ 15L^2\eta^2 \frac{gPK}{M} \sum_{g'=1}^{g-1} \sum_{p'=0}^{P-1} \sum_{m'=1}^{M} \sum_{k'=0}^{K-1} \mathbb{E}\|\mathbf{x}_{g,p',m',k'} - \mathbf{x}\|^2$$

Bounding the third term in the left-hand inequality,

$$5\eta^2 \mathbb{E}\| \sum_{k'=0}^{k-1} \nabla F_{g,m}\left(\mathbf{x}\right) + \sum_{p'=0}^{p-1} \frac{1}{M} \sum_{m=1}^{M} \sum_{k=0}^{K-1} \nabla F_{g,m'}\left(\mathbf{x}\right) + \sum_{g'=1}^{g-1} \sum_{p'=0}^{P-1} \frac{1}{M} \sum_{m=1}^{M} \sum_{k=0}^{K-1} \nabla F_{g',m'}\left(\mathbf{x}\right)$$

$$- \sum_{k'=0}^{k-1} \nabla F_{g}\left(\mathbf{x}\right) - \sum_{p'=0}^{p-1} \frac{1}{M} \sum_{m=1}^{M} \sum_{k=0}^{K-1} \nabla F_{g}\left(\mathbf{x}\right) - \sum_{g'=1}^{g-1} \sum_{p'=0}^{P-1} \frac{1}{M} \sum_{m=1}^{M} \sum_{k=0}^{K-1} \nabla F_{g'}\left(\mathbf{x}\right) \|^2$$

$$\leq 15\eta^2 k \sum_{k'=0}^{k-1} \mathbb{E}\|\nabla F_{g,m}\left(\mathbf{x}\right) - \nabla F_{g}\left(\mathbf{x}\right)\|^2$$

$$+ 15\eta^2 \frac{pK}{M} \sum_{p'=0}^{p-1} \sum_{m'=1}^{M} \sum_{k'=0}^{K-1} \mathbb{E}\|\nabla F_{g,m'}\left(\mathbf{x}\right) - \nabla F_{g}\left(\mathbf{x}\right)\|^2$$

$$+ 15\eta^2 \frac{gPK}{M} \sum_{g'=1}^{g-1} \sum_{p'=0}^{P-1} \sum_{m'=1}^{M} \sum_{k'=0}^{K-1} \mathbb{E}\|\nabla F_{g',m'}\left(\mathbf{x}\right) - \nabla F_{g'}\left(\mathbf{x}\right)\|^2$$

Bounding the fourth term in the left-hand inequality,

$$5\eta^2 \mathbb{E}\| \sum_{k'=0}^{k-1} \nabla F_{g}\left(\mathbf{x}\right) + \sum_{p'=0}^{p-1} \frac{1}{M} \sum_{m=1}^{M} \sum_{k=0}^{K-1} \nabla F_{g}\left(\mathbf{x}\right) + \sum_{g'=1}^{g-1} \sum_{p'=0}^{P-1} \frac{1}{M} \sum_{m=1}^{M} \sum_{k=0}^{K-1} \nabla F_{g'}\left(\mathbf{x}\right)$$

$$- \sum_{k'=0}^{k-1} \nabla F\left(\mathbf{x}\right) - \sum_{p'=0}^{p-1} \frac{1}{M} \sum_{m=1}^{M} \sum_{k=0}^{K-1} \nabla F\left(\mathbf{x}\right) - \sum_{g'=1}^{g-1} \sum_{p'=0}^{P-1} \frac{1}{M} \sum_{m=1}^{M} \sum_{k=0}^{K-1} \nabla F\left(\mathbf{x}\right) \|^2$$

$$\leq 15\eta^2 k \sum_{k'=0}^{k-1} \mathbb{E}\|\nabla F_{g}\left(\mathbf{x}\right) - \nabla F\left(\mathbf{x}\right)\|^2 + 15\eta^2 \frac{pK}{M} \sum_{p'=0}^{p-1} \sum_{m'=1}^{M} \sum_{k'=0}^{K-1} \mathbb{E}\|\nabla F_{g}\left(\mathbf{x}\right) - \nabla F\left(\mathbf{x}\right)\|^2$$

$$+ 15\eta^2 \frac{PK}{M} \sum_{g'=1}^{g-1} \sum_{p'=0}^{P-1} \sum_{m'=1}^{M} \sum_{k'=0}^{K-1} \mathbb{E}\|\nabla F_{g'}(\mathbf{x}) - \nabla F(\mathbf{x})\|^2$$

Compared to the fixed-order case, the last term of the above equation $15\eta^2 \frac{PK}{M} \sum_{g'=1}^{g-1} \sum_{p'=0}^{P-1} \sum_{m'=1}^{M} \sum_{k'=0}^{K-1} \mathbb{E}\|\nabla F_{g'}(\mathbf{x}) - \nabla F(\mathbf{x})\|^2$ is missing a factor of $g$.

Bounding the fifth term in the left-hand inequality,

$$5\eta^2 \mathbb{E}\| \sum_{k'=0}^{k-1} \nabla F(\mathbf{x}) + \sum_{p'=0}^{p-1} \frac{1}{M} \sum_{m=1}^{M} \sum_{k=0}^{K-1} \nabla F(\mathbf{x}) + \sum_{g'=1}^{g-1} \sum_{p'=0}^{P-1} \frac{1}{M} \sum_{m=1}^{M} \sum_{k=0}^{K-1} \nabla F(\mathbf{x})\|^2$$

$$\leq 15\eta^2 k \sum_{k'=0}^{k-1} \mathbb{E}\|\nabla F(\mathbf{x})\|^2 + 15\eta^2 pMK \sum_{p'=0}^{p-1} \frac{1}{M^2} \sum_{m'=1}^{M} \sum_{k'=0}^{K-1} \mathbb{E}\|\nabla F(\mathbf{x})\|^2$$

$$+ 15\eta^2 gPMK \sum_{g'=1}^{g-1} \sum_{p'=0}^{P-1} \frac{1}{M^2} \sum_{m'=1}^{M} \sum_{k'=0}^{K-1} \mathbb{E}\|\nabla F(\mathbf{x})\|^2$$

Substitute these terms into $E_r$,

$$E_r \leq 15\eta^2 \sum_{g,p,m,k} \left( k\sigma^2 + \frac{pK}{M}\sigma^2 + \frac{gPK}{M}\sigma^2 \right.$$

$$+ L^2 k \sum_{k'=0}^{k-1} \mathbb{E}\|\mathbf{x}_{g,p,m,k'} - \mathbf{x}\|^2 + L^2 \frac{pK}{M} \sum_{p'=0}^{p-1} \sum_{m'=1}^{M} \sum_{k'=0}^{K-1} \mathbb{E}\|\mathbf{x}_{g,p',m',k'} - \mathbf{x}\|^2$$

$$+ L^2 \frac{gPK}{M} \sum_{g'=1}^{g-1} \sum_{p'=0}^{P-1} \sum_{m'=1}^{M} \sum_{k'=0}^{K-1} \mathbb{E}\|\mathbf{x}_{g,p',m',k'} - \mathbf{x}\|^2 + k \sum_{k'=0}^{k-1} \mathbb{E}\|\nabla F_{g,m}(\mathbf{x}) - \nabla F_g(\mathbf{x})\|^2$$

$$+ k \sum_{k'=0}^{k-1} \mathbb{E}\|\nabla F_g(\mathbf{x}) - \nabla F(\mathbf{x})\|^2 + \frac{pK}{M} \sum_{p'=0}^{p-1} \sum_{m'=1}^{M} \sum_{k'=0}^{K-1} \mathbb{E}\|\nabla F_g(\mathbf{x}) - \nabla F(\mathbf{x})\|^2$$

$$+ \frac{PK}{M} \sum_{g'=1}^{g-1} \sum_{p'=0}^{P-1} \sum_{m'=1}^{M} \sum_{k'=0}^{K-1} \mathbb{E}\|\nabla F_g(\mathbf{x}) - \nabla F(\mathbf{x})\|^2 + k \sum_{k'=0}^{k-1} \mathbb{E}\|\nabla F(\mathbf{x})\|^2$$

$$\left. + pMK \sum_{p'=0}^{p-1} \frac{1}{M^2} \sum_{m'=1}^{M} \sum_{k'=0}^{K-1} \mathbb{E}\|\nabla F(\mathbf{x})\|^2 + gPMK \sum_{g'=1}^{g-1} \sum_{p'=0}^{P-1} \frac{1}{M^2} \sum_{m'=1}^{M} \sum_{k'=0}^{K-1} \mathbb{E}\|\nabla F(\mathbf{x})\|^2 \right)$$

$$\leq 15\eta^2 GPMK^2\sigma^2 + 15\eta^2 GP^2K^2\sigma^2 + 15\eta^2 G^2P^2K^2\sigma^2$$

$$+ 15\eta^2 GPMK^3\zeta^2 + 15\eta^2 GPMK^3\delta^2 + 15\eta^2 GP^3MK^3\delta^2 + 15\eta^2 G^2P^3MK^3\delta^2$$

$$+ 15\eta^2 GPMK^3\|\nabla F(\mathbf{x})\|^2 + 15\eta^2 GP^3MK^3\|\nabla F(\mathbf{x})\|^2 + 15\eta^2 G^3P^3MK^3\|\nabla F(\mathbf{x})\|^2$$

$$+ 15L^2 G^2P^2K^2\eta^2 E_r$$

Let $c_1 = \frac{1}{1 - 15L^2G^2P^2K^2\eta^2}$, we have

$$E_r \leq 15c_1\eta^2 GPMK^2\sigma^2 + 30c_1\eta^2 G^2P^2K^2\sigma^2 + 15c_1\eta^2 GPMK^3\zeta^2 + 45c_1\eta^2 G^2P^3MK^3\delta^2$$

$$+ 45c_1\eta^2 G^3P^3MK^3\|\nabla F(\mathbf{x})\|^2$$

After substituting $E_r$ into $\mathbb{E}[F(\mathbf{x}^{(r+1)}) - F(\mathbf{x}^{(r)})]$, we can obtain

$$\mathbb{E}[F(\mathbf{x}^{(r+1)}) - F(\mathbf{x}^{(r)})]$$

$$\leq -\eta GPK(\frac{1}{2} - \frac{45}{2}c_1\eta^2 L^2G^2P^2K^2)\mathbb{E}[\|\nabla F(\mathbf{x}^{(r)})\|^2] + \eta^2 L\frac{GPK}{M}\sigma^2$$

$$+\frac{15}{2}c_1\eta^3L^2GPK^2\sigma^2+15c_1\eta^3L^2\frac{G^2P^2K^2}{M}\sigma^2+\frac{15}{2}c_1\eta^3L^2GPK^3\zeta^2+\frac{45}{2}c_1\eta^3L^2G^2P^3K^3\delta^2.$$

Let $c_2=\frac{1}{\frac{1}{2}-\frac{45}{2}c_1\eta^2L^2G^2P^2K^2}$, then

$$\frac{1}{R}\sum_{r=0}^{R-1}\mathbb{E}[\|\nabla F(\mathbf{x}^{(r)})\|^2]\leq\frac{c_2\left(F(\mathbf{x}^{(0)})-F(\mathbf{x}^{(R)})\right)}{\eta RGPK}+c_2\eta L\frac{1}{M}\sigma^2+\frac{15}{2}c_1c_2\eta^2L^2K\sigma^2$$

$$+15c_1c_2\eta^2L^2\frac{GPK}{M}\sigma^2+\frac{15}{2}c_1c_2\eta^2L^2K^2\zeta^2+\frac{45}{2}c_1c_2\eta^2L^2GP^2K^2\delta^2.$$

Let $\tilde{\eta}=\eta GPK$, and $\tilde{\eta}\leq\frac{1}{\sqrt{60L}}$, we have

$$\mathbb{E}[\|\nabla F(\bar{x}^{(R)})\|^2]\lesssim\frac{A}{\tilde{\eta}R}+\frac{L\tilde{\eta}\sigma^2}{GPMK}+\frac{L^2\tilde{\eta}^2\sigma^2}{G^2P^2K}+\frac{L^2\tilde{\eta}^2\sigma^2}{GPMK}+\frac{L^2\tilde{\eta}^2\zeta^2}{G^2P^2}+\frac{L^2\tilde{\eta}^2\delta^2}{G}.$$

$$\mathbb{E}[\|\nabla F(\bar{x}^{(R)})\|^2]=\mathcal{O}\Bigg(\frac{LA}{R}+\frac{(L\sigma^2A)^{1/2}}{\sqrt{GPMKR}}+\frac{(L^2A^2\sigma^2)^{1/3}}{(G^2P^2KR^2)^{1/3}}+\frac{(L^2A^2\sigma^2)^{1/3}}{(GPMKR^2)^{1/3}}$$

$$+\frac{(L^2A^2\zeta^2)^{1/3}}{(G^2P^2R^2)^{1/3}}+\frac{(L^2A^2\delta^2)^{1/3}}{(GR^2)^{1/3}}\Bigg)$$

where $\bar{x}^R$ is defined as a model uniformly sampled from the $x^{(0)},\ldots,x^{(R-1)}$ of previous iterations, and $\lesssim$ hides universal constants.

### E.4 RANDOM-ORDER EXTENSION OF THEOREM 1 FOR THE RING-RING CASE

For Ring-Ring,

$$\mathbf{x}_{g,p,m,k}-\mathbf{x}=\sum_{k'=0}^{k-1}\mathbf{g}_{g,p,m,k'}+\sum_{m'=1}^{m-1}\sum_{k'=0}^{K-1}\mathbf{g}_{g,p,m',k'}+\sum_{p'=0}^{p-1}\sum_{m'=1}^{M}\sum_{k=0}^{K-1}\mathbf{g}_{g,p',m',k'}$$

$$+\sum_{g'=1}^{g-1}\sum_{p'=0}^{P-1}\sum_{m'=1}^{M}\sum_{k=0}^{K-1}\mathbf{g}_{g',p',m',k'}.$$

To bound $E_r$, we first bound $\mathbb{E}\|\mathbf{x}_{g,p,m,k}-\mathbf{x}\|^2$.

$$\mathbb{E}\|\mathbf{x}_{g,p,m,k}-\mathbf{x}\|^2$$

$$=\eta^2\mathbb{E}\|\sum_{k'=0}^{k-1}\mathbf{g}_{g,p,m,k'}+\sum_{m'=1}^{m-1}\sum_{k'=0}^{K-1}\mathbf{g}_{g,p,m',k'}+\sum_{p'=0}^{p-1}\sum_{m'=1}^{M}\sum_{k=0}^{K-1}\mathbf{g}_{g,p',m',k'}$$

$$+\sum_{g'=1}^{g-1}\sum_{p'=0}^{P-1}\sum_{m'=1}^{M}\sum_{k=0}^{K-1}\mathbf{g}_{g',p',m',k'}\|^2$$

$$\leq 5\eta^2\mathbb{E}\|\sum_{k'=0}^{k-1}\mathbf{g}_{g,p,m,k'}+\sum_{m'=1}^{m-1}\sum_{k'=0}^{K-1}\mathbf{g}_{g,p,m',k'}+\sum_{p'=0}^{p-1}\sum_{m'=1}^{M}\sum_{k=0}^{K-1}\mathbf{g}_{g,p',m',k'}$$

$$+\sum_{g'=1}^{g-1}\sum_{p'=0}^{P-1}\sum_{m'=1}^{M}\sum_{k=0}^{K-1}\mathbf{g}_{g',p',m',k'}-\sum_{k'=0}^{k-1}\nabla F_{g,m}\left(\mathbf{x}_{g,p,m,k'}\right)-\sum_{m'=1}^{m-1}\sum_{k'=0}^{K-1}\nabla F_{g,m'}\left(\mathbf{x}_{g,p,m',k'}\right)$$

$$-\sum_{p'=0}^{p-1}\sum_{m'=1}^{M}\sum_{k'=0}^{K-1}\nabla F_{g,m'}\left(\mathbf{x}_{g,p',m',k'}\right)-\sum_{g'=1}^{g-1}\sum_{p'=0}^{P-1}\sum_{m'=1}^{M}\sum_{k'=0}^{K-1}\nabla F_{g,m'}\left(\mathbf{x}_{g',p',m',k'}\right)\|^2$$

$$+5\eta^2\mathbb{E}\|\sum_{k'=0}^{k-1}\nabla F_{g,m}\left(\mathbf{x}_{g,p,m,k'}\right)+\sum_{m'=1}^{m-1}\sum_{k'=0}^{K-1}\nabla F_{g,m'}\left(\mathbf{x}_{g,p,m',k'}\right)$$

$$+ \sum_{p'=0}^{p-1} \sum_{m'=1}^{M} \sum_{k'=0}^{K-1} \nabla F_{g,m'} \left(\mathbf{x}_{g,p',m',k'}\right) + \sum_{g'=1}^{g-1} \sum_{p'=0}^{P-1} \sum_{m'=1}^{M} \sum_{k'=0}^{K-1} \nabla F_{g,m'} \left(\mathbf{x}_{g',p',m',k'}\right)$$

$$- \sum_{k'=0}^{k-1} \nabla F_{g,m} \left(\mathbf{x}\right) - \sum_{m'=1}^{m-1} \sum_{k'=0}^{K-1} \nabla F_{g,m'} \left(\mathbf{x}\right) - \sum_{p'=0}^{p-1} \sum_{m'=1}^{M} \sum_{k'=0}^{K-1} \nabla F_{g,m'} \left(\mathbf{x}\right)$$

$$- \sum_{g'=1}^{g-1} \sum_{p'=0}^{P-1} \sum_{m'=1}^{M} \sum_{k'=0}^{K-1} \nabla F_{g',m'} \left(\mathbf{x}\right) \|^2$$

$$+ 5\eta^2 \mathbb{E} \| \sum_{k'=0}^{k-1} \nabla F_{g,m} \left(\mathbf{x}\right) + \sum_{m'=1}^{m-1} \sum_{k'=0}^{K-1} \nabla F_{g,m'} \left(\mathbf{x}\right) + \sum_{p'=0}^{p-1} \sum_{m'=1}^{M} \sum_{k'=0}^{K-1} \nabla F_{g,m'} \left(\mathbf{x}\right)$$

$$+ \sum_{g'=1}^{g-1} \sum_{p'=0}^{P-1} \sum_{m'=1}^{M} \sum_{k'=0}^{K-1} \nabla F_{g',m'} \left(\mathbf{x}\right) - \sum_{k'=0}^{k-1} \nabla F_g \left(\mathbf{x}\right) - \sum_{m'=1}^{m-1} \sum_{k'=0}^{K-1} \nabla F_g \left(\mathbf{x}\right)$$

$$- \sum_{p'=0}^{p-1} \sum_{m'=1}^{M} \sum_{k'=0}^{K-1} \nabla F_g \left(\mathbf{x}\right) - \sum_{g'=1}^{g-1} \sum_{p'=0}^{P-1} \sum_{m'=1}^{M} \sum_{k'=0}^{K-1} \nabla F_{g'} \left(\mathbf{x}\right) \|^2$$

$$+ 5\eta^2 \mathbb{E} \| \sum_{k'=0}^{k-1} \nabla F_g \left(\mathbf{x}\right) + \sum_{m'=1}^{m-1} \sum_{k'=0}^{K-1} \nabla F_g \left(\mathbf{x}\right) + \sum_{p'=0}^{p-1} \sum_{m'=1}^{M} \sum_{k'=0}^{K-1} \nabla F_g \left(\mathbf{x}\right)$$

$$+ \sum_{g'=1}^{g-1} \sum_{p'=0}^{P-1} \sum_{m'=1}^{M} \sum_{k'=0}^{K-1} \nabla F_{g'} \left(\mathbf{x}\right) - \sum_{k'=0}^{k-1} \nabla F \left(\mathbf{x}\right) - \sum_{m'=1}^{m-1} \sum_{k'=0}^{K-1} \nabla F \left(\mathbf{x}\right)$$

$$- \sum_{p'=0}^{p-1} \sum_{m'=1}^{M} \sum_{k'=0}^{K-1} \nabla F \left(\mathbf{x}\right) - \sum_{g'=1}^{g-1} \sum_{p'=0}^{P-1} \sum_{m'=1}^{M} \sum_{k'=0}^{K-1} \nabla F \left(\mathbf{x}\right) \|^2$$

$$+ 5\eta^2 \mathbb{E} \| \sum_{k'=0}^{k-1} \nabla F \left(\mathbf{x}\right) + \sum_{m'=1}^{m-1} \sum_{k'=0}^{K-1} \nabla F \left(\mathbf{x}\right) + \sum_{p'=0}^{p-1} \sum_{m'=1}^{M} \sum_{k'=0}^{K-1} \nabla F \left(\mathbf{x}\right) + \sum_{g'=1}^{g-1} \sum_{p'=0}^{P-1} \sum_{m'=1}^{M} \sum_{k'=0}^{K-1} \nabla F \left(\mathbf{x}\right) \|^2$$

Bounding the first term in the left-hand inequality,

$$5\eta^2 \mathbb{E} \| \sum_{k'=0}^{k-1} \mathbf{g}_{g,p,m,k'} + \sum_{m'=1}^{m-1} \sum_{k'=0}^{K-1} \mathbf{g}_{g,p,m',k'} + \sum_{p'=0}^{p-1} \sum_{m'=1}^{M} \sum_{k=0}^{K-1} \mathbf{g}_{g,p',m',k}$$

$$+ \sum_{g'=1}^{g-1} \sum_{p'=0}^{P-1} \sum_{m'=1}^{M} \sum_{k=0}^{K-1} \mathbf{g}_{g',p',m',k'} - \sum_{k'=0}^{k-1} \nabla F_{g,m} \left(\mathbf{x}_{g,p,m,k'}\right) - \sum_{m'=1}^{m-1} \sum_{k'=0}^{K-1} \nabla F_{g,m'} \left(\mathbf{x}_{g,p,m',k'}\right)$$

$$- \sum_{p'=0}^{p-1} \sum_{m'=1}^{M} \sum_{k'=0}^{K-1} \nabla F_{g,m'} \left(\mathbf{x}_{g,p',m',k'}\right) - \sum_{g'=1}^{g-1} \sum_{p'=0}^{P-1} \sum_{m'=1}^{M} \sum_{k'=0}^{K-1} \nabla F_{g,m'} \left(\mathbf{x}_{g',p',m',k'}\right) \|^2$$

$$\leq 20\eta^2 \sum_{k'=0}^{k-1} \mathbb{E} \| \mathbf{g}_{g,p,m,k'} - \nabla F_{g,m} \left(\mathbf{x}_{g,p,m,k'}\right) \|^2$$

$$+ 20\eta^2 \sum_{m'=1}^{m-1} \sum_{k'=0}^{K-1} \mathbb{E} \| \mathbf{g}_{g,p,m',k'} - \nabla F_{g,m'} \left(\mathbf{x}_{g,p,m',k'}\right) \|^2$$

$$+ 20\eta^2 \sum_{p'=0}^{p-1} \sum_{m'=1}^{M} \sum_{k'=0}^{K-1} \mathbb{E} \| \mathbf{g}_{g,p',m',k'} - \nabla F_{g,m'} \left(\mathbf{x}_{g,p',m',k'}\right) \|^2$$

$$+ 20\eta^2 \sum_{g'=1}^{g-1} \sum_{p'=0}^{P-1} \sum_{m'=1}^{M} \sum_{k'=0}^{K-1} \mathbb{E} \| \mathbf{g}_{g',p',m',k'} - \nabla F_{g,m'} \left(\mathbf{x}_{g',p',m',k'}\right) \|^2$$

$$\leq 20\eta^2 k\sigma^2 + 20\eta^2 mK\sigma^2 + 20\eta^2 pMK\sigma^2 + 20\eta^2 gPMK\sigma^2$$

Bounding the second term in the left-hand inequality,

$$5\eta^2\mathbb{E}\|\sum_{k'=0}^{k-1}\nabla F_{g,m}\left(\mathbf{x}_{g,p,m,k'}\right) + \sum_{m'=1}^{m-1}\sum_{k'=0}^{K-1}\nabla F_{g,m'}\left(\mathbf{x}_{g,p,m',k'}\right)$$

$$+ \sum_{p'=0}^{p-1}\sum_{m'=1}^{M}\sum_{k'=0}^{K-1}\nabla F_{g,m'}\left(\mathbf{x}_{g,p',m',k'}\right) + \sum_{g'=1}^{g-1}\sum_{p'=0}^{P-1}\sum_{m'=1}^{M}\sum_{k'=0}^{K-1}\nabla F_{g,m'}\left(\mathbf{x}_{g',p',m',k'}\right)$$

$$- \sum_{k'=0}^{k-1}\nabla F_{g,m}\left(\mathbf{x}\right) - \sum_{m'=1}^{m-1}\sum_{k'=0}^{K-1}\nabla F_{g,m'}\left(\mathbf{x}\right) - \sum_{p'=0}^{p-1}\sum_{m'=1}^{M}\sum_{k'=0}^{K-1}\nabla F_{g,m'}\left(\mathbf{x}\right)$$

$$- \sum_{g'=1}^{g-1}\sum_{p'=0}^{P-1}\sum_{m'=1}^{M}\sum_{k'=0}^{K-1}\nabla F_{g',m'}\left(\mathbf{x}\right)\|^2$$

$$\leq 20\eta^2 k\sum_{k'=0}^{k-1}\mathbb{E}\|\nabla F_{g,m}\left(\mathbf{x}_{g,p,m,k'}\right) - \nabla F_{g,m}\left(\mathbf{x}\right)\|^2$$

$$+ 20\eta^2 mK\sum_{m'=1}^{m-1}\sum_{k'=0}^{K-1}\mathbb{E}\|\nabla F_{g,m'}\left(\mathbf{x}_{g,p,m',k'}\right) - \nabla F_{g,m'}\left(\mathbf{x}\right)\|^2$$

$$+ 20\eta^2 pMK\sum_{p'=0}^{p-1}\sum_{m'=1}^{M}\sum_{k'=0}^{K-1}\mathbb{E}\|\nabla F_{g,m'}\left(\mathbf{x}_{g,p',m',k'}\right) - \nabla F_{g,m'}\left(\mathbf{x}\right)\|^2$$

$$+ 20\eta^2 gPMK\sum_{g'=1}^{g-1}\sum_{p'=0}^{P-1}\sum_{m'=1}^{M}\sum_{k'=0}^{K-1}\mathbb{E}\|\nabla F_{g',m'}\left(\mathbf{x}_{g',p',m',k'}\right) - \nabla F_{g',m'}\left(\mathbf{x}\right)\|^2$$

$$\leq 20L^2\eta^2 k\sum_{k'=0}^{k-1}\mathbb{E}\|\mathbf{x}_{g,p,m,k'} - \mathbf{x}\|^2 + 20L^2\eta^2 mK\sum_{m'=1}^{m-1}\sum_{k'=0}^{K-1}\mathbb{E}\|\mathbf{x}_{g,p',m',k'} - \mathbf{x}\|^2$$

$$+ 20L^2\eta^2 pMK\sum_{p'=0}^{p-1}\sum_{m'=1}^{M}\sum_{k'=0}^{K-1}\mathbb{E}\|\mathbf{x}_{g,p',m',k'} - \mathbf{x}\|^2$$

$$+ 20L^2\eta^2 gPMK\sum_{g'=1}^{g-1}\sum_{p'=0}^{P-1}\sum_{m'=1}^{M}\sum_{k'=0}^{K-1}\mathbb{E}\|\mathbf{x}_{g',p',m',k'} - \mathbf{x}\|^2$$

Bounding the third term in the left-hand inequality,

$$5\eta^2\mathbb{E}\|\sum_{k'=0}^{k-1}\nabla F_{g,m}\left(\mathbf{x}\right) + \sum_{m'=1}^{m-1}\sum_{k'=0}^{K-1}\nabla F_{g,m'}\left(\mathbf{x}\right) + \sum_{p'=0}^{p-1}\sum_{m'=1}^{M}\sum_{k'=0}^{K-1}\nabla F_{g,m'}\left(\mathbf{x}\right)$$

$$+ \sum_{g'=1}^{g-1}\sum_{p'=0}^{P-1}\sum_{m'=1}^{M}\sum_{k'=0}^{K-1}\nabla F_{g',m'}\left(\mathbf{x}\right) - \sum_{k'=0}^{k-1}\nabla F_g\left(\mathbf{x}\right) - \sum_{m'=1}^{m-1}\sum_{k'=0}^{K-1}\nabla F_g\left(\mathbf{x}\right)$$

$$- \sum_{p'=0}^{p-1}\sum_{m'=1}^{M}\sum_{k'=0}^{K-1}\nabla F_g\left(\mathbf{x}\right) - \sum_{g'=1}^{g-1}\sum_{p'=0}^{P-1}\sum_{m'=1}^{M}\sum_{k'=0}^{K-1}\nabla F_{g'}\left(\mathbf{x}\right)\|^2$$

$$\leq 20\eta^2 k\sum_{k'=0}^{k-1}\mathbb{E}\|\nabla F_{g,m}\left(\mathbf{x}\right) - \nabla F_g\left(\mathbf{x}\right)\|^2 + 20\eta^2 K\sum_{m'=1}^{m-1}\sum_{k'=0}^{K-1}\mathbb{E}\|\nabla F_{g,m'}\left(\mathbf{x}\right) - \nabla F_g\left(\mathbf{x}\right)\|^2$$

$$+ 20\eta^2 pMK\sum_{p'=0}^{p-1}\sum_{m'=1}^{M}\sum_{k'=0}^{K-1}\mathbb{E}\|\nabla F_{g,m'}\left(\mathbf{x}\right) - \nabla F_g\left(\mathbf{x}\right)\|^2$$

$$+ 20\eta^2 gPMK \sum_{g'=1}^{g-1} \sum_{p'=0}^{P-1} \sum_{m'=1}^{M} \sum_{k'=0}^{K-1} \mathbb{E}\|\nabla F_{g',m'}(\mathbf{x}) - \nabla F_{g'}(\mathbf{x})\|^2$$

Compared to the fixed-order case, the second term of the above equation $20\eta^2 K \sum_{m'=1}^{m-1} \sum_{k'=0}^{K-1} \mathbb{E}\|\nabla F_{g,m'}(\mathbf{x}) - \nabla F_g(\mathbf{x})\|^2$ is missing a factor of $m$.

Bounding the fourth term in the left-hand inequality,

$$5\eta^2 \mathbb{E}\| \sum_{k'=0}^{k-1} \nabla F_g(\mathbf{x}) + \sum_{m'=1}^{m-1} \sum_{k'=0}^{K-1} \nabla F_g(\mathbf{x}) + \sum_{p'=0}^{p-1} \sum_{m'=1}^{M} \sum_{k'=0}^{K-1} \nabla F_g(\mathbf{x})$$

$$+ \sum_{g'=1}^{g-1} \sum_{p'=0}^{P-1} \sum_{m'=1}^{M} \sum_{k'=0}^{K-1} \nabla F_{g'}(\mathbf{x}) - \sum_{k'=0}^{k-1} \nabla F(\mathbf{x}) - \sum_{m'=1}^{m-1} \sum_{k'=0}^{K-1} \nabla F(\mathbf{x})$$

$$- \sum_{p'=0}^{p-1} \sum_{m'=1}^{M} \sum_{k'=0}^{K-1} \nabla F(\mathbf{x}) - \sum_{g'=1}^{g-1} \sum_{p'=0}^{P-1} \sum_{m'=1}^{M} \sum_{k'=0}^{K-1} \nabla F(\mathbf{x}) \|^2$$

$$\leq 20\eta^2 k \sum_{k'=0}^{k-1} \mathbb{E}\|\nabla F_g(\mathbf{x}) - \nabla F(\mathbf{x})\|^2 + 20\eta^2 mK \sum_{m'=1}^{m-1} \sum_{k'=0}^{K-1} \mathbb{E}\|\nabla F_g(\mathbf{x}) - \nabla F(\mathbf{x})\|^2$$

$$+ 20\eta^2 pMK \sum_{p'=0}^{p-1} \sum_{m'=1}^{M} \sum_{k'=0}^{K-1} \mathbb{E}\|\nabla F_g(\mathbf{x}) - \nabla F(\mathbf{x})\|^2$$

$$+ 20\eta^2 PMK \sum_{g'=1}^{g-1} \sum_{p'=0}^{P-1} \sum_{m'=1}^{M} \sum_{k'=0}^{K-1} \mathbb{E}\|\nabla F_{g'}(\mathbf{x}) - \nabla F(\mathbf{x})\|^2$$

Compared to the fixed-order case, the last term of the above equation $20\eta^2 gPMK \sum_{g'=1}^{g-1} \sum_{p'=0}^{P-1} \sum_{m'=1}^{M} \sum_{k'=0}^{K-1} \mathbb{E}\|\nabla F_{g'}(\mathbf{x}) - \nabla F(\mathbf{x})\|^2$ is missing a factor of $g$.

Bounding the fifth term in the left-hand inequality,

$$5\eta^2 \mathbb{E}\| \sum_{k'=0}^{k-1} \nabla F(\mathbf{x}) + \sum_{m'=1}^{m-1} \sum_{k'=0}^{K-1} \nabla F(\mathbf{x}) + \sum_{p'=0}^{p-1} \sum_{m'=1}^{M} \sum_{k'=0}^{K-1} \nabla F(\mathbf{x}) + \sum_{g'=1}^{g-1} \sum_{p'=0}^{P-1} \sum_{m'=1}^{M} \sum_{k'=0}^{K-1} \nabla F(\mathbf{x}) \|^2$$

$$\leq 20\eta^2 k \sum_{k'=0}^{k-1} \mathbb{E}\|\nabla F(\mathbf{x})\|^2 + 20\eta^2 mK \sum_{p'=0}^{p-1} \sum_{m'=1}^{m-1} \sum_{k'=0}^{K-1} \mathbb{E}\|\nabla F(\mathbf{x})\|^2$$

$$+ 20\eta^2 pMK \sum_{p'=0}^{p-1} \sum_{m'=1}^{M} \sum_{k'=0}^{K-1} \mathbb{E}\|\nabla F(\mathbf{x})\|^2 + 20\eta^2 gPMK \sum_{g'=1}^{g-1} \sum_{p'=0}^{P-1} \sum_{m'=1}^{M} \sum_{k'=0}^{K-1} \mathbb{E}\|\nabla F(\mathbf{x})\|^2$$

Substitute these terms into $E_r$,

$$E_r \leq 20\eta^2 \sum_{g,p,m,k} \left( k\sigma^2 + mK\sigma^2 + pMK\sigma^2 + gPMK\sigma^2 + L^2 k \sum_{k'=0}^{k-1} \mathbb{E}\|\mathbf{x}_{g,p,m,k'} - \mathbf{x}\|^2 \right.$$

$$+ L^2 mK \sum_{m'=1}^{m-1} \sum_{k'=0}^{K-1} \mathbb{E}\|\mathbf{x}_{g,p,m',k'} - \mathbf{x}\|^2 + L^2 pMK \sum_{p'=0}^{p-1} \sum_{m'=1}^{M} \sum_{k'=0}^{K-1} \mathbb{E}\|\mathbf{x}_{g,p',m',k'} - \mathbf{x}\|^2$$

$$+ L^2 gPMK \sum_{g'=1}^{g-1} \sum_{p'=0}^{P-1} \sum_{m'=1}^{M} \sum_{k'=0}^{K-1} \mathbb{E}\|\mathbf{x}_{g',p',m',k'} - \mathbf{x}\|^2 + k \sum_{k'=0}^{k-1} \mathbb{E}\|\nabla F_{g,m}(\mathbf{x}) - \nabla F_g(\mathbf{x})\|^2$$

$$+ K \sum_{m'=1}^{m-1} \sum_{k'=0}^{K-1} \mathbb{E}\|\nabla F_{g,m'}(\mathbf{x}) - \nabla F_g(\mathbf{x})\|^2 + mk \sum_{k'=0}^{k-1} \mathbb{E}\|\nabla F_g(\mathbf{x}) - \nabla F(\mathbf{x})\|^2$$

$$+ K \sum_{m'=1}^{m-1} \sum_{k'=0}^{K-1} \mathbb{E}\|\nabla F_g(\mathbf{x}) - \nabla F(\mathbf{x})\|^2 + pMK \sum_{p'=0}^{p-1} \sum_{m'=1}^{M} \sum_{k'=0}^{K-1} \mathbb{E}\|\nabla F_g(\mathbf{x}) - \nabla F(\mathbf{x})\|^2$$

$$+ gPMK \sum_{g'=1}^{g-1} \sum_{p'=0}^{P-1} \sum_{m'=1}^{M} \sum_{k'=0}^{K-1} \mathbb{E}\|\nabla F_{g'}(\mathbf{x}) - \nabla F(\mathbf{x})\|^2 + k \sum_{k'=0}^{k-1} \mathbb{E}\|\nabla F(\mathbf{x})\|^2$$

$$+ mK \sum_{m'=1}^{m-1} \sum_{k'=0}^{K-1} \mathbb{E}\|\nabla F(\mathbf{x})\|^2 + pMK \sum_{p'=0}^{p-1} \sum_{m'=1}^{M} \sum_{k'=0}^{K-1} \mathbb{E}\|\nabla F(\mathbf{x})\|^2$$

$$+ pMK \sum_{g'=1}^{g-1} \sum_{p'=0}^{P-1} \sum_{m'=1}^{M} \sum_{k'=0}^{K-1} \mathbb{E}\|\nabla F(\mathbf{x})\|^2 \Bigg)$$

$$\leq 20\eta^2 GPMK^2\sigma^2 + 20\eta^2 GPM^2K^2\sigma^2 + 20\eta^2 GP^2M^2K^2\sigma^2 + 20\eta^2 G^2P^2M^2K^2\sigma^2$$
$$+ 20\eta^2 GPMK^3\zeta^2 + 20\eta^2 GPM^2K^3\zeta^2 + 20\eta^2 GPMK^3\delta^2$$
$$+ 20\eta^2 GPM^3K^3\delta^2 + 20\eta^2 GP^3M^3K^3\delta^2 + 20\eta^2 G^2P^3M^3K^3\delta^2$$
$$+ 20\eta^2 GPMK^3\|\nabla F(\mathbf{x})\|^2 + 20\eta^2 GPM^3K^3\|\nabla F(\mathbf{x})\|^2 + 20\eta^2 GP^3M^3K^3\|\nabla F(\mathbf{x})\|^2$$
$$+ 20\eta^2 G^3P^3M^3K^3\|\nabla F(\mathbf{x})\|^2 + 20L^2G^2P^2M^2K^2\eta^2 E_r$$

Let $c_1 = \frac{1}{1 - 20L^2G^2P^2M^2K^2\eta^2}$, we have

$$E_r \leq 80c_1\eta^2 G^2P^2M^2K^2\sigma^2 + 40c_1\eta^2 GPM^2K^3\zeta^2 + 80c_1\eta^2 G^2P^3M^3K^3\delta^2$$
$$+ 80c_1\eta^2 G^3P^3M^3K^3\|\nabla F(\mathbf{x})\|^2$$

After substituting $E_r$ into $\mathbb{E}[F(\mathbf{x}^{(r+1)}) - F(\mathbf{x}^{(r)})]$, we can obtain

$$\mathbb{E}[F(\mathbf{x}^{(r+1)}) - F(\mathbf{x}^{(r)})]$$
$$\leq -\eta GPMK(\frac{1}{2} - 40c_1\eta^2 L^2G^2P^2M^2K^2)\mathbb{E}[\|\nabla F(\mathbf{x}^{(r)})\|^2] + \eta^2 LGPMK\sigma^2$$
$$+ 40c_1\eta^3 L^2GP^2M^2K^2\sigma^2 + 20c_1\eta^3 L^2GPM^2K^3\zeta^2 + 40c_1\eta^3 L^2G^2P^3M^3K^3\delta^2.$$

Let $c_2 = \frac{1}{\frac{1}{2} - 40c_1\eta^2 L^2G^2P^2M^2K^2}$, then

$$\frac{1}{R} \sum_{r=0}^{R-1} \mathbb{E}[\|\nabla F(\mathbf{x}^{(r)})\|^2] \leq \frac{c_2\left(F(\mathbf{x}^{(0)}) - F(\mathbf{x}^{(R)})\right)}{\eta RGPMK} + c_2\eta L\sigma^2 + 40c_1c_2\eta^2 L^2GPMK\sigma^2$$
$$+ 20c_1c_2\eta^2 L^2MK^2\zeta^2 + 40c_1c_2\eta^2 L^2GP^2M^2K\delta^2.$$

Let $\tilde{\eta} = \eta GPMK$, and $\tilde{\eta} \leq \frac{1}{10L}$, we have

$$\mathbb{E}[\|\nabla F(\bar{x}^{(R)})\|^2] \lesssim \frac{A}{\tilde{\eta}R} + \frac{L\tilde{\eta}\sigma^2}{GPMK} + \frac{L^2\tilde{\eta}^2\sigma^2}{GPMK} + \frac{L^2\tilde{\eta}^2\zeta^2}{G^2P^2M} + \frac{L^2\tilde{\eta}^2\delta^2}{G}$$

$$\mathbb{E}[\|\nabla F(\bar{x}^{(R)})\|^2] = \mathcal{O}\left( \frac{LA}{R} + \frac{(L\sigma^2 A)^{1/2}}{\sqrt{GPMKR}} + \frac{(L^2A^2\sigma^2)^{1/3}}{(GPMKR^2)^{1/3}} + \frac{(L^2A^2\zeta^2)^{1/3}}{(G^2P^2MR^2)^{1/3}} \right.$$
$$\left. + \frac{(L^2A^2\delta^2)^{1/3}}{(GR^2)^{1/3}} \right)$$

where $\bar{x}^R$ is defined as a model uniformly sampled from the $x^{(0)}, \ldots, x^{(R-1)}$ of previous iterations, and $\lesssim$ hides universal constants.

# F  PROOF OF THEOREM 2

In the following proof, we consider the partial client participation setting, specifically, selecting partial clients without replacement. So we assume that $\pi 1 = \{\pi 1(1), \pi 1(2), \ldots, \pi 1(G)\}$ is a permutation of $\{1, 2, \ldots, G\}$, $\pi 2 = \{\pi 2(1), \pi 2(2), \ldots, \pi 2(M)\}$ is a permutation of $\{1, 2, \ldots, M\}$ in a certain training round. And only the first $S_2$ selected clients $\pi 2 = \{\pi 2(1), \pi 2(2), \ldots, \pi 2(S_2)\}$ within the first $S_1$ selected groups $\{\pi 1(1), \pi 1(2), \ldots, \pi 1(S_1)\}$ will participate in this round. Unless otherwise stated, we use $\mathbb{E}[\cdot]$ to represent the expectation with respect to both types of randomness (i.e., sampling data samples $\xi$ and sampling clients $\pi_2$).

## F.1  FIND THE PER-ROUND RECURSION

**Lemma 2.** *Let Assumptions 1, 2 hold. If the learning rate satisfies $\eta \leq \frac{1}{5Lc_0 S_1 S_2 PK}$, then*

$$\mathbb{E}\left[F(\mathbf{x}^{(r+1)}) - F(\mathbf{x}^{(r)})\right]$$

$$\leq -\eta c_0 S_1 S_2 PK \left(\frac{1}{2} - \frac{5}{2}\eta Lc_0 S_1 S_2 PK\right) \mathbb{E}[\|\nabla F(\mathbf{x}^{(r)})\|^2]$$

$$+ \frac{5}{2}\eta^2 L c_0{}^2 S_1 S_2 PK\sigma^2 + \frac{5}{2}\eta^2 L c_0{}^2 S_1 P^2 S_2{}^2 K^2\zeta^2 + \frac{5}{2}\eta^2 L c_0{}^2 S_1{}^2 P^2 S_2 K^2\delta^2$$

$$+ \eta L^2 c_0 \sum_{s_1=1}^{S_1}\sum_{p=0}^{P-1}\sum_{s_2=1}^{S_2}\sum_{k=0}^{K-1} \mathbb{E}[\|\mathbf{x}_{s_1,p,s_2,k}^{(r)} - \mathbf{x}^{(r)}\|^2].$$

*where $c_0$ is a topology-dependent coefficient that takes values of $1/(S_1 S_2)$, $1/S_1$, $1/S_2$, and $1$ for the Star-Star, Star-Ring, Ring-Star, and Ring-Ring topologies, respectively.*

*Proof.*

$$\mathbb{E}\left[F(\mathbf{x} + \Delta\mathbf{x}) - F(\mathbf{x})\right] \leq \mathbb{E}\left[\langle\nabla F(\mathbf{x}), \Delta\mathbf{x}\rangle\right] + \frac{1}{2}L\mathbb{E}[\|\Delta\mathbf{x}\|^2].$$

The model updates within a single global round for the four topologies can be represented by a unified format,

$$\Delta\mathbf{x}^{(r)} = \mathbf{x}^{(r+1)} - \mathbf{x}^{(r)} = -\eta c_0 \sum_{s_1=1}^{S_1}\sum_{p=0}^{P-1}\sum_{s_2=1}^{S_2}\sum_{k=0}^{K-1} \mathbf{g}_{\pi_1(s_1),p,\pi_2(s_2),k}^{(r)}.$$

After substituting the overall updates $\Delta\mathbf{x}$, we can get

$$\mathbb{E}\left[\langle\nabla F(\mathbf{x}^{(r)}), \Delta\mathbf{x}^{(r)}\rangle\right]$$

$$= \mathbb{E}\left[\left\langle\nabla F(\mathbf{x}^{(r)}), -\eta c_0 \sum_{s_1=1}^{S_1}\sum_{p=0}^{P-1}\sum_{s_2=1}^{S_2}\sum_{k=0}^{K-1} \mathbf{g}_{\pi_1(s_1),p,\pi_2(s_2),k}\right\rangle\right]$$

$$= \mathbb{E}\left[\left\langle\nabla F(\mathbf{x}^{(r)}), -\eta c_0 \sum_{s_1=1}^{S_1}\sum_{p=0}^{P-1}\sum_{s_2=1}^{S_2}\sum_{k=0}^{K-1} \nabla F_{\pi_1(s_1),\pi_2(s_2)}(\mathbf{x}_{s_1,p,s_2,k}^{(r)})\right\rangle\right]$$

$$= \mathbb{E}\left[\left\langle\nabla F(\mathbf{x}^{(r)}), -\eta c_0 \sum_{s_1=1}^{S_1}\sum_{p=0}^{P-1}\sum_{s_2=1}^{S_2}\sum_{k=0}^{K-1} \left(\nabla F_{\pi_1(s_1),\pi_2(s_2)}(\mathbf{x}_{s_1,p,s_2,k}^{(r)}) - \nabla F(\mathbf{x}^{(r)})\right)\right\rangle\right]$$

$$+ \mathbb{E}\left[\left\langle\nabla F(\mathbf{x}^{(r)}), -\eta c_0 \sum_{s_1=1}^{S_1}\sum_{p=0}^{P-1}\sum_{s_2=1}^{S_2}\sum_{k=0}^{K-1} \nabla F(\mathbf{x}^{(r)})\right\rangle\right]$$

$$= -\eta c_0 S_1 S_2 PK\mathbb{E}[\|\nabla F(\mathbf{x}^{(r)})\|^2] + \eta c_0 S_1 S_2 PK\mathbb{E}\left[\left\langle\nabla F(\mathbf{x}^{(r)}),\right.\right.$$

$$c_0 \sum_{s_1=1}^{S_1}\sum_{p=0}^{P-1}\sum_{s_2=1}^{S_2}\sum_{k=0}^{K-1} \left(\nabla F_{\pi_1(s_1),\pi_2(s_2)}(\mathbf{x}_{s_1,p,s_2,k}^{(r)}) - \nabla F_{\pi_1(s_1),\pi_2(s_2)}(\mathbf{x}^{(r)})\right)\right\rangle\Bigg]$$

$$\leq -\eta c_0 S_1 S_2 PK \mathbb{E}[\|\nabla F(\mathbf{x}^{(r)})\|^2] + \frac{1}{2}\eta c_0 S_1 S_2 PK \mathbb{E}[\|\nabla F(\mathbf{x}^{(r)})\|^2]$$

$$+ \frac{1}{2}\eta c_0 S_1 S_2 PK \mathbb{E}\left[\left\|c_0 \sum_{s_1=1}^{S_1}\sum_{p=0}^{P-1}\sum_{s_2=1}^{S_2}\sum_{k=0}^{K-1}\left(\nabla F_{\pi_1(s_1),p,\pi_2(s_2),k}(\mathbf{x}^{(r)}_{s_1,p,s_2,k})\right.\right.\right.$$

$$\left.\left.\left. -\nabla F_{\pi_1(s_1),p,\pi_2(s_2),k}(\mathbf{x}^{(r)})\right)\right\|^2\right]$$

$$= -\frac{1}{2}\eta c_0 S_1 S_2 PK \mathbb{E}[\|\nabla F(\mathbf{x}^{(r)})\|^2]$$

$$+ \frac{1}{2}\eta c_0 S_1 S_2 PK \mathbb{E}\left\|c_0 \sum_{s_1=1}^{S_1}\sum_{p=0}^{P-1}\sum_{s_2=1}^{S_2}\sum_{k=0}^{K-1}\left(\nabla F_{\pi_1(s_1),\pi_2(s_2)}(\mathbf{x}^{(r)}_{s_1,p,s_2,k})-\nabla F_{\pi_1(s_1),\pi_2(s_2)}(\mathbf{x}^{(r)})\right)\right\|^2$$

$$\leq -\frac{1}{2}\eta c_0 S_1 S_2 PK \mathbb{E}[\|\nabla F(\mathbf{x}^{(r)})\|^2] + \frac{1}{2}\eta L^2 c_0 \sum_{s_1=1}^{S_1}\sum_{p=0}^{P-1}\sum_{s_2=1}^{S_2}\sum_{k=0}^{K-1}\mathbb{E}\left[\left\|\mathbf{x}^{(r)}_{s_1,p,s_2,k}-\mathbf{x}^{(r)}\right\|^2\right]$$

For the remaining term, we have

$$\frac{1}{2}L\mathbb{E}[\|\Delta\mathbf{x}\|^2]$$

$$= \frac{1}{2}L\mathbb{E}\left[\left\|\eta c_0 \sum_{s_1=1}^{S_1}\sum_{p=0}^{P-1}\sum_{s_2=1}^{S_2}\sum_{k=0}^{K-1}\mathbf{g}^{(r)}_{\pi_1(s_1),p,\pi_2(s_2),k}\right\|^2\right]$$

$$\leq \frac{5}{2}\eta^2 L\mathbb{E}\left[\left\|c_0 \sum_{s_1=1}^{S_1}\sum_{p=0}^{P-1}\sum_{s_2=1}^{S_2}\sum_{k=0}^{K-1}\left(\mathbf{g}^{(r)}_{\pi_1(s_1),p,\pi_2(s_2),k}-\nabla F_{\pi_1(s_1),\pi_2(s_2)}(\mathbf{x}^{(r)}_{s_1,p,s_2,k})\right)\right\|^2\right]$$

$$+ \frac{5}{2}L\mathbb{E}\left[\left\|c_0 \sum_{s_1=1}^{S_1}\sum_{p=0}^{P-1}\sum_{s_2=1}^{S_2}\sum_{k=0}^{K-1}\left(\nabla F_{\pi_1(s_1),\pi_2(s_2)}(\mathbf{x}^{(r)}_{s_1,p,s_2,k})-\nabla F_{\pi_1(s_1),\pi_2(s_2)}(\mathbf{x}^{(r)})\right)\right\|^2\right]$$

$$+ \frac{5}{2}\eta^2 L\mathbb{E}\left[\left\|c_0 \sum_{s_1=1}^{S_1}\sum_{p=0}^{P-1}\sum_{s_2=1}^{S_2}\sum_{k=0}^{K-1}\left(\nabla F_{\pi_1(s_1),\pi_2(s_2)}(\mathbf{x}^{(r)})-\nabla F_{\pi_1(s_1),\pi_2(s_2)}(\mathbf{x}^{(r)})\right)\right\|^2\right]$$

$$+ \frac{5}{2}\eta^2 L\mathbb{E}\left[\left\|c_0 \sum_{s_1=1}^{S_1}\sum_{p=0}^{P-1}\sum_{s_2=1}^{S_2}\sum_{k=0}^{K-1}\left(\nabla F_{\pi_1(s_1),\pi_2(s_2)}(\mathbf{x}^{(r)})-\nabla F(\mathbf{x}^{(r)})\right)\right\|^2\right]$$

$$+ \frac{5}{2}\eta^2 L\mathbb{E}\left[\left\|c_0 \sum_{s_1=1}^{S_1}\sum_{p=0}^{P-1}\sum_{s_2=1}^{S_2}\sum_{k=0}^{K-1}\nabla F(\mathbf{x}^{(r)})\right\|^2\right]$$

$$\leq \frac{5}{2}\eta^2 Lc_0{}^2 S_1 S_2 PK\sigma^2$$

$$+ \frac{5}{2}\eta^2 L^3 c_0 S_1 S_2 PK \sum_{s_1=1}^{S_1}\sum_{p=0}^{P-1}\sum_{s_2=1}^{S_2}\sum_{k=0}^{K-1}\mathbb{E}[\|\mathbf{x}^{(r)}_{s_1,p,s_2,k}-\mathbf{x}^{(r)}\|^2]$$

$$+ \frac{5}{2}\eta^2 Lc_0{}^2 S_1 P^2 S_2{}^2 K^2\zeta^2$$

$$+ \frac{5}{2}\eta^2 Lc_0{}^2 S_1{}^2 P^2 S_2 K^2\delta^2$$

$$+ \frac{5}{2}\eta^2 Lc_0{}^2 S_1{}^2 S_2{}^2 P^2 K^2\mathbb{E}[\|\nabla F(\mathbf{x}^{(r)})\|^2].$$

Let $\eta \leq \frac{1}{5Lc_0 S_1 S_2 PK}$, we have

$$\mathbb{E}\left[F(\mathbf{x}^{(r+1)})-F(\mathbf{x}^{(r)})\right]$$

$$\leq \mathbb{E}\left[\langle\nabla F(\mathbf{x}^{(r)}),\Delta\mathbf{x}^{(r)}\rangle\right] + \frac{1}{2}L\mathbb{E}[\|\Delta\mathbf{x}^{(r)}\|^2]$$

$$= -\eta c_0 S_1 S_2 PK \left( \frac{1}{2} - \frac{5}{2} \eta L c_0 S_1 S_2 PK \right) \mathbb{E}[\|\nabla F(\mathbf{x}^{(r)})\|^2]$$

$$+ \frac{5}{2} \eta^2 L c_0{}^2 S_1 S_2 PK \sigma^2 + \frac{5}{2} \eta^2 L c_0{}^2 S_1 P^2 S_2{}^2 K^2 \zeta^2 + \frac{5}{2} \eta^2 L c_0{}^2 S_1{}^2 P^2 S_2 K^2 \delta^2$$

$$+ \eta L^2 c_0 \left( \frac{1}{2} + \frac{5}{2} \eta L c_0 S_1 S_2 PK \right) \sum_{s_1=1}^{S_1} \sum_{p=0}^{P-1} \sum_{s_2=1}^{S_2} \sum_{k=0}^{K-1} \mathbb{E}[\|\mathbf{x}_{s_1,p,s_2,k}^{(r)} - \mathbf{x}^{(r)}\|^2]$$

$$\leq -\eta c_0 S_1 S_2 PK \left( \frac{1}{2} - \frac{5}{2} \eta L c_0 S_1 S_2 PK \right) \mathbb{E}[\|\nabla F(\mathbf{x}^{(r)})\|^2]$$

$$+ \frac{5}{2} \eta^2 L c_0{}^2 S_1 S_2 PK \sigma^2 + \frac{5}{2} \eta^2 L c_0{}^2 S_1 P^2 S_2{}^2 K^2 \zeta^2 + \frac{5}{2} \eta^2 L c_0{}^2 S_1{}^2 P^2 S_2 K^2 \delta^2$$

$$+ \eta L^2 c_0 \sum_{s_1=1}^{S_1} \sum_{p=0}^{P-1} \sum_{s_2=1}^{S_2} \sum_{k=0}^{K-1} \mathbb{E}[\|\mathbf{x}_{s_1,p,s_2,k}^{(r)} - \mathbf{x}^{(r)}\|^2].$$

### F.2 PROOF OF THEOREM 2 FOR THE STAR-STAR CASE

For Star-Star,

$$\mathbf{x}_{s_1,p,s_2,k}^{(r)} - \mathbf{x}^{(r)} = -\eta \sum_{k'=0}^{k-1} \mathbf{g}_{\pi_1(s_1),p,\pi_2(s_2),k'} - \eta \sum_{p'=0}^{p-1} \frac{1}{S_2} \sum_{s_2'=1}^{S_2} \sum_{k'=0}^{K-1} \mathbf{g}_{\pi_1(s_1),p',\pi_2(s_2'),k'}.$$

We first bound $\mathbb{E}[\|\mathbf{x}_{s_1,p,s_2,k}^{(r)} - \mathbf{x}^{(r)}\|^2]$.

$$\mathbb{E}[\|\mathbf{x}_{s_1,p,s_2,k}^{(r)} - \mathbf{x}^{(r)}\|^2]$$

$$= \mathbb{E}\left[ \left\| \eta \sum_{k'=0}^{k-1} \mathbf{g}_{\pi_1(s_1),p,\pi_2(s_2),k'} + \eta \sum_{p'=0}^{p-1} \frac{1}{S_2} \sum_{s_2'=1}^{S_2} \sum_{k'=0}^{K-1} \mathbf{g}_{\pi_1(s_1),p',\pi_2(s_2'),k'} \right\|^2 \right]$$

$$\leq 10\eta^2 \mathbb{E}\left[ \left\| \sum_{k'=0}^{k-1} \left( \mathbf{g}_{\pi_1(s_1),p,\pi_2(s_2),k'} - \nabla F_{\pi_1(s_1),\pi_2(s_2)}(\mathbf{x}_{s_1,p',s_2,k'}^{(r)}) \right) \right\|^2 \right]$$

$$+ 10\eta^2 \mathbb{E}\left[ \left\| \sum_{p'=0}^{p-1} \frac{1}{S_2} \sum_{s_2'=1}^{S_2} \sum_{k'=0}^{K-1} \left( \mathbf{g}_{\pi_1(s_1),p',\pi_2(s_2'),k'} - \nabla F_{\pi_1(s_1),\pi_2(s_2')}(\mathbf{x}_{s_1,p',s_2',k'}^{(r)}) \right) \right\|^2 \right]$$

$$+ 10\eta^2 \mathbb{E}\left[ \left\| \sum_{k'=0}^{k-1} \left( \nabla F_{\pi_1(s_1),\pi_2(s_2)}(\mathbf{x}_{s_1,p',s_2,k'}^{(r)}) - \nabla F_{\pi_1(s_1),\pi_2(s_2)}(\mathbf{x}^{(r)}) \right) \right\|^2 \right]$$

$$+ 10\eta^2 \mathbb{E}\left[ \left\| \sum_{p'=0}^{p-1} \frac{1}{S_2} \sum_{s_2'=1}^{S_2} \sum_{k'=0}^{K-1} \left( \nabla F_{\pi_1(s_1),\pi_2(s_2)}(\mathbf{x}_{s_1,p',s_2,k'}^{(r)}) - \nabla F_{\pi_1(s_1),\pi_2(s_2)}(\mathbf{x}^{(r)}) \right) \right\|^2 \right]$$

$$+ 10\eta^2 \mathbb{E}\left[ \left\| \sum_{k'=0}^{k-1} \left( \nabla F_{\pi_1(s_1),\pi_2(s_2)}(\mathbf{x}^{(r)}) - \nabla F_{\pi_1(s_1)}(\mathbf{x}^{(r)}) \right) \right\|^2 \right]$$

$$+ 10\eta^2 \mathbb{E}\left[ \left\| \sum_{p'=0}^{p-1} \frac{1}{S_2} \sum_{s_2'=1}^{S_2} \sum_{k'=0}^{K-1} \left( \nabla F_{\pi_1(s_1),\pi_2(s_2)}(\mathbf{x}^{(r)}) - \nabla F_{\pi_1(s_1)}(\mathbf{x}^{(r)}) \right) \right\|^2 \right]$$

$$+ 10\eta^2 \mathbb{E}\left[ \left\| \sum_{k'=0}^{k-1} \left( \nabla F_{\pi_1(s_1)}(\mathbf{x}^{(r)}) - \nabla F(\mathbf{x}^{(r)}) \right) \right\|^2 \right]$$

$$+ 10\eta^2 \mathbb{E}\left[\left\|\sum_{p'=0}^{p-1} \frac{1}{S_2} \sum_{s_2'=1}^{S_2} \sum_{k'=0}^{K-1} \left(\nabla F_{\pi_1(s_1)}(\mathbf{x}^{(r)}) - \nabla F(\mathbf{x}^{(r)})\right)\right\|^2\right]$$

$$+ 10\eta^2 \mathbb{E}\left[\left\|\sum_{k'=0}^{k-1} \nabla F(\mathbf{x}^{(r)})\right\|^2\right]$$

$$+ 10\eta^2 \mathbb{E}\left[\left\|\sum_{p'=0}^{p-1} \frac{1}{S_2} \sum_{s_2'=1}^{S_2} \sum_{k'=0}^{K-1} \nabla F(\mathbf{x}^{(r)})\right\|^2\right]$$

$$\leq 10\eta^2 k\sigma^2 + 10\eta^2 p\frac{1}{S_2}K\sigma^2$$

$$+ 10\eta^2 k \sum_{k'=0}^{k-1} L^2 \mathbb{E}[\|\mathbf{x}_{s_1,p,s_2,k'}^{(r)} - \mathbf{x}^{(r)}\|^2] + 10\eta^2 pK \sum_{p'=0}^{p-1} \frac{1}{S_2} \sum_{s_2'=1}^{S_2} \sum_{k'=0}^{k-1} L^2 \mathbb{E}[\|\mathbf{x}_{s_1,p',s_2',k'}^{(r)} - \mathbf{x}^{(r)}\|^2]$$

$$+ 10\eta^2 k^2\zeta^2 + 10\eta^2 p^2 \frac{1}{S_2}K^2\zeta^2$$

$$+ 10\eta^2 k^2\delta^2 + 10\eta^2 p^2 K^2\delta^2$$

$$+ 10\eta^2 k^2 \mathbb{E}[\|\nabla F(\mathbf{x}^{(r)})\|^2] + 10\eta^2 p^2 K^2 \mathbb{E}[\|\nabla F(\mathbf{x}^{(r)})\|^2].$$

$$\sum_{s_1=1}^{S_1} \sum_{p=0}^{P-1} \sum_{s_2=1}^{S_2} \sum_{k=0}^{K-1} \mathbb{E}[\|\mathbf{x}_{s_1,p,s_2,k}^{(r)} - \mathbf{x}^{(r)}\|^2]$$

$$\leq 10\eta^2 S_1 P S_2 K^2 \sigma^2 + 10\eta^2 S_1 P^2 K^2 \sigma^2 + 20\eta^2 L^2 P^2 K^2 \sum_{s_1=1}^{S_1} \sum_{p=0}^{P-1} \sum_{s_2=1}^{S_2} \sum_{k=0}^{K-1} \mathbb{E}[\|\mathbf{x}_{s_1,p,s_2,k}^{(r)} - \mathbf{x}^{(r)}\|^2]$$

$$+ 10\eta^2 S_1 P S_2 K^3 \zeta^2 + 10\eta^2 S_1 P^3 K^3 \zeta^2 + 20\eta^2 S_1 P^3 S_2 K^3 \delta^2$$

$$+ 20\eta^2 S_1 P^3 S_2 K^3 \mathbb{E}[\|\nabla F(\mathbf{x}^{(r)})\|^2].$$

Let $c_1 = \frac{1}{1 - 20\eta^2 L^2 P^2 K^2}$,

$$\sum_{s_1=1}^{S_1} \sum_{p=0}^{P-1} \sum_{s_2=1}^{S_2} \sum_{k=0}^{K-1} \mathbb{E}[\|\mathbf{x}_{s_1,p,s_2,k}^{(r)} - \mathbf{x}^{(r)}\|^2]$$

$$\leq 10c_1\eta^2 S_1 P S_2 K^2 \sigma^2 + 10c_1\eta^2 S_1 P^2 K^2 \sigma^2$$

$$+ 10c_1\eta^2 S_1 P S_2 K^3 \zeta^2 + 10c_1\eta^2 S_1 P^3 K^3 \zeta^2 + 20c_1\eta^2 S_1 P^3 S_2 K^3 \delta^2$$

$$+ 20c_1\eta^2 S_1 P^3 S_2 K^3 \mathbb{E}[\|\nabla F(\mathbf{x}^{(r)})\|^2]$$

$$\mathbb{E}\left[F(\mathbf{x}^{(r+1)}) - F(\mathbf{x}^{(r)})\right]$$

$$\leq -\eta PK \left(\frac{1}{2} - \frac{1}{2}\eta LPK - 20c_1\eta^2 L^2 P^2 K^2\right) \mathbb{E}[\|\nabla F(\mathbf{x}^{(r)})\|^2]$$

$$+ \frac{5}{2}\eta^2 L \frac{1}{S_1} P \frac{1}{S_2} K\sigma^2 + 10c_1\eta^3 L^2 PK^2\sigma^2 + 10c_1\eta^3 L^2 P^2 \frac{1}{S_2} K^2\sigma^2$$

$$+ \frac{5}{2}\eta^2 LP^2 \frac{1}{S_2} K^2\zeta^2 + 10c_1\eta^3 L^2 PK^3\zeta^2 + 10c_1\eta^3 L^2 P^3 \frac{1}{S_2} K^3\zeta^2$$

$$+ \frac{5}{2}\eta^2 L \frac{1}{S_1} P^2 K^2\delta^2 + 20c_1\eta^3 L^2 P^3 K^3\delta^2$$

Let $c_2 = \frac{1}{\frac{1}{2} - \frac{5}{2}\eta LPK - 20c_1\eta^2 L^2 P^2 K^2}$,

$$\frac{1}{R}\sum_{r=0}^{R-1}\mathbb{E}[\|\nabla F(\mathbf{x}^{(r)})\|^2] \lesssim \frac{c_2(F(\mathbf{x}^{(0)}) - F(\mathbf{x}^{(R)}))}{\eta RPK} + c_2\eta L\frac{1}{S_1}\frac{1}{S_2}\sigma^2 + c_1c_2\eta^2 L^2 K\sigma^2$$

$$+ c_2\eta LP\frac{1}{S_2}K\zeta^2 + c_1c_2\eta^2 L^2 P\frac{1}{S_2}K\sigma^2 + c_1c_2\eta^2 L^2 K^2\zeta^2$$

$$+ c_2\eta L\frac{1}{S_1}PK\delta^2 + c_1c_2\eta^2 L^2 P^2\frac{1}{S_2}K^2\zeta^2 + c_1c_2\eta^2 L^2 P^2 K^2\delta^2.$$

Let $\tilde{\eta} = \eta PK$ and $\tilde{\eta} \leq \frac{1}{10L}$,

$$\mathbb{E}[\|\nabla F(\bar{x}^{(R)})\|^2] \lesssim \frac{F(\mathbf{x}^{(0)}) - F(\mathbf{x}^{(R)})}{\tilde{\eta}R} + \frac{L\tilde{\eta}\sigma^2}{S_1 PS_2 K} + \frac{L\tilde{\eta}\zeta^2}{S_2} + \frac{L\tilde{\eta}\delta^2}{S_1}$$

$$+ \frac{L^2\tilde{\eta}^2\sigma^2}{P^2 K} + \frac{L^2\tilde{\eta}^2\sigma^2}{PS_2 K} + \frac{L^2\tilde{\eta}^2\zeta^2}{P^2} + \frac{L^2\tilde{\eta}^2\zeta^2}{S_2} + L^2\tilde{\eta}^2\delta^2.$$

$$\mathbb{E}[\|\nabla F(\bar{x}^{(R)})\|^2] = \mathcal{O}\Bigg(\frac{LA}{R} + \frac{(LA\sigma^2)^{1/2}}{\sqrt{S_1 PS_2 KR}} + \frac{(LA\zeta^2)^{1/2}}{\sqrt{S_2 R}} + \frac{(LA\delta^2)^{1/2}}{\sqrt{S_1 R}}$$

$$+ \frac{(L^2 A^2\sigma^2)^{1/3}}{(P^2 KR^2)^{1/3}} + \frac{(L^2 A^2\sigma^2)^{1/3}}{(PS_2 KR^2)^{1/3}} + \frac{(L^2 A^2\zeta^2)^{1/3}}{(P^2 R^2)^{1/3}}$$

$$+ \frac{(L^2 A^2\zeta^2)^{1/3}}{(S_2 R^2)^{1/3}} + \frac{(L^2 A^2\delta^2)^{1/3}}{(R^2)^{1/3}}\Bigg)$$

where $\bar{x}^R$ is defined as a model uniformly sampled from the $x^{(0)}, \ldots, x^{(R-1)}$ of previous iterations, and $\lesssim$ hides universal constants.

### F.3 PROOF OF THEOREM 2 FOR THE STAR-RING CASE

For Star-Ring,

$$\mathbf{x}_{s_1,p,s_2,k}^{(r)} - \mathbf{x}^{(r)} = -\eta\sum_{k'=0}^{k-1}\mathbf{g}_{\pi_1(s_1),p,\pi_2(s_2),k'} - \eta\sum_{p'=0}^{p-1}\frac{1}{S_2}\sum_{s_2=1}^{S_2}\sum_{k'=0}^{K-1}\mathbf{g}_{\pi_1(s_1),p',\pi_2(s_2'),k'}$$

$$- \eta\sum_{p'=0}^{p-1}\frac{1}{S_2}\sum_{s_2'=1}^{S_2}\sum_{k'=0}^{K-1}\mathbf{g}_{\pi_1(s_1),p',\pi_2(s_2'),k'}.$$

We first bound $\mathbb{E}[\|\mathbf{x}_{s_1,p,s_2,k}^{(r)} - \mathbf{x}^{(r)}\|^2]$.

$$\mathbb{E}[\|\mathbf{x}_{s_1,p,s_2,k}^{(r)} - \mathbf{x}^{(r)}\|^2]$$

$$\leq 15\eta^2\mathbb{E}\left[\left\|\sum_{k'=0}^{k-1}\left(\mathbf{g}_{\pi_1(s_1),p,\pi_2(s_2),k'} - \nabla F_{\pi_1(s_1),\pi_2(s_2)}(\mathbf{x}_{s_1,p,s_2,k'}^{(r)})\right)\right\|^2\right]$$

$$+ 15\eta^2\mathbb{E}\left[\left\|\sum_{s_2'=1}^{s_2}\sum_{k'=0}^{K-1}\left(\mathbf{g}_{\pi_1(s_1),p,\pi_2(s_2'),k'} - \nabla F_{\pi_1(s_1),\pi_2(s_2')}(\mathbf{x}_{s_1,p,s_2',k'}^{(r)})\right)\right\|^2\right]$$

$$+ 15\eta^2\mathbb{E}\left[\left\|\sum_{p'=0}^{p-1}\sum_{s_2'=1}^{S_2}\sum_{k'=0}^{K-1}\left(\mathbf{g}_{\pi_1(s_1),p',\pi_2(s_2'),k'} - \nabla F_{\pi_1(s_1),\pi_2(s_2')}(\mathbf{x}_{s_1,p',s_2',k'}^{(r)})\right)\right\|^2\right]$$

$$+ 15\eta^2\mathbb{E}\left[\left\|\sum_{k'=0}^{k-1}\left(\nabla F_{\pi_1(s_1),\pi_2(s_2)}(\mathbf{x}_{s_1,p,s_2,k'}^{(r)}) - \nabla F_{\pi_1(s_1),\pi_2(s_2)}(\mathbf{x}^{(r)})\right)\right\|^2\right]$$

$$+ 15\eta^2 \mathbb{E}\left[\left\|\sum_{s_2'=1}^{s_2}\sum_{k'=0}^{K-1}\left(\nabla F_{\pi_1(s_1),\pi_2(s_2')}(\mathbf{x}_{s_1,p,s_2',k'}^{(r)}) - \nabla F_{\pi_1(s_1),\pi_2(s_2')}(\mathbf{x}^{(r)})\right)\right\|^2\right]$$

$$+ 15\eta^2 \mathbb{E}\left[\left\|\sum_{p'=0}^{p-1}\sum_{s_2'=1}^{S_2}\sum_{k'=0}^{K-1}\left(\nabla F_{\pi_1(s_1),\pi_2(s_2')}(\mathbf{x}_{s_1,p',s_2',k'}^{(r)}) - \nabla F_{\pi_1(s_1),\pi_2(s_2')}(\mathbf{x}^{(r)})\right)\right\|^2\right]$$

$$+ 15\eta^2 \mathbb{E}\left[\left\|\sum_{k'=0}^{k-1}\left(\nabla F_{\pi_1(s_1),\pi_2(s_2)}(\mathbf{x}^{(r)}) - \nabla F_{\pi_1(s_1)}(\mathbf{x}^{(r)})\right)\right\|^2\right]$$

$$+ 15\eta^2 \mathbb{E}\left[\left\|\sum_{s_2'=1}^{s_2}\sum_{k'=0}^{K-1}\left(\nabla F_{\pi_1(s_1),\pi_2(s_2')}(\mathbf{x}^{(r)}) - \nabla F_{\pi_1(s_1)}(\mathbf{x}^{(r)})\right)\right\|^2\right]$$

$$+ 15\eta^2 \mathbb{E}\left[\left\|\sum_{p'=0}^{p-1}\sum_{s_2'=1}^{S_2}\sum_{k'=0}^{K-1}\left(\nabla F_{\pi_1(s_1),\pi_2(s_2')}(\mathbf{x}^{(r)}) - \nabla F_{\pi_1(s_1)}(\mathbf{x}^{(r)})\right)\right\|^2\right]$$

$$+ 15\eta^2 \mathbb{E}\left[\left\|\sum_{k'=0}^{k-1}\left(\nabla F_{\pi_1(s_1)}(\mathbf{x}^{(r)}) - \nabla F(\mathbf{x}^{(r)})\right)\right\|^2\right]$$

$$+ 15\eta^2 \mathbb{E}\left[\left\|\sum_{s_2'=1}^{s_2}\sum_{k'=0}^{K-1}\left(\nabla F_{\pi_1(s_1)}(\mathbf{x}^{(r)}) - \nabla F(\mathbf{x}^{(r)})\right)\right\|^2\right]$$

$$+ 15\eta^2 \mathbb{E}\left[\left\|\sum_{p'=0}^{p-1}\sum_{s_2'=1}^{S_2}\sum_{k'=0}^{K-1}\left(\nabla F_{\pi_1(s_1)}(\mathbf{x}^{(r)}) - \nabla F(\mathbf{x}^{(r)})\right)\right\|^2\right]$$

$$+ 15\eta^2 \mathbb{E}\left[\left\|\sum_{k'=0}^{k-1}\nabla F(\mathbf{x}^{(r)})\right\|^2\right]$$

$$+ 15\eta^2 \mathbb{E}\left[\left\|\sum_{s_2'=1}^{s_2}\sum_{k'=0}^{K-1}\nabla F(\mathbf{x}^{(r)})\right\|^2\right]$$

$$+ 15\eta^2 \mathbb{E}\left[\left\|\sum_{p'=0}^{p-1}\sum_{s_2'=1}^{S_2}\sum_{k'=0}^{K-1}\nabla F(\mathbf{x}^{(r)})\right\|^2\right]$$

$$\leq 15\eta^2 k\sigma^2 + 15\eta^2 s_2 K\sigma^2 + 15\eta^2 p S_2 K\sigma^2$$

$$+ 15\eta^2 L^2 k \sum_{k'=0}^{k-1}\mathbb{E}[\|\mathbf{x}_{s_1,p,s_2,k'}^{(r)} - \mathbf{x}^{(r)}\|^2]$$

$$+ 15\eta^2 L^2 s_2 K \sum_{s_2'=1}^{s_2}\sum_{k'=0}^{k-1}\mathbb{E}[\|\mathbf{x}_{s_1,p,s_2',k'}^{(r)} - \mathbf{x}^{(r)}\|^2]$$

$$+ 15\eta^2 L^2 p s_2 K \sum_{p'=0}^{p-1}\sum_{s_2'=1}^{S_2}\sum_{k'=0}^{K-1}\mathbb{E}[\|\mathbf{x}_{s_1,p',s_2',k'}^{(r)} - \mathbf{x}^{(r)}\|^2]$$

$$+ 15\eta^2 k^2\zeta^2 + 15\eta^2 s_2 K^2\zeta^2 + 15\eta^2 p^2 S_2 K^2\zeta^2$$

$$+ 15\eta^2 k^2\zeta_2^2 + 15\eta^2 s_2^2 K^2\delta^2 + 15\eta^2 p^2 s_2^2 K^2\delta^2$$

$$+ 15\eta^2 K^2 \mathbb{E}[\|\nabla F(\mathbf{x}^{(r)})\|^2] + 15\eta^2 s_2^2 K^2 \mathbb{E}[\|\nabla F(\mathbf{x}^{(r)})\|^2] + 15\eta^2 p^2 S_2^2 K^2 \mathbb{E}[\|\nabla F(\mathbf{x}^{(r)})\|^2].$$

$$\sum_{s_1=1}^{S_1} \sum_{p=0}^{P-1} \sum_{s_2=1}^{S_2} \sum_{k=0}^{K-1} \mathbb{E}[\|\mathbf{x}_{s_1,p,s_2,k}^{(r)} - \mathbf{x}^{(r)}\|^2]$$

$$\leq 45\eta^2 S_1 P^2 S_2^2 K^2 \sigma^2$$

$$+ 45\eta^2 L^2 P^2 S_2^2 K^2 \sum_{s_1=1}^{S_1} \sum_{p=0}^{P-1} \sum_{s_2=1}^{S_2} \sum_{k=0}^{K-1} \mathbb{E}[\|\mathbf{x}_{s_1,p,s_2,k}^{(r)} - \mathbf{x}^{(r)}\|^2]$$

$$+ 45\eta^2 S_1 P^3 S_2^2 K^3 \zeta^2$$

$$+ 45\eta^2 S_1 P^3 S_2^3 K^3 \delta^2$$

$$+ 45\eta^2 S_1 P^3 S_2^3 K^3 \mathbb{E}[\|\nabla F(\mathbf{x}^{(r)})\|^2].$$

Let $c_1 = \frac{1}{1 - 45\eta^2 L^2 P^2 S_2^2 K^2}$,

$$\sum_{s_1=1}^{S_1} \sum_{p=0}^{P_1} \sum_{s_2=1}^{S_2} \sum_{k=0}^{K-1} \mathbb{E}[\|\mathbf{x}_{s_1,p,s_2,k}^{(r)} - \mathbf{x}^{(r)}\|^2]$$

$$\leq 45c_1\eta^2 S_1 P^2 S_2^2 K^2 \sigma^2 + 45c_1\eta^2 S_1 P^3 S_2^2 K^3 \zeta^2 + 45c_1\eta^2 S_1 P^3 S_2^3 K^3 \delta^2$$

$$+ 45c_1\eta^2 S_1 P^3 S_2^3 K^3 \mathbb{E}[\|\nabla F(\mathbf{x}^{(r)})\|^2].$$

Let $c_2 = \frac{1}{\frac{1}{2} - \frac{5}{2}\eta L P S_2 K - 45c_1\eta^2 L^2 P^2 S_2^2 K^2}$,

$$\frac{1}{R} \sum_{k=0}^{R-1} \mathbb{E}[\|\nabla F(\mathbf{x}^{(r)})\|^2] \leq c_2 \frac{F(\mathbf{x}^{(0)}) - F(\mathbf{x}^{(R)})}{\eta R P S_2 K}$$

$$+ \frac{5}{2}c_2\eta L \frac{1}{S_1}\sigma^2 + 45c_1c_2\eta^2 L^2 P S_2 K \sigma^2$$

$$+ \frac{5}{2}c_2\eta L P K \zeta^2 + 45c_1c_2\eta^2 L^2 P^2 S_2 K^2 \zeta^2$$

$$+ \frac{5}{2}c_2\eta L \frac{1}{S_1} P S_2 K \delta^2 + 45c_1c_2\eta^2 L^2 P^2 S_2^2 K^2 \delta^2.$$

Let $\tilde{\eta} = \eta P S_2 K$ and $\tilde{\eta} \leq \frac{1}{45L}$,

$$\mathbb{E}[\|\nabla F(\bar{x}^{(R)})\|^2] \lesssim \frac{F(\mathbf{x}^{(0)}) - F(\mathbf{x}^{(R)})}{\tilde{\eta} R} + \frac{L\tilde{\eta}\sigma^2}{S_1 P S_2 K} + \frac{L\tilde{\eta}\zeta^2}{S_2} + \frac{L\tilde{\eta}\delta^2}{S_1}$$

$$+ \frac{L^2\tilde{\eta}^2\sigma^2}{P S_2 K} + \frac{L^2\tilde{\eta}^2\zeta^2}{S_2} + L^2\tilde{\eta}^2\delta^2.$$

$$\mathbb{E}[\|\nabla F(\bar{x}^{(R)})\|^2] = \mathcal{O}\left( \frac{LA}{R} + \frac{(LA\sigma^2)^{1/2}}{\sqrt{S_1 P S_2 K R}} + \frac{(LA\zeta^2)^{1/2}}{\sqrt{S_2 R}} + \frac{(LA\delta^2)^{1/2}}{\sqrt{S_1 R}} \right.$$

$$\left. + \frac{(L^2 A^2 \sigma^2)^{1/3}}{(P S_2 K R^2)^{1/3}} + \frac{(L^2 A^2 \zeta^2)^{1/3}}{(S_2 R^2)^{1/3}} + \frac{(L^2 A^2 \delta^2)^{1/3}}{(R^2)^{1/3}} \right)$$

where $\bar{x}^R$ is defined as a model uniformly sampled from the $x^{(0)}, \ldots, x^{(R-1)}$ of previous iterations, and $\lesssim$ hides universal constants.

### F.4 PROOF OF THEOREM 2 FOR THE RING-STAR CASE

For Ring-Star,

$$\mathbf{x}_{s_1,p,s_2,k}^{(r)} - \mathbf{x}^{(r)} = -\eta \sum_{k'=0}^{k-1} \mathbf{g}_{s_1,p,s_2,k'}^{(r)} - \eta \sum_{p'=0}^{p-1} \frac{1}{S_2} \sum_{s_2'=1}^{S_2} \sum_{k'=0}^{K-1} \mathbf{g}_{s_1,p',s_2',k'}^{(r)}$$

$$- \eta \sum_{s_1'=1}^{s_1} \sum_{p'=0}^{P-1} \frac{1}{S_2} \sum_{s_2'=1}^{S_2} \sum_{k'=0}^{k-1} \mathbf{g}_{s_1',p',s_2,k'}^{(r)}.$$

We first bound $\mathbb{E}[\|\mathbf{x}_{s_1,p,s_2,k}^{(r)} - \mathbf{x}^{(r)}\|^2]$.

$$\mathbb{E}[\|\mathbf{x}_{s_1,p,s_2,k}^{(r)} - \mathbf{x}^{(r)}\|^2]$$

$$\leq 15\eta^2 \mathbb{E}\left[\left\|\sum_{k'=0}^{k-1} \left(\mathbf{g}_{s_1,p,s_2,k'}^{(r)} - \nabla F_{\pi_1(s_1),\pi_2(s_2)}(\mathbf{x}_{s_1,p,s_2,k'}^{(r)})\right)\right\|^2\right]$$

$$+ 15\eta^2 \mathbb{E}\left[\left\|\sum_{p'=0}^{p-1} \frac{1}{S_2} \sum_{s_2'=1}^{S_2} \sum_{k'=0}^{K-1} \left(\mathbf{g}_{s_1,p',s_2',k'}^{(r)} - \nabla F_{\pi_1(s_1),\pi_2(s_2')}(\mathbf{x}_{s_1,p',s_2',k'}^{(r)})\right)\right\|^2\right]$$

$$+ 15\eta^2 \mathbb{E}\left[\left\|\sum_{s_1'=1}^{s_1} \sum_{p'=0}^{P-1} \frac{1}{S_2} \sum_{s_2'=1}^{S_2} \sum_{k'=0}^{k-1} \left(\mathbf{g}_{s_1',p',s_2,k'}^{(r)} - \nabla F_{\pi_1(s_1'),\pi_2(s_2')}(\mathbf{x}_{s_1',p',s_2',k'}^{(r)})\right)\right\|^2\right]$$

$$+ 15\eta^2 \mathbb{E}\left[\left\|\sum_{k'=0}^{k-1} \left(\nabla F_{\pi_1(s_1),\pi_2(s_2)}(\mathbf{x}_{s_1,p,s_2,k'}^{(r)}) - \nabla F_{\pi_1(s_1),\pi_2(s_2)}(\mathbf{x}^{(r)})\right)\right\|^2\right]$$

$$+ 15\eta^2 \mathbb{E}\left[\left\|\sum_{p'=0}^{p-1} \frac{1}{S_2} \sum_{s_2'=1}^{S_2} \sum_{k'=0}^{K-1} \left(\nabla F_{\pi_1(s_1),\pi_2(s_2')}(\mathbf{x}_{s_1,p',s_2',k'}^{(r)}) - \nabla F_{\pi_1(s_1),\pi_2(s_2')}(\mathbf{x}^{(r)})\right)\right\|^2\right]$$

$$+ 15\eta^2 \mathbb{E}\left[\left\|\sum_{s_1'=1}^{s_1} \sum_{p'=0}^{P-1} \frac{1}{S_2} \sum_{s_2'=1}^{S_2} \sum_{k'=0}^{k-1} \left(\nabla F_{\pi_1(s_1'),\pi_2(s_2')}(\mathbf{x}_{s_1',p',s_2',k'}^{(r)}) - \nabla F_{\pi_1(s_1'),\pi_2(s_2')}(\mathbf{x}^{(r)})\right)\right\|^2\right]$$

$$+ 15\eta^2 \mathbb{E}\left[\left\|\sum_{k'=0}^{k-1} \left(\nabla F_{\pi_1(s_1),\pi_2(s_2)}(\mathbf{x}^{(r)}) - \nabla F_{\pi_1(s_1)}(\mathbf{x}^{(r)})\right)\right\|^2\right]$$

$$+ 15\eta^2 \mathbb{E}\left[\left\|\sum_{p'=0}^{p-1} \frac{1}{S_2} \sum_{s_2'=1}^{S_2} \sum_{k'=0}^{K-1} \left(\nabla F_{\pi_1(s_1),\pi_2(s_2')}(\mathbf{x}^{(r)}) - \nabla F_{\pi_1(s_1)}(\mathbf{x}^{(r)})\right)\right\|^2\right]$$

$$+ 15\eta^2 \mathbb{E}\left[\left\|\sum_{s_1'=1}^{s_1} \sum_{p'=0}^{P-1} \frac{1}{S_2} \sum_{s_2'=1}^{S_2} \sum_{k'=0}^{k-1} \left(\nabla F_{\pi_1(s_1'),\pi_2(s_2')}(\mathbf{x}^{(r)}) - \nabla F_{\pi_1(s_1')}(\mathbf{x}^{(r)})\right)\right\|^2\right]$$

$$+ 15\eta^2 \mathbb{E}\left[\left\|\sum_{k'=0}^{k-1} \left(\nabla F_{\pi_1(s_1)}(\mathbf{x}^{(r)}) - \nabla F(\mathbf{x}^{(r)})\right)\right\|^2\right]$$

$$+ 15\eta^2 \mathbb{E}\left[\left\|\sum_{p'=0}^{p-1} \frac{1}{S_2} \sum_{s_2'=1}^{S_2} \sum_{k'=0}^{K-1} \left(\nabla F_{\pi_1(s_1)}(\mathbf{x}^{(r)}) - \nabla F(\mathbf{x}^{(r)})\right)\right\|^2\right]$$

$$+ 15\eta^2 \mathbb{E}\left[\left\|\sum_{s_1'=1}^{s_1} \sum_{p'=0}^{P-1} \frac{1}{S_2} \sum_{s_2'=1}^{S_2} \sum_{k'=0}^{k-1} \left(\nabla F_{\pi_1(s_1')}(\mathbf{x}^{(r)}) - \nabla F(\mathbf{x}^{(r)})\right)\right\|^2\right]$$

$$+ 15\eta^2 \mathbb{E}\left[\left\|\sum_{k'=0}^{k-1} \nabla F(\mathbf{x}^{(r)})\right\|^2\right]$$

$$+ 15\eta^2\mathbb{E}\left[\left\|\sum_{p'=0}^{p-1}\frac{1}{S_2}\sum_{s_2'=1}^{S_2}\sum_{k'=0}^{K-1}\nabla F(\mathbf{x}^{(r)})\right\|^2\right]$$

$$+ 15\eta^2\mathbb{E}\left[\left\|\sum_{s_1'=1}^{s_1}\sum_{p'=0}^{P-1}\frac{1}{S_2}\sum_{s_2'=1}^{S_2}\sum_{k'=0}^{k-1}\nabla F(\mathbf{x}^{(r)})\right\|^2\right]$$

$$\leq 15\eta^2 K\sigma^2 + 15\eta^2 p\frac{1}{S_2}K\sigma^2 + 15\eta^2 s_1 P\frac{1}{S_2}K\sigma^2$$

$$+ 15\eta^2 L^2 k\sum_{k'=0}^{k-1}\mathbb{E}[\|\mathbf{x}_{s_1,p,s_2,k'}^{(r)} - \mathbf{x}^{(r)}\|^2]$$

$$+ 15\eta^2 L^2 p\sum_{p'=0}^{p-1}\frac{1}{S_2}\sum_{s_2'=1}^{S_2}K\sum_{k'=0}^{K-1}\mathbb{E}[\|\mathbf{x}_{s_1,p',s_2',k'}^{(r)} - \mathbf{x}^{(r)}\|^2]$$

$$+ 15\eta^2 L^2 s_1\sum_{s_1'=1}^{s_1}P\sum_{p'=0}^{P-1}\frac{1}{S_2}\sum_{s_2'=1}^{S_2}K\sum_{k'=0}^{K-1}\mathbb{E}[\|\mathbf{x}_{s_1',p',s_2',k'}^{(r)} - \mathbf{x}^{(r)}\|^2]$$

$$+ 15\eta^2 k^2\zeta^2 + 15\eta^2 p^2\frac{1}{s_2}K^2\zeta^2 + 15\eta^2 s_1^2 p^2\frac{1}{s_2}K^2\zeta^2$$

$$+ 15\eta^2 K^2 s_2^2\delta^2 + 15\eta^2 p^2 K^2 s_2^2\delta^2 + 15\eta^2 s_1^2 p^2 K^2 s_2^2\delta^2$$

$$+ 15\eta^2 K^2\mathbb{E}[\|\nabla F(\mathbf{x}^{(r)})\|^2] + 15\eta^2 p^2 K^2\mathbb{E}[\|\nabla F(\mathbf{x}^{(r)})\|^2] + 15\eta^2 s_1^2 p^2 K^2\mathbb{E}[\|\nabla F(\mathbf{x}^{(r)})\|^2].$$

$$\sum_{s_1=1}^{S_1}\sum_{p=0}^{P_1}\sum_{s_2=1}^{S_2}\sum_{k=0}^{K-1}\mathbb{E}[\|\mathbf{x}_{s_1,p,s_2,k}^{(r)} - \mathbf{x}^{(r)}\|^2]$$

$$\leq 15\eta^2 S_1 P S_2 K^2\sigma^2 + 30\eta^2 S_1^2 P^2 K^2\sigma^2$$

$$+ 45\eta^2 S_1^2 P^2 K^2\sum_{s_1=1}^{S_1}\sum_{p=0}^{P-1}\sum_{s_2=1}^{S_2}\sum_{k=0}^{K-1}\mathbb{E}[\|\mathbf{x}_{s_1,p,s_2,k}^{(r)} - \mathbf{x}^{(r)}\|^2]$$

$$+ 15\eta^2 S_1 P S_2 K^3\zeta^2 + 30\eta^2 S_1^3 P^3 K^3\zeta^2$$

$$+ 45\eta^2 S_1^2 P^3 S_2 K^3\delta^2$$

$$+ 45\eta^2 S_1^3 P^3 S_2 K^3\mathbb{E}[\|\nabla F(\mathbf{x}^{(r)})\|^2].$$

Let $c_1 = \frac{1}{1-45\eta^2 L^2 S_1^2 P^2 K^2}$,

$$\sum_{s_1=1}^{S_1}\sum_{p=0}^{P-1}\sum_{s_2=1}^{S_2}\sum_{k=0}^{K-1}\mathbb{E}[\|\mathbf{x}_{s_1,p,s_2,k}^{(r)} - \mathbf{x}^{(r)}\|^2]$$

$$\leq 15c_1\eta^2 S_1 P S_2 K^2\sigma^2 + 30c_1\eta^2 S_1^2 P^2 K^2\sigma^2$$

$$+ 15c_1\eta^2 S_1 P S_2 K^3\zeta^2 + 30c_1\eta^2 S_1^3 P^3 K^3\zeta^2$$

$$+ 45c_1\eta^2 S_1^2 P^3 S_2 K^3\delta^2 + 45c_1\eta^2 S_1^3 P^3 S_2 K^3\mathbb{E}[\|\nabla F(\mathbf{x}^{(r)})\|^2].$$

$$\mathbb{E}[F(\mathbf{x}^{(r+1)}) - F(\mathbf{x}^{(r)})]$$

$$\leq -\eta S_1 P K\left(\frac{1}{2} - \frac{5}{2}\eta L S_1 P K - 45c_1\eta^2 L^2 S_1^2 P^2 K^2\right)\mathbb{E}[\|\nabla F(\mathbf{x}^{(r)})\|^2]$$

$$+ \frac{5}{2}\eta^2 L S_1 P\frac{1}{S_2}K\sigma^2 + 15c_1\eta^3 L^2 S_1 P K^2\sigma^2 + 30c_1\eta^3 L^2 S_1^2 P^2\frac{1}{S_2}K^2\sigma^2$$

$$+ \frac{5}{2}\eta^2 LS_1^2 P^2 \frac{1}{S_2} K^2 \zeta^2 + 15c_1 \eta^3 L^2 S_1 PK^3 \zeta^2 + 30c_1 \eta^3 L^2 S_1^3 P^3 \frac{1}{S_2} K^3 \zeta^2$$

$$+ \frac{5}{2}\eta^2 LS_1 P^2 K^2 \delta^2 + 45c_1 \eta^3 L^2 S_1^2 P^3 K^3 \delta^2.$$

Let $c_2 = \frac{1}{\frac{1}{2} - \frac{5}{2}\eta LS_1 pK - 45c_1 c_2 \eta^2 L^2 S_1^2 P^2 K^2}$,

$$\frac{1}{R}\sum_{r=0}^{R-1}\mathbb{E}[\|\nabla F(\mathbf{x}^{(r)})\|^2] \leq \frac{c_2 \mathbb{E}[F(\mathbf{x}^{(0)}) - F(\mathbf{x}^{(R)})]}{\eta R S_1 PK}$$

$$+ \frac{5}{2}c_2\eta L \frac{1}{S_2}\sigma^2 + 15c_1 c_2\eta^2 L^2 K\sigma^2 + 30c_1 c_2\eta^2 L^2 S_1 P \frac{1}{S_2} K\sigma^2$$

$$+ \frac{5}{2}c_2\eta LS_1 P\frac{1}{S_2}K\zeta^2 + 15c_1 c_2\eta^2 L^2 K^2 \zeta^2 + 30c_1 c_2\eta^2 L^2 S_1^2 P^2 \frac{1}{S_2} K^2 \zeta^2$$

$$+ \frac{5}{2}c_2\eta LPK\delta^2 + 45c_1 c_2\eta^2 L^2 S_1 P^2 K^2 \delta^2.$$

Let $\tilde{\eta} = \eta S_1 PK$ and $\tilde{\eta} \leq \frac{1}{15L}$,

$$\mathbb{E}[\|\nabla F(\bar{x}^{(R)})\|^2] \lesssim \frac{F(\mathbf{x}^{(0)}) - F(\mathbf{x}^{(R)})}{\tilde{\eta}R} + \frac{L\tilde{\eta}\sigma^2}{S_1 PS_2 K} + \frac{L\tilde{\eta}\zeta^2}{S_2} + \frac{L\tilde{\eta}\delta^2}{S_1}$$

$$+ \frac{L^2\tilde{\eta}^2\sigma^2}{S_1{}^2 P^2 K} + \frac{L^2\tilde{\eta}^2\sigma^2}{S_1 PS_2 K} + \frac{L^2\tilde{\eta}^2\zeta^2}{S_1{}^2 P^2} + \frac{L^2\tilde{\eta}^2\zeta^2}{S_2} + \frac{L^2\tilde{\eta}^2\delta^2}{S_1}.$$

$$\mathbb{E}[\|\nabla F(\bar{x}^{(R)})\|^2] = \mathcal{O}\Big( \frac{LA}{R} + \frac{(LA\sigma^2)^{1/2}}{\sqrt{S_1 PS_2 KR}} + \frac{(LA\zeta^2)^{1/2}}{\sqrt{S_2 R}} + \frac{(LA\delta^2)^{1/2}}{\sqrt{S_1 R}} + \frac{(L^2 A^2\sigma^2)^{1/3}}{(S_1^2 P^2 KR^2)^{1/3}}$$

$$+ \frac{(L^2 A^2\sigma^2)^{1/3}}{(S_1 PS_2 KR^2)^{1/3}} + \frac{(L^2 A^2\zeta^2)^{1/3}}{(S_1^2 P^2 R^2)^{1/3}} + \frac{(L^2 A^2\zeta^2)^{1/3}}{(S_2 R^2)^{1/3}} + \frac{(L^2 A^2\delta^2)^{1/3}}{(S_1 R^2)^{1/3}} \Big)$$

where $\bar{x}^R$ is defined as a model uniformly sampled from the $x^{(0)}, \ldots, x^{(R-1)}$ of previous iterations, and $\lesssim$ hides universal constants.

## F.5 PROOF OF THEOREM 2 FOR THE RING-RING CASE

For Ring-Ring,

$$\mathbf{x}_{s_1,p,s_2,k}^{(r)} - \mathbf{x}^{(r)} = -\eta \sum_{k'=0}^{k-1} \mathbf{g}_{s_1,p,s_2,k'}^{(r)} - \eta \sum_{s_2'=1}^{s_2}\sum_{k'=0}^{K-1} \mathbf{g}_{s_1,p,s_2',k'}^{(r)}$$

$$- \eta \sum_{p'=0}^{p-1}\sum_{s_2'=1}^{S_2}\sum_{k'=0}^{K-1} \mathbf{g}_{s_1,p',s_2',k'}^{(r)} - \eta \sum_{s_1'=1}^{s_1}\sum_{p'=0}^{P-1}\sum_{s_2'=1}^{S_2}\sum_{k'=0}^{K-1} \mathbf{g}_{s_1',p',s_2',k'}^{(r)}.$$

We first bound $\mathbb{E}[\|\mathbf{x}_{s_1,p,s_2,k}^{(r)} - \mathbf{x}^{(r)}\|^2]$.

$$\mathbb{E}[\|\mathbf{x}_{s_1,p,s_2,k}^{(r)} - \mathbf{x}^{(r)}\|^2]$$

$$\leq 20\eta^2 \mathbb{E}\left[\left\|\sum_{k'=0}^{k-1}\Big(\mathbf{g}_{s_1,p,s_2,k'}^{(r)} - \nabla F_{\pi_1(s_1),\pi_2(s_2)}(\mathbf{x}_{s_1,p,s_2,k'}^{(r)})\Big)\right\|^2\right]$$

$$+ 20\eta^2 \mathbb{E}\left[\left\|\sum_{s_2'=1}^{s_2}\sum_{k'=0}^{K-1}\Big(\mathbf{g}_{s_1,p,s_2',k'}^{(r)} - \nabla F_{\pi_1(s_1),\pi_2(s_2')}(\mathbf{x}_{s_1,p,s_2',k'}^{(r)})\Big)\right\|^2\right]$$

$$+ 20\eta^2 \mathbb{E}\left[\left\|\sum_{p'=0}^{p-1}\sum_{s_2'=1}^{S_2}\sum_{k'=0}^{K-1}\Big(\mathbf{g}_{s_1,p',s_2',k'}^{(r)} - \nabla F_{\pi_1(s_1),\pi_2(s_2')}(\mathbf{x}_{s_1,p',s_2',k'}^{(r)})\Big)\right\|^2\right]$$

$$+ 20\eta^2 \mathbb{E}\left[\left\|\sum_{s_1'=1}^{s_1}\sum_{p'=0}^{P-1}\sum_{s_2'=1}^{S_2}\sum_{k'=0}^{K-1}\left(\mathbf{g}_{s_1',p',s_2',k'}^{(r)} - \nabla F_{\pi_1(s_1'),\pi_2(s_2')}(\mathbf{x}_{s_1',p',s_2',k'}^{(r)})\right)\right\|^2\right]$$

$$\leq 20\eta^2 \mathbb{E}\left[\left\|\sum_{k'=0}^{k-1}\left(\nabla F_{\pi_1(s_1),\pi_2(s_2)}(\mathbf{x}_{s_1,p,s_2,k'}^{(r)}) - \nabla F_{\pi_1(s_1),\pi_2(s_2)}(\mathbf{x}^{(r)})\right)\right\|^2\right]$$

$$+ 20\eta^2 \mathbb{E}\left[\left\|\sum_{s_2'=1}^{s_2}\sum_{k'=0}^{K-1}\left(\nabla F_{\pi_1(s_1),\pi_2(s_2')}(\mathbf{x}_{s_1,p,s_2',k'}^{(r)}) - \nabla F_{\pi_1(s_1),\pi_2(s_2')}(\mathbf{x}^{(r)})\right)\right\|^2\right]$$

$$+ 20\eta^2 \mathbb{E}\left[\left\|\sum_{p'=0}^{p-1}\sum_{s_2'=1}^{S_2}\sum_{k'=0}^{K-1}\left(\nabla F_{\pi_1(s_1),\pi_2(s_2')}(\mathbf{x}_{s_1,p',s_2',k'}^{(r)}) - \nabla F_{\pi_1(s_1),\pi_2(s_2')}(\mathbf{x}^{(r)})\right)\right\|^2\right]$$

$$+ 20\eta^2 \mathbb{E}\left[\left\|\sum_{s_1'=1}^{s_1}\sum_{p'=0}^{P-1}\sum_{s_2'=1}^{S_2}\sum_{k'=0}^{K-1}\left(\nabla F_{\pi_1(s_1'),\pi_2(s_2')}(\mathbf{x}_{s_1',p',s_2',k'}^{(r)}) - \nabla F_{\pi_1(s_1'),\pi_2(s_2')}(\mathbf{x}^{(r)})\right)\right\|^2\right]$$

$$\leq 20\eta^2 \mathbb{E}\left[\left\|\sum_{k'=0}^{k-1}\left(\nabla F_{\pi_1(s_1),\pi_2(s_2)}(\mathbf{x}^{(r)}) - \nabla F_{\pi_1(s_1)}(\mathbf{x}^{(r)})\right)\right\|^2\right]$$

$$+ 20\eta^2 \mathbb{E}\left[\left\|\sum_{s_2'=1}^{s_2}\sum_{k'=0}^{K-1}\left(\nabla F_{\pi_1(s_1),\pi_2(s_2')}(\mathbf{x}^{(r)}) - \nabla F_{\pi_1(s_1)}(\mathbf{x}^{(r)})\right)\right\|^2\right]$$

$$+ 20\eta^2 \mathbb{E}\left[\left\|\sum_{p'=0}^{p-1}\sum_{s_2'=1}^{S_2}\sum_{k'=0}^{K-1}\left(\nabla F_{\pi_1(s_1),\pi_2(s_2')}(\mathbf{x}^{(r)}) - \nabla F_{\pi_1(s_1)}(\mathbf{x}^{(r)})\right)\right\|^2\right]$$

$$+ 20\eta^2 \mathbb{E}\left[\left\|\sum_{s_1'=1}^{s_1}\sum_{p'=0}^{P-1}\sum_{s_2'=1}^{S_2}\sum_{k'=0}^{K-1}\left(\nabla F_{\pi_1(s_1'),\pi_2(s_2')}(\mathbf{x}^{(r)}) - \nabla F_{\pi_1(s_1')}(\mathbf{x}^{(r)})\right)\right\|^2\right]$$

$$\leq 20\eta^2 \mathbb{E}\left[\left\|\sum_{k'=0}^{k-1}\left(\nabla F_{\pi_1(s_1)}(\mathbf{x}^{(r)}) - \nabla F(\mathbf{x}^{(r)})\right)\right\|^2\right]$$

$$+ 20\eta^2 \mathbb{E}\left[\left\|\sum_{s_2'=1}^{s_2}\sum_{k'=0}^{K-1}\left(\nabla F_{\pi_1(s_1)}(\mathbf{x}^{(r)}) - \nabla F(\mathbf{x}^{(r)})\right)\right\|^2\right]$$

$$+ 20\eta^2 \mathbb{E}\left[\left\|\sum_{p'=0}^{p-1}\sum_{s_2'=1}^{S_2}\sum_{k'=0}^{K-1}\left(\nabla F_{\pi_1(s_1)}(\mathbf{x}^{(r)}) - \nabla F(\mathbf{x}^{(r)})\right)\right\|^2\right]$$

$$+ 20\eta^2 \mathbb{E}\left[\left\|\sum_{s_1'=1}^{s_1}\sum_{p'=0}^{P-1}\sum_{s_2'=1}^{S_2}\sum_{k'=0}^{K-1}\left(\nabla F_{\pi_1(s_1')}(\mathbf{x}^{(r)}) - \nabla F(\mathbf{x}^{(r)})\right)\right\|^2\right]$$

$$\leq 20\eta^2 \mathbb{E}\left[\left\|\sum_{k'=0}^{k-1}\nabla F(\mathbf{x}^{(r)})\right\|^2\right]$$

$$+ 20\eta^2 \mathbb{E}\left[\left\|\sum_{s_2'=1}^{s_2} \sum_{k'=0}^{K-1} \nabla F(\mathbf{x}^{(r)})\right\|^2\right]$$

$$+ 20\eta^2 \mathbb{E}\left[\left\|\sum_{p'=0}^{p-1} \sum_{s_2'=1}^{S_2} \sum_{k'=0}^{K-1} \nabla F(\mathbf{x}^{(r)})\right\|^2\right]$$

$$+ 20\eta^2 \mathbb{E}\left[\left\|\sum_{s_1'=1}^{s_1} \sum_{p'=0}^{P-1} \sum_{s_2'=1}^{S_2} \sum_{k'=0}^{K-1} \nabla F(\mathbf{x}^{(r)})\right\|^2\right]$$

$$\leq 20\eta^2 k\sigma^2 + 20\eta^2 s_2 K\sigma^2 + 20\eta^2 p S_2 K\sigma^2 + 20\eta^2 s_1 P S_2 K\sigma^2$$

$$+ 20\eta^2 L^2 k \sum_{k'=0}^{k-1} \mathbb{E}[\|\mathbf{x}_{s_1,p,s_2,k'}^{(r)} - \mathbf{x}^{(r)}\|^2]$$

$$+ 20\eta^2 L^2 (s_2 - 1) K \sum_{s_2'=1}^{s_2-1} \sum_{k'=0}^{K-1} \mathbb{E}[\|\mathbf{x}_{s_1,p,s_2',k'}^{(r)} - \mathbf{x}^{(r)}\|^2]$$

$$+ 20\eta^2 L^2 p \sum_{p'=0}^{p-1} S_2 \sum_{s_2'=1}^{S_2} K \sum_{k'=0}^{K-1} \mathbb{E}[\|\mathbf{x}_{s_1,p',s_2',k'}^{(r)} - \mathbf{x}^{(r)}\|^2]$$

$$+ 20\eta^2 L^2 (s_1 - 1) \sum_{s_1'=1}^{s_1-1} P \sum_{p'=0}^{P-1} S_2 \sum_{s_2'=1}^{S_2} K \sum_{k=0}^{K-1} \mathbb{E}[\|\mathbf{x}_{s_1',p',s_2',k'}^{(r)} - \mathbf{x}^{(r)}\|^2]$$

$$+ 20\eta^2 k^2 \zeta^2 + 20\eta^2 (s_2 - 1) K^2 \zeta^2 + 20\eta^2 p^2 S_2 K^2 \zeta^2 + 20\eta^2 (s_1 - 1) P^2 S_2 K^2 \zeta^2$$

$$+ 20\eta^2 k^2 \delta^2 + 20\eta^2 (s_2 - 1) K^2 \delta^2 + 20\eta^2 p^2 S_2^2 K^2 \delta^2 + 20\eta^2 (s_1 - 1) P^2 S_2^2 K^2 \delta^2$$

$$+ 20\eta^2 k^2 \mathbb{E}[\|\nabla F(\mathbf{x}^{(r)})\|^2] + 20\eta^2 (s_2 - 1) K^2 \mathbb{E}[\|\nabla F(\mathbf{x}^{(r)})\|^2]$$

$$+ 20\tilde{\eta}^2 p^2 S_2^2 K^2 \mathbb{E}[\|\nabla F(\mathbf{x}^{(r)})\|^2] + 20\tilde{\eta}^2 (s_1 - 1)^2 P^2 S_2^2 K^2 \mathbb{E}[\|\nabla F(\mathbf{x}^{(r)})\|^2].$$

Let $c_1 = \frac{1}{1 - 80\eta^2 S_1^2 P^2 S_2^2 K^2}$,

$$\sum_{s_1=1}^{S_1} \sum_{p=0}^{P-1} \sum_{s_2=1}^{S_2} \sum_{k=0}^{K-1} \mathbb{E}[\|\mathbf{x}_{s_1,p,s_2,k}^{(r)} - \mathbf{x}^{(r)}\|^2]$$

$$\leq 80 c_1 \eta^2 S_1^2 P^2 S_2^2 K^2 \sigma^2$$

$$+ 80 c_1 \eta^2 S_1^2 P^2 S_2^2 K^2 \sum_{s_1=1}^{S_1} \sum_{p=0}^{P-1} \sum_{s_2=1}^{S_2} \sum_{k=0}^{K-1} \mathbb{E}[\|\mathbf{x}_{s_1,p,s_2,k}^{(r)} - \mathbf{x}^{(r)}\|^2]$$

$$+ 80 c_1 \eta^2 S_1^3 P^3 S_2^2 K^3 \zeta^2$$

$$+ 80 c_1 \eta^2 S_1^2 P^3 S_2^3 K^3 \delta^2$$

$$+ 80 c_1 \eta^2 S_1^3 P^3 S_2^3 K^3 \mathbb{E}[\|\nabla F(\mathbf{x}^{(r)})\|^2].$$

$$\mathbb{E}[F(\mathbf{x}^{(r+1)}) - F(\mathbf{x}^{(r)})]$$

$$\leq -\eta S_1 P S_2 K \left(\frac{1}{2} - \frac{5}{2}\eta L S_1 P S_2 K - 80 c_1 c_2 \eta^2 L^2 S_1^2 P^2 S_2^2 K^2\right) \mathbb{E}[\|\nabla F(\mathbf{x}^{(r)})\|^2]$$

$$+ \frac{5}{2}\eta^2 L S_1 P S_2 K \sigma^2 + 80 c_1 \eta^3 L^2 S_1^2 P^2 S_2^2 K^2 \sigma^2$$

$$+ \frac{5}{2}\eta^2 L S_1^2 P^2 S_2 K^2 \zeta^2 + 80 c_1 \eta^3 L^2 S_1^3 P^3 S_2^2 K^3 \zeta^2$$

$$+ \frac{5}{2}\eta^2 LS_1 P^2 S_2^2 K^2 \delta^2 + 80c_1\eta^3 L^2 S_1^2 P^3 S_2^3 K^3 \delta^2.$$

Let $c_2 = \frac{1}{\frac{1}{2} - \frac{5}{2}\eta LS_1 PS_2 K - 80c_1 c_2 \eta^3 L^2 S_1^2 P^2 S_2^2 K^2}$,

$$\frac{1}{R}\sum_{r=0}^{R-1} \mathbb{E}[\|\nabla F(\mathbf{x}^{(r)})\|^2] \leq \frac{c_2\left(F(\mathbf{x}^{(0)}) - F(\mathbf{x}^{(R)})\right)}{\eta R S_1 PS_2 K} + \frac{5}{2}c_2\eta L\sigma^2 + 80c_1 c_2\eta^2 L^2 S_1 PS_2 K\sigma^2$$

$$+ \frac{5}{2}c_2\eta LS_1 PK\zeta^2 + 80c_1 c_2\eta^2 L^2 S_1^2 P^2 S_2 K^2 \zeta^2$$

$$+ \frac{5}{2}c_2\eta LPS_2 K\delta^2 + 80c_1 c_2\eta^2 L^2 S_1 P^2 S_2^2 K^2 \delta^2.$$

Let $\tilde{\eta} = \eta S_1 PS_2 K$ and $\tilde{\eta} \leq \frac{1}{20L}$,

$$\mathbb{E}[\|\nabla F(\bar{x}^{(R)})\|^2] \lesssim \frac{F(\mathbf{x}^{(0)}) - F(\mathbf{x}^{(R)})}{\tilde{\eta}R} + \frac{L\tilde{\eta}\sigma^2}{S_1 PS_2 K} + \frac{L\tilde{\eta}\zeta^2}{S_2} + \frac{L\tilde{\eta}\delta^2}{S_1}$$

$$+ \frac{L^2\tilde{\eta}^2\sigma^2}{S_1 PS_2 K} + \frac{L^2\tilde{\eta}^2\zeta^2}{S_2} + \frac{L^2\tilde{\eta}^2\delta^2}{S_1}.$$

$$\mathbb{E}[\|\nabla F(\bar{x}^{(R)})\|^2] = \mathcal{O}\left(\frac{LA}{R} + \frac{(LA\sigma^2)^{1/2}}{\sqrt{S_1 PS_2 KR}} + \frac{(LA\zeta^2)^{1/2}}{\sqrt{S_2 R}} + \frac{(LA\delta^2)^{1/2}}{\sqrt{S_1 R}}\right.$$

$$\left. + \frac{(L^2 A^2\sigma^2)^{1/3}}{(S_1 PS_2 KR^2)^{1/3}} + \frac{(L^2 A^2\zeta^2)^{1/3}}{(S_2 R^2)^{1/3}} + \frac{(L^2 A^2\delta^2)^{1/3}}{(S_1 R^2)^{1/3}}\right)$$

where $\bar{x}^R$ is defined as a model uniformly sampled from the $x^{(0)}, \ldots, x^{(R-1)}$ of previous iterations, and $\lesssim$ hides universal constants.

