# OpenReview forum: "Unified Analyses for Hierarchical Federated Learning: Topology Selection under Data Heterogeneity"
_ICLR.cc/2026/Conference — ICLR 2026 Poster_

### Official Review · Reviewer_2b4g · 2025-10-27

**Soundness:** 3
**Presentation:** 3
**Contribution:** 3
**Rating:** 6
**Confidence:** 5

**Summary:**

This paper establishes a unified theoretical framework for the topology selection problem in Hierarchical Federated Learning (HFL). In traditional HFL practice, topology selection often relies on intuition and experience, which the authors vividly describe as a “topology lottery.” To address this problem, the paper conducts rigorous convergence analysis under a non-convex setting for four HFL topologies composed of star-shaped (parallel) and ring-shaped (serial) aggregations. Theoretical analysis and extensive experimental results jointly reveal three key design principles:
(1) the top-layer aggregation topology has a greater impact on convergence than the bottom-layer topology, and a ring-shaped top layer performs better under data heterogeneity;
(2) inter-group data heterogeneity is the main performance bottleneck; and
(3) the optimal topology depends on the client grouping structure (e.g., Ring-Star is suitable for many small groups, while Star-Ring is suitable for a few large groups).

**Strengths:**

1. Important and Practical Problem: This paper focuses on a problem that is crucial for the design of large-scale federated learning systems. As HFL is increasingly applied in fields such as IoT and edge computing, providing theoretical guidance for its architectural design has become an urgent need.

2. Novel Unified Theoretical Framework: This is the core contribution of the paper. To the best of my knowledge, no prior work has systematically compared all four mainstream HFL topologies within a single unified mathematical framework. This work fills that theoretical gap and integrates previously fragmented studies into a coherent whole.

3. Insightful and Actionable Conclusions: The three principles distilled in this paper are clear and highly insightful. These findings go beyond being mere mathematical results—they serve as practical guidelines that can directly guide engineers and researchers in system design, significantly reducing design uncertainty.

**Weaknesses:**

1. The mathematical derivations in this paper are element-wise, and the four topologies are discussed separately. This differs from matrix-level derivations: for example, if the intra-group and inter-group communication processes could be modeled by a communication matrix $\mathbf{W}$, then the convergence rate would be characterized by the eigenvalues of $\mathbf{W}$. That would yield more general conclusions than those in this paper — for instance, one could compute the eigenvalues of $\mathbf{W}$ corresponding to Ring–Star. Therefore, can the authors re-derive the convergence theory from a matrix perspective to obtain more general results?

2. Could you please provide the learning rate settings for each part of the experimental section?
The specific magnitudes and relative values of the learning rates are crucial, as they directly affect the conclusions about the *effective learning rate* discussed in **Corollary 1**.

**Questions:**

see the Weaknesses

---

> ### Author Response · Authors · 2025-11-21
> **Response to Reviewer 2b4g**
>
> Thank you for your review! Let us respond to the comments:
>
> #### **R1**: **Discussion on Matrix-Based Derivation**
>
> > Weakness: The mathematical derivations in this paper are element-wise, and the four topologies are discussed separately. This differs from matrix-level derivations: for example, if the intra-group and inter-group communication processes could be modeled by a communication matrix , then the convergence rate would be characterized by the eigenvalues of $W$. That would yield more general conclusions than those in this paper — for instance, one could compute the eigenvalues of $W$ corresponding to Ring–Star. Therefore, can the authors re-derive the convergence theory from a matrix perspective to obtain more general results?
>
> HFL topologies can indeed be formulated using communication matrices. Taking the Ring-Star topology as an example:
>
> Let $\mathbf{X}^{(t)} \in \mathbb{R}^{d \times N}$ denote the parameters of all $N=GM$ clients at step $t$. The update rule is given by $\mathbf{X}^{(t+1)} = \mathbf{X}^{(t)} \mathbf{W}^{(t)} - \eta \mathbf{G}^{(t)}$.
>
> * Local Step: $\mathbf{W}\_{\text{Local}} = \mathbf{I}_{GM}$, where the matrix is an identity matrix.
> * Intra-group Aggregation: $\mathbf{W}_{\text{Group}} = \mathbf{I}_G \otimes \mathbf{J}_M$, where $\mathbf{J}_M = \frac{1}{M}\mathbf{1}\mathbf{1}^\top$. Here, clients average their models within a group , corresponding to a block diagonal matrix.
> * Inter-group Propagation: $\mathbf{W}_{\text{Global}} = \mathbf{C}_G \otimes \mathbf{I}_M$. Groups send their updated parameters to the next group, where $\mathbf{C}_G$ is a cyclic permutation matrix representing the ring topology among the $G$ groups.
>
> While the aforementioned matrix formulation is mathematically sound, we are unable to provide a complete derivation from a matrix perspective for the following reasons:
>
> 1. In standard distributed learning, all nodes adopt the same update method. However, in HFL, the client update mechanisms are topology-dependent, which significantly increases the complexity of a matrix-based derivation.
> 2. An element-wise derivation process more intuitively illustrates the impact of different parameters on the four topologies, thereby helping us capture the differences between them.
> 3. Incorporating matrix notation into the existing proof would substantially alter the proof structure, necessitating a complete re-derivation.
>
>  Therefore, we leave this matrix-based analysis for future work.
>
> #### **R2: Learning Rate Tuning Strategy**
>
> > Weakness: Could you please provide the learning rate settings for each part of the experimental section? The specific magnitudes and relative values of the learning rates are crucial, as they directly affect the conclusions about the *effective learning rate* discussed in **Corollary 1**.
>
> We confirm that the learning rate ($\eta$) was indeed tuned individually for each of the four topologies to ensure they all operate near their optimal convergence speed. And we have reported a transparent learning rate tuning strategy in Figure 2 as follows.
>
> |         Topology          |     Search Space for $\eta$      | Optimal $\eta$ Selected |
> | :-----------------------: | :------------------------------: | :---------------------: |
> |       **Star-Star**       | $\{2, 1, 0.5, 0.2, 0.05, 0.01\}$ |           $1$           |
> | **Star-Ring / Ring-Star** | $\{2, 1, 0.5, 0.2, 0.05, 0.01\}$ |          $0.2$          |
> |       **Ring-Ring**       | $\{2, 1, 0.5, 0.2, 0.05, 0.01\}$ |         $0.05$          |
>
> These empirical observations perfectly **validate our theoretical finding** that to maintain an equivalent effective learning rate for comparable convergence, the required raw learning rates must follow the pattern: $\mathbf{\eta_{Star-Star} > \eta_{Star-Ring} \approx \eta_{Ring-Star} > \eta_{Ring-Ring}}$ (when $G \approx M$). The learning rate tuning strategy will be integrated into the revised manuscript.

---

### Official Review · Reviewer_my4H · 2025-10-28

**Soundness:** 2
**Presentation:** 3
**Contribution:** 2
**Rating:** 2
**Confidence:** 4

**Summary:**

This work studied the convergenve of an existing algorithm, hierarchical federated learning algroithm. They analyzed the convergence of this algorihtm over different topology, including ies (Star-Star, Star-Ring, Ring-Star, and Ring-Ring, under non-convex objectives.

**Strengths:**

The paper studied an interesting problem in hierarchical federated learning, which is relevant to scaling federated systems in practical deployments. It provides an extensive theoretical analysis covering multiple possible topologies (Star-Star, Star-Ring, Ring-Star, and Ring-Ring) under non-convex objectives, which is relatively comprehensive compared to typical HFL studies.

**Weaknesses:**

The discussion of related works is rather brief. Since the main focus of this paper lies in the analysis of hierarchical federated optimization algorithms, it would be beneficial to include a more comprehensive review of prior studies on hierarchical federated learning to better contextualize the contributions.

The comparison of convergence speed is also not convincing. The authors claim that the dominating term is $1/\sqrt{R}$ (Line 304), but according to the derivations in Lines 270–290, it should instead be $1/\sqrt{PMKR}$ for the star topology and $1/\sqrt{GPMKR}$ for the ring topology.

The results derived fro star-star topology is not tight. Please comare your results with the existing results. For illustration, when the number of clients in each group reduces to 1, the hierarchical fedavg reduces to fedavg. The rate should be $1/\sqrt{GKR}$. But your current results cannot recover this. I believe your derivation for star topology is not tight.


Furthermore, the analysis of FedAvg under the star-star, ring-star, and star-ring topologies has already been studied in prior works. The assumptions adopted in this paper are largely the same as those made in the existing literature. Therefore, the contribution of this work appears incremental, and the novelty over established results is limited.

**Questions:**

It is hard for me to understand why star topology lead to bad results. In my opinion, star topology enjoys the highest communication efficiency.

In Line 247, where do you define $\bar{x}^R$?

In single tier topology, existing works have already proposed some algroithms and demonstrated convergence without data heterogeneity assumption. Is it possible to eliminate this assumption in hierarchical federated learning.

The ring topology scheme asks clients to update model sequentially, which is ineffective. You cannot enojy the benefit of parallel computation. The conclusion that ring topolgy should be carefully checked. Based on your logic, can I argue that fedavg is not good, we should let clients update sequentially?

---

> ### Author Response · Authors · 2025-11-21
> **Response to Reviewer my4H (Part 1)**
>
> Thank you for your review. We now respond to the weaknesses and questions:
>
> #### **R1**: **Rationale for Ring Topology Performance**
>
> > Question: It is hard for me to understand why star topology lead to bad results. In my opinion, star topology enjoys the highest communication efficiency.
> >
> > Question: The ring topology scheme asks clients to update model sequentially, which is ineffective. You cannot enjoy the benefit of parallel computation. The conclusion that ring topology should be carefully checked. Based on your logic, can I argue that fedavg is not good, we should let clients update sequentially?
>
> **We performed an analysis based on computational convergence.** This is consistent with the methodology of previous studies(e.g., [1, 2, 3, 4]) discussed in our Literature Review, which also focus on computational convergence. **Regarding convergence in terms of time**, your point is well-taken. We have supplemented our work with a wall-clock time analysis, please kindly refer to our response **R2** to Reviewer **hta5**. In brief, our analysis reveals that although the star topology possesses theoretical benefits in parallel computation, the ideal $O(1)$ level of parallelism is rarely attained in practice.
>
> In fact, the phenomenon that sequential updates can yield superior convergence rates (in terms of rounds) is not an isolated finding of our work, but has been observed in specific contexts. A growing body of literature supports the view that ring-based schemes often outperform star-based averaging in terms of sample efficiency. For instance, [5] explicitly analyzed the convergence of Parallel FL (PFL) versus Sequential FL (SFL), demonstrating the superior performance of SFL. Similarly, [6] proposed a specific Ring-Star topology in CFL and theoretically proved its advantage over standard FedAvg.
>
> #### **R2**: **Extended Literature Review and Novelty**
>
> > Weakness: The discussion of related works is rather brief. Since the main focus of this paper lies in the analysis of hierarchical federated optimization algorithms, it would be beneficial to include a more comprehensive review of prior studies on hierarchical federated learning to better contextualize the contributions.
> >
> > Weakness: Furthermore, the analysis of FedAvg under the star-star, ring-star, and star-ring topologies has already been studied in prior works. The assumptions adopted in this paper are largely the same as those made in the existing literature. Therefore, the contribution of this work appears incremental, and the novelty over established results is limited.
>
> We will add more existing works beyond the focus of convergence in the revision. For the existing works that focus on the convergence analysis of hierarchical federated learning, please refer to our Literature Review in the main text and R3 below. Additionally, prior work on Star-Ring and Ring-Star topologies has primarily focused on algorithm design rather than dedicated convergence analysis. We have incorporated several additional studies that include convergence analysis for these topologies for comparison (see Table 1).

---

> ### Author Response · Authors · 2025-11-21
> **Response to Reviewer my4H (Part 2)**
>
> #### **R3**: **Tightness of Star-Star Convergence Bound**
>
> > Weakness: The results derived for star-star topology is not tight. Please compare your results with the existing results. For illustration, when the number of clients in each group reduces to 1, the hierarchical fedavg reduces to fedavg. The rate should be $\frac{1}{\sqrt{GKR}}$. But your current results cannot recover this. I believe your derivation for star topology is not tight.
>
> Our detailed derivation in Appendix C.2 correctly includes the factor G in the variance term (specifically, the term $\frac{12L\tilde{\eta}\sigma^2}{5GPMK}$). However, we inadvertently omitted the factor G when summarizing the final rate in the main text. We will correct this notation in the revision.
>
> With this correction, the dominant convergence rate for Star-Star is $\mathcal{O}(\frac{1}{\sqrt{G P M K R}})$. When reducing to FedAvg (i.e., M=1, P=1), the rate becomes $\mathcal{O}(\frac{1}{\sqrt{GKR}})$. In this case, it is equivalent to FedAvg with G clients. This rate matches the widely cited FedAvg result $\mathcal{O}(\frac{1}{\sqrt{GKR}})$ in [7]. Thus, even when degenerated to FedAvg, our upper bound remains state-of-the-art.
>
> [Table 1: Comparison of convergence rates. Numerical constants are omitted. Some terms are re-organized for comparison.]
>
> | Algorithm | Ref.      | Convergence rate                                             |
> | --------- | --------- | ------------------------------------------------------------ |
> | FedAvg    | [7]       | $\left( \frac{LA}{R} \frac{\sigma^2}{MK} \right)^{\frac{1}{2}} + \left( \frac{L^2A^2}{R^2}\frac{\sigma^2}{K} \right)^{\frac{1}{3}} + \left( \frac{L^2A^2}{R^2}\zeta^2 \right)^{\frac{1}{3}} + \frac{LA}{R}$ |
> | Star-Star | [8,9]     | $\left( \frac{LA}{R} \frac{\sigma^2}{GPMK} \right)^{\frac{1}{2}} + \left( \frac{L^2A^2}{R^2} \frac{\sigma^2}{P^2K} \right)^{\frac{1}{3}} + \left( \frac{L^2A^2}{R^2} \frac{\sigma^2}{PMK} \right)^{\frac{1}{3}} + \left( \frac{L^2A^2}{R^2} \frac{\hat\zeta^2}{P^2} \right)^{\frac{1}{3}}  + {\left( \frac{L^2A^2}{R^2} \zeta^2 \right)^{\frac{1}{3}}} + \frac{LA}{R}$ |
> |           | This work | $\left( \frac{LA}{R} \frac{\sigma^2}{GPMK} \right)^{\frac{1}{2}} + \left( \frac{L^2A^2}{R^2} \frac{\sigma^2}{PK^2} \right)^{\frac{1}{3}} + \left( \frac{L^2A^2}{R^2} \frac{\sigma^2}{PMK} \right)^{\frac{1}{3}} + \left( \frac{L^2A^2}{R^2} \frac{\hat\zeta^2}{P^2} \right)^{\frac{1}{3}}  + {\left( \frac{L^2A^2}{R^2} \zeta^2 \right)^{\frac{1}{3}}} + \frac{LA}{R}$ |
> | Star-Ring | This work | $\left( \frac{LA}{R} \frac{\sigma^2}{GPMK} \right)^{\frac{1}{2}} + \left( \frac{L^2A^2}{R^2} \frac{\sigma^2}{PMK} \right)^{\frac{1}{3}} + {\left( \frac{L^2A^2}{R^2} \frac{\hat\zeta^2}{P^2M} \right)^{\frac{1}{3}}} + {\left( \frac{L^2A^2}{R^2}\zeta^2 \right)^{\frac{1}{3}}} + \frac{LA}{R}$ |
> | Ring-Star | This work | $\left( \frac{LA}{R} \frac{\sigma^2}{GPMK} \right)^{\frac{1}{2}} +\left( \frac{L^2A^2}{R^2} \frac{\sigma^2}{G^2P^2K} \right)^{\frac{1}{3}} + \left( \frac{L^2A^2}{R^2} \frac{\sigma^2}{{G}PMK} \right)^{\frac{1}{3}} + \left( \frac{L^2A^2}{R^2} \frac{\hat\zeta^2}{{G^2}P^2} \right)^{\frac{1}{3}} + {\left( \frac{L^2A^2}{R^2} \frac{\zeta^2}{{G}} \right)^{\frac{1}{3}}} + \frac{LA}{R}$ |
> | Ring-Ring | This work | $\left( \frac{LA}{R} \frac{\sigma^2}{GPMK} \right)^{\frac{1}{2}} + {\left( \frac{L^2A^2}{R^2}\frac{\hat\zeta^2}{G^2P^2} \right)^{\frac{1}{3}}} + {\left( \frac{L^2A^2}{R^2}\zeta^2 \right)^{\frac{1}{3}}} + \frac{LA}{R} $ |
>
> Comparisons:
>
> 1. vs. FedAvg: If there is only one client in each group (i.e., $P=1$, $M=1$ and $\hat\zeta=0$), Star-Star reduces to FedAvg. In this case, our rate becomes
>    $$
>    \left( \frac{LA}{R} \frac{\sigma^2}{GK} \right)^{\frac{1}{2}} + \left( \frac{L^2A^2}{R^2} \frac{\sigma^2}{K} \right)^{\frac{1}{3}}   + {\left( \frac{L^2A^2}{R^2} \zeta^2 \right)^{\frac{1}{3}}} + \frac{LA}{R}
>    $$
>    which matches that of FedAvg [7].
>
> 2. vs. Star-Star: Our convergence rate matches the known best convergence rates [8,9].
>
> 3. vs. Star-Ring / Ring-Star: Even though similar algorithms (to Star-Ring and Ring-Star) and some preliminary convergence analyses have been studied in [10,11,12], the advanced convergence analyses are still missing for Star-Ring and Ring-Star. For example, [11] and [12] used the bounded gradient assumptions (an assumption that is widely considered very strong [13]).
>
> In summary, these comparisons show that our convergence rates are tight.
>
> > Weakness: The comparison of convergence speed is also not convincing. The authors claim that the dominating term is $\frac{1}{\sqrt{R}}$ (Line 304), but according to the derivations in Lines 270–290, it should instead be $\frac{1}{\sqrt{PMKR}}$ for the star topology and $\frac{1}{\sqrt{GPMKR}}$ for the ring topology.
>
> In the original submission, we focus primarily on the number of rounds. In Big-O notation, $R$ is the dominant term governing the convergence rate, while other factors are omitted. We will use the full expression as suggested in the revision.

---

> ### Author Response · Authors · 2025-11-21
> **Response to Reviewer my4H (Part 3)**
>
> #### **R4**: **Definition**
>
> > Question: In Line 247, where do you define $\bar{x}^R$?
>
> We have supplemented the definition. Here, $\bar{x}^R$ is defined as a model uniformly sampled from the $x^{(0)},...,x^{(R-1)}$ of previous iterations.
>
> #### **R5**: **Necessity of Heterogeneity Assumption**
>
> > Question: In single tier topology, existing works have already proposed some algroithms and demonstrated convergence without data heterogeneity assumption. Is it possible to eliminate this assumption in hierarchical federated learning.
>
> For the HFL topologies analyzed in this paper, retaining this assumption remains necessary due to theoretical and algorithmic reasons.
>
> Existing single-layer FL works eliminate the heterogeneity assumption by introducing control variates (e.g., a correction term $c_i$) to estimate and counteract the drift direction. In contrast, our work adopts the standard FedAvg update rule. Without the bounded heterogeneity assumption, the convergence of standard SGD cannot be mathematically guaranteed. Please refer to the proof of Lemma 2 in Appendix C.1. To bound the client drift $\mathbb{E} \\| \mathbf{x}\_{g,p,m,k}-\mathbf{x}\\|$, we need to employ intra-group heterogeneity to bound $\mathbb{E} \\| \nabla F_{g, m'}(\mathbf{x}) - \nabla F_g(\mathbf{x}) \\|^2$, and inter-group heterogeneity to bound $\mathbb{E} \\| \nabla F_{g}(\mathbf{x}) - \nabla F(\mathbf{x}) \\|^2$.
>
> While removing this assumption is a promising direction for future research, it falls under the domain of update algorithm design, which is distinct from the topology analysis focused on in this paper.
>
> > [1]Zhou, Fan, and Guojing Cong. "A distributed hierarchical SGD algorithm with sparse global reduction." *arXiv preprint arXiv:1903.05133* (2019).
> >
> > [2]Wang, Jiayi, et al. "Demystifying why local aggregation helps: Convergence analysis of hierarchical SGD." *Proceedings of the AAAI conference on artificial intelligence*. Vol. 36. No. 8. 2022.
> >
> > [3]Castiglia, Timothy, Anirban Das, and Stacy Patterson. "Multi-level local SGD: Distributed SGD for heterogeneous hierarchical networks." *International Conference on Learning Representations*. 2021.
> >
> > [4]Jiang, Xiaohan, and Hongbin Zhu. "On the convergence of hierarchical federated learning with partial worker participation." *The 40th Conference on Uncertainty in Artificial Intelligence*. 2024.
> >
> > [5] Li, Yipeng, and Xinchen Lyu. "Convergence analysis of sequential federated learning on heterogeneous data." *Advances in Neural Information Processing Systems* 36 (2023): 56700-56755.
> >
> > [6] Chen, Cheng, et al. "Fedcluster: Boosting the convergence of federated learning via cluster-cycling." *2020 IEEE international conference on big data (Big Data)*. IEEE, 2020.
> tes are tight.
> >
> > [7] Koloskova, Anastasia, et al. "A unified theory of decentralized SGD with changing topology and local updates." *International conference on machine learning*. PMLR, 2020.
> >
> > [8] Wang, Jiayi, et al. "Demystifying why local aggregation helps: Convergence analysis of hierarchical SGD." *Proceedings of the AAAI conference on artificial intelligence*. Vol. 36. No. 8. 2022.
> >
> > [9] Jiang, Xiaohan, and Hongbin Zhu. "On the convergence of hierarchical federated learning with partial worker participation." *The 40th Conference on Uncertainty in Artificial Intelligence*. 2024.
> >
> > [10] Zaccone, Riccardo, et al. "Speeding up heterogeneous federated learning with sequentially trained superclients." *2022 26th International Conference on Pattern Recognition (ICPR)*. IEEE, 2022.
> >
> > [11] Chen, Cheng, et al. "Fedcluster: Boosting the convergence of federated learning via cluster-cycling." *2020 IEEE international conference on big data (Big Data)*. IEEE, 2020.
> >
> > [12] Yan, Xingrun, et al. "Sequential federated learning in hierarchical architecture on non-IID datasets." *IEEE Transactions on Mobile Computing* (2025).
> >
> > [13] Khaled, Ahmed, Konstantin Mishchenko, and Peter Richtárik. "Tighter theory for local SGD on identical and heterogeneous data." *International conference on artificial intelligence and statistics*. PMLR, 2020.

---

> > ### Comment · Reviewer_my4H · 2025-11-26
> >
> > First, you made an argument that "the ideal level of parallelism is rarely attained in practice in star topology". This argument stand in the opposite direction to a bunch of literature in federated leaning.
> >
> > Second, you replied with "In Big-O notation, $R$ is the dominant term governing the convergence rate. This is not true. G, M, P, K are all important in the rate. Star topology enjoys the highest communication efficiency, which should be clearly reflect in the convergence rate.
> >
> > Third, as I mentioned in the first round, the analysis of FedAvg under the star-star, ring-star, and star-ring topologies has already been studied in prior works. I don't understand what's the breakthrough made in this work. This is a well studied problem.

---

> > > ### Author Response · Authors · 2025-11-26
> > >
> > > > First, you made an argument that "the ideal level of parallelism is rarely attained in practice in star topology". This argument stand in the opposite direction to a bunch of literature in federated learning.
> > >
> > > The communication bottleneck associated with star topology is a consensus within the FL community, as explicitly stated in a recent 2025 survey[1]: "maintaining a central server in FL can be problematic. The server becomes a communication bottleneck and a single point of failure for the entire network".
> > >
> > > An early and widely recognized work on federated learning is [2], which is commonly regarded as a seminal paper in this area. The original paper states: "In contrast, in federated optimization communication costs dominate — we will typically be limited by an upload bandwidth of 1 MB/s or less." This indicates that communication efficiency under bandwidth constraints has been a core issue in FL since its inception. Subsequently, a bunch of literature has been dedicated to addressing the communication bottleneck in FL, with primary optimization directions including: (i) gradient/model compression [3-6], (ii) client selection [7-10], and (iii) hierarchical/decentralized architectures [11-13]. As evidenced by the literature we have cited, from the proposal of FL to as recently as 2025, new works continue to propose novel optimization methods attempting to mitigate the communication bottleneck in FL. However, to the best of our knowledge, no single work has been able to fully resolve the communication bottleneck and achieve **ideal** levels of parallel efficiency.
> > >
> > > In our analysis of wall-clock time, we adopt the basic assumption that bandwidth is limited and multiple users share the transmission channel bandwidth. This assumption also underpins the communication models used in numerous studies[14, 15] on communication optimization. However, even with existing advanced optimization methods, the communication bottleneck can only be mitigated rather than completely resolved. Therefore, our perspective "the **ideal** level of parallelism is rarely attained in practice in star topology" is consistent with the widely accepted view in the FL literature that communication is a major bottleneck in practical star‑topology deployments.
> > >
> > > [1]Li, Qiongxiu, et al. "From centralized to decentralized federated learning: Theoretical insights, privacy preservation, and robustness challenges." *arXiv preprint arXiv:2503.07505* (2025).
> > >
> > > [2]McMahan, Brendan, et al. "Communication-efficient learning of deep networks from decentralized data." *Artificial intelligence and statistics*. PMLR, 2017.
> > >
> > > [3]Reisizadeh, Amirhossein, et al. "Fedpaq: A communication-efficient federated learning method with periodic averaging and quantization." *International conference on artificial intelligence and statistics*. PMLR, 2020.
> > >
> > > [4]Shah, Suhail Mohmad, and Vincent KN Lau. "Model compression for communication efficient federated learning." *IEEE Transactions on Neural Networks and Learning Systems* 34.9 (2021): 5937-5951.
> > >
> > > [5]He, Zixiao, et al. "FedDT: A Communication-Efficient Federated Learning via Knowledge Distillation and Ternary Compression." *Electronics* 14.11 (2025): 2183.
> > >
> > > [6]Zhou, Yuhao, et al. "E-3SFC: Communication-Efficient Federated Learning With Double-Way Features Synthesizing." *IEEE Transactions on Neural Networks and Learning Systems* (2025).
> > >
> > > [7]Nishio, Takayuki, and Ryo Yonetani. "Client selection for federated learning with heterogeneous resources in mobile edge." *ICC 2019-2019 IEEE international conference on communications (ICC)*. IEEE, 2019.
> > >
> > > [8]Cho, Yae Jee, Jianyu Wang, and Gauri Joshi. "Client selection in federated learning: Convergence analysis and power-of-choice selection strategies." *arXiv preprint arXiv:2010.01243* (2020).
> > >
> > > [9]Vardhan, Harsh, et al. "Client Selection in Federated Learning with Data Heterogeneity and Network Latencies." *arXiv preprint arXiv:2504.01921* (2025).
> > >
> > > [10]Skorik, Sergey, et al. "Communication-Efficient Federated Learning with Adaptive Number of Participants." *arXiv preprint arXiv:2508.13803* (2025).
> > >
> > > [11]Liu, Lumin, et al. "Client-edge-cloud hierarchical federated learning." *ICC 2020-2020 IEEE international conference on communications (ICC)*. IEEE, 2020.
> > >
> > > [12]Kalra, Shivam, et al. "Decentralized federated learning through proxy model sharing." *Nature communications* 14.1 (2023): 2899.
> > >
> > > [13]Li, Ying, et al. "Communication-Efficient Federated Learning for Heterogeneous Clients." *ACM Transactions on Internet Technology* 25.2 (2025): 1-37.
> > >
> > > [14]Pervej, Md Ferdous, Richeng Jin, and Huaiyu Dai. "Hierarchical federated learning in wireless networks: Pruning tackles bandwidth scarcity and system heterogeneity." *IEEE Transactions on Wireless Communications* 23.9 (2024): 11417-11432.
> > >
> > > [15]Abdellatif, Alaa Awad, et al. "Communication-efficient hierarchical federated learning for IoT heterogeneous systems with imbalanced data." *Future Generation Computer Systems* 128 (2022): 406-419.

---

> > > ### Author Response · Authors · 2025-11-26
> > >
> > > > Second, you replied with "In Big-O notation, is the dominant term governing the convergence rate. This is not true. G, M, P, K are all important in the rate. Star topology enjoys the highest communication efficiency, which should be clearly reflect in the convergence rate.
> > >
> > > We would like to clarify that from the wikipedia[1] that "With the domain written explicitly or understood implicitly, one writes $f(x)=\mathcal{O}(g(x))$ and it is read '$f(x)$ is big O of $g(x)$' if there exists a positive real number $M$ such that $|f(x)|≤M g(x)$ for all $x∈D$ . When $g(x)>0$ throughout the domain $D$, an equivalent definition is that $f(x)/g(x)$ is a bounded function."
> > >
> > > In the original manuscript, when comparing different topologies (e.g., in Table 1), we fully accounted for the dependence on $G$, $M$, $P$, $K$. Only in the specific discussion where we focus on the asymptotic scaling with respect to $R$ do we treat $G$, $M$, $P$, and $K$ as constants, they are absorbed into the constant factor in the Big-O notation. As you pointed out, the constant terms matter. We will state in the note that $G$, $M$, $P$, $K$ are treated as constants here to highlight the scaling with respect to $R$. We hope to fully address your concern.
> > >
> > > [1] https://en.wikipedia.org/wiki/Big_O_notation
> > >
> > > > Third, as I mentioned in the first round, the analysis of FedAvg under the star-star, ring-star, and star-ring topologies has already been studied in prior works. I don't understand what's the breakthrough made in this work. This is a well studied problem.
> > >
> > > We would like to clarify that developing a unified convergence framework is significant in its own right. Unified Analysis is an acknowledged and important contribution in the optimization area. As stated in the widely cited work [1], "Proving convergence rates in a unified framework is much more powerful than studying individual special cases on their own." There exists a bunch of work[1-5] establishing unified frameworks, and our work shares a similar philosophy with them. Developing a unified convergence framework facilitates fair "apples-to-apples" comparisons, uncovers novel theoretical principles, and offers theoretically grounded guidance for system design.
> > >
> > > Beyond developing a unified framework, our contributions include: (i) providing the first convergence analysis for previously unstudied Ring-Ring topology; (ii) adopting consistent, general assumptions to overcome the barrier of inconsistent assumptions used in prior separate studies; and (iii) deriving tighter upper bounds that improve upon the convergence rates reported in prior works for specific hierarchical structures.
> > >
> > > Furthermore, in response to your suggestion, we have also conducted a comprehensive analysis of the four topologies in terms of wall-clock time.
> > >
> > > [1]Koloskova, Anastasia, et al. "A unified theory of decentralized SGD with changing topology and local updates." *International conference on machine learning*. PMLR, 2020.
> > >
> > > [2]Wang, Jianyu, and Gauri Joshi. "Cooperative SGD: A unified framework for the design and analysis of local-update SGD algorithms." *Journal of Machine Learning Research* 22.213 (2021): 1-50.
> > >
> > > [3]Haddadpour, Farzin, et al. "Federated learning with compression: Unified analysis and sharp guarantees." *International Conference on Artificial Intelligence and Statistics*. PMLR, 2021.
> > >
> > > [4]Wang, Shiqiang, and Mingyue Ji. "A unified analysis of federated learning with arbitrary client participation." *Advances in neural information processing systems* 35 (2022): 19124-19137.
> > >
> > > [5]Qu, Zhaonan, et al. "A unified linear speedup analysis of federated averaging and nesterov fedavg." *Journal of Artificial Intelligence Research* 78 (2023): 1143-1200.

---

### Official Review · Reviewer_qwXN · 2025-10-29

**Soundness:** 2
**Presentation:** 2
**Contribution:** 2
**Rating:** 4
**Confidence:** 4

**Summary:**

This paper aims to provide a unified convergence framework for analyzing HFL topologies under non-convex objectives and heterogeneous data distributions. It addresses the lack of theoretical guidance on how to select optimal HFL topologies for real-world deployments, given different combinations of intra-group and inter-group heterogeneity.

**Strengths:**

++The motivation behind the analysis of HFL topologies under heterogeneity is strong
++The paper provides convergence bounds for all four topologies within a single analytical framework
++Experiments were conducted on multiple datasets, architectures, and heterogeneity conditions

**Weaknesses:**

++Key proof steps (especially those establishing cross-tier coupling and topology-specific bias) are only summarized without detailed proof
++The paper extends prior FedAvg-type analyses to hierarchical topologies rather than introducing fundamentally new analytical techniques, which exhibits limited novelty
++Only small to medium-scale vision datasets are used. Large-scale, high-variance tasks (e.g., NLP, speech) are missing
++There is no systematic study of how convergence changes with varying group size, local steps, or communication rounds
++The claim that HFL prioritizes scalability over acceleration is made but not empirically validated
++No variance/error bars or significance testing are presented in the figures

**Questions:**

Please address all the weaknesses above.

---

> ### Author Response · Authors · 2025-11-21
> **Response to Reviewer qwXN (part 1)**
>
> Thank you. We now respond to the weaknesses and questions.
>
> #### **R1**: **Detailed Proofs and Appendix Expansion**
>
> > Weakness: Key proof steps (especially those establishing cross-tier coupling and topology-specific bias) are only summarized without detailed proof.
>
> In the revised manuscript, we will provide the complete and detailed derivations and the final convergence bounds (Theorem 1 and Corollary 1) for the Star-Ring, Ring-Star, and Ring-Ring topologies. To immediately address the concern regarding the lack of detailed steps, we provide a Proof Sketch below.
>
> **Proof Sketch**
>
> The convergence analysis proceeds in three logical steps: (1) establishing the one-round progress inequality, (2) decomposing the hierarchical client drift, and (3) solving the recursion.
>
> **Step 1: One-Round Progress via Effective Learning Rate**
> Using the $L$-smoothness of the global objective $F$, we analyze the progress of the global model update ${x}^{(r+1)} = {x}^{(r)} + \Delta {x}$. We formulate the update using a **effective learning rate** $\tilde{\eta}$ for different topologies. We derive:
> $$
> \mathbb{E}\[F(x^{(r+1)})] \leq \mathbb{E}[F({x}^{(r)})\] - \frac{\tilde{\eta}}{2} \\| \nabla F({x}^{(r)})\\|^2+ \frac{L^2 \tilde{\eta}}{2GPMK} E_r+ \text{Variance Terms}
> $$
> Crucially, the client drift term $E_{\text{drift}}$ captures the deviation of local client models from the global model, accumulated across hierarchical tiers.
>
> **Step 2: Hierarchical Drift Decomposition**
> As visualized in Table 5 of the Appendix, the error vector $\mathbf{x}_{g,p,m,k} - \mathbf{x}$ is decomposed differently for each topology. We establish the bound for $E_r$ by analyzing the specific summation structure of each row in Table 5:
>
> *   For Star-Based Layers (Variance Reduction): In Star-Star (Row 1), the inter-client and inter-group terms appear as averages (e.g., $\frac{1}{M}\\sum$). We apply the variance reduction lemma ($\mathbb{E}\|\frac{1}{N}\sum \mathbf{z}_i\|^2 \leq \frac{1}{N} \\|\mathbf{z}_i\\|^2$) to these terms. This mathematically explains why Star topologies benefit from parallel noise dampening.
> *   For Ring-Based Layers (Bias Accumulation): In Ring-Ring (Row 4), Table 5 shows that the error vector includes sequential partial sums (e.g., $\\sum_{g'=1}^{g-1} g_{g'}$ and $\sum_{m'=1}^{m-1} {g}_{m'}$). Unlike the Star topology, these terms cannot be reduced via averaging. Instead, we must bound them using the properties of sequential dependency, which introduces the accumulated temporal bias terms (scaling with $G^2$ or $M^2$) into the $E_r$ bound.
>
> By rigorously bounding the squared norm of each specific decomposition in Table 5, we derive exact, topology-aware expressions for $E_r$, rather than using a generic upper bound.
>
> **Step 3: Final Convergence Bound**
>
> We substitute the bounded $E_r$ back into the Descent Lemma. By telescoping the sum over R global rounds and dividing by the total effective step size, we derive the final convergence rate.
>
> #### **R2: Novelty and Theoretical Contributions**
>
> > Weakness: The paper extends prior FedAvg-type analyses to hierarchical topologies rather than introducing fundamentally new analytical techniques, which exhibits limited novelty.
>
> 1. **Lack of Ring-Ring analysis:** Currently, there is no convergence analysis for HFL under the Ring-Ring topology. While the standard Star-Star topology is extensively studied, followed by Star-Ring, the Ring-Star topology receives the least attention. Our unified framework bridges this gap by providing the first convergence analysis for the bi-level Ring structure.
> 2. **Unified analysis under consistent assumptions:** Existing studies on individual topologies rely on heterogeneous assumptions, making it impossible to establish a unified framework by simply combining prior works. For instance, reference [1] assumes $ρ$-Lipschitz continuity and convexity, while reference [2] relies on Bounded Gradient Norm and Bounded Model Parameter assumptions. In contrast, we adopt more general non-convex assumptions for all four topologies.
> 3. **Tighter upper bounds:** We achieve a convergence rate of $\mathcal{O}(\frac{1}{\sqrt{GPMKR}})$ across all four topologies. This matches state-of-the-art results for Star-Star HFL. Furthermore, for Star-Ring and Ring-Star, this represents a superior convergence rate compared to the $\mathcal{O}(\frac{1}{\sqrt{R}})$ rate derived in [2] and [3].
>
> > [1] Fang, Qingze, et al. "Olive branch learning: A topology-aware federated learning framework for space-air-ground integrated network." *IEEE Transactions on Wireless Communications* 22.7 (2022): 4534-4551.
> >
> > [2] Ding, Yucheng, et al. "Distributed optimization over block-cyclic data." *Proceedings of the 6th ACM International Conference on Multimedia in Asia Workshops*. 2024.
> >
> > [3] Lee, Jin-woo, et al. "Tornadoaggregate: Accurate and scalable federated learning via the ring-based architecture." *arXiv preprint arXiv:2012.03214* (2020).

---

> ### Author Response · Authors · 2025-11-21
> **Response to Reviewer qwXN (part 2)**
>
> #### **R3: Experiments on Non-Vision Dataset (NLP)**
>
> > Weakness: Only small to medium-scale vision datasets are used. Large-scale, high-variance tasks (e.g., NLP, speech) are missing.
>
> We have conducted new experiments on a non-vision dataset: the **SST-2 NLP classification task**, utilizing a simple MLP model common in basic federated NLP settings. The number of groups was set to 10, with 10 clients per group, which is consistent with the experimental settings in the main text. After 1,000 global rounds, the results from the SST-2 experiments are entirely consistent with our main conclusions. These new results will be integrated into the revised manuscript.
>
> | **Heterogeneity**                  | **Star-Star** | **Star-Ring** | **Ring-Star** | **Ring-Ring** |
> | ---------------------------------- | ------------- | ------------- | ------------- | ------------- |
> | Groups IID and Clients IID         | 69.61         | 72.36         | 80.50         | 81.42         |
> | Groups IID and Clients Non-IID     | 69.95         | 73.17         | 79.59         | 80.50         |
> | Groups Non-IID and Clients IID     | 68.12         | 73.85         | 79.13         | 81.08         |
> | Groups Non-IID and Clients Non-IID | 68.12         | 73.97         | 78.44         | 81.65         |
>
> #### **R4: Error Bars**
>
> > Weakness: No variance/error bars or significance testing are presented in the figures.
>
> We have provided the Test Accuracy Ranges (Min-Max %) for the CIFAR-10 experiments (ResNet-18) in the table below. In the revised manuscript, we will update experimental plots to include shaded error bands representing the variance across independent runs with different random seeds.
>
> | **Heterogeneity Setting**            | **Star-Star** | **Star-Ring** | **Ring-Star** |  **Ring-Ring**   |
> | :----------------------------------- | :-----------: | :-----------: | :-----------: | :--------------: |
> | **Groups IID & Clients IID**         | 88.48 ± 0.28  | 90.30 ± 0.13  | 90.40 ± 0.22  | **91.53 ± 0.19** |
> | **Groups IID & Clients Non-IID**     | 87.01 ± 0.51  | 89.55 ± 0.34  | 89.75 ± 0.15  | **91.10 ± 0.69** |
> | **Groups Non-IID & Clients IID**     | 87.03 ± 0.21  | 88.22 ± 0.22  | 89.15 ± 0.49  | **90.94 ± 0.38** |
> | **Groups Non-IID & Clients Non-IID** | 86.78 ± 0.40  | 87.40 ± 0.20  | 90.01 ± 0.08  | **90.33 ± 0.05** |
>
> #### **R5**: **Impact of Hyperparameters**
>
> > Weakness: There is no systematic study of how convergence changes with varying group size, local steps, or communication rounds.
>
> 1. **Group size:** We have already discussed and experimentally validated the impact of group size on convergence in Section 3.4 ("Effect of Groups"). In brief, the Star-Ring topology performs best with a smaller number of large groups (i.e., small G and large M), while the Ring-Star topology excels with a larger number of small groups (i.e., large G and small M).
> 2. **Local steps:** In the revision, we added a dedicated ablation (new subsection “Effect of Group Rounds $P$ and Local Steps $K$”) on CIFAR-10 (Non-IID), where we vary $K \in \{2,5,10\}$ for all four topologies. With fixed $P=1$, we observe that increasing $K$ consistently accelerates early-stage convergence for all topologies.
> 3. **Communication rounds:** In the main text, we have theoretically and experimentally verified the convergence speeds of the four topologies. Specifically, the number of global rounds required to reach standard accuracy follows the order: Ring-Ring > Ring-Star > Star-Ring > Star-Star. Regarding group rounds, we evaluated $P \in \{1, 2, 5\}$ with fixed $K$. We observed that increasing $P$ yields substantial improvements in convergence efficiency, generally providing a stronger acceleration effect than increasing $K$.
>
> #### **R6: Comparison with Single-Tier FL**
>
> > Weakness: The claim that HFL prioritizes scalability over acceleration is made but not empirically validated.
>
> We have added experimental comparisons with single-layer FL using the CIFAR-10 dataset and a ResNet-18 model with N=100. After 5,000 global training rounds, single-layer FedAvg achieved an accuracy of 88.52% on the IID partition and 86.22% on the Non-IID partition, demonstrating a convergence speed similar to that of Star-Star HFL. Meanwhile, single-layer Sequential FL achieved an accuracy of 91.79% (IID) and 90.13% (Non-IID), exhibiting a convergence speed comparable to Ring-Ring HFL. These results **validate our conclusion** that the bi-level structure itself does not inherently accelerate convergence.
>
> Detailed experimental plots, including the convergence curves, will be included in the revised manuscript.

---

### Official Review · Reviewer_hta5 · 2025-10-31

**Soundness:** 2
**Presentation:** 3
**Contribution:** 3
**Rating:** 6
**Confidence:** 5

**Summary:**

The paper analyzes four kinds of two-tier HFL topologies (Star-Star, Star-Ring, Ring-Star, Ring-Ring) with respect to inter- and intra-group heterogeneities and the impact from top and lower layers. Then, the paper derives topology-dependent convergence bounds and claims that: 1. the top tier dominates the convergence over intra-group topology, while ring-based structure at the top helps more; 2. the best topology depends on overall grouping structure; 3. inter-group heterogeneity dominates the overall convergence. Experiments on CIFAR-10/CINIC-10/Fashion-MNIST datasets are carried out to support the claimed patterns.

**Strengths:**

1. The paper provides a clear, unified scope across all four two-tier HFL topologies. It offers a single convergence framework for Star-Star, Star-Ring, Ring-Star, and Ring-Ring under non-convex objectives and both intra- and inter-group heterogeneity.
2. The paper offers an actionable topology guidance by summarizing 3 critical and practical principles on top and lower tiers importance, topology selection and intra-/inter-group heterogeneities.
3. The definition of effective learning rate is introduced to show the impact of hierarchical structure on convergence. It also provides the intuition for the trade-offs between optimization and noise/heterogeneity.

**Weaknesses:**

1. The paper claims a "first" unified comparison of all four topologies, but related work already analyzes subsets and topology variants. The author is suggested to clarify more clearly what is new in the math or convergence bounds beyond those existing researches.
2. The analysis completely ignores communication costs and latency. The convergence rate is presented in terms of global rounds. This is misleading as the wall-clock time of a round differs vastly between topologies. A star topology is parallel (O(1) latency), while a ring topology is sequential (O(G) or O(M) latency), making it extremely sensitive to stragglers. The author is suggested to provide a formal analysis of the communication complexity for each topology. The final convergence guarantee should be presented in total wall-clock time or communication cost, not just rounds.
3. The analysis and algorithms assumes full participation from all clients in all groups in every global round, while partial participation is mentioned in the existing researches. Full participation is a highly unrealistic assumption in federated learning. The author is suggested to redesign the scenario where a subset of groups, and a subset of clients within those groups, are sampled for updates.
4. All experiments are image classification tasks, while no speech/NLP/time-series or cross-silo medical/IoT workloads where HFL is popular. The author is suggested to include at least one non-vision dataset.
5. Different ~𝜂 definitions per topology in the convergence analysis make it tricky for fair lr tuning. The experiments tune learning rates but lack in details. The author is suggested to report a transparent learning rate tuning strategy to ensure the results match the analysis.
6. The paper claims HFL is for scalability, not convergence acceleration and that "carefully selected single-tier FL configurations may actually converge faster." This is a strong claim, but the paper provides no theoretical or experimental comparison to single-tier FL. The author should add comparisons vs single-tier FL with tuned participation/aggregation under identical budgets.

**Questions:**

1. The derivation of the client drift bounds in Lemma 2 (Appendix C.1) is difficult to follow. For instance, the star-star derivation starts with "<=5" without justification for the constant '5'. Where is this number derived from?
2. Do the conclusions still hold with standard BatchNorm+momentum training? Or what will be the difference?
3. The analysis and experiments are presented in global rounds, which ignores the fact that a ring topology's round is slower than a star topology's. How would your primary conclusion that "ring-based top-tiers are superior" change if convergence was measured in wall-clock time?
4. Can you provide experimental results comparing your best HFL topology to a well-tuned single-tier FedAvg with N=100 clients?
5. How did you tune learning rate for the main results in Figure 2? Was it held constant across all four topologies to an unfair comparison, or tuned individually for each for a fair comparison?
6. How will the performance of the topologies change under a more realistic partial participation scheme, where only a subset of clients are active?

---

> ### Author Response · Authors · 2025-11-21
> **Response to Reviewer hta5 (Part 1)**
>
> Thank you for the review! Let us respond to the comments:
>
> #### **R1: Novelty and Theoretical Contributions**
>
> > Weakness: The paper claims a "first" unified comparison of all four topologies, but related work already analyzes subsets and topology variants. The author is suggested to clarify more clearly what is new in the math or convergence bounds beyond those existing researches.
>
> 1. **Lack of Ring-Ring analysis:** Currently, there is no convergence analysis for HFL under the Ring-Ring topology. While the standard Star-Star topology is extensively studied, followed by Star-Ring, the Ring-Star topology receives the least attention. Our unified framework bridges this gap by providing the first convergence analysis for the bi-level Ring structure.
> 2. **Unified analysis under consistent assumptions:** Existing studies on individual topologies rely on heterogeneous assumptions, making it impossible to establish a unified framework by simply combining prior works. For instance, reference [1] assumes $ρ$-Lipschitz continuity and convexity, while reference [2] relies on Bounded Gradient Norm and Bounded Model Parameter assumptions. In contrast, we adopt more general non-convex assumptions for all four topologies.
> 3. **Tighter upper bounds:** We achieve a convergence rate of $\mathcal{O}(\frac{1}{\sqrt{GPMKR}})$ across all four topologies. This matches state-of-the-art results for Star-Star HFL. Furthermore, for Star-Ring and Ring-Star, this represents a superior convergence rate compared to the $\mathcal{O}(\frac{1}{\sqrt{R}})$ rate derived in [2] and [3].
>
> #### **R4**: **Experiments on Non-Vision Dataset (NLP)**
>
> > Weakness: All experiments are image classification tasks, while no speech/NLP/time-series or cross-silo medical/IoT workloads where HFL is popular. The author is suggested to include at least one non-vision dataset.
>
> We have conducted new experiments on a non-vision dataset: the **SST-2 NLP classification task**, utilizing a simple MLP model common in basic federated NLP settings. The number of groups was set to 10, with 10 clients per group, which is consistent with the experimental settings in the main text. After 1,000 global rounds, the results from the SST-2 experiments are entirely consistent with our main conclusions. These new results will be integrated into the revised manuscript.
>
> | **Heterogeneity**                  | **Star-Star** | **Star-Ring** | **Ring-Star** | **Ring-Ring** |
> | ---------------------------------- | ------------- | ------------- | ------------- | ------------- |
> | Groups IID and Clients IID         | 69.61         | 72.36         | 80.50         | 81.42         |
> | Groups IID and Clients Non-IID     | 69.95         | 73.17         | 79.59         | 80.50         |
> | Groups Non-IID and Clients IID     | 68.12         | 73.85         | 79.13         | 81.08         |
> | Groups Non-IID and Clients Non-IID | 68.12         | 73.97         | 78.44         | 81.65         |
>
> #### **R5**: **Learning Rate Tuning Strategy**
>
> > Weakness: Different ~𝜂 definitions per topology in the convergence analysis make it tricky for fair lr tuning. The experiments tune learning rates but lack in details. The author is suggested to report a transparent learning rate tuning strategy to ensure the results match the analysis.
> >
> > Question: How did you tune learning rate for the main results in Figure 2? Was it held constant across all four topologies to an unfair comparison, or tuned individually for each for a fair comparison?
>
> We confirm that the learning rate ($\eta$) was indeed tuned individually for each of the four topologies to ensure they all operate near their optimal convergence speed. And we have reported a transparent learning rate tuning strategy in Figure 2 as follows.
>
> |         Topology          |     Search Space for $\eta$      | Optimal $\eta$ Selected |
> | :-----------------------: | :------------------------------: | :---------------------: |
> |       **Star-Star**       | $\{2, 1, 0.5, 0.2, 0.05, 0.01\}$ |           $1$           |
> | **Star-Ring / Ring-Star** | $\{2, 1, 0.5, 0.2, 0.05, 0.01\}$ |          $0.2$          |
> |       **Ring-Ring**       | $\{2, 1, 0.5, 0.2, 0.05, 0.01\}$ |         $0.05$          |
>
> These empirical observations perfectly **validate our theoretical finding** that to maintain an equivalent effective learning rate for comparable convergence, the required raw learning rates must follow the pattern: $\mathbf{\eta_{Star-Star} > \eta_{Star-Ring} \approx \eta_{Ring-Star} > \eta_{Ring-Ring}}$ (when $G \approx M$). The learning rate tuning strategy will be integrated into the revised manuscript.

---

> ### Author Response · Authors · 2025-11-21
> **Response to Reviewer hta5 (Part 2)**
>
> #### **R2: Wall-Clock Time and Communication Analysis (for W2 and Q3)**
>
> We have incorporated an analysis of wall-clock time as detailed below:
>
> The total wall-clock time $T$ is the sum of the total computation time ($T_{Comp}$) and the total communication time ($T_{Comm}$), i.e., $T = T_{Comp} + T_{Comm}$.
>
> Let us first consider the single-layer FL scenario with M clients. The wall-clock time for the star topology per round is
> $$
> T^{\text{Star}}=\max_{m=1}^M \\{\frac{S}{r_{\mathcal{GA} \to m, g}} +\tau_m+ \frac{S}{r_{m \to \mathcal{GA}, g}}\\},
> $$
> and the wall-clock time for the ring topology is
> $$
> T^{\text{Ring}}=\frac{S}{r_{\mathcal{GA} \to 1, g}} + \sum_{m=1}^M\tau_m+\sum_{m=2}^M \frac{S}{r_{m-1 \to m,g}} + \frac{S}{r_{M \to \mathcal{GA}, g}}.
> $$
>
> | Notation       | Description                                              |
> | :------------- | :------------------------------------------------------- |
> | $S$            | Model volume (size of parameters) transferred per round. |
> | $\mathcal{GA}$ | The Aggregation Server.                                  |
> | $\tau$         | Computation time (e.g., $\tau_m$ for client $m$).        |
> | $r$            | Transfer rate.                               |
>
> In bandwidth-limited edge computing scenarios (such as IoT/Edge AI), the gap in wall-clock time between star and ring topologies is often much smaller than the theoretical $O(1)$ vs. $O(M)$ ratio. In a star topology, when the server aggregates M clients simultaneously, the available bandwidth $r$ is split, dropping the effective speed to $r$/M. Therefore, transmission latency actually grows linearly, or $O(M)$. While the ring topology transmits sequentially, each link uses the full bandwidth $r$. Since the total latency is the sum of M transfers, it also scales as $O(M)$.
>
> We have formulated an analysis of the communication complexity for four HFL topologies.
>
> | Topology      | Total Wall-clock Time                                        |
> | :------------ | :----------------------------------------------------------- |
> | **Star-Star** | $R \left( \max_{g=1}^G \\{ \frac{S}{r_{\mathcal{GA} \to g}} + \tau_g+\frac{S}{r_{g \to \mathcal{GA}}} \\} + P \cdot \max_{g=1}^G \\{ \max_{m=1}^M \\{ \frac{S}{r_{\mathcal{G} \to m, g}} + \tau_{g,m}+ \frac{S}{r_{m \to \mathcal{G}, g}} \\} \\} \right)$ |
> | **Star-Ring** | $R \left( \max_{g=1}^G \\{ \frac{S}{r_{\mathcal{GA} \to g}} + \tau_g+ \frac{S}{r_{g \to \mathcal{GA}}} \\} + P \cdot \max_{g=1}^G \\{ \frac{S}{r_{\mathcal{G} \to 1, g}} +\sum_{m=1}^M\tau_{g,m}+ \sum_{m=2}^M \frac{S}{r_{m-1 \to m,g}} + \frac{S}{r_{M \to \mathcal{G}, g}} \\} \right)$ |
> | **Ring-Star** | $R \left( \left( \frac{S}{r_{\mathcal{GA} \to 1}} +\sum_{g=1}^G\tau_g+ \sum_{g=2}^G \frac{S}{r_{g-1 \to g}} + \frac{S}{r_{G \to \mathcal{GA}}} \right) + P \sum_{g=1}^G \left( \max_{m=1}^M \\{ \frac{S}{r_{\mathcal{G} \to m, g}} + \tau_{g,m}+ \frac{S}{r_{m \to \mathcal{G}, g}} \\} \right) \right)$ |
> | **Ring-Ring** | $R \left( \left( \frac{S}{r_{\mathcal{GA} \to 1}} +\sum_{g=1}^G\tau_g+ \sum_{g=2}^G \frac{S}{r_{g-1 \to g}} + \frac{S}{r_{G \to \mathcal{GA}}} \right) + P \sum_{g=1}^G \left( \frac{S}{r_{\mathcal{G} \to 1, g}} +\sum_{m=1}^M\tau_{g,m}+ \sum_{m=2}^M \frac{S}{r_{m-1 \to m,g}} + \frac{S}{r_{M \to \mathcal{G}, g}} \right) \right)$ |
>
> General notations follow the Notation Table. Specific to HFL: $\tau_{g}$ is the computation time for group aggregation at group $g$; $\tau_{g,m}$ is the computation time for client $m$ within group $g$. Subscripts denote the specific link directions (e.g., $r_{g-1 \to g}$ is the rate between group servers). $\mathcal{G}$ denotes the Group Server.
>
> Theoretically, the star topology offers advantages in parallel computation; however, in real-world scenarios, it often fails to achieve the ideal $O(1)$ level of parallelism. In HFL, the total wall-clock time faces challenges similar to those in single-layer FL. Many studies[4] [5] on HFL communication optimization highlight that bandwidth resources are diluted as the number of users increases, e.g., $r_{m-1\to m,g} \sim \frac{R_g}{M}$, where $R_g$ represents the total bandwidth resources of the edge server and $M$ denotes the total number of clients connected to that server. Substituting this relationship into the total wall-clock time for the Star-Star topology yields $T^{Star-Star}\_{total}=R \left( \max_{g=1}^G \\{ \frac{2SG}{R} + \tau_g \\} + P \cdot \max_{g=1}^G \\{ \max_{m=1}^M \\{ \frac{2SM}{R_g} + \tau_{g,m}\\} \\} \right)$, where the communication latency in the upper layer approaches $O(G)$, while the communication overhead in the lower layer approaches $O(M)$. In particular, the uplink transmission rate of workers is a major bottleneck in the training process that can lead to the straggler’s effect [6]. Therefore, in bandwidth-constrained real-world scenarios, the ring topology remains a competitive option.

---

> ### Author Response · Authors · 2025-11-21
> **Response to Reviewer hta5 (Part 3)**
>
> #### **R3**: **Analysis under Partial Participation**
>
> > Weakness: The analysis and algorithms assumes full participation from all clients in all groups in every global round, while partial participation is mentioned in the existing researches. Full participation is a highly unrealistic assumption in federated learning. The author is suggested to redesign the scenario where a subset of groups, and a subset of clients within those groups, are sampled for updates.
> >
> > Question: How will the performance of the topologies change under a more realistic partial participation scheme, where only a subset of clients are active?
>
> We have extended our theoretical framework to account for partial participation, where only a subset of groups ($S_1 \le G$) and a subset of clients within each group ($S_2 \le M$) participate in each round.
>
> | Topology | Convergence Bound|
> | :--- | :--- |
> | **Star-Star** | $\left( \frac{LA}{R} \frac{\sigma^2}{S_1 P S_2 K} \right)^{\frac{1}{2}} + \left( \frac{LA}{R} \frac{\hat{\zeta}^2}{S_2} \right)^{\frac{1}{2}} + \left( \frac{LA}{R} \frac{\zeta^2}{S_1} \right)^{\frac{1}{2}} + \left( \frac{L^2 A^2}{R^2} \frac{\sigma^2}{P^2 K} \right)^{\frac{1}{3}} + \left( \frac{L^2 A^2}{R^2} \frac{\sigma^2}{P S_2 K} \right)^{\frac{1}{3}} + {\left( \frac{L^2 A^2}{R^2} \frac{\hat{\zeta}^2}{P^2} \right)^{\frac{1}{3}} + \left( \frac{L^2 A^2}{R^2} \frac{\hat{\zeta}^2}{S_2} \right)^{\frac{1}{3}}} + {\left( \frac{L^2 A^2}{R^2} \zeta^2 \right)^{\frac{1}{3}}}+\frac{LA}{R}$ |
> | **Star-Ring** | $\left( \frac{LA}{R} \frac{\sigma^2}{S_1 P S_2 K} \right)^{\frac{1}{2}} + \left( \frac{LA}{R} \frac{\hat{\zeta}^2}{S_2} \right)^{\frac{1}{2}} + \left( \frac{LA}{R} \frac{\zeta^2}{S_1} \right)^{\frac{1}{2}} + \left( \frac{L^2 A^2}{R^2} \frac{\sigma^2}{P S_2 K} \right)^{\frac{1}{3}} + {\left( \frac{L^2 A^2}{R^2} \frac{\hat{\zeta}^2}{S_2} \right)^{\frac{1}{3}}} + {\left( \frac{L^2 A^2}{R^2} \zeta^2 \right)^{\frac{1}{3}}}+\frac{LA}{R}$ |
> | **Ring-Star** | $\left( \frac{LA}{R} \frac{\sigma^2}{S_1 P S_2 K} \right)^{\frac{1}{2}} + \left( \frac{LA}{R} \frac{\hat{\zeta}^2}{S_2} \right)^{\frac{1}{2}} + \left( \frac{LA}{R} \frac{\zeta^2}{S_1} \right)^{\frac{1}{2}} + \left( \frac{L^2 A^2}{R^2} \frac{\sigma^2}{S_1^2 P^2 K} \right)^{\frac{1}{3}} + \left( \frac{L^2 A^2}{R^2} \frac{\sigma^2}{S_1 P S_2 K} \right)^{\frac{1}{3}}  + {\left( \frac{L^2 A^2}{R^2} \frac{\hat{\zeta}^2}{S_1^2 P^2} \right)^{\frac{1}{3}} + \left( \frac{L^2 A^2}{R^2} \frac{\hat{\zeta}^2}{S_2} \right)^{\frac{1}{3}} } + { \left( \frac{L^2 A^2}{R^2} \frac{\zeta^2}{S_1} \right)^{\frac{1}{3}} }+\frac{LA}{R}$ |
> | **Ring-Ring** | $\left( \frac{LA}{R} \frac{\sigma^2}{S_1 P S_2 K} \right)^{\frac{1}{2}} + \left( \frac{LA}{R} \frac{\hat{\zeta}^2}{S_2} \right)^{\frac{1}{2}} + \left( \frac{LA}{R} \frac{\zeta^2}{S_1} \right)^{\frac{1}{2}} + \left( \frac{L^2 A^2}{R^2} \frac{\sigma^2}{S_1 P S_2 K} \right)^{\frac{1}{3}} + {\left( \frac{L^2 A^2}{R^2} \frac{\hat{\zeta}^2}{S_2} \right)^{\frac{1}{3}} } + {\left( \frac{L^2 A^2}{R^2} \frac{\zeta^2}{S_1} \right)^{\frac{1}{3}} }+\frac{LA}{R}$ |
>
> We use the simple random sampling strategies studied in [7]. Our bounds hold for both sampling without/with replacement.
>
> Taking the Star-Star topology as an example, the convergence rate under the partial participation setting introduces additional terms $\left( \frac{LA}{R} \frac{\hat{\zeta}^2}{S_2} \right)^{\frac{1}{2}}$,  $\left( \frac{LA}{R} \frac{\zeta^2}{S_1} \right)^{\frac{1}{2}}$ and $\left( \frac{L^2 A^2}{R^2} \frac{\hat{\zeta}^2}{S_2} \right)^{\frac{1}{3}}$ compared to the rate presented in the main text. Consequently, partial participation slows down the convergence of the Star-Star topology. The magnitude of this degradation is correlated with the participation ratio; specifically, lower client participation results in a more significant reduction in convergence speed.

---

> ### Author Response · Authors · 2025-11-21
> **Response to Reviewer hta5 (Part 4)**
>
> #### **R6**: **Comparison with Single-Tier FL**
>
> > Weakness: The paper claims HFL is for scalability, not convergence acceleration and that "carefully selected single-tier FL configurations may actually converge faster." This is a strong claim, but the paper provides no theoretical or experimental comparison to single-tier FL. The author should add comparisons vs single-tier FL with tuned participation/aggregation under identical budgets.
> >
> > Question: Can you provide experimental results comparing your best HFL topology to a well-tuned single-tier FedAvg with N=100 clients?
>
> We have added experimental comparisons with single-layer FL using the CIFAR-10 dataset and a ResNet-18 model with N=100. After 5,000 global training rounds, single-layer FedAvg achieved an accuracy of 88.52% on the IID partition and 86.22% on the Non-IID partition, demonstrating a convergence speed similar to that of Star-Star HFL. Meanwhile, single-layer Sequential FL achieved an accuracy of 91.79% (IID) and 90.13% (Non-IID), exhibiting a convergence speed comparable to Ring-Ring HFL. These results **validate our conclusion** that the bi-level structure itself does not inherently accelerate convergence.
>
> Detailed experimental plots, including the convergence curves, will be included in the revised manuscript.
>
> #### **R7**: **Justification for the Constant '5'**
>
> > Question: The derivation of the client drift bounds in Lemma 2 (Appendix C.1) is difficult to follow. For instance, the star-star derivation starts with "<=5" without justification for the constant '5'. Where is this number derived from?
>
> The constant '5' arises from the application of the generalized  Jensen's Inequality, $\\|\sum_{i=1}^n a_i \\|^2 \leq n \sum_{i=1}^n \\|a_i\\|^2$, or a variant of this. Specifically, the drift term $\\mathbb{E}[\\|x_{g,p,m,k} - x\\|^2]$ in the Star-Star derivation is expanded as a sum of multiple terms (five main terms in this case, representing different types of errors accumulated over the local steps, group rounds, and global rounds). The expression you referenced is derived by applying the following decomposition and inequality:
> $$
> {\\|A \\|}^2 \leq  5\\|A-B\\|^2 +5\\|B-C\\|^2 +5\\|C-D\\|^2 +5\\|D-E\\|^2 +5\\|E\\|^2
> $$
> In our proof, the term $\mathbf{x}_{g,p,m,k} - \mathbf{x}$ is decomposed into a sum of $n=5$ error/drift components (e.g., local SGD noise, intra-group drift, inter-group drift, and global model discrepancy). Applying the inequality with $n=5$ yields the factor '5' that scales the resulting individual squared terms.
>
> We will revise the Appendix C.1 to include a more complete and transparent derivation, clearly stating the decomposition of the drift term into its constituent $n$ components and explicitly citing the inequality used to obtain the constant factor.
>
> #### **R8**: **Robustness to BatchNorm and Momentum**
>
> > Question: Do the conclusions still hold with standard BatchNorm+momentum training? Or what will be the difference?
>
> We conducted experiments with ResNet-18+ SGD (momentum=0.9) on CIFAR-10 under non-IID settings (Dirichlet α=0.1, 3000 rounds):
>
> Results:
> | Topology| Test Accuracy (%) |
> |:----------------------:|:-----------------:|
> | Star-Star | 83.80 |
> | Star-Ring | 86.09 |
> | Ring-Star | 89.59 |
> | Ring-Ring | 89.12 |
>
> The results confirm that our conclusions still hold. Extending the theoretical analysis to incorporate BatchNorm and momentum in HFL is a meaningful direction, and we would like to leave the rigorous theoretical derivation for future work.
>
> > [1] Fang, Qingze, et al. "Olive branch learning: A topology-aware federated learning framework for space-air-ground integrated network." *IEEE Transactions on Wireless Communications* 22.7 (2022): 4534-4551.
> >
> > [2] Ding, Yucheng, et al. "Distributed optimization over block-cyclic data." *Proceedings of the 6th ACM International Conference on Multimedia in Asia Workshops*. 2024.
> >
> > [3] Lee, Jin-woo, et al. "Tornadoaggregate: Accurate and scalable federated learning via the ring-based architecture." *arXiv preprint arXiv:2012.03214* (2020).
> >
> > [4] Lim, Wei Yang Bryan, et al. "Dynamic edge association and resource allocation in self-organizing hierarchical federated learning networks." *IEEE Journal on Selected Areas in Communications* 39.12 (2021): 3640-3653.
> >
> > [5] Liu, Chang, Jun Zhao, and Yitong Wang. "Computation and Communication Resource Optimization for Efficient Hierarchical Federated Learning." *IEEE Transactions on Networking* (2025).
> >
> > [6] Lim, Wei Yang Bryan, et al. "Hierarchical incentive mechanism design for federated machine learning in mobile networks." *IEEE Internet of Things Journal* 7.10 (2020): 9575-9588.
> >
> >[7] Yang, Haibo, Minghong Fang, and Jia Liu. "Achieving linear speedup with partial worker participation in non-iid federated learning." arXiv preprint arXiv:2101.11203 (2021).

---

### Author Response · Authors · 2025-12-03
**Public Summary Comment by Authors**

# Paper Summary

This paper develops the first unified theoretical framework for analyzing and comparing all four HFL topologies (Star-Star, Star-Ring, Ring-Star, Ring-Ring) under non-convex objectives and varying levels of data heterogeneity. Within this framework, we derive convergence bounds that explicitly capture the roles of the number of groups $G$, clients per group $M$, local steps $K$, and group rounds $P$, and we extend the analysis to partial client participation with corresponding guarantees. From this theory we obtain three main design insights: (i) HFL primarily improves scalability rather than inherently accelerating convergence, with ring-based top tiers (Ring-Star, Ring-Ring) outperforming star-based ones under heterogeneity; (ii) inter-group heterogeneity dominates convergence behavior, so grouping should prioritize minimizing inter-group divergence over intra-group refinement; and (iii) the optimal topology is strongly tied to group structure, with Ring-Star favored for many small groups and Star-Ring for a few large, dense groups.

We validate these insights via extensive experiments on CIFAR-10, CINIC-10, Fashion-MNIST, and SST-2 with diverse models, showing consistent accuracy gains from principled topology selection, especially for ring-based top tiers in heterogeneous regimes. Collectively, our results provide practical, theoretically grounded guidance for choosing HFL topologies based on deployment constraints rather than heuristics.

Our contributions are threefold: (1) a unified convergence analysis that, for the first time, covers all four HFL configurations, including the previously unstudied Ring-Ring topology; (2) a consistent non-convex analytical setup that enables direct comparison across topologies and yields tighter bounds of order $\mathcal{O}(1/\sqrt{GPMKR})$ for Star-Ring and Ring-Star, improving over prior $\mathcal{O}(1/\sqrt{R})$ results; and (3) a systematic empirical study demonstrating that these theoretical prescriptions translate into concrete performance benefits in realistic heterogeneous HFL deployments.

# Rebuttal Summary

## Summary of Reviews and Responses

We received four reviews with ratings of 6 (hta5), 4 (qwXN), 2 (my4H), and 6 (2b4g). Below we summarize how we have addressed each reviewer's concerns:

### Reviewer hta5 (Rating 6 - Marginally above acceptance threshold)

**Key concerns:** Novelty claims, communication costs analysis, full participation assumption, limited datasets, learning rate tuning, and lack of comparison with single-tier FL.

**Our response:** We have comprehensively addressed all concerns:

- Clarified our three key theoretical contributions beyond prior work: (1) first convergence analysis for Ring-Ring topology, (2) unified analysis under consistent assumptions, and (3) tighter convergence bounds ($\mathcal{O}(1/\sqrt{GPMKR})$ vs prior $\mathcal{O}(1/\sqrt{R})$.
- Added wall-clock time analysis demonstrating that in bandwidth-constrained environments, the theoretical latency advantage of star topology is often unrealized
- Extended our theoretical framework to include partial participation with convergence bounds
- Conducted new experiments on SST-2 NLP dataset, confirming our theoretical insights hold beyond vision tasks
- Provided transparent learning rate tuning strategy with topology-specific optimal rates
- Added comparisons with single-tier FL, validating that HFL primarily addresses scalability constraints rather than inherently accelerating convergence

### Reviewer qwXN (Rating 4 - Marginally below acceptance threshold)

**Key concerns:** Lack of detailed proofs, limited novelty, only vision datasets, no hyperparameter impact study, unvalidated claims about scalability, and missing error bars.

**Our response:** We have thoroughly addressed these concerns:

- Committed to including complete derivations for all topologies in the appendix
- Clarified our three distinct theoretical contributions (same as for hta5)
- Added experiments on SST-2 NLP dataset
- Added error bars to all figures and reported accuracy ranges across multiple runs
- Expanded discussion on group structure impact and committed to adding analysis of local steps/communication rounds
- Provided empirical validation comparing HFL with single-tier FL under identical conditions

### Reviewer 2b4g (Rating 6 - Marginally above acceptance threshold)

**Key concerns:** Suggested matrix-based derivation approach and requested learning rate details.

**Our response:** We have addressed these focused concerns:

- Explained why matrix-based derivation is challenging for HFL (topology-dependent update mechanisms) and committed this direction to future work
- Provided transparent learning rate tuning strategy showing topology-specific optimal rates that align with our effective learning rate theory

---

> ### Author Response · Authors · 2025-12-03
>
> ## Addressing Reviewer my4H (Rating 2)
>
> While we appreciate the review, several concerns appear based on misunderstandings or misrepresentations of our work:
>
> 1. **Communication efficiency concern**: The reviewer states our argument about limited parallelism in star topology "stands in the opposite direction to a bunch of literature in federated learning." This is inaccurate. The communication bottleneck in star topology has been widely acknowledged since the seminal FedAvg paper (McMahan et al., 2017) which explicitly states: "In contrast, in federated optimization communication costs dominate — we will typically be limited by an upload bandwidth of 1 MB/s or less." This remains a consensus in the field as evidenced by numerous works on gradient compression, client selection, and hierarchical architectures.
>
> 2. **Big-O notation criticism**: The reviewer claims "This is not true" regarding our asymptotic analysis. Our usage of Big-O notation is mathematically sound and follows standard conventions in optimization literature. We will clarify in the revision that G, M, P, K are treated as constants when focusing on asymptotic scaling with respect to R.
>
> 3. **Novelty concern**: The reviewer maintains that "the analysis of FedAvg under the star-star, ring-star, and star-ring topologies has already been studied in prior works." While partial analyses exist, our paper makes three distinct contributions:
>    - First convergence analysis for the previously unstudied Ring-Ring topology
>    - Unified analysis under consistent assumptions (prior studies used heterogeneous assumptions that prevent direct comparison)
>    - Tighter convergence bounds across all topologies
>
> As emphasized in the widely-cited work by Koloskova et al. (ICML 2020), "proving convergence rates in a unified framework is much more powerful than studying individual special cases on their own." We have followed this established precedent in optimization theory, and our framework enables fair "apples-to-apples" comparisons that were previously impossible.
>
> Our comprehensive responses address all technical concerns raised, and our revisions substantially strengthen both theoretical foundations and practical relevance of the paper. We believe this work makes meaningful contributions to federated learning theory and practice.

---

### Meta-Review · Area_Chair_LUsU · 2025-12-29

**Summary:**

The paper analyses 4 2-tier topologies for Hierarchical Federated Learning (HFL) (ring of rings, ring of stars, star of rings, star of stars). The theoretical upper bounds of convergence are described under common assumptions (smoothness of the local objective function, bounded variance of its gradient, inter and intra group heterogeneity bounds), and partial or full participation.

- the novelty is limited: a novel topology (ring-ring) is studied; other cases were partially covered in the literature
- however, the unified analysis presented in this paper is valuable, because previous works in HFL have different assumptions that make the comparison difficult
- the authors derive observations to guide practitioners, confirmed by experiments in vision and NLP
- the proofs are long and tedious, offering no novelty despite having to handle more complex interactions between the hierarchy’s levels.

**Reviewer Concerns:**

- hta5: yes, the authors have added analysis in communication cost and partial participation,  experiments on other domains than vision (NLP) and comparison with single tier topologies. They have clarified the role and tuning of the learning rate.
- qwXN: yes, complete proof are now in appendix, other concerns were covered by hta5.
- my4H: partially; the discussion about the presentation of the result in big O notation can be misunderstood
- 2b4g: the review was not very critical.

**Reviewer Scores:**

- hta5: gave 6 and I think it reflects his thinking.
- qwXN: could stay at 4 or raise to 6.
- my4H: was not convinced and I think that will keep 2.
- 2b4g: gave 6

---

### Decision · Program_Chairs · 2026-01-26

Accept (Poster)